# ICE-BeeM: Identifiable Conditional Energy-Based Deep Models Based on Nonlinear ICA

**Ilyes Khemakhem** [1]     **Ricardo P. Monti** [1]     **Diederik P. Kingma** [2]     **Aapo Hyvärinen** [3,4]

[1] Gatsby Unit, University College London
[2] Google Research
[3] Université Paris-Saclay, Inria
[4] University of Helsinki
ilyesk@gatsby.ucl.ac.uk

## Abstract

We consider the identifiability theory of probabilistic models and establish sufficient conditions under which the representations learned by a very broad family of conditional energy-based models are unique in function space, up to a simple transformation. In our model family, the energy function is the dot-product between two feature extractors, one for the dependent variable, and one for the conditioning variable. We show that under mild conditions, the features are unique up to scaling and permutation. Our results extend recent developments in nonlinear ICA, and in fact, they lead to an important generalization of ICA models. In particular, we show that our model can be used for the estimation of the components in the framework of Independently Modulated Component Analysis (IMCA), a new generalization of nonlinear ICA that relaxes the independence assumption. A thorough empirical study shows that representations learned by our model from real-world image datasets are identifiable, and improve performance in transfer learning and semi-supervised learning tasks.

## 1   Introduction

A central question in unsupervised deep learning is how to learn nonlinear representations that are a faithful reconstruction of the true latent variables behind the data. This allows us to learn representations that are semantically meaningful, interpretable and useful for downstream tasks. Identifiability is fundamental for meaningful and principled disentanglement, and in applications such as causal discovery. However, this is a very difficult task: by definition, we never observe the latent variables; the only information directly available to us is given by the observed variables. Learning the true representations is only possible when the representation is identifiable: if, in the limit of infinite data, only a single representation function can fit the data. Conversely, if, in the limit of infinite data, multiple representation functions can fit the data, then the true representation function is unidentifiable.

Until recently (Hyvärinen and Morioka, 2016, 2017), results relating to identifiability of (explicit and implicit) latent variable models were mainly constrained to linear models (*e.g.*, as in linear ICA), as it was acknowledged that the flexibility of nonlinear mappings could yield arbitrary latent variables which fulfill model assumptions such as independence (Hyvärinen and Pajunen, 1999). However, it is now understood that nonlinear deep latent variable models can be identifiable provided we observe some additional auxiliary variables such that the latent variables are conditionally independent given the auxiliary variable. The approach was introduced using self-supervised learning by Hyvärinen et al. (2019), and Khemakhem et al. (2020) explicited a connection between nonlinear ICA and the framework of variational autoencoders. It was shortly followed by work by Sorrenson et al. (2020),

where a similar connection was made to flow-based models (Rezende and Mohamed, 2015). This signals the importance of identifiability in popular deep generative models.

We extend this trend to a broad family of (unnormalized) conditional energy-based models (EBM), using insight from the nonlinear ICA theory. EBMs offer unparalleled flexibility, mainly because they do not require the modeled densities to be normalized nor easy to sample from. In fact, the energy model we suggest will have universal approximation capabilities. The energy function we will consider is defined in two steps: we learn two feature extractors, parameterized by neural networks, one for each of the observed variables (dependent and conditioning); then, we set the energy function to be the dot-product of the learned features. The modeled conditional densities are defined to be the exponential of the negative energy function.

A first important contribution of this paper is to provide a set of sufficient mild conditions to be satisfied by the feature extractors, which would guarantee their identifiability: they learn representations that are unique up to a linear transformation. In addition, by slightly altering the definition of the energy function, we prove the linear transformation is essentially a permutation. These conditions are functional, *i.e.* they abstract away the architecture of the networks. As a concrete example, we provide a neural network architecture based on fully connected layers, for which the functional conditions hold, and is thus identifiable. Moreover, we do not make any assumptions on the distributions of the learned representations. Effectively, this makes our family of models very flexible and adaptable to practical problems. We call this model Identifiable Conditional Energy-Based deep Models, or ICE-BeeM for short.

Our second contribution is to develop a framework we call Independently Modulated Component Analysis (IMCA): a deep latent variable model where the latents are non-independent (thus generalizing nonlinear ICA), with an arbitrary global dependency structure. Nonlinear ICA research has formalized the trade-off between expressivity of the mapping between latents to observations (from linear to nonlinear) and distributional assumptions over latent variables (from independent to conditionally independent given auxiliary variables). However, the need for (conditional) independence in order to obtain identifiability results may sometimes be seen as a limitation, for example in the context of learning disentangled representations. Therefore, it would be important to relax the assumption of independence while maintaining identifiability. This was achieved before in the linear case (Monti and Hyvärinen, 2018; Hyvärinen and Hurri, 2004), and we show how it may be achieved in the nonlinear setting. We show how our ICE-BeeM can estimate this generative model, thus connecting both the generative and non-generative views.

Finally, we show empirically that ICE-BeeM learns identifiable representations from real-world image datasets. As a further, rather different application of our results, we show how identifiability of ICE-BeeM can be leveraged for transfer learning and semi-supervised learning. In fact, we believe that the identifiability results are generally important for principled application of EBMs, whether for the purposes of disentanglement or otherwise.

## 2 Identifiable conditional energy-based deep models

In this section, we define ICE-BeeM , and study its properties. All proofs can be found in Appendix C.

### 2.1 Model definition

We collect a dataset of observations of tuples $(\mathbf{x}, \mathbf{y})$, where $\mathbf{x} \in \mathcal{X} \subset \mathbb{R}^{d_x}$ is the main variable of interest, also called the dependent variable, and $\mathbf{y} \in \mathcal{Y} \subset \mathbb{R}^{d_y}$ is an auxiliary variable also called the conditioning variable.

Consider two feature extractors $\mathbf{f}_{\boldsymbol{\theta}}(\mathbf{x}) \in \mathbb{R}^{d_z}$ and $\mathbf{g}_{\boldsymbol{\theta}}(\mathbf{y}) \in \mathbb{R}^{d_z}$, which we parameterize by neural networks, and $\boldsymbol{\theta}$ is the vector of weights and biases. To alleviate notations, we will drop $\boldsymbol{\theta}$ when it's clear which quantities we refer to. These feature extractors are used to define the conditional energy function $\mathcal{E}_{\boldsymbol{\theta}}(\mathbf{x}|\mathbf{y}) = \mathbf{f}_{\boldsymbol{\theta}}(\mathbf{x})^T \mathbf{g}_{\boldsymbol{\theta}}(\mathbf{y})$.

The parameter $\boldsymbol{\theta}$ lives in the space $\Theta$ which is defined such that the normalizing constant $Z(\mathbf{y}; \boldsymbol{\theta}) = \int_{\mathcal{X}} \exp(-\mathcal{E}_{\boldsymbol{\theta}}(\mathbf{x}|\mathbf{y})) \mathrm{d}\mathbf{x} < \infty$ is finite. Our family of conditional energy-based models has the form:

$$p_{\boldsymbol{\theta}}(\mathbf{x}|\mathbf{y}) = \frac{\exp(-\mathbf{f}_{\boldsymbol{\theta}}(\mathbf{x})^T \mathbf{g}_{\boldsymbol{\theta}}(\mathbf{y}))}{Z(\mathbf{y}; \boldsymbol{\theta})} \tag{1}$$

As we will see later, this choice of energy function is not restrictive, as our model has powerful theoretical guarantees: universal approximation capabilities and strong identifiability properties. There exists a multitude of methods we can use to estimate this model (Hyvärinen, 2005; Gutmann and Hyvärinen, 2010; Ceylan and Gutmann, 2018; Uehara et al., 2020). In this work, we will use Flow Contrastive Estimation (Gao et al., 2019) and Denoising Score Matching (Vincent, 2011), which are discussed and extended to the conditional case in Appendix B.

## 2.2 Identifiability

As stated earlier, we want our model to learn meaningful representations of the dependent and conditioning variables. In particular, when learning two different models of the family (1) from the same dataset, we want the learned features to be very similar.

This similarity between representations is better expressed as equivalence relations on the parameters $\boldsymbol{\theta}$ of the network, which would characterize the form of identifiability we will end up with for our energy model. This notion of identifiability up to equivalence class was introduced by Khemakhem et al. (2020) to address the fact that there typically exist many choices of neural network parameters $\boldsymbol{\theta}$ that map to the same point in function-space. In our case, it is given by the following definitions:

**Definition 1** (Weak identifiability). *Let $\sim_w^{\mathbf{f}}$ and $\sim_w^{\mathbf{g}}$ be equivalence relations on $\Theta$ defined as:*

$$\boldsymbol{\theta} \sim_w^{\mathbf{f}} \boldsymbol{\theta}' \Leftrightarrow \forall \mathbf{x}, \mathbf{f}_{\boldsymbol{\theta}}(\mathbf{x}) = \mathbf{A}\mathbf{f}_{\boldsymbol{\theta}'}(\mathbf{x}) + \mathbf{c}$$
$$\boldsymbol{\theta} \sim_w^{\mathbf{g}} \boldsymbol{\theta}' \Leftrightarrow \forall \mathbf{x}, \mathbf{g}_{\boldsymbol{\theta}}(\mathbf{y}) = \mathbf{B}\mathbf{g}_{\boldsymbol{\theta}'}(\mathbf{y}) + \mathbf{e} \tag{2}$$

*where $\mathbf{A}$ and $\mathbf{B}$ are $(d_z \times d_z)$-matrices of rank at least $\min(d_z, d_x)$ and $\min(d_z, d_y)$ respectively, and $\mathbf{c}$ and $\mathbf{e}$ are vectors.*

**Definition 2** (Strong identifiability). *Let $\sim_s^{\mathbf{f}}$ and $\sim_s^{\mathbf{g}}$ be the equivalence relations on $\Theta$ defined as:*

$$\boldsymbol{\theta} \sim_s^{\mathbf{f}} \boldsymbol{\theta}' \Leftrightarrow \forall i, \forall \mathbf{x}, f_{i,\boldsymbol{\theta}}(\mathbf{x}) = a_i f_{\sigma(i),\boldsymbol{\theta}'}(\mathbf{x}) + c_i$$
$$\boldsymbol{\theta} \sim_s^{\mathbf{g}} \boldsymbol{\theta}' \Leftrightarrow \forall i, \forall \mathbf{x}, g_{i,\boldsymbol{\theta}}(\mathbf{x}) = b_i g_{\gamma(i),\boldsymbol{\theta}'}(\mathbf{x}) + e_i \tag{3}$$

*where $\sigma$ and $\gamma$ are permutations of $[\![1, n]\!]$, $a_i$ and $b_i$ are non-zero scalars and $c_i$ and $e_i$ are scalars.*

Two parameters are thus considered equivalent if they parameterize two feature extractors that are equal up to a linear transformation (2) or a scaled permutation (3). The subscripts $w$ and $s$ stand for weak and strong, respectively. Special cases are discussed in Appendix C.1.

Identifiability in the context of probability densities modeled by neural networks can be seen as a study of degeneracy of the networks. In applications where the representations are used in a downstream classification task, the weak identifiability (2) may be enough. It guarantees that the hyperplanes defining the boundaries between classes in the feature space are consistent, up to a global rotation, and thus the downstream task may be unaffected. Strong identifiability (3), on the other hand, is crucial in applications where such rotation is undesirable. For example, Monti et al. (2019) propose an algorithm for causal discovery based on independence tests between the observations and latent variables learnt by solving a nonlinear ICA task. The tested independences only hold for the true latent noise variables. Were one to learn the latents only up to a rotation, such causal analysis method would not work at all.

### 2.2.1 Weak identifiability

This initial form of identifiability requires very few assumptions on the feature extractors $\mathbf{f}$ and $\mathbf{g}$. In fact, the conditions we develop here are easy to satisfy in practice, and we will see how in Section 2.3. Most importantly, our result also covers the case where the number of features is larger than the number of observed variables. As far as we know, this is the first identifiability result that extends to *overcomplete* representations in the *nonlinear* setting. The following theorem summarizes the main result. Intuition behind the conditions, as well as a proof under milder assumptions, can be found in Appendix C.2.

**Theorem 1.** *Let $\sim_w^{\mathbf{f}}$ and $\sim_w^{\mathbf{g}}$ be the equivalence relations in (2). Assume that for any choice of parameter $\boldsymbol{\theta}$:*

    *1. The feature extractor $\mathbf{f}_{\boldsymbol{\theta}}$ is differentiable, and its Jacobian $\mathbf{J}_{\mathbf{f}_{\boldsymbol{\theta}}}$ is full rank.[1]*

2. *There exist $d_z + 1$ points $\mathbf{y}^0, \dots, \mathbf{y}^{d_z}$ such that the matrix $\mathbf{R}_{\boldsymbol{\theta}} = \left(\mathbf{g}_{\boldsymbol{\theta}}(\mathbf{y}^1) - \mathbf{g}_{\boldsymbol{\theta}}(\mathbf{y}^0), \dots, \mathbf{g}_{\boldsymbol{\theta}}(\mathbf{y}^{d_z}) - \mathbf{g}_{\boldsymbol{\theta}}(\mathbf{y}^0)\right)$ of size $d_z \times d_z$ is invertible.*

*then $p_{\boldsymbol{\theta}}(\mathbf{x}|\mathbf{y}) = p_{\boldsymbol{\theta}'}(\mathbf{x}|\mathbf{y}) \implies \boldsymbol{\theta} \sim_w^{\mathbf{f}} \boldsymbol{\theta}'$.*

*With $\mathbf{f}_{\boldsymbol{\theta}}$ and $\mathbf{g}_{\boldsymbol{\theta}}$ switched, the same conclusion applies to $\mathbf{g}_{\boldsymbol{\theta}}$: $p_{\boldsymbol{\theta}}(\mathbf{x}|\mathbf{y}) = p_{\boldsymbol{\theta}'}(\mathbf{x}|\mathbf{y}) \implies \boldsymbol{\theta} \sim_w^{\mathbf{g}} \boldsymbol{\theta}'$.*

*Finally, if both assumptions 1 and 2 are satisfied by both feature extractors $\mathbf{f}_{\boldsymbol{\theta}}$ and $\mathbf{g}_{\boldsymbol{\theta}}$, then the matrices $\mathbf{A}$ and $\mathbf{B}$ in (2) have full row rank equal to $d_z$.*

### 2.2.2 Strong identifiability

We propose two different alterations to our energy function which will both allow for the stronger form of identifiability defined by $\sim_s^{\mathbf{f}}$ and $\sim_s^{\mathbf{g}}$ in (3). We will focus on $\mathbf{f}$, but the same results hold for $\mathbf{g}$ by a simple transposition of assumptions. Importantly, we will suppose that the output dimension $d_z$ is smaller than the input dimension $d_x$.

The first is based on restricting the feature extractor $\mathbf{f}$ to be non-negative. It will induce constraints on the matrix $\mathbf{A}$ defining the equivalence relation $\sim_w^{\mathbf{f}}$: loosely speaking, if $\mathbf{A}$ induces a rotation in space, then it will violate the non-negativity constraint, since the only rotation that maps the positive orthant of the plan to itself is the identity.

The second alteration is based on augmenting $\mathbf{f}$ by its square, effectively resulting in the $2d_z$-dimensional feature extractor $\tilde{\mathbf{f}}(\mathbf{x}) = (\dots, f_i(\mathbf{x}), f_i^2(\mathbf{x}), \dots) \in \mathbb{R}^{2d_z}$. This augmented feature map is combined with a $2d_z$-dimensional feature map $\tilde{\mathbf{g}}(\mathbf{y}) \in \mathbb{R}^{2d_z}$ for the conditioning variable $\mathbf{y}$, to define an augmented energy function $\tilde{\mathcal{E}}(\mathbf{x}|\mathbf{y}) = \tilde{\mathbf{f}}(\mathbf{x})^T \tilde{\mathbf{g}}(\mathbf{y})$. The advantage of this approach is that it doesn't require the feature extractors to be positive. However, it makes the effective size of the feature extractor equal to $2d_z$.

Identifiability results derived from these two alterations are summarized by the following theorem.

**Theorem 2.** *Assume that $d_z \leq d_x$ and that the assumptions of Theorem 1 hold. Further assume that, for any choice of parameter $\boldsymbol{\theta}$, either one of the following conditions hold:*

3. *The feature extractor $\mathbf{f}_{\boldsymbol{\theta}}$ is surjective, and its image is $\mathbb{R}_+^{d_x}$.*

4. *The feature extractor $\mathbf{f}_{\boldsymbol{\theta}}$ is differentiable and surjective, its Jacobian $\mathbf{J}_{\mathbf{f}_{\boldsymbol{\theta}}}$ is full rank; there exist $2d_z + 1$ points $\mathbf{y}^0, \dots, \mathbf{y}^{2d_z}$ such that the matrix $\tilde{\mathbf{R}}_{\boldsymbol{\theta}} = \left(\tilde{\mathbf{g}}_{\boldsymbol{\theta}}(\mathbf{y}^1) - \tilde{\mathbf{g}}_{\boldsymbol{\theta}}(\mathbf{y}^0), \dots, \tilde{\mathbf{g}}_{\boldsymbol{\theta}}(\mathbf{y}^{2d_z}) - \tilde{\mathbf{g}}_{\boldsymbol{\theta}}(\mathbf{y}^0)\right)$ of size $2d_z \times 2d_z$ is invertible; and we use the augmented energy function $\tilde{\mathcal{E}}(\mathbf{x}|\mathbf{y})$ in the definition of the model.*

*Then $p_{\boldsymbol{\theta}}(\mathbf{x}|\mathbf{y}) = p_{\boldsymbol{\theta}'}(\mathbf{x}|\mathbf{y}) \implies \boldsymbol{\theta} \sim_s^{\mathbf{f}} \boldsymbol{\theta}'$ where $\sim_s^{\mathbf{f}}$ is defined in (3).*

A more general form of the Theorem is provided in Appendix C.5. This theorem is fundamental as it proves very strong identifiability results for a conditional deep energy-based model. As far as we know, our results require the least amount of assumptions in recent theoretical work for functional identifiability of deep learning models (Khemakhem et al., 2020; Sorrenson et al., 2020). Most importantly, we do not make any assumption on the distribution of the latent features.

### 2.3 An identifiable neural network architecture

In this section, we give a concrete example of a neural network architecture that satisfies the functional assumptions of Theorem 1. We suppose that each of the networks $\mathbf{f}$ and $\mathbf{g}$ are parameterized as multi-layer perceptrons (MLP). More specifically, consider an MLP with $L$ layers, where each layer consists of a linear mapping with weight matrix $\mathbf{W}_l \in \mathbb{R}^{d_l \times d_{l-1}}$ and bias $\mathbf{b}_l \in \mathbb{R}^{d_l}$, followed by an activation function $h_l$. Consider the following architecture:

(a.) The activation functions $h_l$ are LeakyReLUs, $\forall l \in [\![1, L-1]\!]$.[2]

(b.) The weight matrices $\mathbf{W}_l$ are full rank (its rank is equal to its smaller dimension), $\forall l \in [\![1, L]\!]$.

(c.) The row dimension of the weight matrices are either monotonically increasing or decreasing: $d_l \geq d_{l+1}, \forall l \in [\![0, L-1]\!]$ or $d_l \leq d_{l+1}, \forall l \in [\![0, L-1]\!]$.

(d.) All submatrices of $\mathbf{W}_l$ of size $d_l \times d_l$ are invertible if $d_l < d_{l+1}$, $\forall l \in [\![0, L-1]\!]$.

This architecture satisfies the assumptions of Theorems 1 and 2, as is stated by the propositions below.

**Proposition 1.** *Consider an MLP* $\mathbf{f}$ *whose architecture satisfies assumptions (a.), (b.) and (c.), then* $\mathbf{f}$ *satisfies Assumption 1. If in addition,* $d_L \leq d_0$*, then* $\mathbf{f}$ *satisfies Assumption 4. Finally, if on top of that, we apply a ReLU to the output of the network, then* $\mathbf{f}$ *satisfies Assumption 3.*

**Proposition 2.** *Consider a nonlinear MLP* $\mathbf{g}$ *whose architecture satisfies assumptions (a.), (b.), and (d.).[3] Then,* $\mathbf{g}$ *satisfies Assumptions 2 and 4.*

While assumptions (a.)-(d.) might seem a bit restrictive, they serve the important goal of giving sufficient *architectural* conditions that correspond to the purely *functional* assumptions of Theorems 1 and 2. Note that the full rank assumptions are necessary to ensure that the learnt representations are not degenerate, since we lose information with low rank matrices. In practice, random initialization of floating point parameters, which are then optimized with stochastic updates (SGD), will result in weight matrices that are almost certainly full rank.

## 2.4 Universal approximation capability

With such a potentially overcomplete network, we can further achieve universal approximation of the data distribution. It might initially seem that this is an impossible endeavor given the somehow restricted form of the energy function. However, if we also consider the dimension $d_z$ of $\mathbf{f}$ and $\mathbf{g}$ as an additional architectural parameter that we can change at will, then we can always find an arbitrarily good approximation of the conditional probability density function:

**Theorem 3.** *Let* $p(\mathbf{x}|\mathbf{y})$ *be a conditional probability density. Assume that* $\mathcal{X}$ *and* $\mathcal{Y}$ *are compact Hausdorff spaces, and that* $p(\mathbf{x}|\mathbf{y}) > 0$ *almost surely* $\forall (\mathbf{x}, \mathbf{y}) \in \mathcal{X} \times \mathcal{Y}$*. Then for each* $\varepsilon > 0$*, there exists* $(\boldsymbol{\theta}, d_z) \in \Theta \times \mathbb{N}$*, where* $d_z$ *is the dimension of the feature extractor, such that* $\sup_{(\mathbf{x}, \mathbf{y}) \in \mathcal{X} \times \mathcal{Y}} |p_{\boldsymbol{\theta}}(\mathbf{x}|\mathbf{y}) - p(\mathbf{x}|\mathbf{y})| < \varepsilon$*.*

This means that our model is capable of approximating any conditional distribution that is positive on its compact support arbitrarily well. In practice, the optimal dimension $d_z$ of the feature extractors can be estimated using cross-validation for instance. It is possible that to achieve a near perfect approximation, we require a value of $d_z$ that is larger than the dimension of the input. This is why it is crucial that our identifiability result from Theorem 1 covers the overcomplete case as well, and highlights the importance of our contribution in comparison to previous identifiable deep models.

## 3 Independently modulated component analysis

Next, we show how ICE-BeeM relates to a generative, latent variable model. We develop here a novel framework that generalizes nonlinear ICA to non-independent latent variables, and show how we can use our energy model to estimate them.

**Model definition**   Assume we observe a random variable $\mathbf{x} \in \mathbb{R}^{d_x}$ as a result of a nonlinear transformation $\mathbf{h}$ of a latent variable $\mathbf{z} \in \mathbb{R}^{d_z}$. We assume the distribution of $\mathbf{z}$ is conditioned on an auxiliary variable $\mathbf{y} \in \mathbb{R}^{d_y}$, which is also observed:

$$\mathbf{z} \sim p(\mathbf{z}|\mathbf{y}) \quad , \quad \mathbf{x} = \mathbf{h}(\mathbf{z}) \tag{4}$$

We will suppose here that $d_x = d_z = d$. The proofs, as well as an extension to $d_z < d_x$, can be found in Appendix D. The main modeling assumption we make on the latent variable is that its density has the following form:

$$p(\mathbf{z}|\mathbf{y}) = \mu(\mathbf{z}) e^{\sum_{i=1}^{d_z} \mathbf{T}_i(z_i)^T \boldsymbol{\lambda}_i(\mathbf{y}) - \Gamma(\mathbf{y})} \tag{5}$$

where $\mu(\mathbf{z})$ is a base measure and $\Gamma(\mathbf{y})$ is the conditional normalizing constant. Crucially, the exponential term factorizes across components: the sufficient statistic $\mathbf{T}$ of this exponential family is composed of $d$ functions that are each a function of only one component $z_i$ of the latent variable $\mathbf{z}$.

Equations (4) and (5) together define a nonparametric model with parameters $(\mathbf{h}, \mathbf{T}, \boldsymbol{\lambda}, \mu)$. For the special case $\mu(\mathbf{z}) = \prod_i \mu_i(z_i)$, the distribution of $\mathbf{z}$ factorizes across dimensions, and the components

$z_i$ are independent. Then the generative model gives rise to a nonlinear ICA model, and it was studied to a great depth by Khemakhem et al. (2020).

We propose to generalize such earlier models by allowing for an arbitrary base measure $\mu(\mathbf{z})$, *i.e.* the components of the latent variable are no longer independent, as $\mu$ doesn't necessarily factorize across dimensions. We call this new framework *Independently Modulated Component Analysis* (IMCA). We show in Appendix E that the strong identifiability guarantees developed for nonlinear ICA can be extended to IMCA, yielding a more general and more flexible principled framework for representation learning and disentanglement.

**Estimation by ICE-BeeM**     Guided by the strong identifiability results above, we suggest augmenting our feature extractor $\mathbf{f}$ by output activation functions, resulting in the modified feature map $\tilde{\mathbf{f}}(\mathbf{x}) = (\mathbf{H}_1(f_1(\mathbf{x})), \ldots, \mathbf{H}_d(f_d(\mathbf{x})))$. In Section 2.2.2 for instance, we used $\mathbf{H}_i(x) = (x, x^2)$. These output nonlinearities play the role of sufficient statistics to the learnt representation $\mathbf{f}_{\boldsymbol{\theta}}(\mathbf{x})$, and have a double purpose: to allow for strong identifiability results, and to match the dimensions of the components $\mathbf{T}_i$ of sufficient statistic in (5), as formalized by the following theorem.

**Theorem 4.**  *Assume:*

*(i) The observed data follows the exponential IMCA model of equations (4)-(5).*

*(ii) The mixing function $\mathbf{h}$ is a $\mathcal{D}^2$-diffeomorphism.[4]*

*(iii) The sufficient statistics $\mathbf{T}_i$ are twice differentiable, and the functions $T_{ij} \in \mathbf{T}_i$ are linearly independent on any subset of $\mathcal{X}$ of measure greater than zero. Furthermore, they all satisfy $\dim(\mathbf{T}_i) \geq 2$, $\forall i$; or $\dim(\mathbf{T}_i) = 1$ and $\mathbf{T}_i$ is non-monotonic $\forall i$.*

*(iv) There exist $k + 1$ distinct points $\mathbf{y}^0, \ldots, \mathbf{y}^k$ such that the matrix $\mathbf{L} = (\boldsymbol{\lambda}(\mathbf{y}_1) - \boldsymbol{\lambda}(\mathbf{y}_0), \ldots, \boldsymbol{\lambda}(\mathbf{y}_k) - \boldsymbol{\lambda}(\mathbf{y}_0))$ of size $k \times k$ is invertible, where $k = \sum_{i=1}^{d} \dim(\mathbf{T}_i)$.*

*(v) We use a consistent estimator to fit the model (1) to the conditional density $p(\mathbf{x}|\mathbf{y})$, where we assume the feature extractor $\mathbf{f}(\mathbf{x})$ to be a $\mathcal{D}^2$-diffeomorphism and $d$-dimensional, and the vector-valued pointwise nonlinearities $\mathbf{H}_i$ to be differentiable and $k$-dimensional, and their dimensions to be chosen from $(\dim(\mathbf{T}_1), \ldots, \dim(\mathbf{T}_d))$ without replacement.*

*Then, in the limit of infinite data,   $\mathbf{H}_i(f_i(\mathbf{x})) = \mathbf{A}_i \mathbf{T}_{\gamma(i)}(z_{\gamma(i)}) + \mathbf{b}_i$ where $\gamma$ is a permutation of $[\![1, d]\!]$ such that $\dim(\mathbf{H}_i) = \dim(\mathbf{T}_{\gamma(i)})$ and $\mathbf{A}_i$ is an invertible square matrix; that is: we can recover the latent variables up to a block permutation linear transformation and point-wise nonlinearities.*

# 4   Relation to previous work on nonlinear ICA

Our results greatly extend existing identifiability results and models. The closest latent variable model identifiability theory to ours is that of nonlinear ICA theory (Hyvärinen and Morioka, 2016; Hyvärinen et al., 2019; Khemakhem et al., 2020). These works formalized a trade-off between distributional assumptions over latent variables (from linear and independent to nonlinear but conditionally independent given auxiliary variables) that would lead to identifiability.

On this front, our first contribution was to identify that conditional independence is not necessary for identifiability, and to propose the more general IMCA framework. Our proofs extend previous ones to the non-independent case, and are the most general to date, even considering linear ICA theory. In fact, as a second contribution, our conditional EBM generalizes previous results by completely dropping any distributional assumptions on the representations—which are ubiquitous in the latent variable case.

Third, most of our theoretical results hold for overcomplete representations, which means that unlike the earlier works cited above, our model can be shown to even have universal approximation capabilities. Fourth, while recent identifiability theory focused on providing functional conditions for identifiability, such work is a bit removed from the reality of neural network training. Our results on network architectures are the first step towards bridging the gap between theory and practice.

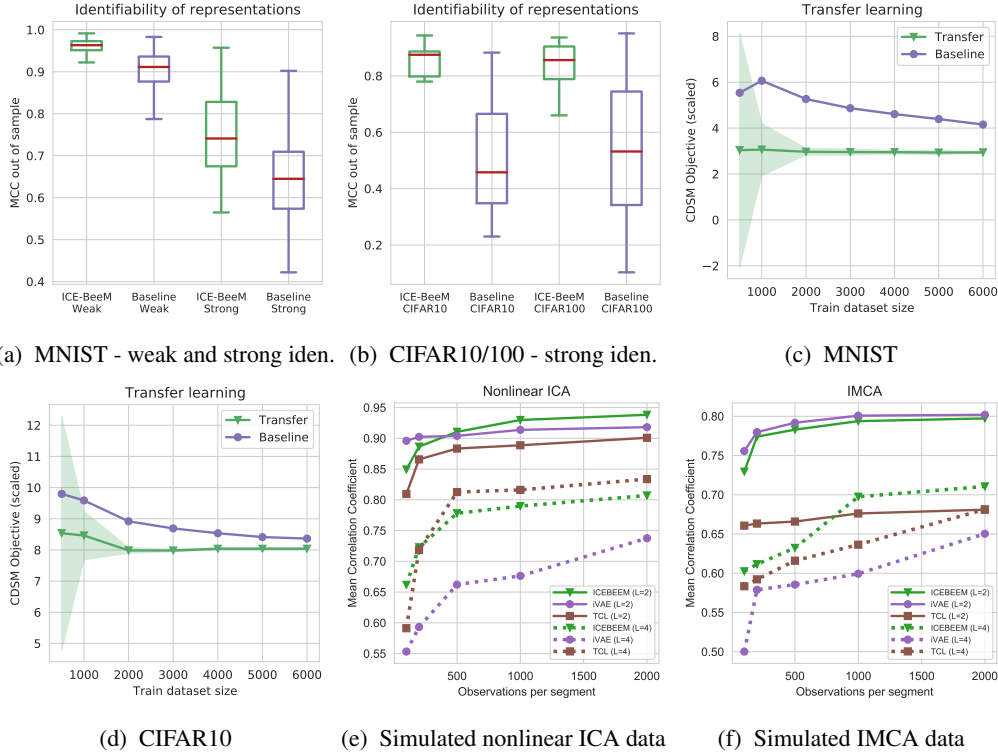

Figure 1: $(a) - (b)$ Quantifying the identifiability of learnt representations using MCC (higher is better). $(c) - (d)$ Transfer learning onto unseen classes using denoising score matching objective (lower is better). $(e) - (f)$ Simulations on artificial nonlinear ICA/IMCA data (higher is better).

## 5 Experiments

### 5.1 Identifiability of representations on image datasets

We explore the importance of identifiability and the applicability of ICE-BeeM in a series of experiments on image datasets (MNIST, FashionMNIST, CIFAR10 and CIFAR100). First, we investigate the identifiability of ICE-BeeM by comparing representations obtained from different random initializations, using an unconditional EBM as a baseline. We further present applications to transfer and semi-supervised learning, where we find identifiability leads to significant improvements. The different architectures used throughout these experiments are described in Appendix A.1. Code for reproducibility is available here.

**Quantifying identifiability** We start by empirically validating Theorems 1 and 2 on image datasets. Briefly, these theorems provided conditions for weak and strong identifiability of latent representations, respectively. We propose to study the weak and strong identifiability properties of both conditional and unconditional EBMs by training such models multiple times using distinct random initializations. We subsequently compute the mean correlation coefficient (MCC, see Appendix A.2) between learned representations obtained via distinct random initializations; consistent high MCCs indicate the model is identifiable. In the context of weak identifiability, we consider the MCC up to a linear transformation, $\mathbf{A}$, as defined in (2). Throughout experiments, we employ CCA to learn the linear mapping $\mathbf{A}$. However, our main interest is studying the strong identifiability of EBM architectures, defined in (3). To this end we consider the MCC directly on inferred representations (i.e., without a linear mapping $\mathbf{A}$). Both an ICE-BeeM model and an unconditional EBM were trained on three distinct image datasets: MNIST, CIFAR 10 and 100. For each dataset, we train models using 20 distinct random initializations and compare inferred representations. Conditional denoising score matching (CDSM, see Appendix B.1) was employed to train all networks. Results presented in

Table 1: $(a)$ Transfer learning. $(b)$ Semi-supervised learning.

(a) CDSM objective (lower is better)

| Dataset | $\mathbf{f} \cdot \mathbf{g}_{\boldsymbol{\theta}}$ | $\mathbf{f} \cdot \mathbf{1}$ | $\mathbf{f}_{\boldsymbol{\theta}} \cdot \mathbf{g}_{\boldsymbol{\theta}}$ | $\mathbf{f}_{\boldsymbol{\theta}} \cdot \mathbf{1}$ |
|---|---|---|---|---|
| MNIST | **2.95** | 23.43 | 4.22 | 3.64 |
| CIFAR10 | **8.03** | 23.08 | 8.37 | 8.16 |

(b) Classification Accuracy (higher is better)

| Dataset | ICE-BeeM | Uncond. EBM |
|---|---|---|
| FMNIST | **77.07 ± 1.39** | 56.33 ± 3.18 |
| CIFAR10 | **64.42 ± 1.09** | 51.88 ± 1.33 |

Figures [1a-1b] show that for ICE-BeeM, the representations were more consistent, both in the weak and the strong case, thus validating our theory. See Appendix A.3 for further details and experiments.

**Application to transfer learning** Second, we present an application of ICE-BeeM to transfer learning. We suppose that the auxiliary variable $y \in \mathbb{R}$ is the index of a dataset or a task. We propose an intuitively appealing approach, where we approximate the unnormalized log-pdf in $y$-th dataset $\log p(\mathbf{x}|y)$ by a linear combination of a learned "basis" functions $f_{i,\boldsymbol{\theta}}$ as $\log p(\mathbf{x}; y) + \log Z(\boldsymbol{\theta}) \approx \sum_{i=1}^{k} g_i(y) f_{i,\boldsymbol{\theta}}(\mathbf{x})$, where the $g_i(y)$ are scalar parameters that act as coefficients in the basis $(f_{i,\boldsymbol{\theta}})$. For a new unseen dataset $y_{\text{new}}$, reducing the transfer learning to the estimation of the $g_i(y_{\text{new}})$ clearly requires that we have estimated the true $f_i$, at least up to a linear transformation. This is exactly what can be achieved by ICE-BeeM based on weak identifiability. To this end, an ICE-BeeM model was trained on classes 0-7 of MNIST and CIFAR10 using the CDSM objective. After training, we fix $\mathbf{f}$ and learn $\mathbf{g}_{\boldsymbol{\theta}}(y_{\text{new}})$ for the unseen classes (we denote this by $\mathbf{f} \cdot \mathbf{g}_{\boldsymbol{\theta}}$; unseen classes are 8 & 9). We allow $\mathbf{g}_{\boldsymbol{\theta}}$ to be parameterized by a vector for each class, which leads to a drastic simplification for the new classes. We compare against a baseline where both $\mathbf{f}_{\boldsymbol{\theta}}$ and $\mathbf{g}_{\boldsymbol{\theta}}$ are trained directly on data from unseen classes only (*i.e.* there is no transfer learning—denoted $\mathbf{f}_{\boldsymbol{\theta}} \cdot \mathbf{g}_{\boldsymbol{\theta}}$). Results are presented in Figures [1c] and [1d] where we vary the sample size of the unseen classes and report the CDSM objective. Overall, the use of a *pretrained* $\mathbf{f}$ network improves performance, demonstrating effective transfer learning. We also compare against a baseline where we just evaluate the pretrained $\mathbf{f}$ on the new classes, while fixing $\mathbf{g} = \mathbf{1}$ (without learning the new coefficients—denoted $\mathbf{f} \cdot \mathbf{1}$); and a baseline where we estimate an unconditional EBM using new classes only (no transfer—denoted $\mathbf{f}_{\boldsymbol{\theta}} \cdot \mathbf{1}$). The average CDSM scores are reported in Table [1a], where the transfer learning with an identifiable EBM (i.e., using ICE-BeeM ) performs best. See Appendix A.4 for further details and experiments. We note here that based on strong identifiability, we could impose sparsity on the coefficients $g_i(y)$, which might improve the results even further.

**Application to semi-supervised learning** Finally, we also highlight the benefits of identifiability in the context of semi-supervised learning. We compared training both an identifiable ICE-BeeM model and an unconditional (non-identifiable) EBM on classes 0-7 and employing the learned features $\mathbf{f}_{\boldsymbol{\theta}}$ to classify unseen classes 8-9 using a logistic regression. In both cases, training proceeded via CDSM. Table [1b] reports the classification accuracy over unseen classes. We note that ICE-BeeM obtains significantly higher classification accuracy, which we attribute to the identifiable nature of its representations. See Appendix A.5 for further details and experiments.

## 5.2 IMCA and nonlinear ICA simulations

We run a series of simulations comparing ICE-BeeM to previous nonlinear ICA methods such as iVAE (Khemakhem et al., 2020) and TCL (Hyvärinen and Morioka, 2016). We generate non-stationary 5-dimensional synthetic datasets, where data is divided into segments, and the conditioning variable $\mathbf{y}$ is defined to be a segment index. First, we let the data follow a nonlinear ICA model, which is a special case of equation (4) where the base measure, $\mu(\mathbf{z})$, is factorial. Following Hyvärinen and Morioka (2016), the $\mathbf{z}$ are generated according to isotropic Gaussian distributions with distinct precisions $\boldsymbol{\lambda}(\mathbf{y})$ determined by the segment index. Second, we let the data follow an IMCA model where the base measure $\mu(\mathbf{z})$ is *not factorial*. We set it to be a Gaussian term with a *fixed* but *non-diagonal* covariance matrix. More specifically, we randomly generate an invertible and symmetric matrix $\boldsymbol{\Sigma}_0 \in \mathbb{R}^{d \times d}$, such that $\mu(\mathbf{z}) \propto e^{-0.5\mathbf{z}^T \Sigma_0^{-1} \mathbf{z}}$. The covariance matrix of each segment is now equal to $\Sigma(\mathbf{y}) = (\Sigma_0^{-1} + \text{diag}(\boldsymbol{\lambda}(\mathbf{y})))^{-1}$, meaning the latent variables are no longer conditionally independent. In both cases, a randomly initialized neural network with varying number of layers,

$L \in \{2, 4\}$, was employed to generate the nonlinear mixing function **h**. The data generation process and the employed architectures are detailed in Appendix A.6.

In the case of ICE-BeeM, conditional flow contrastive estimation (CFCE, see Appendix B.2) was employed to estimate network parameters. To evaluate the performance of the method, we compute the mean correlation coefficient (MCC, see Appendix A.2) between the true latent variables and the recovered latents estimated by all three methods. Results for nonlinear ICA are provided in Figure [1e], where we note that ICE-BeeM performs competitively with respect to both iVAE and TCL. We note that as the depth of the mixing network, $L$, increases the performance of all methods decreases. Results for IMCA are provided in Figure [1f] where ICE-BeeM outperforms alternative nonlinear ICA methods, particularly when $L = 4$. This is because such other methods implicitly assume latent variables are conditionally independent and are therefore misspecified, whereas in ICE-BeeM , no distributional assumptions on the latent space are made.

## 6    Conclusion

We proposed a new *identifiable conditional energy-based deep model*, or ICE-BeeM for short, for unsupervised representation learning. This is probably the first energy-based model to benefit from rigorous identifiability results. Crucially, the model benefits from the tremendous flexibility and generality of EBMs. We even prove a universal approximation capability for the model.

We further prove a fundamental connection between EBMs and latent variable models, showing that ICE-BeeM is able to estimate nonlinear ICA, as a special case. In fact, it can even estimate a generalized version where the components do not need to be independent: they only need to be independently modulated by another variable such as a time index, history or noisy labels.

Empirically, we showed on real-world image datasets that our model learns identifiable representations in the sense that the representations do not change arbitrarily from one run to another, and that such representations improve performance in a transfer learning and semi-supervised learning applications.

Identifiability is fundamental for meaningful and principled disentanglement; it is necessary to make any interpretation of the features meaningful; it is also crucial in such applications as causal discovery (Monti et al., 2019) and transfer learning. The present results go further than any identifiability results hitherto and extend them to the EBM framework. We believe this paves the way for many new applications of EBMs, by giving them a theoretically sound basis.

## Broader Impact

This work is mainly theoretical, and aims to provide theoretical guarantees for the identifiability of a large family of deep models. Identifiability is very important, as it is key for reproducible science and interpretable results. For instance, if the networks behind search engines were identifiable, then their results would be consistent for most users. In addition, using perfectly identifiable networks in real life applications eliminates the randomness and arbitrariness of the system, and gives more control to the operator.

In general, identifiability is a desirable property. The system we develop here does not make any decisions, and thus can not exhibit any bias. Our theoretical guarantees abstract away the nature of the data and the practical implementation. Therefore, our work doesn't encourage the use of biased data or networks with potentially dangerous consequences.

## Acknowledgments and Disclosure of Funding

I.K. and R.P.M. were supported by the Gatsby Charitable Foundation. A.H. was supported by a Fellowship from CIFAR, and from the DATAIA convergence institute as part of the "Programme d'Investissement d'Avenir", (ANR-17-CONV-0003) operated by Inria.

## Footnotes

[1]Its rank is equal to its smaller dimension.

[2]A *LeakyReLU* has the form $h_l(x) = \max(0, x) + \alpha \min(0, x), \ \alpha \in (0, 1)$.

[3]The particular case of linear MLPs is discussed in Appendix C.4.

[4]That is: invertible, all second order cross-derivatives of the function and its inverse exist.

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
