[Supplementary Material 1]

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

We divide the Appendix into 5 main sections:

- Section A: we give extensive details on the experimental setup, as well as additional experiments;
- Section B: we discuss the estimation algorithms we used with ICE-BeeM and how they can be extended to the conditional setting;
- Section C: we prove the identifiability of ICE-BeeM and its universal approximation capability;
- Section D: we show how ICE-BeeM estimates IMCA;
- Section E: we provide a thorough theoretical analysis of the IMCA framework and draw parallels to the identifiability results in nonlinear ICA.

## A  Experimental protocol

### A.1  Model architecture details

In this section, we describe the neural network architectures used for the experiments of Section 5.1, on the image datasets (MNIST, FashionMNIST, CIFAR10 and CIFAR100). Code to reproduce these experiments can be found in the supplementary material.

We can distinguish three different types of configurations:

1. A series of fully connected layers — denoted *MLP*. This configuration satisfies the assumptions of Section 2.3.
2. A mix of convolutional and fully connected layers — denoted *ConvMLP*. We expect this configuration to work better than an MLP for images.
3. A variant of a RefineNet (Lin et al., 2017), following Song and Ermon (2019), which implements skip connections to help low level information reach the top layers — denoted for simplicity *Unet* (RefineNets are modern variants of U-net architectures). This configuration is very advanced and complicated, and serves to test if identifiable representations can be learnt for modern architectures.

The detailed architectures are in Table [2].

After choosing one of the configurations, we can further chose to reduce the dimensionality of the features ($d_z < d_x$), to use it in conjunction with positive features (condition 3 of Theorem 2) or with augmented features (condition 4 of Theorem 2). This results in the following nomenclature, where we will take as an example a *ConvMLP* network:

- If we reduce the dimension of the latent space ($d_z < d_x$)—for example $d_z = 50$, we denote the configuration by *ConvMLP-50*.
- If we used positive features, we denote the configuration by *ConvMLP-p*.
- If we used augmented features, we denote the configuration by *ConvMLP-a*.
- We can also have a mix of the above, for examples *ConvMLP-50p*.
- We can also have non of the above, in which case we simply write *ConvMLP*—implying that $d_z = d_x$.

We summarize the configurations used for the different experiments of Section 5.1 in Table [3].

For all the experiments, we used the Adam optimizer (Kingma and Ba, 2014) to update the parameters of the networks. We used a learning rate of 0.001, and $(\beta_1, \beta_2) = (0.9, 0.999)$; `amsgrad` was turned off, as well as weight decay. Data was fed to the networks in mini-batches of size 63, and the

Table 2: Architecture detail

| Configuration | Architecture | Comment |
|---|---|---|
| | Input: $d_x = w \times w \times n_c$ | $n_c$: channels, $w$: width/height |
| | | MNIST: $n_c = 1$, $w = 28$ |
| | | FashionMNIST: $n_c = 1$, $w = 28$ |
| | | CIFAR10: $n_c = 3$, $w = 32$ |
| | | CIFAR100: $n_c = 3$, $w = 32$ |
| | Output: $d_z$ | |
| *MLP* | Input: $d_x$ | |
| | FC 512, LeakyReLU(0.1) | |
| | FC 384, LeakyReLU(0.1) | |
| | Dropout(0.1) | |
| | FC 256, LeakyReLU(0.1) | |
| | FC 256, LeakyReLU(0.1) | |
| | FC $d_z$ | |
| *ConvMLP* | Input: $d_x = w \times w \times c$ | stride 1 for all conv. layers |
| | Conv $w \times w \times 32$, BatchNorm, ReLU | padding 1, filter size 3 |
| | Conv $w \times w \times 64$, BatchNorm, ReLU | padding 1, filter size 3 |
| | MaxPool $\frac{w}{2} \times \frac{w}{2} \times 64$ | |
| | Conv $\frac{w}{2} \times \frac{w}{2} \times 128$, BatchNorm, ReLU | padding 1, filter size 3 |
| | Conv $\frac{w}{2} \times \frac{w}{2} \times 256$, BatchNorm, ReLU | padding 1, filter size 3 |
| | MaxPool $\frac{w}{4} \times \frac{w}{4} \times 256$ | |
| | Conv $1 \times 1 \times 256$ | padding 0, filter size $\frac{w}{4}$ |
| | Dropout(0.1) | |
| | FC 256, LeakyReLU(0.1) | |
| | FC $d_z$ | |
| *Unet* | Input: $d_x = w \times w \times n_c$ | stride 1 for all conv. layers |
| | Conv $w \times w \times 64$ | padding 1, filter size 3 |
| | 4-cascaded RefinNet | see Song and Ermon (2019) |
| | \|   activation: ELU | exponential LU |
| | \|   normalization: InstanceNorm+ | see Song and Ermon (2019) |
| | InstanceNorm+, ELU | |
| | Conv $w \times w \times n_c$ | padding 1, filter size 3 |
| | FC $d_z$ | only if $d_z < d_x$ |

Table 3: Architectures used in the experiments

| Fig./Tab. | Dataset | Description | Configuration |
|---|---|---|---|
| Fig. [1a] | MNIST | Quantifying quality of representations | *Unet-a* |
| Fig. [1b] | CIFAR10 | Quantifying quality of representations | *Unet* |
| Fig. [1b] | CIFAR100 | Quantifying quality of representations | *Unet* |
| Fig. [1c] | MNIST | Transfer learning | *ConvMLP-50* |
| Fig. [1d] | CIFAR10 | Transfer learning | *ConvMLP-90* |
| Tab. [1a] | MNIST | Transfer learning | *ConvMLP-50* |
| Tab. [1a] | CIFAR10 | Transfer learning | *ConvMLP-90* |
| Tab. [1b] | FashionMNIST | Semi-supervised learning | *ConvMLP-50* |
| Tab. [1b] | CIFAR10 | Semi-supervised learning | *ConvMLP-50p* |

training was done for 5000 iterations (no visible improvements in the results were observed after this many iterations). For CIFAR10 and CIFAR100 experiments, we introduced a random horizontal flip to the data, with probability $0.5$.

We used conditional denoising score matching (CDSM, Appendix B.1) to train the energy models. The noise parameter used is $\sigma = 0.01$.

## A.2 The MCC metric

To quantify identifiability, we use the mean correlation coefficient (MCC) metric, which computes the maximum linear correlations up to permutation of components. To obtain the value of this metric between two vectors $\mathbf{x}$ and $\mathbf{y}$, we first calculate all pairs of correlation coefficients between the components $x_i$ of $\mathbf{x}$, and the components $y_j$ of $\mathbf{y}$. Since the order of the components in each vector can be arbitrary, we have to account for possible permutations between the indices $i$ and $j$. This is done by solving a linear sum assignment problem (for instance, using the auction algorithm). We finally average over all correlation coefficients (after finding the right permutation). This makes the MCC metric invariant by permutation and component-wise transformations (as a consequence of the transformation invariance of the correlation coefficient).

To better understand this metric, let's consider the following example. Let $\mathbf{x} \in \mathbb{R}^2$ be a bivariate random variable such that $x_1 \perp\!\!\!\perp x_2$, and let $\mathbf{y} = (x_2^2, x_1^2)$. If we don't account for any permutations, then the average correlation is equal to $\frac{1}{2} \sum_i \text{corr}(x_i, y_i) = 0$ because $x_1 \perp\!\!\!\perp x_2$. In reality, though, $\mathbf{y}$ and $\mathbf{x}$ are perfectly correlated, since the value of $\mathbf{x}$ completely determines that of $\mathbf{y}$. Thus, we have to find the optimal permutation of the elements of $\mathbf{y}$ in order to maximize the average correlation. The MCC does this by computing all pair-wise correlations, and finding the assignment that maximizes the average correlation.

When the latent ground truth is known (Section 5.2—IMCA and nonlinear ICA simulations, for instance), we can test for identifiability of the components by comparing the recovered latents to this ground truth. A high MCC means that we recovered the true latents.

When the ground truth is unknown (Section 5.1—real image datasets), we compare pairs of learnt representations, each from a different random initialization. A consistently high MCC means that changing the random state of the model doesn't drastically change the learnt representations.

## A.3 Quality of representations

We argued that conditioning enables EBMs to learn identifiable representations. The results in Section 5.1 validate this. The plots presented in Figures [1a] and [1b] were produced using the *Unet* configuration, described in Table [3]. This architecture is complex and deep, and involves multiple layers for which a thorough theoretical analysis is very difficult, unlike MLPs for instance. In addition, the dimension of the latent space was chosen to be equal to that of the input space. Intuitively, we would expect that the chance of learning arbitrary representations increases as we increase the number of features because this increases the entropy of the system.

This allows us to challenge the capabilities of ICE-BeeM, and test its limits. We concluded from the results that the theory presented here does benefit modern deep learning architectures. This experiment serves to empirically validate our theoretical result, and is the first of its kind in recent identifiability literature, which focused on validating the theory on simulated data with well know ground truth.

The matrix $\mathbf{A}$ in equation (2) and the permutation $\sigma$ in equation (3) were learnt from the first half of the test partition for each dataset. The evaluation of the MCCs was done on the remaining half of the test dataset.

We present further plots detailing the quality of the learnt representations on MNIST, FashionMNIST, CIFAR10 and CIFAR100 for a variety of different configurations in Figures [2] and [3].

(a) MNIST - *ConvMLP-50/50p*  (b) MNIST - *ConvMLP-90/90p*  (c) MNIST - *ConvMLP-200/200p*

(d) FMNIST - *ConvMLP-50/50p*  (e) FMNIST - *ConvMLP-90/90p*  (f) FMNIST - *ConvMLP-200/200p*

(g) C10 - *ConvMLP-50/50p*  (h) C10 - *ConvMLP-90/90p*  (i) C10 - *ConvMLP-200/200p*

(j) C100 - *ConvMLP-50/50p*  (k) C100 - *ConvMLP-90/90p*  (l) C100 - *ConvMLP-200/200p*

Figure 2: Further experiments on the *strong* identifiability of learnt representations using the *ConvMLP* architecture on image datasets — C10/100 stands for CIFAR10 and CIFAR100, respectively.

| (a) MNIST - *Unet* | (b) CIFAR10 - *Unet* | (c) CIFAR100 - *Unet* |

Figure 3: Further experiments on the identifiability of representations using the *Unet* architecture on image datasets.

### A.4 Transfer learning experiments

#### A.4.1 Intuition

As a practical application of our framework where identifiability is important, we consider meta-learning, in particular multi-task and transfer learning. Assume we have $N$ datasets, which could be, *e.g.*, different subjects in biomedical settings, or different image datasets. This fits well with our framework, where $y = 1, \ldots, N$ is now the index of the dataset, or "task". The key question in such a setting is how we can leverage all the observations to better model each single dataset, and especially transfer knowledge of existing models to a new dataset.

To this end, we propose an intuitively appealing approach, where we approximate the unnormalized log-pdf in $y$-th dataset $p(\mathbf{x}; y)$ by a linear combination of a learned "basis" functions $f_{i,\boldsymbol{\theta}}$ as

$$\log p(\mathbf{x}; y) + \log Z(\boldsymbol{\theta}) \approx \sum_{i=1}^{k} g_i(y) f_{i,\boldsymbol{\theta}}(\mathbf{x}) \tag{6}$$

where the $g_i(y)$ are scalar parameters as a function of $y$, which act as coefficients in the basis $(f_{i,\boldsymbol{\theta}})$. This linear approximation is nothing else than a special case of ICE-BeeM, but here, we interpret such an approximation as a linear approximation in log-pdf space. In fact, what we are doing is a kind of PCA in the set of probability distributions $p(\mathbf{x}; y)$. Such "probability space" PCA allows the models for the different datasets to learn from each other, as in the classical idea of denoising by projection onto the PCA subspace.

In transfer learning, we observe a new dataset, with distribution $p(\mathbf{x}; y_{\text{new}})$ for $y_{\text{new}} = N + 1$. Based on our decomposition, we approximate $p(\mathbf{x}; y_{\text{new}})$ as in (6). This leads to a drastic simplification: we can learn the basis functions $f_{i,\boldsymbol{\theta}}$ from the first $N$ datasets, then we only need to estimate the $k$ scalar parameters $g_i(y_{\text{new}})$ for the new dataset. The coefficients are likely to be sparse as well, which provides an additional penalty.

Reducing the transfer learning to estimation of the $g_i(y_{\text{new}})$ clearly requires that we have estimated the true $f_i$ up to a linear transformation, which is the weaker form of identifiability in Theorem 1. Moreover, using a sparsity penalty is only meaningful if we have the true $f_i$ without any linear mixing, which requires the stronger identifiability in Theorem 2.

Training can be done by any method for EBM estimation. In particular, it is very easy by score matching because equation (6) is an exponential family for fixed $f_i$ (Hyvärinen, 2007).

#### A.4.2 Further experiments

The pre-training was done on labels 0-7 from the train partition for MNIST, FashionMNIST and CIFAR10, and on labels 0-84 from the train partition for CIFAR100. The second (transfer) step was done on labels 8-9 from the train partition for MNIST, FashionMNIST and CIFAR10, and on and labels 85-99 the train partition for CIFAR100.

| (a) Transfer learning, $\mathbf{f}_{\boldsymbol{\theta}}$ fixed | (b) Baseline, both $\mathbf{f}_{\boldsymbol{\theta}}$ and $\mathbf{g}_{\boldsymbol{\theta}}$ estimated |

Figure 4: Further results for transfer learning experiments on MNIST. In the case of transfer learning 99 out of a hundred returned digits are class 8 compared to only 58 in the baseline.

We considered a subset of size 6000 to produce the values in Table [1a]. This table should be read in conjunction with Figures [1c]-[1d] for a proper evaluation of performance.

We present further plots and results of transfer learning experiments in Figures [5]-[6] and Table [4] ran on MNIST, FashionMNIST, CIFAR10 and CIFAR100 for a variety of different configurations. for different configurations and datasets. We considered a subset of size 6000 to produce the values in Table 4. We expect the baseline where we don't perform transfer learning to perform comparatively for such a subset size: transfer learning is mostly important when data is scarce. For the complete picture, this table should be read in conjunction with Figures [5]-[6].

As an additional way to visualize the results, Figure [4a] shows unseen MNIST samples (taken across all possible classes) which are assigned high confidence of belonging to the "new" class 8 after transfer learning, indicating that the ICE-BeeM model has learnt a reasonable distribution over unseen classes. By comparison the case where no transfer learning is employed (Figure [4b]), incorrectly assigns high confidences to other digits.

## A.5 Semi-supervised learning

In this experiment, we train both an identifiable ICE-BeeM model and an unconditional (non-identifiable) EBM on classes 0-7. The purpose of this step is to learn a feature extractor $\mathbf{f}_{\boldsymbol{\theta}}$ that is able of learning meaningful features from the images. To test the quality of the features learnt by both models (the ICE-BeeM, and the unconditional EBM), we use the feature map $\mathbf{f}_{\boldsymbol{\theta}}$ to classify unseen samples from classes 8-9. Results show that ICE-BeeM outperforms the unconditional baseline in this classification task. We attribute this to the identifiability of ICE-BeeM: our model seems to be performing a principled form of disentanglement by learning features that are faithful to the unknown factors of variation in the data.

Training was done on labels 0-7, using the train partition for MNIST, FashionMNIST and CIFAR10. Evaluation was done on labels 8-9, using the test partition for all three datasets. This data was in turn partitioned for the classification into a train and test split. The split proportion is $15\%$ for MNIST and FashionMNIST, and $33\%$ for CIFAR10 and CIFAR100.

We present further results for the semi-supervised learning experiments in Table [5], ran on MNIST, FashionMNIST, CIFAR10 for a variety of different configurations.

## A.6 IMCA and nonlinear ICA simulations

We give here more detail on the data generation process for the simulations in Section 5.2, as well as the architectures used.

(a) MNIST - *ConvMLP-50*

(b) MNIST - *ConvMLP-200*

(c) FMNIST - *ConvMLP-90*

(d) FMNIST - *ConvMLP-90p*

(e) CIFAR10 - *ConvMLP-200*

(f) CIFAR10 - *ConvMLP-200p*

(g) CIFAR100 - *ConvMLP-50*

(h) CIFAR100 - *ConvMLP-50p*

Figure 5: Further transfer learning — the dataset/configuration combo are reported in the captions.

(a) MNIST - *Unet*

(b) MNIST - *Unet-a*

(c) FMNIST - *Unet*

(d) FMNIST - *Unet-a*

(e) CIFAR10 - *Unet*

(f) CIFAR10 - *Unet-a*

(g) CIFAR100 - *Unet*

(h) CIFAR100 - *Unet-a*

Figure 6: Further transfer learning — the dataset/configuration combo are reported in the captions.

Table 4: Transfer learning — CDSM score (lower is better)

| Dataset | Configuration | $\mathbf{f} \cdot \mathbf{g}_{\theta}$ | $\mathbf{f} \cdot \mathbf{1}$ | $\mathbf{f}_{\theta} \cdot \mathbf{g}_{\theta}$ | $\mathbf{f}_{\theta} \cdot \mathbf{1}$ |
|---|---|---|---|---|---|
| MNIST | *ConvMLP-50* | $2.95 \pm 0.02$ | $23.43 \pm 0.04$ | $4.22 \pm 0.15$ | $3.64 \pm 0.10$ |
| | *ConvMLP-50p* | $2.79 \pm 0.00$ | $796.99 \pm 0.86$ | $10.13 \pm 4.74$ | $3.63 \pm 0.09$ |
| | *ConvMLP-90* | $2.94 \pm 0.01$ | $12.18 \pm 0.03$ | $4.29 \pm 0.13$ | $3.67 \pm 0.12$ |
| | *ConvMLP-90p* | $3.03 \pm 0.01$ | $694.94 \pm 1.03$ | $10.22 \pm 4.63$ | $3.70 \pm 0.12$ |
| | *ConvMLP-200* | $2.91 \pm 0.01$ | $27.70 \pm 0.02$ | $4.29 \pm 0.12$ | $3.74 \pm 0.09$ |
| | *ConvMLP-200p* | $2.95 \pm 0.01$ | $805.45 \pm 3.56$ | $12.08 \pm 3.79$ | $3.71 \pm 0.13$ |
| | *Unet* | $2.23 \pm 0.01$ | $10.04 \pm 0.01$ | $3.44 \pm 0.03$ | $2.97 \pm 0.25$ |
| | *Unet-a* | $2.29 \pm 0.01$ | $6.18 \pm 0.00$ | $3.44 \pm 0.02$ | $6.27 \pm 4.21$ |
| | *Unet-p* | $14.00 \pm 0.01$ | $14.08 \pm 0.00$ | $11.97 \pm 4.01$ | $6.14 \pm 4.17$ |
| | *Unet-50a* | $2.61 \pm 0.02$ | $14.24 \pm 0.01$ | $3.79 \pm 0.56$ | $2.92 \pm 0.20$ |
| | *MLP-50* | $13.99 \pm 0.01$ | $13.99 \pm 0.01$ | $14.00 \pm 0.01$ | $14.00 \pm 0.01$ |
| | *MLP-50p* | $13.99 \pm 0.01$ | $14.00 \pm 0.01$ | $14.00 \pm 0.01$ | $14.00 \pm 0.01$ |
| | *MLP-90* | $14.00 \pm 0.01$ | $14.00 \pm 0.01$ | $14.00 \pm 0.01$ | $13.99 \pm 0.01$ |
| | *MLP-90p* | $13.99 \pm 0.01$ | $14.00 \pm 0.01$ | $14.00 \pm 0.01$ | $14.00 \pm 0.01$ |
| | *MLP-200* | $13.99 \pm 0.01$ | $14.00 \pm 0.01$ | $14.00 \pm 0.01$ | $14.00 \pm 0.01$ |
| | *MLP-200p* | $13.99 \pm 0.01$ | $13.99 \pm 0.01$ | $14.00 \pm 0.01$ | $14.00 \pm 0.01$ |
| FMNIST | *ConvMLP-50* | $7.88 \pm 0.01$ | $9.82 \pm 0.03$ | $7.88 \pm 0.07$ | $7.18 \pm 0.25$ |
| | *ConvMLP-50p* | $8.00 \pm 0.02$ | $197.84 \pm 2.27$ | $7.92 \pm 0.18$ | $7.10 \pm 0.24$ |
| | *ConvMLP-90* | $8.09 \pm 0.02$ | $10.86 \pm 0.04$ | $7.88 \pm 0.05$ | $7.14 \pm 0.24$ |
| | *ConvMLP-90p* | $7.94 \pm 0.01$ | $197.93 \pm 2.33$ | $7.87 \pm 0.13$ | $7.13 \pm 0.20$ |
| | *ConvMLP-200* | $7.98 \pm 0.00$ | $15.86 \pm 0.01$ | $7.91 \pm 0.16$ | $7.17 \pm 0.21$ |
| | *ConvMLP-200p* | $7.86 \pm 0.01$ | $196.14 \pm 2.07$ | $7.81 \pm 0.15$ | $7.11 \pm 0.15$ |
| | *Unet* | $6.47 \pm 0.02$ | $277.56 \pm 1.06$ | $6.52 \pm 0.03$ | $6.46 \pm 0.07$ |
| | *Unet-a* | $6.60 \pm 0.02$ | $24.62 \pm 0.02$ | $6.52 \pm 0.02$ | $6.41 \pm 0.01$ |
| | *MLP-50* | $13.99 \pm 0.01$ | $14.00 \pm 0.01$ | $13.99 \pm 0.01$ | $14.00 \pm 0.01$ |
| | *MLP-200* | $13.99 \pm 0.01$ | $14.00 \pm 0.01$ | $13.99 \pm 0.01$ | $14.00 \pm 0.01$ |
| CIFAR10 | *ConvMLP-50* | $8.02 \pm 0.01$ | $32.09 \pm 0.07$ | $8.36 \pm 0.03$ | $8.15 \pm 0.03$ |
| | *ConvMLP-50p* | $8.04 \pm 0.02$ | $412.15 \pm 2.54$ | $8.35 \pm 0.04$ | $8.17 \pm 0.01$ |
| | *ConvMLP-90* | $8.03 \pm 0.01$ | $23.08 \pm 0.04$ | $8.37 \pm 0.02$ | $8.16 \pm 0.05$ |
| | *ConvMLP-90p* | $8.05 \pm 0.01$ | $408.51 \pm 2.30$ | $8.37 \pm 0.04$ | $8.16 \pm 0.01$ |
| | *ConvMLP-200* | $8.02 \pm 0.02$ | $13.35 \pm 0.01$ | $8.41 \pm 0.07$ | $8.13 \pm 0.03$ |
| | *ConvMLP-200p* | $8.06 \pm 0.01$ | $509.09 \pm 2.31$ | $8.35 \pm 0.02$ | $8.11 \pm 0.03$ |
| | *Unet* | $7.29 \pm 0.01$ | $118.93 \pm 0.34$ | $7.51 \pm 0.05$ | $9.21 \pm 3.43$ |
| | *Unet-a* | $7.18 \pm 0.01$ | $18.73 \pm 0.01$ | $7.48 \pm 0.09$ | $7.47 \pm 0.13$ |
| | *Unet-50a* | $7.30 \pm 0.05$ | $16.41 \pm 0.00$ | $7.64 \pm 0.26$ | $7.27 \pm 0.03$ |
| | *MLP-50* | $16.00 \pm 0.00$ | $16.00 \pm 0.00$ | $16.00 \pm 0.00$ | $16.00 \pm 0.00$ |
| | *MLP-200* | $16.00 \pm 0.01$ | $16.00 \pm 0.00$ | $16.00 \pm 0.01$ | $16.00 \pm 0.00$ |
| CIFAR100 | *ConvMLP-50* | $8.25 \pm 0.01$ | $45.19 \pm 0.15$ | $8.69 \pm 0.04$ | $8.59 \pm 0.02$ |
| | *ConvMLP-50p* | $8.24 \pm 0.01$ | $2560.77 \pm 7.15$ | $8.68 \pm 0.04$ | $8.61 \pm 0.04$ |
| | *ConvMLP-90* | $8.23 \pm 0.01$ | $8.74 \pm 0.01$ | $8.68 \pm 0.05$ | $8.61 \pm 0.03$ |
| | *ConvMLP-90p* | $8.25 \pm 0.01$ | $3018.50 \pm 7.27$ | $8.65 \pm 0.02$ | $8.58 \pm 0.03$ |
| | *ConvMLP-200* | $8.26 \pm 0.01$ | $42.80 \pm 0.09$ | $8.69 \pm 0.06$ | $8.59 \pm 0.03$ |
| | *ConvMLP-200p* | $8.18 \pm 0.01$ | $3827.36 \pm 16.14$ | $8.65 \pm 0.07$ | $8.63 \pm 0.05$ |
| | *Unet* | $7.41 \pm 0.02$ | $106.28 \pm 0.75$ | $7.77 \pm 0.05$ | $8.38 \pm 0.55$ |
| | *Unet-a* | $7.39 \pm 0.02$ | $11.15 \pm 0.01$ | $7.82 \pm 0.42$ | $9.35 \pm 3.33$ |
| | *Unet-50a* | $7.54 \pm 0.01$ | $15.95 \pm 0.00$ | $7.97 \pm 0.13$ | $7.60 \pm 0.05$ |
| | *MLP-50p* | $16.00 \pm 0.01$ | $16.00 \pm 0.00$ | $16.00 \pm 0.00$ | $16.00 \pm 0.00$ |
| | *MLP-200p* | $16.00 \pm 0.01$ | $16.00 \pm 0.00$ | $16.00 \pm 0.00$ | $16.00 \pm 0.00$ |

Table 5: Semi-supervised learning — classification accuracy (higher is better)

| Dataset | Configuration | ICE-BeeM | Unconditional EBM |
|---|---|---|---|
| MNIST | *ConvMLP-50* | $76.98 \pm 1.61$ | $62.82 \pm 1.48$ |
| | *ConvMLP-50p* | $88.46 \pm 1.14$ | $66.58 \pm 2.64$ |
| | *ConvMLP-90* | $78.93 \pm 1.51$ | $71.61 \pm 1.71$ |
| | *ConvMLP-90p* | $78.66 \pm 1.91$ | $69.13 \pm 1.49$ |
| | *ConvMLP-200* | $81.21 \pm 2.6$ | $71.48 \pm 2.23$ |
| | *ConvMLP-200p* | $77.38 \pm 1.32$ | $68.99 \pm 1.68$ |
| | *MLP-50* | $91.74 \pm 1.72$ | $85.77 \pm 1.14$ |
| | *MLP-50p* | $92.21 \pm 1.74$ | $84.56 \pm 1.1$ |
| | *MLP-90* | $95.17 \pm 0.46$ | $85.91 \pm 2.07$ |
| | *MLP-90p* | $94.97 \pm 0.7$ | $85.97 \pm 1.61$ |
| | *MLP-200* | $94.36 \pm 1.28$ | $89.26 \pm 1.7$ |
| | *MLP-200p* | $91.81 \pm 2.33$ | $90.87 \pm 1.05$ |
| | *Unet* | $97.79 \pm 0.34$ | $98.39 \pm 0.68$ |
| | *Unet-a* | $97.18 \pm 0.5$ | $97.79 \pm 0.78$ |
| | *Unet-50a* | $97.52 \pm 0.4$ | $97.92 \pm 0.49$ |
| | *Unet-20a* | $95.64 \pm 0.7$ | $92.08 \pm 1.71$ |
| FMNIST | *ConvMLP-50* | $77.07 \pm 1.39$ | $56.33 \pm 3.18$ |
| | *ConvMLP-50p* | $71.67 \pm 1.85$ | $57.6 \pm 2.24$ |
| | *ConvMLP-90* | $74.13 \pm 1.86$ | $57.73 \pm 3.12$ |
| | *ConvMLP-90p* | $70.87 \pm 1.13$ | $60.07 \pm 2.9$ |
| | *ConvMLP-200* | $81.4 \pm 1.93$ | $68.27 \pm 2.78$ |
| | *ConvMLP-200p* | $78.47 \pm 0.96$ | $57.47 \pm 2.62$ |
| | *MLP-50* | $98.07 \pm 1.06$ | $90.47 \pm 1.56$ |
| | *MLP-50p* | $97.6 \pm 0.53$ | $90.47 \pm 1.56$ |
| | *MLP-90* | $97.8 \pm 0.34$ | $94.4 \pm 0.53$ |
| | *MLP-90p* | $97.8 \pm 0.34$ | $94.4 \pm 0.53$ |
| | *MLP-200* | $98.6 \pm 0.49$ | $94.87 \pm 0.96$ |
| | *MLP-200p* | $98.6 \pm 0.65$ | $95.33 \pm 1.05$ |
| | *Unet* | $99.67 \pm 0.3$ | $99.93 \pm 0.13$ |
| | *Unet-a* | $99.53 \pm 0.16$ | $99.87 \pm 0.16$ |
| CIFAR10 | *ConvMLP-50* | $69.36 \pm 2.23$ | $56.39 \pm 1.0$ |
| | *ConvMLP-50p* | $64.42 \pm 1.09$ | $51.88 \pm 1.33$ |
| | *ConvMLP-90* | $68.24 \pm 2.0$ | $52.82 \pm 0.95$ |
| | *ConvMLP-90p* | $66.18 \pm 1.01$ | $52.33 \pm 1.73$ |
| | *ConvMLP-200* | $64.73 \pm 1.36$ | $54.18 \pm 1.09$ |
| | *ConvMLP-200p* | $66.3 \pm 0.99$ | $54.48 \pm 1.28$ |
| | *MLP-50* | $68.73 \pm 1.35$ | $70.27 \pm 2.67$ |
| | *MLP-50p* | $69.82 \pm 1.78$ | $69.36 \pm 2.3$ |
| | *MLP-90* | $71.58 \pm 1.21$ | $72.85 \pm 1.16$ |
| | *MLP-90p* | $71.12 \pm 1.64$ | $72.85 \pm 1.16$ |
| | *MLP-200* | $72.39 \pm 1.92$ | $72.97 \pm 1.75$ |
| | *MLP-200p* | $70.94 \pm 1.25$ | $71.97 \pm 2.29$ |
| | *Unet* | $80.27 \pm 4.0$ | $80.58 \pm 0.9$ |
| | *Unet-a* | $80.48 \pm 1.45$ | $80.48 \pm 1.45$ |
| | *Unet-50a* | $77.64 \pm 1.02$ | $73.79 \pm 0.81$ |
| | *Unet-20a* | $74.21 \pm 0.73$ | $68.82 \pm 0.67$ |

**Data generation**   We generate 5-dimensional synthetic datasets following the nonlinear ICA model which is a special case of equation (4) where the base measure, $\mu(\mathbf{z})$, is factorial. In particular, we set it to $\mu(\mathbf{z}) = 1$. As such, latent variables are conditionally independent given segment labels. The sources are divided into $M = 8$ segments, and the conditioning variable $\mathbf{y}$ is defined to be the segment index, uniformly drawn from the integer set $[\![1, M]\!]$. Following Hyvärinen and Morioka (2016), the $\mathbf{z}$ are generated according to isotropic Gaussian distributions with distinct precisions $\boldsymbol{\lambda}(\mathbf{y})$ determined by the segment index. Second, we perform the same experiment but on data generated from an IMCA model where the base measure $\mu(\mathbf{z})$ is *not factorial*. More specifically, we randomly generate an invertible and symmetric matrix $\boldsymbol{\Sigma}_0 \in \mathbb{R}^{d \times d}$, such that $\mu(\mathbf{z}) \propto e^{-0.5\mathbf{z}^T \Sigma_0^{-1} \mathbf{z}}$. As before, we define $\boldsymbol{\lambda}(\mathbf{y})$ to be the distinct conditional precisions. The precision matrix of each segment is now equal to $\Sigma(\mathbf{y})^{-1} = \Sigma_0^{-1} + \text{diag}(\boldsymbol{\lambda}(\mathbf{y}))^{-1}$, meaning the latent variables are no longer conditionally independent.

For both nonlinear ICA and IMCA data, a randomly initialized neural network with varying number of layers, $L \in \{2, 4\}$, was employed to generate the nonlinear mixing function $\mathbf{h}$. Leaky ReLU with negative slope equal to $0.1$ was employed as the activation function in order to ensure the network was invertible. The hidden dimensions of the mixing network are equal to the latent dimension $d_x$, and the output dimension is $d_x = d_z$.

**Baseline methods**   The first baseline we compare to is TCL (Hyvärinen and Morioka, 2016), which is a self-supervised method for nonlinear ICA based on the nonstationarity of the sources. TCL learns to invert the mixing function $\mathbf{h}$, by performing a surrogate classification task, where the goal is to classify original observations against their segment indices in a multinomial classification task. Its theory is premised on the fact that the feature extractor used for the classification has to extract meaningful latents in order to perform well in the classification task.

The second baseline is iVAE (Khemakhem et al., 2020), a nonlinear ICA method which uses an identifiable VAE to recover the independent sources. Its theory is premised on the consistency of maximum likelihood training, and on the flexibility of VAEs in approximating densities. They show that given enough data, the variational posterior learns to approximate the true posterior distribution, and can thus be used to invert the mixing function. The iVAE, like a regular VAE, is trained by maximizing the ELBO (Kingma and Welling, 2013).

**Training of ICE-BeeM via flow contrastive estimation**   To demonstrate that ICE-BeeM can be trained by any method for training EBMs, we switched from denoising score matching to flow contrastive estimation (FCE, Appendix B.2). As a contrastive flow, we used a normalizing flow model (Rezende and Mohamed, 2015), with an isotropic and tractable base distribution. It is then transformed by a 10-layer flow, where each layer is made of a succession of a neural spline flow (Durkan et al., 2019), an invertible $1 \times 1$ convolution (Kingma and Dhariwal, 2018), and an ActNorm layer (Kingma and Dhariwal, 2018). The flow parameters are updated by and Adam optimizer, with a learning rate of $10^{-5}$.

**Used architectures**   The architectures used to produce Figures [1e] and [1f] are summarized by Table [6].

# B   Estimation algorithms

It is important to note that the identifiability results presented above apply to conditional EBMs in general. As such, we may employ any of the wide variety of methods which have been proposed for the estimation of unnormalized EBMs. In this work we used two different options with good results for both: flow contrastive estimation (Gao et al., 2019) and denoising score matching (Vincent, 2011). Both methods can also be extended to the conditional case in a straightforward fashion.

Flow-contrastive estimation (FCE) can be seen as an extension of noise-contrastive estimation (Gutmann and Hyvärinen, 2012, NCE), which seeks to learn unnormalized EBMs by solving a surrogate classification task. The proposed classification task seeks to discriminate between the true data and some synthetic noise data based on the log-odds ratio of the EBM and the noise distribution. However, a limitation of NCE is the need to specify a noise distribution which can be sampled from and whose log-density can be evaluated pointwise but which also shares some of the empirical

Table 6: Architectures used in the simulations

| Model | Optimizer | Architecture | |
|---|---|---|---|
| | | Input | $d_x = 5$ |
| | | Condition | one hot encoded $d_y = M = 8$ |
| | | Latent | $d_z = d_x = 5$ |
| | | Num. layers | $L \in \{2, 4\}$ |
| ICE-BeeM | Adam<br>lr $3.10^{-4}$ | $\mathbf{f}_{\boldsymbol{\theta}}$ | $(L+1)$-layer MLP<br>batch norm after each FC layer<br>hidden dim 32, LeakyReLU(0.1) act |
| | | $\mathbf{g}_{\boldsymbol{\theta}}$ | $(d_z \times d_y)$ learnable matrix |
| iVAE | Adam<br>lr $10^{-3}$ | Encoder | $p(\mathbf{z}|\mathbf{x})$ Normal<br>3-layer MLP<br>hidden dim $2d_x$, LeakyReLU(0.1) act |
| | | Decoder | $p(\mathbf{x}|\mathbf{z}, \mathbf{y})$ Normal<br>3-layer MLP<br>hidden dim $2d_x$, LeakyReLU(0.1) act |
| | | Prior | $p(\mathbf{z}|\mathbf{y})$ Normal<br>3-layer MLP<br>hidden dim $2d_x$, LeakyReLU(0.1) act |
| TCL | Momentum 0.9<br>lr 0.01<br>exp decay 0.1 | | $L$-layer MLP<br>FC $2d_x$, maxout(2)<br>$(L-2)\times$ [FC $d_x$, maxout(2)]<br>FC $d_x$, absolute value |

properties of the observed data. To address this concern Gao et al. (2019) propose to employ a flow model as the contrast noise distribution. FCE seeks to simultaneously learn both an unnormalized EBM as well as a flow model for the contrast noise in an alternating fashion. We naturally get a conditional version for FCE by learning a conditional EBM (Gao et al., 2019, eq. 12).

Score matching is another well-known method for learning unnormalized models (Hyvärinen, 2005). However, its computational implementation in deep networks is problematic, which is why Vincent (2011) proposed a stochastic approximation which can be interpreted as denoising the data, and which works efficiently in deep networks (Saremi et al., 2018; Song and Ermon, 2019).

### B.1 Conditional denoising score matching

We extend the original score matching objective to the conditional setting in a natural way: for a fixed $\mathbf{y}$, we compute the unconditional score matching objective: $J(\boldsymbol{\theta}, \mathbf{y}) = \mathbb{E}_{p(\mathbf{x}|\mathbf{y})} \|\nabla_{\mathbf{x}} \log p_{\boldsymbol{\theta}}(\mathbf{x}|\mathbf{y}) - \nabla_{\mathbf{x}} \log p(\mathbf{x}|\mathbf{y})\|^2$, and then average over all values of $\mathbf{y}$. The expression of the conditional score matching objective is then:

$$\mathcal{J}_{\text{CSM}}(\boldsymbol{\theta}) = \mathbb{E}_{p(\mathbf{x},\mathbf{y})} \|\nabla_{\mathbf{x}} \log p_{\boldsymbol{\theta}}(\mathbf{x}|\mathbf{y}) - \nabla_{\mathbf{x}} \log p(\mathbf{x}|\mathbf{y})\|^2 \tag{7}$$

We build on the recent developments by Vincent (2011), and introduce a conditional denoising score matching objective by replacing the unknown density by a kernel density estimator. Formally, given observations $\mathcal{D} = \{(\mathbf{x}^{(1)}, \mathbf{y}^{(1)}), \dots, (\mathbf{x}^{(N)}, \mathbf{y}^{(N)})\}$, we first derive nonparamteric kernel density estimates of $p(\mathbf{x}, \mathbf{y})$ and $p(\mathbf{y})$, which we then use to derive the estimate for $p(\mathbf{x}|\mathbf{y})$ using the product rule. These estimates have the forms:

$$q_b(\mathbf{y}) = \mathbb{E}_{\mathbf{y}' \sim q_{\mathcal{D}}} [l_b(\mathbf{y}|\mathbf{y}')] \tag{8}$$

$$q_{ab}(\mathbf{x}, \mathbf{y}) = \mathbb{E}_{(\mathbf{x}',\mathbf{y}') \sim q_{\mathcal{D}}} [k_a(\mathbf{x}|\mathbf{x}')l_b(\mathbf{y}|\mathbf{y}')] \tag{9}$$

$$q_{ab}(\mathbf{x}|\mathbf{y}) = \frac{q_{ab}(\mathbf{x}, \mathbf{y})}{q_b(\mathbf{y})} \tag{10}$$

where $k_a$ and $l_b$ are bounded kernel functions defined on $\mathcal{X}$ and $\mathcal{Y}$ and with bandwidths[5] $a$ and $b$, respectively. In the following, we assume that the bandwidth sequences are equal ($a = b = \sigma$).

We replace $p(\mathbf{x}, \mathbf{y})$ and $p(\mathbf{x}|\mathbf{y})$ in (7) by their estimates $q_\sigma(\mathbf{x}, \mathbf{y})$ and $q_\sigma(\mathbf{x}|\mathbf{y})$, to arrive at the new objective

$$\mathcal{J}_{\mathrm{CSM}_\sigma}(\boldsymbol{\theta}) = \mathbb{E}_{q_\sigma(\mathbf{x},\mathbf{y})} \|\nabla_\mathbf{x} \log p_{\boldsymbol{\theta}}(\mathbf{x}|\mathbf{y}) - \nabla_\mathbf{x} \log q_\sigma(\mathbf{x}|\mathbf{y})\|^2 \qquad (11)$$

which is the conditional score matching objective when applied to the nonparametric estimates of the unknown target density. We will show below that it is equivalent to a simpler objective, in which we only need to compute gradients of the conditioning kernel $k_\sigma(\mathbf{x}|\mathbf{y})$:

$$\mathcal{J}_{\mathrm{CDSM}_\sigma}(\boldsymbol{\theta}) = \mathbb{E}\|\nabla_\mathbf{x} \log p_{\boldsymbol{\theta}}(\mathbf{x}|\mathbf{y}) - \nabla_\mathbf{x} \log k_\sigma(\mathbf{x}|\mathbf{x}')\|^2 \qquad (12)$$

where the expectation is taken with respect to $p_\mathcal{D}(\mathbf{x}', \mathbf{y}')k_\sigma(\mathbf{x}|\mathbf{x}')l_\sigma(\mathbf{y}|\mathbf{y}')$. We call this objective conditional denoising score matching. Its extrema landscape is the same as $\mathcal{J}_{\mathrm{CSM}_\sigma}$, but it has the advantage of being simpler to evaluate and interpret.

Above, we presented this objective when $k_\sigma$ is the Gaussian kernel, and $l_\sigma$ is simply the identity kernel.

**From CSM to CDSM** We will show here that the stochastic approximation used in denoising score matching can also be used for the conditional case to get to the CDSM objective (12) from the CSM objective (11):

$$\mathcal{J}_{\mathrm{CSM}_\sigma}(\boldsymbol{\theta}) = \mathbb{E}_{q_\sigma(\mathbf{x},\mathbf{y})} \left\| \nabla_\mathbf{x} \log \frac{p_{\boldsymbol{\theta}}(\mathbf{x}|\mathbf{y})}{q_\sigma(\mathbf{x}|\mathbf{y})} \right\|^2 = \mathbb{E}_{q_\sigma(\mathbf{x},\mathbf{y})} \|\nabla_\mathbf{x} \log p_{\boldsymbol{\theta}}(\mathbf{x}|\mathbf{y})\|^2 - S(\boldsymbol{\theta}) + C_1 \qquad (13)$$

where $C_1$ is a constant term that only depends on $q_\sigma(\mathbf{x}|\mathbf{y})$, and

$$
\begin{aligned}
S(\boldsymbol{\theta}) &= \mathbb{E}_{q_\sigma(\mathbf{x},\mathbf{y})} \langle \nabla_\mathbf{x} \log p_{\boldsymbol{\theta}}(\mathbf{x}|\mathbf{y}), \nabla_\mathbf{x} \log q_\sigma(\mathbf{x}|\mathbf{y}) \rangle \\
&= \int q_\sigma(\mathbf{x}, \mathbf{y}) \langle \nabla_\mathbf{x} \log p_{\boldsymbol{\theta}}(\mathbf{x}|\mathbf{y}), \frac{\nabla_\mathbf{x} q_\sigma(\mathbf{x}|\mathbf{y})}{q_\sigma(\mathbf{x}|\mathbf{y})} \rangle \mathrm{d}\mathbf{x}\mathrm{d}\mathbf{y} \\
&= \int q_\sigma(\mathbf{y}) \langle \nabla_\mathbf{x} \log p_{\boldsymbol{\theta}}(\mathbf{x}|\mathbf{y}), \nabla_\mathbf{x} q_\sigma(\mathbf{x}|\mathbf{y}) \rangle \mathrm{d}\mathbf{x}\mathrm{d}\mathbf{y} \\
&= \int q_\sigma(\mathbf{y}) \langle \nabla_\mathbf{x} \log p_{\boldsymbol{\theta}}(\mathbf{x}|\mathbf{y}), \nabla_\mathbf{x} \frac{\int p_\mathcal{D}(\mathbf{x}', \mathbf{y}')k_\sigma(\mathbf{x}|\mathbf{x}')l_\sigma(\mathbf{y}|\mathbf{y}')\mathrm{d}\mathbf{x}'\mathrm{d}\mathbf{y}'}{q_\sigma(\mathbf{y})} \rangle \mathrm{d}\mathbf{x}\mathrm{d}\mathbf{y} \\
&= \int \int p_\mathcal{D}(\mathbf{x}', \mathbf{y}')l_\sigma(\mathbf{y}|\mathbf{y}')k_\sigma(\mathbf{x}|\mathbf{x}') \langle \nabla_\mathbf{x} \log p_{\boldsymbol{\theta}}(\mathbf{x}|\mathbf{y}), \nabla_\mathbf{x} \log k_\sigma(\mathbf{x}|\mathbf{x}') \rangle \mathrm{d}\mathbf{x}'\mathrm{d}\mathbf{y}'\mathrm{d}\mathbf{x}\mathrm{d}\mathbf{y} \\
&= \mathbb{E}_{p_\mathcal{D}(\mathbf{x}',\mathbf{y}')k_\sigma(\mathbf{x}|\mathbf{x}')l_\sigma(\mathbf{y}|\mathbf{y}')} \langle \nabla_\mathbf{x} \log p_{\boldsymbol{\theta}}(\mathbf{x}|\mathbf{y}), \nabla_\mathbf{x} \log k_\sigma(\mathbf{x}|\mathbf{x}') \rangle
\end{aligned}
$$

Plugging this back into equation (13), we find that

$$
\begin{aligned}
\mathcal{J}_{\mathrm{CSM}_\sigma}(\boldsymbol{\theta}) &= \mathbb{E}\|\nabla_\mathbf{x} \log p_{\boldsymbol{\theta}}(\mathbf{x}|\mathbf{y}) - \nabla_\mathbf{x} \log k_\sigma(\mathbf{x}|\mathbf{x}')\|^2 + C_1 - C_2 \\
&= \mathcal{J}_{\mathrm{CDSM}_\sigma}(\boldsymbol{\theta}) + C_1 - C_2
\end{aligned}
$$

where the expectation is with respect to $p_\mathcal{D}(\mathbf{x}', \mathbf{y}')k_\sigma(\mathbf{x}|\mathbf{x}')l_\sigma(\mathbf{y}|\mathbf{y}')$ and $C_2$ is another constant that is only a function of $k_\sigma(\mathbf{x}|\mathbf{x}')$. $\qquad\square$

### B.2 Conditional flow contrastive estimation

As described above, FCE learns the parameter for the density $p_{\boldsymbol{\theta}}$ of an EBM by performing a surrogate classification task: noise is generated from a noise distribution $q_{\boldsymbol{\alpha}}$ which is parameterized as a flow model, and a logistic regression is performed to classify observation into real data samples or noise samples. The objective function is simply the log-odds:

$$\mathcal{J}_{\mathrm{FCE}}(\boldsymbol{\theta}, \boldsymbol{\alpha}) = \mathbb{E}_{p_{\mathrm{data}}(\mathbf{x})} \log \frac{p_{\boldsymbol{\theta}}(\mathbf{x})}{q_{\boldsymbol{\alpha}}(\mathbf{x}) + p_{\boldsymbol{\theta}}(\mathbf{x})} + \mathbb{E}_{q_{\boldsymbol{\alpha}}(\mathbf{x})} \log \frac{q_{\boldsymbol{\alpha}}(\mathbf{x})}{q_{\boldsymbol{\alpha}}(\mathbf{x}) + p_{\boldsymbol{\theta}}(\mathbf{x})} \qquad (14)$$

This objective is minimized with respect to $\boldsymbol{\theta}$ and maximized with respect to $\boldsymbol{\alpha}$: the EBM and the flow model are playing a min-max game. This objective can be extended to the conditional case

naturally: we replace the model density by the conditional density $p_{\boldsymbol{\theta}}(\mathbf{x}|\mathbf{y})$. In the conditional case, it follows that noise samples should also be associated with a conditioning variable, $\mathbf{y}$. One way this can be achieved is by also considering a conditional flow. This also has the additional benefit that an improved flow should lead to better estimation of EBM. Alternatively, a standard (non-conditional) flow could be employed. This would require marginalizing over the conditioning variable, $\mathbf{y}$. The objective simply becomes:

$$\mathcal{J}_{\text{CFCE}}(\boldsymbol{\theta}, \boldsymbol{\alpha}) = \mathbb{E}_{p_{\text{data}(\mathbf{x}, \mathbf{y})}} \log \frac{p_{\boldsymbol{\theta}}(\mathbf{x}|\mathbf{y})}{q_{\boldsymbol{\alpha}}(\mathbf{x}, \mathbf{y}) + p_{\boldsymbol{\theta}}(\mathbf{x}|\mathbf{y})} + \mathbb{E}_{q_{\boldsymbol{\alpha}}(\mathbf{x}, \mathbf{y})} \log \frac{q_{\boldsymbol{\alpha}}(\mathbf{x}, \mathbf{y})}{q_{\boldsymbol{\alpha}}(\mathbf{x}, \mathbf{y}) + p_{\boldsymbol{\theta}}(\mathbf{x}|\mathbf{y})} \quad (15)$$

We can write the flow density as $q_{\boldsymbol{\alpha}}(\mathbf{x}, \mathbf{y}) = p(\mathbf{y}) q_{\boldsymbol{\alpha}}(\mathbf{x}|\mathbf{y})$. This is particularly useful when the conditioning variable $\mathbf{y}$ is discrete, like for instance the index of a dataset or a segment, as we can sample draw a index from a uniform distribution, and use the conditional flow to sample an observation.

## C  Identifiability of the conditional energy-based model

Recall the form of our conditional energy model

$$p_{\boldsymbol{\theta}}(\mathbf{x}|\mathbf{y}) = Z(\mathbf{y}; \boldsymbol{\theta})^{-1} \exp\left(-\mathbf{f}_{\boldsymbol{\theta}}(\mathbf{x})^T \mathbf{g}_{\boldsymbol{\theta}}(\mathbf{y})\right) \quad (16)$$

We present in this section the proofs for the different forms of identifiability that is guaranteed for the feature extractors $\mathbf{f}$ and $\mathbf{g}$. We will focus on the proofs for the feature extractor $\mathbf{f}$, as the proofs for the feature extractor $\mathbf{g}$ are very similar. For the rest of the Appendix, we will denote by $d = d_x$, $m = d_y$ and $n = d_z$.

### C.1  More on the equivalence relations

The relation $\sim_w^{\mathbf{f}}$ in equation (2) is an equivalence relation in the strict term only if $\mathbf{A}$ is full rank. If $\mathbf{A}$ is not full rank (which is only possible if $d_z > d_x$, given the rest of assumptions), then it is not necessarily symmetric. This is not a real problem, and can be fixed by changing the definition to: there exists $\mathbf{A}_1, \mathbf{A}_2$ such that $\mathbf{f}_{\boldsymbol{\theta}} = \mathbf{A}_1 \mathbf{f}_{\boldsymbol{\theta}'} + \mathbf{c}_1$ and $\mathbf{f}_{\boldsymbol{\theta}'} = \mathbf{A}_2 \mathbf{f}_{\boldsymbol{\theta}} + \mathbf{c}_2$. We present the simpler version in the paper for clarity.

### C.2  Proof of Theorem 1

We start by proving the main theoretical result of this paper, which applies to all dimensions of the feature extractor. Alternative and weaker assumptions are discussed after the proof.

**Theorem 1** (Identifiable conditional EBMs)**.** *Assume:*

1. *The feature extractor $\mathbf{f}$ is differentiable, and its Jacobian $\mathbf{J_f}$ is full rank.*

2. *There exist $n + 1$ points $\mathbf{y}^0, \ldots, \mathbf{y}^n$ such that the matrix*
$$\mathbf{R} = \left(\mathbf{g}(\mathbf{y}^1) - \mathbf{g}(\mathbf{y}^0), \ldots, \mathbf{g}(\mathbf{y}^n) - \mathbf{g}(\mathbf{y}^0)\right) \quad (17)$$
   *of size $n \times n$ is invertible.*

*then*
$$p_{\boldsymbol{\theta}}(\mathbf{x}|\mathbf{y}) = p_{\boldsymbol{\theta}'}(\mathbf{x}|\mathbf{y}) \implies \boldsymbol{\theta} \sim_w^{\mathbf{f}} \boldsymbol{\theta}'$$
*where $\sim_w^{\mathbf{f}}$ is defined as follows:*
$$\boldsymbol{\theta} \sim_w^{\mathbf{f}} \boldsymbol{\theta}' \Leftrightarrow \mathbf{f}_{\boldsymbol{\theta}}(\mathbf{y}) = \mathbf{A} \mathbf{f}_{\boldsymbol{\theta}'}(\mathbf{y}) + \mathbf{c} \quad (18)$$
$\mathbf{A}$ *is a $(d_z \times d_z)$-matrix of rank at least $\min(d_z, d_x)$.*

*If, instead or in addition, we assume that:*

3. *The feature extractor $\mathbf{g}$ is differentiable, and its Jacobian $\mathbf{J_g}$ is full rank.*

4. *There exist $n + 1$ points $\mathbf{x}^0, \ldots, \mathbf{x}^n$ such that the matrix*
$$\mathbf{Q} = \left(\mathbf{f}(\mathbf{x}^1) - \mathbf{f}(\mathbf{x}^0), \ldots, \mathbf{f}(\mathbf{x}^n) - \mathbf{f}(\mathbf{x}^0)\right)$$
   *of size $n \times n$ is invertible.*

*then*

$$p_{\boldsymbol{\theta}}(\mathbf{x}|\mathbf{y}) = p_{\boldsymbol{\theta}'}(\mathbf{x}|\mathbf{y}) \implies \boldsymbol{\theta} \sim^{\mathbf{g}}_{w} \boldsymbol{\theta}'$$

*where $\sim^{\mathbf{g}}_{w}$ is defined as follows:*

$$\boldsymbol{\theta} \sim^{\mathbf{g}}_{w} \boldsymbol{\theta}' \Leftrightarrow \mathbf{g}_{\boldsymbol{\theta}}(\mathbf{y}) = \mathbf{B}\mathbf{g}_{\boldsymbol{\theta}'}(\mathbf{y}) + \mathbf{e} \tag{19}$$

**B** *is a $(d_z \times d_z)$-matrix of rank at least $\min(d_z, d_x)$.*

*Finally, if $d_z \geq \max(d_x, d_y)$ and all assumptions 1- 4 hold, then the matrices* **A** *and* **B** *have full rank (equal to $d_z$).*

*Proof.* We will only prove this theorem for the feature extractor $\mathbf{f}$. The proof for $\mathbf{g}$ is very similar. Suppose assumptions 1 and 2 hold.

Consider two parameters $\boldsymbol{\theta}$ and $\tilde{\boldsymbol{\theta}}$ such that

$$p_{\boldsymbol{\theta}}(\mathbf{x}|\mathbf{y}) = p_{\tilde{\boldsymbol{\theta}}}(\mathbf{x}|\mathbf{y}) \tag{20}$$

Then, by applying the logarithm to both sides, we get:

$$\log Z(\mathbf{y}; \boldsymbol{\theta}) - \mathbf{f}_{\boldsymbol{\theta}}(\mathbf{x})^T \mathbf{g}_{\boldsymbol{\theta}}(\mathbf{y}) = \log Z(\mathbf{y}; \tilde{\boldsymbol{\theta}}) - \mathbf{f}_{\tilde{\boldsymbol{\theta}}}(\mathbf{x})^T \mathbf{g}_{\tilde{\boldsymbol{\theta}}}(\mathbf{y}) \tag{21}$$

Consider the points $\mathbf{y}^0, \ldots, \mathbf{y}^n$ provided by assumption 2 for $\mathbf{g}_{\boldsymbol{\theta}}$. We plug each of these points in (21) to obtain $n + 1$ such equations. We subtract the first equation for $\mathbf{y}^0$ from the remaining $n$ equations, and write the resulting equations in matrix form:

$$\mathbf{R}\mathbf{f}_{\boldsymbol{\theta}}(\mathbf{x}) = \tilde{\mathbf{R}}\mathbf{f}_{\tilde{\boldsymbol{\theta}}}(\mathbf{x}) + \mathbf{b} \tag{22}$$

where $\mathbf{R} = (\ldots, \mathbf{g}_{\boldsymbol{\theta}}(\mathbf{y}^l) - \mathbf{g}_{\boldsymbol{\theta}}(\mathbf{y}^0), \ldots)$, $\tilde{\mathbf{R}} = (\ldots, \mathbf{g}_{\tilde{\boldsymbol{\theta}}}(\mathbf{y}^l) - \mathbf{g}_{\tilde{\boldsymbol{\theta}}}(\mathbf{y}^0), \ldots)$, and $\mathbf{b} = (\ldots, \log \frac{Z(\mathbf{y}^l; \boldsymbol{\theta})}{Z(\mathbf{y}^l; \tilde{\boldsymbol{\theta}})} - \log \frac{Z(\mathbf{y}^0; \boldsymbol{\theta})}{Z(\mathbf{y}^0; \tilde{\boldsymbol{\theta}})}, \ldots)$. Since $\mathbf{R}$ is invertible (by assumption 2), we multiply by its inverse from the left to get:

$$\mathbf{f}_{\boldsymbol{\theta}}(\mathbf{x}) = \mathbf{A}\mathbf{f}_{\tilde{\boldsymbol{\theta}}}(\mathbf{x}) + \mathbf{c} \tag{23}$$

where $\mathbf{A} = \mathbf{R}^{-1}\tilde{\mathbf{R}}$ and $\mathbf{c} = \mathbf{R}^{-1}\mathbf{b}$. Now since $\mathbf{f}_{\boldsymbol{\theta}}$ is differentiable and its Jacobian is full rank (assumption 1), by differentiating the last equation we deduce that $\text{rank}(\mathbf{A}) \geq \min(n, d)$, which in turn proves that $\boldsymbol{\theta} \sim^{\mathbf{f}}_{w} \tilde{\boldsymbol{\theta}}$.

Finally, suppose that in addition, assumptions 4 holds. Then there exists $\mathbf{x}^0, \ldots \mathbf{x}^n$ such that $\mathbf{Q} := (\ldots, \mathbf{f}_{\boldsymbol{\theta}}(\mathbf{x}^i) - \mathbf{f}_{\boldsymbol{\theta}}(\mathbf{x}^0), \ldots)$. Plugging these $n + 1$ points into equation (23), and subtracting the first equation for $\mathbf{x}^0$ from the remaining $n$ equations, we get

$$\mathbf{Q} = \mathbf{A}(\ldots, \mathbf{f}_{\tilde{\boldsymbol{\theta}}}(\mathbf{x}^i) - \mathbf{f}_{\tilde{\boldsymbol{\theta}}}(\mathbf{x}^0), \ldots) \tag{24}$$

Since $\mathbf{Q}$ is an $n \times n$ invertible matrix, we conclude that $\mathbf{A}$ is also invertible, which concludes the proof. $\square$

**Intuition behind assumption 2** Assumption 2 requires that the conditioning feature extractor $\mathbf{g}$ has an image that is rich enough. Intuitively, this relaxes the amount of flexibility the main feature extractor $\mathbf{f}$ would need to have if $\mathbf{g}$ were to be very simple. It implies that the search for $\mathbf{f}$ will be naturally restricted to a smaller space, for which we can prove identifiability.

**Proof under weaker assumptions** Assumption 1 of full rank Jacobian can be weakened without changing the conclusion of Theorem 1. In fact, this assumption is only used right after equation (23) to prove that the matrix $\mathbf{A}$ has a rank that is at least equal to $\min(n, d)$. Suppose instead that

1.' There exists a point $\mathbf{x}^0 \in \mathbb{R}^d$ where the Jacobian $\mathbf{J}_{\mathbf{f}_{\boldsymbol{\theta}}}$ of $\mathbf{f}_{\boldsymbol{\theta}}$ exists and is invertible

Then by computing the differential of equation (23) at $\mathbf{x}^0$ (assuming that $\mathbf{J}_{\mathbf{f}_{\tilde{\boldsymbol{\theta}}}}(\mathbf{x}^0)$ exists), we can make the same conclusion on the rank of $\mathbf{A}$.

In fact, this condition can be scrapped altogether if we relax the definition of the equivalence class in Appendix C.1 to have no conditions on the ranks of matrices $\mathbf{A}_1$ and $\mathbf{A}_2$. This however comes at the expense of a relatively weak, and potentially meaningless, equivalence class.

Finally, assumption 2 of Theorem 1 can be replaced by requiring the Jacobian of $\mathbf{g}_{\theta}$ to be differentiable and full rank in at least one point, but this requires the conditioning variable to be continuous.

### C.3 Proof of Proposition 1

**Proposition 1.** *Consider an MLP with $L$ layers, where each layer consists of a linear mapping with weight matrix $\mathbf{W}_l \in \mathbb{R}^{d_l \times d_{l-1}}$ and bias $\mathbf{b}_l$, followed by an activation function. Assume*

- *a. All activation functions are LeakyReLUs.*

- *b. All weight matrices $\mathbf{W}_l$ are full rank.*

- *c. The row dimension of the weight matrices are either monotonically increasing or decreasing: $d_l \geq d_{l+1}, \forall l \in [\![0, L-1]\!]$ or $d_l \leq d_{l+1}, \forall l \in [\![0, L-1]\!]$.*

*Then the MLP has a full rank Jacobian almost everywhere. If in addition, $d_L \leq d_0$, then the MLP is surjective.*

*Proof.* Denote by $\mathbf{x}$ the input to the MLP, and by $\mathbf{x}^l$ the output of layer $l$,

$$\mathbf{x}^0 = \mathbf{x} \tag{25}$$

$$\overline{\mathbf{x}}^l = \mathbf{W}_l \mathbf{x}^{l-1} + \mathbf{b}_l \tag{26}$$

$$\mathbf{x}^l = h(\mathbf{W}_l \mathbf{x}^{l-1} + \mathbf{b}_l) = h(\overline{\mathbf{x}}^l) \tag{27}$$

$$h(y) = \alpha y \mathbf{1}_{y<0} + y \mathbf{1}_{y>0} \tag{28}$$

with $h$ in equation (27) is applied to each element of its input, and $\alpha \in (0, 1)$.

Denote by $\mathbf{v}^l \in \mathbb{R}^{d_l}$ the vector whose elements are

$$v_k^l = h'(\overline{x}_k^l) = \begin{cases} 1 \text{ if } \overline{x}_k^l > 0 \\ \alpha \text{ if } \overline{x}_k^l < 0 \end{cases} \tag{29}$$

which is undefined if $\overline{x}_k^l = 0$, and by $\mathbf{V}_l = \text{diag}(\mathbf{v}^l)$. Note that $\mathbf{V}_l$ is a function of its input, and thus of $\mathbf{x}$, but we keep this implicit for simplicity. Using these notations, and the fact that $h$ is piece-wise linear, we can write,

$$\mathbf{x}^L = h(\overline{\mathbf{x}}^L) = \mathbf{V}_L \overline{\mathbf{x}}^L = \mathbf{V}_L \mathbf{W}_L \mathbf{x}^{L-1} + \mathbf{V}_L \mathbf{b}_{L-1} = \cdots = \overline{\mathbf{V}}^L \mathbf{x} + \overline{\mathbf{b}}^L \tag{30}$$

where $\overline{\mathbf{V}}^l = \mathbf{V}_l \mathbf{W}_l \mathbf{V}_{l-1} \mathbf{W}_{l-1} \ldots \mathbf{V}_1 \mathbf{W}_1$, $\overline{\mathbf{b}}^0 = 0$ and $\overline{\mathbf{b}}^l = \mathbf{V}_l \mathbf{b}_l + \mathbf{V}_l \mathbf{W}_l \overline{\mathbf{b}}^{l-1}$. This is of course only possible if $\overline{x}_k^l \neq 0$ for all $l \in [\![1, L]\!]$ and all $k \in [\![1, d_l]\!]$. As such, define the set

$$\mathcal{N} = \bigcup_{l=1}^{L} \bigcup_{k=1}^{d_l} \left\{ \mathbf{x} \in \mathbb{R}^d | \overline{x}_k^l = 0 \right\} = \bigcup_{l=1}^{L} \bigcup_{k=1}^{d_l} \left\{ \mathbf{x} \in \mathbb{R}^d | (\overline{\mathbf{v}}_k^l)^T \mathbf{x} + \overline{b}_k^l = 0 \right\} \tag{31}$$

where $\overline{\mathbf{v}}_k^l$ is the $k$-th row of $\overline{\mathbf{V}}^l$. For each $\mathbf{x} \notin \mathcal{N}$, we have that $\mathbf{V}_l$ is full rank, and, using Lemma 2, $\overline{\mathbf{V}}^l$ is also a full rank matrix.

While it is true that $\overline{b}_k^l$ and $\overline{\mathbf{v}}_k^l$ are functions of $\mathbf{x}$, yet they only take a finite number of values. Thus, the set $\left\{ \mathbf{x} \in \mathbb{R}^d | (\overline{\mathbf{v}}_k^l)^T \mathbf{x} + \overline{b}_k^l = 0 \right\}$ is included in the union over all the values taken by $\overline{b}_k^j$ and $\overline{\mathbf{v}}_k^j$ up to layer $l$. For each of these values, the set becomes a dot product between a row of $\overline{\mathbf{V}}^j$ which is independent of the input $\mathbf{x}$, and is nonzero because $\overline{\mathbf{V}}^j$ is full rank; such set has measure zero in $\mathbb{R}^d$. Thus, $\mathcal{N}$ is included in a finite union of sets of measure zero, which implies that it also has measure zero.

Now, for all $\mathbf{x} \notin \mathcal{N}$, $\frac{\partial \mathbf{x}^L}{\partial \mathbf{x}}$ exists, and can be computed using the chain rule:

$$\frac{\partial \mathbf{x}^L}{\partial \mathbf{x}} = \prod_{l=L}^{1} \frac{\partial \mathbf{x}^l}{\partial \mathbf{x}^{l-1}} = \prod_{l=L}^{1} \frac{\partial \mathbf{x}^l}{\partial \overline{\mathbf{x}}^l} \frac{\partial \overline{\mathbf{x}}^l}{\partial \mathbf{x}^{l-1}} = \prod_{l=L}^{1} \mathbf{V}_l \mathbf{W}_l = \overline{\mathbf{V}}^L \tag{32}$$

which is full rank. Thus, the MLP has a full rank Jacobian almost everywhere.

The surjectivity is easy to prove since $h$ is surjective and so is $\overline{\mathbf{x}}^l$ as a function of $\mathbf{x}^{l-1}$ if $d_{l-1} \geq d_l$ and $\text{rank}(\mathbf{W}_l) = d_l$. $\square$

**Lemma 1.** *Denote by $\sigma_{min}(\mathbf{A})$ the smallest singular value of a matrix $\mathbf{A}$. Let $\mathbf{M}$ be an $m \times n$ matrix, and $\mathbf{N}$ be an $n \times p$ matrix, such that $m \leq n \leq p$ or $m \geq n \geq p$. Then $\sigma_{min}(\mathbf{MN}) \geq \sigma_{min}(\mathbf{M})\sigma_{min}(\mathbf{N})$.*

*Proof.* The proof in the case $m \geq n \geq p$ can be found in (Arbel et al., 2018, Lemma 10), but we provide a proof here for completeness, and for the other case $m \leq n \leq p$.

Let $\mathbb{R}^n_* := \mathbb{R}^n \setminus \{0\}$, and $\lambda_{min}(\mathbf{A})$ the smallest eigenvalue of $\mathbf{A}$. Recall that for a matrix $\mathbf{A} \in \mathbb{R}^{n \times m}$, with $m \geq n$,

$$\sigma_{\min}(\mathbf{A}) = \sqrt{\lambda_{\min}(\mathbf{A}^T\mathbf{A})} = \sqrt{\inf_{\mathbf{x}\in\mathbb{R}^n_*} \frac{\mathbf{x}^T\mathbf{A}^T\mathbf{A}\mathbf{x}}{\mathbf{x}^T\mathbf{x}}} = \inf_{\mathbf{x}\in\mathbb{R}^n_*} \frac{\|\mathbf{A}\mathbf{x}\|}{\|\mathbf{x}\|} \tag{33}$$

Thus, if the null space of $\mathbf{N}$ is non trivial, then $\sigma_{\min}(\mathbf{N}) = 0$, and the inequality is satisfied. Otherwise, we have $\mathbf{N}\mathbf{x} \neq 0$, $\forall \mathbf{x} \in \mathbb{R}^n_*$,

$$\begin{aligned}
\sigma_{\min}(\mathbf{MN}) &= \inf_{\mathbf{x}\in\mathbb{R}^p_*} \frac{\|\mathbf{MNx}\|}{\|\mathbf{x}\|} \\
&= \inf_{\mathbf{x}\in\mathbb{R}^p_*} \frac{\|\mathbf{MNx}\|\,\|\mathbf{Nx}\|}{\|\mathbf{Nx}\|\,\|\mathbf{x}\|} \\
&\geq \left(\inf_{\mathbf{x}\in\mathbb{R}^p_*} \frac{\|\mathbf{MNx}\|}{\|\mathbf{Nx}\|}\right)\left(\inf_{\mathbf{x}\in\mathbb{R}^p_*} \frac{\|\mathbf{Nx}\|}{\|\mathbf{x}\|}\right) \\
&\geq \left(\inf_{\mathbf{x}\in\mathbb{R}^n_*} \frac{\|\mathbf{Mx}\|}{\|\mathbf{x}\|}\right)\left(\inf_{\mathbf{x}\in\mathbb{R}^p_*} \frac{\|\mathbf{Nx}\|}{\|\mathbf{x}\|}\right) \\
&= \sigma_{\min}(\mathbf{M})\sigma_{\min}(\mathbf{N})
\end{aligned}$$

If, instead, $\mathbf{A} \in \mathbb{R}^{m \times n}$ with $m \leq n$, then

$$\sigma_{\min}(\mathbf{A}) = \sqrt{\lambda_{\min}(\mathbf{A}\mathbf{A}^T)} = \sqrt{\inf_{\mathbf{x}\in\mathbb{R}^m_*} \frac{\mathbf{x}^T\mathbf{A}\mathbf{A}^T\mathbf{x}}{\mathbf{x}^T\mathbf{x}}} = \inf_{\mathbf{x}\in\mathbb{R}^m_*} \frac{\|\mathbf{A}^T\mathbf{x}\|}{\|\mathbf{x}\|} \tag{34}$$

Similarly, if the null space of $\mathbf{M}^T$ is non trivial, then $\sigma_{\min}(\mathbf{M}^T) = \sigma_{\min}(\mathbf{M}) = 0$, and the inequality holds. Otherwise, we have $\mathbf{M}^T\mathbf{x} \neq 0$, $\forall \mathbf{x} \in \mathbb{R}^m_*$,

$$\begin{aligned}
\sigma_{\min}(\mathbf{MN}) &= \inf_{\mathbf{x}\in\mathbb{R}^m_*} \frac{\|\mathbf{N}^T\mathbf{M}^T\mathbf{x}\|}{\|\mathbf{x}\|} \\
&= \inf_{\mathbf{x}\in\mathbb{R}^m_*} \frac{\|\mathbf{N}^T\mathbf{M}^T\mathbf{x}\|\,\|\mathbf{M}^T\mathbf{x}\|}{\|\mathbf{M}^T\mathbf{x}\|\,\|\mathbf{x}\|} \\
&\geq \left(\inf_{\mathbf{x}\in\mathbb{R}^m_*} \frac{\|\mathbf{N}^T\mathbf{M}^T\mathbf{x}\|}{\|\mathbf{M}^T\mathbf{x}\|}\right)\left(\inf_{\mathbf{x}\in\mathbb{R}^m_*} \frac{\|\mathbf{M}^T\mathbf{x}\|}{\|\mathbf{x}\|}\right) \\
&\geq \left(\inf_{\mathbf{x}\in\mathbb{R}^n_*} \frac{\|\mathbf{N}^T\mathbf{x}\|}{\|\mathbf{x}\|}\right)\left(\inf_{\mathbf{x}\in\mathbb{R}^m_*} \frac{\|\mathbf{M}^T\mathbf{x}\|}{\|\mathbf{x}\|}\right) \\
&= \sigma_{\min}(\mathbf{N})\sigma_{\min}(\mathbf{M})
\end{aligned}$$

which concludes the proof. $\qquad\square$

**Lemma 2.** *Consider a finite sequence of matrices $(\mathbf{M}_i)_{1 \leq i \leq p}$, with $\mathbf{M}_i \in \mathbb{R}^{n_{i-1} \times n_i}$. If $\mathbf{M}_i$ is full rank for all $i \in [\![1, p]\!]$, and either $n_0 \leq n_1 \leq \ldots \leq n_p$ or $n_0 \geq n_1 \geq \ldots \geq n_p$, then the product $\mathbf{M}_1\mathbf{M}_2 \ldots \mathbf{M}_p$ is also full rank.*

*Proof.* If two matrices $\mathbf{M}_1$ and $\mathbf{M}_2$ with ordered dimensions are full rank, then $\sigma_{\min}(\mathbf{M}_1) > 0$ and $\sigma_{\min}(\mathbf{M}_2) > 0$. According to Lemma 1, this implies that $\sigma_{\min}(\mathbf{M}_1\mathbf{M}_2) > 0$, and that $\mathbf{M}_1\mathbf{M}_2$ is full rank. The proof for $p \geq 3$ is done by induction on $p$. $\qquad\square$

### C.4 Proof of Proposition 2

**Linear MLPs** The particular case of linear feature extractors is quite interesting. If $d_z \leq d_y$ and the feature extractor $\mathbf{g}$ satisfies the assumptions of Proposition 1, then assumption 2 is trivially satisfied.

On the other hand, if $d_z > d_y$, then assumption 2 can't hold when the network is linear. This signals that it is important to use *deep* nonlinear networks to parameterize the feature extractors, at least in the overcomplete case.

**Proposition 2.** *Consider an MLP* **g** *with L layers, where each layer consists of a linear mapping with weight matrix* $\mathbf{W}_l \in \mathbb{R}^{d_l \times d_{l-1}}$ *and bias* $\mathbf{b}_l$, *followed by an activation function. Assume*

     *a. All activation functions are LeakyReLUs.*

     *b. All weight matrices* $\mathbf{W}_l$ *are full rank.*

     *c. All submatrices of* $\mathbf{W}_l$ *of size* $d_l \times d_l$ *are invertible if* $d_l < d_{l+1}$.

*Then there exist* $d_L + 1$ *points* $\mathbf{y}^0, \ldots, \mathbf{y}^{d_L}$ *such that the matrix* $\mathbf{R} = \left( \mathbf{g}(\mathbf{y}^1) - \mathbf{g}(\mathbf{y}^0), \ldots, \mathbf{g}(\mathbf{y}^{d_L}) - \mathbf{g}(\mathbf{y}^0) \right)$ *is invertible.*

*Proof.* Let $\mathbf{y}^0$ be an arbitrary point in $\mathbb{R}^{d_0}$. Without loss of generality, suppose that $\mathbf{g}(\mathbf{y}^0) = 0$. This is because $\mathbf{y} \mapsto \mathbf{g}(\mathbf{y}) - \mathbf{g}(\mathbf{y}^0)$ is still an MLP that satisfies all the assumptions above. If for any choice of points $\mathbf{y}^1$ to $\mathbf{y}^{d_L}$, the matrix $\mathbf{R}$ defined above isn't invertible, then this means that $\mathbf{g}(\mathbb{R}^{d_0})$ is necessarily included in a subspace of $\mathbb{R}^{d_L}$ of dimension at most $d_L - 1$. In other words, this would imply that the functions $g_1, \ldots, g_{d_L}$ are not linearly independent. However, this is in contradiction with the result of Lemma 8, which stipulates that $g_1, \ldots, g_{d_L}$ are linearly independent, provided all weight matrices satisfy the assumptions of the lemma (which are the same as the assumptions made in this proposition).

Thus, we can conclude that there exist $d_L + 1$ points $\mathbf{y}^0, \ldots, \mathbf{y}^{d_L}$ such that the matrix $\mathbf{R} = \left( \mathbf{g}(\mathbf{y}^1) - \mathbf{g}(\mathbf{y}^0), \ldots, \mathbf{g}(\mathbf{y}^{d_L}) - \mathbf{g}(\mathbf{y}^0) \right)$ is invertible. $\qquad\square$

**Proof under weaker conditions**    Note that the proof argument used for the overcomplete case can be used for the undercomplete as well. This same argument can be proved for ReLU as the nonlinearity instead of LeakyReLU. We chose to give the proof for, and suggest to use the latter because it is needed for Proposition 1.

**Lemma 3.** *Let* $\mathbf{A}$ *be an* $n \times n$ *invertible matrix. Denote by* $\mathbf{a}_n$ *the* $n$-*th row of* $\mathbf{A}$. *Then the matrix* $\mathbf{B} \in \mathbb{R}^{n+1, n+1}$ *such that*

$$\mathbf{B} = \left( \begin{array}{c|c} & \begin{matrix} \gamma_1 \\ \vdots \\ \gamma_{n-1} \\ \lambda \end{matrix} \\ \hline \mathbf{a}_n & 1 \end{array} \right) \quad \text{with } \mathbf{A} \text{ in the upper-left block} \tag{35}$$

*is invertible for any choice of* $\gamma_1, \ldots, \gamma_{n-1}$, *and for* $\lambda \neq 1$.

*Proof.* Denote by $\mathbf{b}_i$ the $i$-th row of $\mathbf{B}$. Let $\alpha_1, \ldots, \alpha_{n+1}$ such that

$$\sum_{i=1}^{n+1} \alpha_i \mathbf{b}_i = 0 \tag{36}$$

Then in particular, by looking at the first $n$ lines of this vectorial equation, we have that $\sum_{i=1}^{n-1} \alpha_i \mathbf{a}_i + (\alpha_n + \alpha_{n+1}) \mathbf{a}_n = 0$. Since $\mathbf{A}$ is invertible, its rows are linearly independent, and thus $\alpha_n = -\alpha_{n+1}$ and $\alpha_i = 0, \forall i < n$. Plugging this back into equation (36), and looking closely at the last equation, we have that $(1 - \lambda)\alpha_n = 0$, and we conclude that $\alpha_{n+1} = \alpha_n = 0$ (because $\lambda \neq 1$), and that $\mathbf{B}$ is invertible. $\qquad\square$

**Lemma 4.** *Consider* $n$ *affine functions* $f_i : \mathbf{x} \in \mathbb{R}^d \mapsto \mathbf{a}_i^T \mathbf{x} + b_i$, *such that the matrix* $\mathbf{A} \in \mathbb{R}^{n \times d}$ *whose rows are the* $\mathbf{a}_i$ *is full column rank, and all its submatrices of size* $d \times d$ *are invertible if* $d < n$. *Then there exist* $n$ *non-empty regions* $\mathcal{H}_1, \ldots, \mathcal{H}_n$ *of* $\mathbb{R}^d$ *defined by the signs of the functions* $f_i$ *(for instance,* $\mathcal{H} = \{\mathbf{x} \in \mathbb{R}^n | \forall i, f_i(\mathbf{x}) > 0\}$*) such that the matrix* $\mathbf{S}^n \in \mathbb{R}^{n \times n}$ *defined as* $S_{i,j}^n = \text{sign}_{\mathbf{x} \in \mathcal{H}_i}(f_j(\mathbf{x}))$ *is invertible.*

*Proof.* We will prove this Lemma by induction on $n$ the number of functions $f_i$. Denote by $V_i = \{\mathbf{x} \in \mathbb{R}^d | f_i(\mathbf{x}) = 0\}$. The sign of $f_i$ changes if we cross the hyperplan $V_i$.

First, suppose that $n = 2$. By assumption, we now that $\mathbf{a}_1 \not\propto \mathbf{a}_2$, and thus the hyperplans $V_1$ and $V_2$ are not parallel and divide $\mathbb{R}^d$ into 4 regions. This implies that the regions $\mathcal{H}_1 = \{\mathbf{x} \in \mathbb{R}^d | \mathbf{a}_1^T \mathbf{x} + b_1 > 0, \mathbf{a}_2^T \mathbf{x} + b_2 > 0\}$ and $\mathcal{H}_2 = \{\mathbf{x} \in \mathbb{R}^d | \mathbf{a}_1^T \mathbf{x} + b_1 > 0, \mathbf{a}_2^T \mathbf{x} + b_2 < 0\}$ are not empty.

Second, suppose that there exists $n$ regions $\mathcal{H}_1, \ldots \mathcal{H}_n$ such that the the matrix $\mathbf{S}^n$ is invertible. Consider the affine function $f_{n+1} = \mathbf{a}_{n+1}^T \mathbf{x} + b_{n+1}$. The hyperplan $V_{n+1} = \{\mathbf{x} \in \mathbb{R}^d | f_{n+1}(\mathbf{x}) = 0\}$ intersects at least one of the regions $\mathcal{H}_1, \ldots \mathcal{H}_n$. This is because $(\ldots, \mathbf{a}_i, \ldots)_{i \in J}$ are linearly independent for any $J$ of size $\min(d, n+1)$ such that $n + 1 \in J$, and thus there exists $i_0$ such that $\mathbf{a}_{n+1} \not\propto \mathbf{a}_{i_0}$. Suppose without loss of generality that this region is $\mathcal{H}_n$. Denote by $\tilde{\mathcal{H}}_n = \{\mathbf{x} \in \mathbb{R}^n | \mathbf{x} \in \mathcal{H}_n, f_{n+1}(\mathbf{x}) < 0\} \subset \mathcal{H}_n$. Now consider the matrix $\tilde{\mathbf{S}}^n$ such that $\tilde{S}_{n,j}^n = \mathrm{sign}_{\mathbf{x} \in \tilde{\mathcal{H}}_n}(f_j(\mathbf{x}))$ and $\tilde{S}_{i,j}^n = S_{i,j}^n$. Because $\tilde{\mathcal{H}}_n \subset \mathcal{H}_n$, we have that $\mathrm{sign}_{\mathbf{x} \in \mathcal{H}_n}(f_j(\mathbf{x})) = \mathrm{sign}_{\mathbf{x} \in \tilde{\mathcal{H}}_n}(f_j(\mathbf{x}))$ and thus $\tilde{\mathbf{S}}^n = \mathbf{S}^n$, which implies that $\tilde{\mathbf{S}}^n$ is also invertible. Now define $\mathcal{H}_{n+1} = \{\mathbf{x} \in \mathbb{R}^n | \mathbf{x} \in \mathcal{H}_n, f_{n+1}(\mathbf{x}) > 0\} \subset \mathcal{H}_n$. Again, the inclusion implies that $\mathrm{sign}_{\mathbf{x} \in \mathcal{H}_n}(f_j(\mathbf{x})) = \mathrm{sign}_{\mathbf{x} \in \tilde{\mathcal{H}}_n}(f_j(\mathbf{x}))$. Finally, consider the regions $\mathcal{H}_1, \ldots, \mathcal{H}_{n-1}, \tilde{\mathcal{H}}_n, \mathcal{H}_{n+1}$, and the matrix $\mathbf{S}^{n+1}$ defined on those regions. Then

$$\mathbf{S}^{n+1} = \begin{pmatrix} & & & u_1 \\ & \mathbf{S}^n & & \vdots \\ & & & u_{n-1} \\ & & & -1 \\ \hline & \mathbf{s}_n^n & & 1 \end{pmatrix} \tag{37}$$

where $u_i = \mathrm{sign}_{\mathbf{x} \in \mathcal{H}_i} f_{n+1}(\mathbf{x})$ and $\mathbf{s}_n^n$ is the $n$-th line of $\mathbf{S}^n$. According to Lemma 3, $\mathbf{S}^{n+1}$ is invertible, which achieves the proof. $\qquad \square$

**Lemma 5.** *Let $h$ denote a LeakyReLU activation function with slope $\lambda \in [0, 1)$ (if $\lambda = 0$, then $h$ is simply a ReLU). Consider $n$ piece-wise affine functions $g_i : \mathbf{x} \in \mathbb{R}^d \mapsto h(\mathbf{a}_i^T \mathbf{x} + b_i)$, such that the matrix $\mathbf{A} \in \mathbb{R}^{n \times d}$ whose rows are the $\mathbf{a}_i$ is full column rank, and all its submatrices of size $d \times d$ are invertible if $d < n$. Then the functions $g_1, \ldots, g_n$ are linearly independent, and their generalized slopes (as piece-wise affine functions) are also linearly independent.*

*Proof.* Let $f_i = \mathbf{a}_i^T \mathbf{x} + b_i$ such that $g_i = h(f_i) = \mathbf{1}_{f_i \geq 0} f_i + \mathbf{1}_{f_i < 0} \lambda f_i$.

The assumptions of Lemma 4 are met for the function $f_1, \ldots, f_n$, and we conclude that there exists n regions $\mathcal{H}_1, \ldots, \mathcal{H}_n$ such that $\mathbf{S}^n = \left(\mathrm{sign}_{\mathbf{x} \in \mathcal{H}_i}(f_j(\mathbf{x}))\right)_{i,j}$ is invertible. Define the matrix $\tilde{\mathbf{S}}$ where we replace all entries of $\mathbf{S}^n$ by $\lambda$ if they are equal to $-1$. Then $\tilde{\mathbf{S}}$ is invertible (in fact, to see this, consider the proof of the previous lemma with the slightly unconventional choice of sign function $\mathrm{sign}(x) = \lambda$ if $x < 0$).

Now consider $\alpha_1, \ldots, \alpha_n$ such that

$$\sum_{i=1}^n \alpha_i g_i = 0 \tag{38}$$

Let $k \in [\![1, n]\!]$, and evaluate this equation at $\mathbf{x} \in \mathcal{H}_k$. After taking the gradient with respect to $\mathbf{x}$, we get

$$\sum_i (\mathbf{1}_{\mathbf{x} \in \mathcal{H}_k, f_i(\mathbf{x}) \geq 0} + \lambda \mathbf{1}_{\mathbf{x} \in \mathcal{H}_k, f_i(\mathbf{x}) < 0}) \alpha_i \mathbf{a}_i = 0 \tag{39}$$

Denote by $\tilde{\mathbf{s}}_k$ the $k$-th line of the matrix $\tilde{\mathbf{S}}$, and define $\mathbf{e}_l = (\alpha_1 a_{1,l}, \ldots, \alpha_n a_{n,l}) \in \mathbb{R}^n$. We can write the $l$-th line of equation (39) as:

$$\tilde{\mathbf{s}}_k^T \mathbf{e}_l = 0 \tag{40}$$

Collating these equations for a fixed $l$ and $k \in [\![1, n]\!]$, we get

$$\tilde{\mathbf{S}} \mathbf{e}_l = 0 \tag{41}$$

which implies that $\mathbf{e}_l = 0$ because $\mathbf{S}$ is invertible. In particular, $\alpha_i a_{i,l} = 0$ for all $i \in [\![1, n]\!]$ and $l \in [\![1, d]\!]$. This implies that $\mathbf{A}_J^T \boldsymbol{\alpha}_J = 0$, where $J \subset [\![1, n]\!]$ of size $\min(n, d)$, $\mathbf{A}_J = (a_{i,l})_{i \in J, l \in [\![1, d]\!]} \in \mathbb{R}^{d \times d}$ is a submatrix of $\mathbf{A}$ and $\boldsymbol{\alpha}_J = (\alpha_i)_{i \in J} \in \mathbb{R}^d$. Since we know, by assumption, that $\mathbf{A}_J$ is invertible for any choice of set of indices $J$ (relevant when $n > d$), we conclude that $\boldsymbol{\alpha} = 0$ and that the functions $g_1, \ldots, g_n$ are linearly independent.

Each function $g_i$ is a piece-wise affine function, with a "generalized slope" equal to $\tilde{\mathbf{a}}_i(\mathbf{x}) = (\mathbf{1}_{f_i \geq 0}(\mathbf{x}) + \lambda \mathbf{1}_{f_i < 0}(\mathbf{x}))\mathbf{a}_i$. As a corollary of the independence of $g_1, \ldots g_n$, we can conclude that the slopes $\tilde{\mathbf{a}}_1(\mathbf{x}), \ldots, \tilde{\mathbf{a}}_n(\mathbf{x})$ are also independent. $\qquad\square$

**Lemma 6.** *Let* $\mathbf{f} = (f_1, \ldots, f_n)$ *be a vector-valued function defined on* $\mathbb{R}^d$. *We suppose that* $f_1, \ldots, f_n$ *are linearly independent piece-wise affine functions, and that their generalized slopes* $\mathbf{a}_1(\mathbf{x}), \ldots, \mathbf{a}_n(\mathbf{x})$ *are also linearly independent. Consider* $m$ *piece-wise affine functions* $g_i : \mathbf{x} \in \mathbb{R}^d \mapsto \mathbf{c}_i^T \mathbf{f}(\mathbf{x}) + d_i$, *such that the matrix* $\mathbf{C} \in \mathbb{R}^{m \times n}$ *whose rows are the* $\mathbf{c}_i$ *is full column rank, and all its submatrices of size* $n \times n$ *are invertible if* $n < m$. *Then there exist* $m$ *non-empty regions* $\mathcal{K}_1, \ldots, \mathcal{K}_m$ *of* $\mathbb{R}^d$ *defined by the signs of the functions* $g_i$ *such that the matrix* $\mathbf{T}^m \in \mathbb{R}^{m \times m}$ *defined as* $T_{i,j}^m = \operatorname{sign}_{\mathbf{x} \in \mathcal{K}_i}(g_j(\mathbf{x}))$ *is invertible.*

*Proof.* Denote by $\tilde{\mathbf{c}}_i(\mathbf{x})$ the generalized slope of the p.w. affine function $g_i$: $\tilde{\mathbf{c}}_i(\mathbf{x}) = \sum_j c_{i,j} \mathbf{a}_j(\mathbf{x})$. The key is to show than under the assumptions made here, the slopes $(\ldots, \tilde{\mathbf{c}}_i(\mathbf{x}), \ldots)_{i \in J}$ are linearly independent for any choice of subset $J \subset [\![1, m]\!]$ of size $\min(m, n)$.

If $m > n$, chose a subset $J \in [\![1, m]\!]$ of size $n$, and let $(\alpha_i)_{i \in J}$ such that $\sum_{i \in J} \alpha_i \tilde{\mathbf{c}}_i(\mathbf{x}) = 0$. By replacing $\tilde{\mathbf{c}}_i$ by its expression, we get: $\sum_j (\sum_i \alpha_i c_{i,j}) \mathbf{a}_j(\mathbf{x}) = 0$. Since $\mathbf{a}_1, \ldots, \mathbf{a}_n$ are linearly independent, we conclude that $\sum_{i \in J} \alpha_i c_{i,j} = 0$ for all $j \in [\![1, n]\!]$. This, along with the full rank assumption on $\mathbf{C}$ prove that $(\alpha_i)_{i \in J} = 0$ and that $(\ldots, \tilde{\mathbf{c}}_i(\mathbf{x}), \ldots)_{i \in J}$ are linearly independent. We can use the same argument if, instead, $m \leq n$, where $J = [\![1, m]\!]$, and conclude.

The rest of the proof follows the same argument of the proof of Lemma 4: we proceed by induction on $m$. For $m = 2$, we know that $\tilde{\mathbf{c}}_1 \not\propto \tilde{\mathbf{c}}_2$, and so the "generalized hyperplans" defined by these two vectors divide $\mathbb{R}^d$ into at least 3 different regions, 2 of which yield a matrix $\mathbf{T}^2$ that is invertible. Then, if the result hold for $m$, then the hyperplan defined by the generalized slope of the $(m+1)$-th p.w. affine function $g_{m+1}$ necessarily intersects one of the regions $\mathcal{K}_1, \ldots, \mathcal{K}_m$ since for any subset $J$ of size $\min(m+1, n)$ s.t. $(m+1) \in J$, the generalized slopes $(\ldots, \tilde{\mathbf{c}}_i(\mathbf{x}), \ldots)_{i \in J}$ are linearly independent. The rest is identical to Lemma 4. $\qquad\square$

**Lemma 7.** *Let* $h$ *denote a LeakyReLU activation function with slope* $\lambda \in [0, 1)$ *(if* $\lambda = 0$, *then* $h$ *is simply a ReLU), and* $\mathbf{f} = (f_1, \ldots, f_n)$ *be a vector-valued function defined on* $\mathbb{R}^d$. *We suppose that* $f_1, \ldots, f_n$ *are linearly independent piece-wise affine functions, and that their generalized slopes* $\mathbf{a}_1(\mathbf{x}), \ldots, \mathbf{a}_n(\mathbf{x})$ *are also linearly independent. Consider* $m$ *piece-wise affine functions* $g_i : \mathbf{x} \in \mathbb{R}^d \mapsto h(\mathbf{c}_i^T \mathbf{f}(\mathbf{x}) + d_i)$, *such that the matrix* $\mathbf{C} \in \mathbb{R}^{m \times n}$ *whose rows are the* $\mathbf{c}_i$ *is full column rank, and all its submatrices of size* $n \times n$ *are invertible if* $n < m$. *Then the functions* $g_1, \ldots, g_m$ *are linearly independent, and their generalized slopes are also linearly independent.*

*Proof.* Let $\tilde{g}_i = \mathbf{c}_i^T \mathbf{f} + d_i$ such that $g_i = h(\tilde{g}_i)$. The assumptions of Lemma 6 are met for the functions $\tilde{g}_1, \ldots, \tilde{g}_m$, and we conclude that there exists m regions $\mathcal{K}_1, \ldots, \mathcal{K}_m$ such that $\mathbf{T}^m = \left(\operatorname{sign}_{\mathbf{x} \in \mathcal{K}_i}(\tilde{g}_j(\mathbf{x}))\right)_{i,j}$ is invertible. Let $\tilde{\mathbf{T}}$ the invertible matrix equal to $\mathbf{T}^m$ after substituting $-1$ for $\lambda$.

Now consider $\alpha_1, \ldots, \alpha_m$ such that $\sum_{i=1}^m \alpha_i g_i = 0$ After taking the gradient with respect to $\mathbf{x}$, we get:

$$\sum_j (\sum_i \alpha_i (\mathbf{1}_{\tilde{g}_i \geq 0}(\mathbf{x}) + \lambda \mathbf{1}_{\tilde{g}_i < 0}(\mathbf{x})) c_{i,j}) \alpha_j(\mathbf{x}) = 0 \qquad (42)$$

Since $\mathbf{a}_1, \ldots, \mathbf{a}_n$ are independent, we conclude that $\sum_i \alpha_i (\mathbf{1}_{\tilde{g}_i \geq 0}(\mathbf{x}) + \lambda \mathbf{1}_{\tilde{g}_i < 0}(\mathbf{x})) c_{i,j}$ for all $j \in [\![1, m]\!]$. This in turn implies that

$$\sum_i \alpha_i (\mathbf{1}_{\tilde{g}_i \geq 0}(\mathbf{x}) + \lambda \mathbf{1}_{\tilde{g}_i < 0}(\mathbf{x})) \mathbf{c}_i = 0 \qquad (43)$$

Let $k \in [\![1, m]\!]$, and evaluate the last equation at $\mathbf{x} \in \mathcal{K}_k$:

$$\sum_i (\mathbf{1}_{\mathbf{x} \in \mathcal{H}_k, f_i(\mathbf{x}) \geq 0} + \lambda \mathbf{1}_{\mathbf{x} \in \mathcal{H}_k, f_i(\mathbf{x}) < 0}) \alpha_i \mathbf{c}_i = 0 \qquad (44)$$

This last equation is similar to equation (39), and we can use the same argument used for the proof of Lemma 5 here (using $\tilde{\mathbf{T}}$ instead of $\tilde{\mathbf{S}}$) and deduce that $\alpha_i = 0$ for all $i$.

We conclude that $g_1, \ldots, g_m$ are linearly independent, and so are their generalized slopes as a consequence. $\qquad\square$

**Lemma 8.** *Let $\mathbf{f}^L = (f_1^L, \ldots, f_{d_L}^L)$ be the output of an L-layer MLP (we assume that $L \geq 2$: there is at least one nonlinearity) that satisfies:*

(a.) *All activation functions are LeakyReLUs with slope $\lambda \in [0, 1)$ (if $\lambda = 0$, then the activation function is simply a ReLU).*

(b.) *All weight matrices $\mathbf{W}_l \in \mathbb{R}^{d_{l+1} \times d_l}$ are full rank, and all submatrices of $\mathbf{W}_l$ of size $d_l \times d_l$ are invertible if $d_l < d_{l+1}$.*

*Then $f_1^L, \ldots, f_{d_L}^L$ are linearly independent. In addition, all the intermediate features $(f_1^l, \ldots, f_{d_l}^l)$ are also linearly independent.*

*Proof.* We prove the Lemma by induction on the number of layers $L \geq 2$. If $L = 2$, then by Lemma 5, we conclude that $f_1, \ldots, f_n$ are independent. If we suppose the result hold for $L \geq 2$, we can use Lemma 7 to prove that it also holds for $L + 1$. Finally, since all layers satisfy the same conditions, the conclusion also applies to intermediate layers. $\qquad\square$

## C.5 Proof of Theorem 2

We will decompose Theorem 2 into two sub-theorems, which will make the proof easier to understand, but also more adaptable into future work. Each of these sub-theorems corresponds to one of the assumptions.

### C.5.1 Positive features

We will prove here a more general version where we assume that each component $f_i$ of the feature extractor $\mathbf{f}$ has a global minimum that is reached, instead of being necessarily non-negative.

**Theorem 2a.** *Assume the assumptions of Theorem 1 hold. Further assume that $n \leq d$, and that each $f_i$ has a global minimum that is reached at least in the limit, and the feature extractor $\mathbf{f} = (f_1, \ldots, f_n)$ is surjective onto the set that is defined by the lower bounds of the $f_i$. Then*

$$p_{\boldsymbol{\theta}}(\mathbf{x}|\mathbf{y}) = p_{\boldsymbol{\theta}'}(\mathbf{x}|\mathbf{y}) \implies \boldsymbol{\theta} \sim_s \boldsymbol{\theta}'$$

*where $\sim_s$ is defined as follows:*

$$\boldsymbol{\theta} \sim_s \boldsymbol{\theta}' \Leftrightarrow \forall i, f_{i,\boldsymbol{\theta}}(\mathbf{x}) = a_i f_{\sigma(i),\boldsymbol{\theta}'}(\mathbf{x}) + b_i \tag{45}$$

*where $\sigma$ is a permutation of $[\![1, n]\!]$, $a_i$ is a non zero scalar and $b_i$ is a scalar.*

*Proof.* Consider two different parameters $\boldsymbol{\theta}$ and $\tilde{\boldsymbol{\theta}}$ such that:

$$p_{\boldsymbol{\theta}}(\mathbf{x}|\mathbf{y}) = p_{\tilde{\boldsymbol{\theta}}}(\mathbf{x}|\mathbf{y}) \tag{46}$$

To simplify notations, denote by $\mathbf{f} = \mathbf{f}_{\boldsymbol{\theta}}$ and $\tilde{\mathbf{f}} = \mathbf{f}_{\tilde{\boldsymbol{\theta}}}$. We start the proof from the conclusion of Theorem 1, since its assumptions hold:

$$\mathbf{f}(\mathbf{x}) = \mathbf{A}\tilde{\mathbf{f}}(\mathbf{x}) + \mathbf{c} \tag{47}$$

where $\mathbf{A}$ is an invertible $n \times n$ matrix and $\mathbf{c}$ a constant vector. Without loss of generality, we can suppose that $f_i$ has an infimum equal to zero, simply by subtracting $\inf f_i$, and including in $\mathbf{c}$, and similarly for $\tilde{\mathbf{f}}$. We will also suppose that the infima are reached, as the next argument would hold if we change exact minima by limits.

Now since $\mathbf{f} \geq 0$ and is surjective, then there exists $\mathbf{x}_0 \in \mathbb{R}^d$ such that $\mathbf{f}(\mathbf{x}_0) = 0$. This implies that $\mathbf{c} = -\mathbf{A}\tilde{\mathbf{f}}(\mathbf{x}_0)$, and that $\mathbf{f}(\mathbf{x}) = \mathbf{A}(\tilde{\mathbf{f}}(\mathbf{x}) - \tilde{\mathbf{f}}(\mathbf{x}_0))$. Define $\mathbf{h}(\mathbf{x}) = \tilde{\mathbf{f}}(\mathbf{x}) - \tilde{\mathbf{f}}(\mathbf{x}_0)$. We know that $\tilde{\mathbf{f}} \geq 0$ and is surjective, and so $\mathbf{h}$ is also surjective, and its image includes $\mathbb{R}_+^n$. Let $\mathbf{I} = (\mathbf{e}_1, \ldots, \mathbf{e}_n)$ be the matrix of canonical basis vectors, or positive scalar multiples of the canonical basis vectors $\mathbf{e}_i$. These must be mapped to the non-negative quadrant, so $\mathbf{A}\mathbf{I}$ must be non-negative, which implies that $\mathbf{A}$ must be non-negative.

Denote by $\mathbf{B} = \mathbf{A}^{-1}$. $\mathbf{B}$ is also non-negative for the same reasons described above. Denote the **rows** of $\mathbf{A}$ by $\mathbf{a}_i$ and the **columns** of $\mathbf{B}$ by $\mathbf{b}_j$. We have by definition of inverse:

$$\mathbf{a}_i^T \mathbf{b}_j = \delta_{ij} \tag{48}$$

where if $i = j$ then $\delta_{ij} = 1$, else $\delta_{ij} = 0$. Now, assume there is a row $\mathbf{a}_k$ which has at least two non-zero entries. By the property above, $d - 1$ of the vectors $\mathbf{b}_j$ must have zero dot-product with that vector. By non-negativity of $\mathbf{B}$ and $\mathbf{A}$, those $d - 1$ vectors must have zeros in the at least two indices corresponding to the non-zeros of $\mathbf{a}_k$. But that means they can only span a $d - 2$-dimensional subspace, and all the $\mathbf{b}_j$ together can only span a $d - 1$-dimensional subspace. This is in contradiction of the invertibility of $\mathbf{B}$. Thus, each $\mathbf{a}_i$ can have only one non-zero entry, which, together with the invertibility of $\mathbf{A}$, proves it is a scaled permutation matrix.

Thus, there exists a permutation $\sigma$ of $[\![1, n]\!]$, such that $f_i(\mathbf{x}) = a_{i,\sigma(i)} \tilde{f}_{\sigma(i)}(\mathbf{x}) + c_i$, which concludes the proof. $\qquad\square$

### C.5.2 Augmented features

**Theorem 2b.** *Assume that $n \leq d$, and that:*

1. *The feature extractor $\mathbf{f}$ is differentiable and surjective, and its Jacobian $\mathbf{J_f}$ is full rank.*

2. *There exist $2n + 1$ points $\mathbf{y}^0, \ldots, \mathbf{y}^{2n}$ such that the matrix*
$$\tilde{\mathbf{R}} = \left( \tilde{\mathbf{g}}(\mathbf{y}^1) - \tilde{\mathbf{g}}(\mathbf{y}^0), \ldots, \tilde{\mathbf{g}}(\mathbf{y}^{2n}) - \tilde{\mathbf{g}}(\mathbf{y}^0) \right) \tag{49}$$
*of size $2n \times 2n$ is invertible.*

*Then*
$$p_{\boldsymbol{\theta}}(\mathbf{x}|\mathbf{y}) = p_{\boldsymbol{\theta}'}(\mathbf{x}|\mathbf{y}) \implies \boldsymbol{\theta} \sim_s \boldsymbol{\theta}'$$
*where $\sim_s$ is defined in (45).*

*Proof.* Similarly to the proof of Theorem 2a, we pass the features $f_i$ through the nonlinear function $\mathbf{H}_i(f_i) = (f_i, f_i^2)$ which produces the augmented features $\tilde{\mathbf{f}}$ introduced in section 2.2.2.

Consider two different parameters $\boldsymbol{\theta}$ and $\tilde{\boldsymbol{\theta}}$ such that:
$$p_{\boldsymbol{\theta}}(\mathbf{x}|\mathbf{y}) = p_{\tilde{\boldsymbol{\theta}}}(\mathbf{x}|\mathbf{y}) \tag{50}$$
Since we have similar assumptions to Theorem 1, we will skip the first part of the proof and make the same conclusion, where the equivalence up to linear transformation here applies to $\mathbf{H}(\mathbf{f}_{\boldsymbol{\theta}})$ and $\mathbf{H}(\mathbf{f}_{\tilde{\boldsymbol{\theta}}})$:

$$\mathbf{H}(\mathbf{f}_{\boldsymbol{\theta}}(\mathbf{x})) = \mathbf{A}\mathbf{H}(\mathbf{f}_{\tilde{\boldsymbol{\theta}}}(\mathbf{x})) + \mathbf{c} \tag{51}$$
where $\mathbf{A}$ is a $2n \times 2n$ matrix of rank at least $n$ because $\mathbf{J_f}$ and $\mathbf{J_H}$ are full rank ($\mathbf{A}$ is not necessarily invertible yet, but this will be proven later) and $\mathbf{c}$ a constant vector. By replacing $\mathbf{H}$ by its expression, we get:
$$\begin{pmatrix} \mathbf{f}_{\boldsymbol{\theta}}(\mathbf{x}) \\ \mathbf{f}_{\boldsymbol{\theta}}^2(\mathbf{x}) \end{pmatrix} = \begin{pmatrix} \mathbf{A}^{(1)} & \mathbf{A}^{(2)} \\ \mathbf{A}^{(3)} & \mathbf{A}^{(4)} \end{pmatrix} \begin{pmatrix} \mathbf{f}_{\tilde{\boldsymbol{\theta}}}(\mathbf{x}) \\ \mathbf{f}_{\tilde{\boldsymbol{\theta}}}^2(\mathbf{x}) \end{pmatrix} + \begin{pmatrix} \boldsymbol{\alpha} \\ \boldsymbol{\beta} \end{pmatrix} \tag{52}$$
where each $\mathbf{A}^{(i)}$ is an $n \times n$ matrix, and $\mathbf{c} = (\boldsymbol{\alpha}, \boldsymbol{\beta})$. To simplify notations, denote by $\mathbf{h} = \mathbf{f}_{\tilde{\boldsymbol{\theta}}}$. We will also drop reference to $\boldsymbol{\theta}$ and $\tilde{\boldsymbol{\theta}}$. The first $n$ lines in the previous equation are:
$$f_i(\mathbf{x}) = \sum_{j=1}^{n} A_{ij}^{(1)} h_j(\mathbf{x}) + A_{ij}^{(2)} h_j^2(\mathbf{x}) + \alpha_i \tag{53}$$

and the last $n$ lines are:
$$f_i^2(\mathbf{x}) = \sum_{j=1}^{n} A_{ij}^{(3)} h_j(\mathbf{x}) + A_{ij}^{(4)} h_j^2(\mathbf{x}) + \beta_i \tag{54}$$

Fix an index $i$ in equations (53) and (54). To alleviate notations and reduce the number of subscripts and superscripts, we introduce $a_j = A_{ij}^{(1)}$, $b_j = A_{ij}^{(2)}$, $c_j = A_{ij}^{(3)}$, $d_j = A_{ij}^{(4)}$, $\alpha = \alpha_i$ and $\beta = \beta_i$. This proof is done in 5 steps. Note that the surjectivity assumption is key for the rest of the proof, and it requires that we set the dimension of the feature extractor to be lower than the dimension of the observations.

By equating equations (54) and (53) after squaring, we get, using our new notations:
$$\left( \sum_j a_j h_j(\mathbf{x}) + b_j h_j^2(\mathbf{x}) + \alpha \right)^2 = \sum_j c_j h_j(\mathbf{x}) + d_j h_j^2(\mathbf{x}) + \beta \tag{55}$$

**Step 1** First, since $\mathbf{h}$ is surjective, there exists a point where it is equal to zero. Evaluating equation (55) at this point shows that $\beta = \alpha^2$.

**Step 2** Second, the left hand side of equation (55) has terms raised to the power 4. These terms grow to infinity much faster than the rest of the terms of the rhs and the lhs. It is thus equal to zero. More rigorously, consider the vectors $\mathbf{e}_l(y) = (0, \ldots, y, \ldots, 0) \in \mathbb{R}^n$ where the only non zero entry is $y$ at the $l$-th position. Each of these vectors has a preimage by $\mathbf{h}$ (since it is surjective), which we denote by $\mathbf{x}_l(y)$. By evaluating equation (55) at each of these points, we get

$$(a_l y + b_l y^2 + \alpha)^2 = c_l y + d_l y^2 + \beta \tag{56}$$

Divide both sides of this equation by $y^4$, then take the limit $y \to \infty$. The right hand side will converge to 0, while the left hand side will converge to $b_l$, which shows that $b_l = 0$. By doing this process for all $l \in [\![1, n]\!]$, we can show that $\mathbf{b} = 0$.

**Step 3** So far, we've shown that (55) becomes, after expanding the square in the lhs, and writing $\sum_j a_j h_j(\mathbf{x}) = \mathbf{a}^T \mathbf{h}(\mathbf{x})$:

$$(\mathbf{a}^T \mathbf{h}(\mathbf{x}))^2 + 2\alpha \mathbf{a}^T \mathbf{h}(\mathbf{x}) + \alpha^2 = \sum_j c_j h_j(\mathbf{x}) + d_j h_j^2(\mathbf{x}) + \alpha^2 \tag{57}$$

Let's again consider the vectors $\mathbf{e}_l(y)$ from earlier, and their preimages $\mathbf{x}_l(y)$. By evaluating (57) at the points $\mathbf{x}_l(y)$, we get

$$a_l^2 y^2 + 2\alpha a_l y + \alpha^2 = c_l y + d_l y^2 + \alpha^2 \tag{58}$$

Divide both sides by $y$, and take the limit $y \to 0$. The lhs converges to $2\alpha a_l$, while the rhs converges to $c_l$. Since this is valid for all $l \in [\![1, n]\!]$, we conclude that $\mathbf{c} = 2\alpha \mathbf{a}$. It also follows that $\mathbf{d} = \mathbf{a}^2$.

**Step 4** Injecting this back into equation (57), and writing $\sum_j d_j h_j^2(\mathbf{x}) = \mathbf{h}(\mathbf{x})^T \operatorname{diag}(\mathbf{d}) \mathbf{h}(\mathbf{x})$, we are left with:

$$(\mathbf{a}^T \mathbf{h}(\mathbf{x}))^2 = \mathbf{h}(\mathbf{x})^T \operatorname{diag}(\mathbf{d}) \mathbf{h}(\mathbf{x}) \tag{59}$$

By applying the trace operator to both sides of this equation, and rearranging terms, we get

$$\operatorname{trace}\left( \left( \mathbf{a}\mathbf{a}^T - \operatorname{diag}(\mathbf{d}) \right) \mathbf{h}(\mathbf{x}) \mathbf{h}(\mathbf{x})^T \right) = 0 \tag{60}$$

which is of the form $\operatorname{trace}(\mathbf{C}^T \mathbf{B}(\mathbf{x})) = 0$. This is a dot product on the space $\mathcal{S}_n$ of $n \times n$ symmetric matrices (both $\mathbf{C}$ and $\mathbf{B}(\mathbf{x})$ are symmetric!), which is a vector space of dimension $\frac{n(n+1)}{2}$. If we can show that the matrix $\mathbf{C}$ is orthogonal to a basis of $\mathcal{S}_n$, then we can conclude that $\mathbf{C} = 0$.

For this, let $(\mathbf{e}_j)_{1 \le j \le n}$ be the Euclidean basis of $\mathbb{R}^n$, where each vector $\mathbf{e}_j$ has one non-zero entry equal to 1 at index $j$, and let $(\mathbf{E}_{ij})_{1 \le i \le n, 1 \le j \le n}$ be the Euclidean basis of $\mathbb{R}^{n \times n}$, where each matrix $\mathbf{E}_{ij}$ has only one non-zero entry equal to 1 at row $i$ and column $j$.

Now since $\mathbf{h}$ is surjective, there exists $\mathbf{x}_j$ such that $\mathbf{h}(\mathbf{x}_j) = \mathbf{e}_j$, and $\mathbf{h}(\mathbf{x}_j)\mathbf{h}(\mathbf{x}_j)^T = \mathbf{e}_j \mathbf{e}_j^T = \mathbf{E}_{jj}$. The $n$ different $\mathbf{x}_j$ give us our first $n$ matrices we will use to construct a basis of $\mathcal{S}_n$. We now need to find $\frac{n(n-1)}{2}$ remaining basis matrices. For this, consider the sums $(\mathbf{e}_j + \mathbf{e}_l)_{1 \le j < l \le n}$, of which there is exactly $\frac{n(n-1)}{2}$. Each of these sums of vectors have a preimage $\mathbf{x}_{j,l}$ by $\mathbf{h}$, and $\mathbf{h}(\mathbf{x}_{j,l})\mathbf{h}(\mathbf{x}_{j,l})^T = (\mathbf{e}_j + \mathbf{e}_l)(\mathbf{e}_j + \mathbf{e}_l)^T = \mathbf{E}_{jj} + \mathbf{E}_{ll} + (\mathbf{E}_{il} + \mathbf{E}_{li})$, which is a matrix in $\mathcal{S}_n$ that is linearly independent of all $\mathbf{E}_{jj}$, and all other $(\mathbf{e}_s + \mathbf{e}_t)(\mathbf{e}_s + \mathbf{e}_t)^T$ where $(s, t) \ne (j, l)$ because they have non-zero entries at different rows and columns.

We have then found a total of $\frac{n(n+1)}{2}$ different vectors $(\mathbf{x}_1, \ldots, \mathbf{x}_n, \mathbf{x}_{1,2}, \ldots, \mathbf{x}_{n-1,n})$ such that their images by $\mathbf{h}\mathbf{h}^T$ form a basis of $\mathcal{S}_n$. If we now evaluate equation (60) at each of these points, we find that the matrix $\mathbf{a}\mathbf{a}^T - \operatorname{diag}(\mathbf{d})$ is orthogonal to a basis of $\mathcal{S}_n$, which implies that it is necessarily equal to 0. This in turn implies that $\mathbf{a}\mathbf{a}^T$ is a diagonal matrix, and that $a_j a_l = 0$ for all $j \ne l$, which implies that at most one $a_j$ is non-zero.

**Step 5** So far, we have proven that, among other things, $A_{i,j}^{(2)} = 0$ for all $i, j$. We now go back to equation (53), which we can write as:

$$\mathbf{f}(\mathbf{x}) = \mathbf{A}^{(1)} \mathbf{h}(\mathbf{x}) + \boldsymbol{\alpha} \tag{61}$$

Both $\mathbf{f}$ and $\mathbf{h}$ are differentiable, and according to assumption 2, $J_{\mathbf{f}}$ has rank $n$ (it is full rank and $n \le d$). Thus, by differentiating the last equation, we conclude that $\mathbf{A}^{(1)}$ has rank $n$, and is thus invertible.

**Conclusion** We've shown that $f_i(\mathbf{x}) = a_j h_j(\mathbf{x}) + \alpha_i$, where $a_j = A_{ij}^{(1)}$. This is valid for all $i \in [\![1, n]\!]$. Now since $\mathbf{A}^{(1)}$ is invertible, the non-zero entry $A_{ij}^{(1)}$ has to be in a different column for each row, otherwise some rows will be linearly dependent. Thus, there exists a permutation $\sigma$ of $[\![1, n]\!]$, such that $A_{i\sigma(i)}^{(1)} \neq 0$, and we deduce that

$$f_i(\mathbf{x}) = a_{\sigma(i)} h_{\sigma(i)}(\mathbf{x}) + \alpha_i \tag{62}$$

which concludes the proof.

From the second conclusion of step 3, we have that $\mathbf{d} = \mathbf{a}^2$. Combined with the fact that exactly one element of $\mathbf{a}$ is nonzero such that $\mathbf{A}^{(1)}$ is full rank, this implies that $\mathbf{A}^{(4)}$ is also full rank, which in turn means that $\mathbf{A}$ is full rank. $\qquad\square$

### C.6 Proof of Theorem 3

**Theorem 3.** *Let $p(\mathbf{x}|\mathbf{y})$ be a conditional probability density. Assume that $\mathcal{X}$ and $\mathcal{Y}$ are compact Hausdorff spaces, and that $p(\mathbf{x}|\mathbf{y}) > 0$ almost surely $\forall (\mathbf{x}, \mathbf{y}) \in \mathcal{X} \times \mathcal{Y}$. Then for each $\varepsilon > 0$, there exists $(\boldsymbol{\theta}, n) \in \Theta \times \mathbb{N}$, where $n$ is the dimension of the feature extractor, such that $\sup_{\mathbf{x}, \mathbf{y}} |p_{\boldsymbol{\theta}}(\mathbf{x}|\mathbf{y}) - p(\mathbf{x}|\mathbf{y})| < \varepsilon$.*

*Proof.* We consider here two cases.

**Continuous auxiliary variable** Recall the form of our model:

$$\log p_{\boldsymbol{\theta}}(\mathbf{x}|\mathbf{y}) = -\log Z(\mathbf{y}) - \mathbf{f}(\mathbf{x})^T \mathbf{g}(\mathbf{y}) \tag{63}$$

By parameterizing each of $f_i, g_i$ as neural networks, these functions can approximate continuous function on their respective domains arbitrarily well. According to Lemma 9, this implies that any continuous function on $\mathcal{X} \times \mathcal{Y}$ can be approximated arbitrarily well by a term of the form $-\mathbf{f}(\mathbf{x})^T \mathbf{g}(\mathbf{y})$.

Thus, any continuous function can be approximated by $\log p_{\boldsymbol{\theta}}(\mathbf{x}|\mathbf{y}) + \log Z(\mathbf{y})$ for some $\boldsymbol{\theta}$, where $Z(\mathbf{y})$ captures the difference in scale between the function in question and the normalized density $p_{\boldsymbol{\theta}}(\mathbf{x}|\mathbf{y})$. We apply this result to $\log p(\mathbf{x}|\mathbf{y})$: for any $\varepsilon > 0$, there exists $(\boldsymbol{\theta}, n) \in \Theta \times \mathbb{N}$ such that:

$$\sup_{\mathbf{x}, \mathbf{y}} \left| \log p(\mathbf{x}|\mathbf{y}) + \sum_{i=1}^{n} f_i(\mathbf{x}; \boldsymbol{\theta}) g_i(\mathbf{y}; \boldsymbol{\theta}) \right| < \varepsilon \tag{64}$$

Since $p(\mathbf{x}|\mathbf{y}) > 0$ a.s. on $\mathcal{X} \times \mathcal{Y}$, $\log p(\mathbf{x}|\mathbf{y})$ is finite and bounded. So is the term $-\sum_{i=1}^{n} f_i(\mathbf{x}; \boldsymbol{\theta}) g_i(\mathbf{y}; \boldsymbol{\theta})$. We can then use the fact that $\exp$ is Lipschitz on compacts to conclude for $p(\mathbf{x}|\mathbf{y})$, to conclude that:

$$\sup_{\mathbf{x}, \mathbf{y}} |p(\mathbf{x}|\mathbf{y}) - p_{\boldsymbol{\theta}}(\mathbf{x}|\mathbf{y})| < K\varepsilon \tag{65}$$

where $K$ is the Lipschitz constant of $\exp$, which concludes the proof.

**Discrete auxiliary variable** If $\mathbf{y}$ is discrete and $\mathcal{Y}$ is compact, then $\mathbf{y}$ only takes finitely many values. In this case, we do not need Lemma 9 for the proof. $\mathbf{g}(\mathbf{y})$ can simply be a lookup table, and we learn different approximations for each fixed value of $\mathbf{y}$, since $\mathbf{f}$ has the universal approximation capability, which concludes the proof. $\qquad\square$

Denote by $\mathcal{C}(\mathcal{X})$ (respectively $\mathcal{C}(\mathcal{Y})$ and $\mathcal{C}(\mathcal{X} \times \mathcal{Y})$) the Banach algebra of continuous functions from $\mathcal{X}$ (respectively $\mathcal{Y}$ and $\mathcal{X} \times \mathcal{Y}$) to $\mathbb{R}$. For any subsets of functions $\mathcal{F}_{\mathcal{X}} \subset \mathcal{C}(\mathcal{X})$ and $\mathcal{F}_{\mathcal{Y}} \subset \mathcal{C}(\mathcal{Y})$, let $\mathcal{F}_{\mathcal{X}} \otimes \mathcal{F}_{\mathcal{Y}} := \{\sum_{i=1}^{n} f_i g_i | n \in \mathbb{N}, f_i \in \mathcal{F}_{\mathcal{X}}, g_i \in \mathcal{F}_{\mathcal{Y}}\}$ be the set of *all linear combinations* of products of functions from $\mathcal{F}_{\mathcal{X}}$ and $\mathcal{F}_{\mathcal{Y}}$ to $\mathbb{R}$. The energy function defining our model belongs to this last set. Finally, universal approximation is expressed in terms of density: for instance, the set of functions $\mathcal{F}_{\mathcal{X}}$ have universal approximation of $\mathcal{C}(\mathcal{X})$ if it is dense in it, *i.e.* for any function in $\mathcal{C}(\mathcal{X})$, we can always find a limit of a sequence of functions of $\mathcal{F}_{\mathcal{X}}$ that converges to it. We mathematically express density by writing $\overline{\mathcal{F}_{\mathcal{X}}} = \mathcal{C}(\mathcal{X})$.

Let $\mathcal{F}_{\mathcal{X}}$ (respectively $\mathcal{F}_{\mathcal{Y}}$) be the set of deep neural networks with input in $\mathcal{X}$ (respectively in $\mathcal{Y}$). The universal approximation capability is summarised in the following Lemma.

**Lemma 9** (Universal approximation capability). *Suppose the following:*

*(i) $\mathcal{X}$ and $\mathcal{Y}$ are compact Hausdorff spaces.*

*(ii) $\overline{\mathcal{F}_{\mathcal{X}}} = \mathcal{C}(\mathcal{X})$ and $\overline{\mathcal{F}_{\mathcal{Y}}} = \mathcal{C}(\mathcal{Y})$*

*then $\overline{\mathcal{F}_{\mathcal{X}} \otimes \mathcal{F}_{\mathcal{Y}}} = \mathcal{C}(\mathcal{X} \times \mathcal{Y})$. All completions here are with respect to the infinity norm.*

*Proof.* We prove this theorem in two steps:

1. We first prove that $\mathcal{F}_{\mathcal{X}} \otimes \mathcal{F}_{\mathcal{Y}}$ is dense in $\mathcal{C}(\mathcal{X}) \otimes \mathcal{C}(\mathcal{Y})$ using the hypotheses of Theorem 3.
2. we prove that $\mathcal{C}(\mathcal{X}) \otimes \mathcal{C}(\mathcal{Y})$ is dense in $\mathcal{C}(\mathcal{X} \times \mathcal{Y})$ using Theorem 5.

**Step 1**    Let $\varepsilon > 0$. Let $h \in \mathcal{C}(\mathcal{X}) \otimes \mathcal{C}(\mathcal{Y})$. Then there exists $k \in \mathbb{N}$ and functions $f_i \in \mathcal{C}(\mathcal{X})$ and $g_i \in \mathcal{C}(\mathcal{Y})$ such that $h = \sum_{i=1}^{k} f_i g_i$. For each $i$, since $\mathcal{F}_{\mathcal{Y}}$ dense in $\mathcal{C}(\mathcal{Y})$, there exists $\tilde{g}_i \in \mathcal{F}_{\mathcal{Y}}$ such that $\|g_i - \tilde{g}_i\|_\infty < \frac{\varepsilon}{2k\|f_i\|_\infty}$. From $\mathcal{F}_{\mathcal{X}}$ dense in $\mathcal{C}(\mathcal{X})$, there exists $\tilde{f}_i \in \mathcal{F}_{\mathcal{X}}$ such that $\|f_i - \tilde{f}_i\|_\infty < \frac{\varepsilon}{2k\|\tilde{g}_i\|_\infty}$. We then have

$$\|f_i g_i - \tilde{f}_i \tilde{g}_i\|_\infty = \|f_i g_i - f_i \tilde{g}_i + f_i \tilde{g}_i - \tilde{f}_i \tilde{g}_i\|_\infty \le \|f_i\|_\infty \|g_i - \tilde{g}_i\|_\infty + \|\tilde{g}_i\|_\infty \|f_i - \tilde{f}_i\|_\infty < \frac{\varepsilon}{k} \quad (66)$$

Using this, we conclude that

$$\|h - \sum_{i=1}^{k} \tilde{f}_i \tilde{g}_i\|_\infty \le \sum_{i=1}^{k} \|f_i g_i - \tilde{f}_i \tilde{g}_i\|_\infty < \varepsilon \quad (67)$$

which proves that $\mathcal{F}_{\mathcal{X}} \otimes \mathcal{F}_{\mathcal{Y}}$ is dense in $\mathcal{C}(\mathcal{X}) \otimes \mathcal{C}(\mathcal{Y})$.

**Step 2**    We will use the Stone-Weirstrass theorem for this step. It is enough to show that:

(i) $\mathcal{X} \times \mathcal{Y}$ is a compact Hausdorff space.
(ii) $\mathcal{C}(\mathcal{X}) \otimes \mathcal{C}(\mathcal{Y}) \subset \mathcal{C}(\mathcal{X} \times \mathcal{Y})$.
(iii) $\mathcal{C}(\mathcal{X}) \otimes \mathcal{C}(\mathcal{Y})$ is a unital sub-algebra of $\mathcal{C}(\mathcal{X} \times \mathcal{Y})$ (see Definition 3).
(iv) $\mathcal{C}(\mathcal{X}) \otimes \mathcal{C}(\mathcal{Y})$ separates points in $\mathcal{X} \times \mathcal{Y}$ (see Definition 3).

To prove $(i)$, we use the fact that every finite product of compact spaces is a compact space, and every finite product of Hausdorff spaces is a Hausdorff space. Points $(ii)$ and $(iii)$ are easy to verify. To prove $(iv)$, let $(\mathbf{x}, \mathbf{y})$ and $(\mathbf{x}', \mathbf{y}')$ be distinct points in $\mathcal{X} \times \mathcal{Y}$. Assume that $\mathbf{x} \ne \mathbf{x}'$ (we proceed similarly if $\mathbf{y} \ne \mathbf{y}'$). Define the continuous function $f \in \mathcal{C}(\mathcal{X})$ such that $f(\mathbf{x}) \ne 0$ and $f(\mathbf{x}') = 0$. Then for $g = 1 \in \mathcal{C}(\mathcal{Y})$, we have $f(\mathbf{x})g(\mathbf{y}) = f(\mathbf{x}) \ne 0 = f(\mathbf{x}')g(\mathbf{y}')$.

All the conditions required to use the Stone-Weirestrass Theorem are verified, and we can conclude that $\mathcal{C}(\mathcal{X}) \otimes \mathcal{C}(\mathcal{Y})$ is dense in $\mathcal{C}(\mathcal{X} \times \mathcal{Y})$

**Conclusion**    Combining the results of steps 1 and 2, we conclude that $\mathcal{F}_{\mathcal{X}} \otimes \mathcal{F}_{\mathcal{Y}}$ is dense in $\mathcal{C}(\mathcal{X} \times \mathcal{Y})$. $\square$

**Definition 3.** *Let $K$ be a compact Hausdorff space. Consider the Banach algebra $\mathcal{C}(K)$ equipped with the supremum norm $\|f\|_\infty = \sup_{t \in K} |f(t)|$. Then:*

1. *$\mathcal{A} \in \mathcal{C}(K)$ is a unital sub-algebra if:*

   *(i) $1 \subset \mathcal{A}$.*
   *(ii) for all $f, g \in \mathcal{A}$ and $\alpha, \beta \in \mathbb{R}$, we have $\alpha f + \beta g \in \mathcal{A}$ and $fg \in \mathcal{A}$.*

2. *$\mathcal{A} \subset \mathcal{C}(K)$ separates points of $K$ if $\forall s, t \in K$ such that $s \ne t$, $\exists f \in \mathcal{A}$ s.t. $f(s) \ne f(t)$.*

**Theorem 5** (Stone-Weirstrass). *Let $K$ be a compact Hausdorff space, and $\mathcal{A}$ a unital sub-algebra of $\mathcal{C}(K)$ which separates points of $K$. Then $\mathcal{A}$ is dense in $\mathcal{C}(K)$.*

*Proof.* A proof to this theorem can be found in many references, for instance Brosowski and Deutsch (1981). $\square$

# D  Latent variable estimation in generative models

Recall the generative model of IMCA: we observe a random variable $\mathbf{x} \in \mathbb{R}^d$ as a result of a nonlinear transformation $\mathbf{h}$ of a latent variable $\mathbf{z} \in \mathbb{R}^d$ whose distribution is conditioned on an auxiliary variable $\mathbf{y}$ that is also observed:

$$\mathbf{z} \sim p(\mathbf{z}|\mathbf{y}) \tag{68}$$
$$\mathbf{x} = \mathbf{h}(\mathbf{z}) \tag{69}$$

We assume the latent variable in the IMCA model has a density of the form

$$p(\mathbf{z}|\mathbf{y}) = \mu(\mathbf{z})e^{\sum_i \mathbf{T}_i(z_i)^T \boldsymbol{\lambda}_i(\mathbf{y}) - \Gamma(\mathbf{y})} \tag{70}$$

where $\mu$ is not necessarily factorial.

Further, we will suppose that the density $p(\mathbf{z}|\mathbf{y})$ belongs to the following subclass of the exponential families, introduced by Khemakhem et al. (2020):

**Definition 4** (Strongly exponential). *We say that an exponential family distribution is* strongly exponential *if for any subset $\mathcal{X}$ of $\mathbb{R}$ the following is true:*

$$\left(\exists\,\boldsymbol{\theta} \in \mathbb{R}^k \,|\, \forall x \in \mathcal{X}, \langle\, \mathbf{T}(x), \boldsymbol{\theta} \,\rangle = const\right) \implies (\Lambda(\mathcal{X}) = 0 \text{ or } \boldsymbol{\theta} = 0) \tag{71}$$

*where $\Lambda$ is the Lebesgue measure.*

If we suppose that only $n$ out of $d$ components of the latent variable are modulated by the auxiliary variable $\mathbf{y}$ (equivalently, if we suppose that the parameters $\boldsymbol{\lambda}_{n+1:d}(\mathbf{y})$ are constant), then we can write its density as

$$p(\mathbf{z}|\mathbf{y}) = \mu(\mathbf{z})e^{\sum_{i=1}^{n} \mathbf{T}_i(z_i)^T \boldsymbol{\lambda}_i(\mathbf{y}) - \Gamma(\mathbf{y})} \tag{72}$$

The term $e^{\sum_{i=n+1}^{d} \mathbf{T}_i(z_i)^T \boldsymbol{\lambda}_i}$ is absorbed into $\mu(\mathbf{z})$. This last expression will be useful for dimensionality reduction.

To estimate the latent variables of the IMCA model, we fit an augmented version of our energy model

$$p_{\boldsymbol{\theta}}(\mathbf{x}|\mathbf{y}) = Z(\mathbf{y};\boldsymbol{\theta})^{-1} \exp\left(-\mathbf{H}(\mathbf{f}_{\boldsymbol{\theta}}(\mathbf{x}))^T \mathbf{g}_{\boldsymbol{\theta}}(\mathbf{y})\right) \tag{73}$$

where $\mathbf{H}(\mathbf{f}(\mathbf{x})) = (\mathbf{H}_1(f_1(\mathbf{x})), \ldots, \mathbf{H}_d(f_d(\mathbf{x})))$, and each $\mathbf{H}_l$ is a (nonlinear) output activation. An example of such map is $\mathbf{H}_l(x) = (x, x^2)$.

In this section, we present the proofs for the estimation of the Independently Modulated Component Analysis by an identifiable energy model. These proofs are based on similar ideas and techniques to previous proofs, but are different enough that we can't forgo them.

## D.1  Assumptions

We prove dimensionality reduction capability in Theorem 6. We will decompose Theorem 4 into two sub-theorems, which will make the proof easier to understand, but also more adaptable into future work. For the sake of clarity, we will separate its assumptions into smaller assumptions, and refer to them when needed in the proofs.

   (i) The observed data follows the exponential IMCA model of equations (68)-(70).
  (ii) The mixing function $\mathbf{h} : \mathbb{R}^d \to \mathbb{R}^d$ in (69) is invertible.
 (iii) The sufficient statistics $\mathbf{T}_i$ in (70) are differentiable, and the functions $T_{ij} \in \mathbf{T}_i$ are linearly independent on any subset of $\mathcal{X}$ of measure greater than zero.
  (iv) There exist $k+1$ distinct points $\mathbf{y}^0, \ldots, \mathbf{y}^k$ such that the matrix

$$\mathbf{L} = (\boldsymbol{\lambda}(\mathbf{y}_1) - \boldsymbol{\lambda}(\mathbf{y}_0), \ldots, \boldsymbol{\lambda}(\mathbf{y}_k) - \boldsymbol{\lambda}(\mathbf{y}_0))$$

  of size $k \times k$ is invertible, where $k = \sum_{i=1}^{d} \dim(\mathbf{T}_i)$.
   (v) We fit the model (73) to the conditional density $p(\mathbf{x}|\mathbf{y})$, where we assume the feature extractor $\mathbf{f}(\mathbf{x})$ to be differentiable, $d$-dimensional, and the pointwise nonlinearitiy $\mathbf{H}$ to be differentiable and $k$-dimensional, and the dimension of its vector-valued components $\mathbf{H}_l$ to be chosen from $(\dim(\mathbf{T}_1), \ldots, \dim(\mathbf{T}_d))$ without replacement.

(vi) The sufficient statistic in (70) is twice differentiable and $\dim(\mathbf{T}_l) \geq 2$, $\forall l$.

(vii) The mixing function $\mathbf{h}$ is a $\mathcal{D}^2$-diffeomorphisms.

(viii) The feature extractor $\mathbf{f}$ in (73) is a $\mathcal{D}^2$-diffeomorphism.

(vi') $\dim(\mathbf{T}_l) = 1$ and $\mathbf{T}_l$ is non-monotonic $\forall l$.

(vii') The mixing function $\mathbf{h}$ is a $\mathcal{C}^1$-diffeomorphism.

(viii') The feature extractor $\mathbf{f}$ in (73) is a $\mathcal{C}^1$-diffeomorphism, and the nonlinearities $\mathbf{H}_l$ have a unique extremum.

(ix) Only $n \leq d$ components of the latent variable are modulated, and its density has the form (72).

(x) The feature extractor $\mathbf{f}$ has the form $\mathbf{f}(\mathbf{x}) = (\mathbf{f}_1(\mathbf{x}), \mathbf{f}_2(\mathbf{x}))$ where $\mathbf{f}_1(\mathbf{x}) \in \mathbb{R}^n$, and the auxiliary feature extractor $\mathbf{g}$ has the form $\mathbf{g}(\mathbf{y}) = (\mathbf{g}_1(\mathbf{y}), \mathbf{g}_2)$ where $\mathbf{g}_1(\mathbf{y}) \in \mathbb{R}^n$ and $\mathbf{g}_2$ is constant.

## D.2   Lemmas

We rely on the following Lemmas from Khemakhem et al. (2020), which we state below in the interest of completeness.

**Lemma 10.** *Consider an exponential family distribution with $k \geq 2$ components. Then the components of the sufficient statistic $\mathbf{T}$ are linearly independent.*

**Lemma 11.** *Consider a* strongly exponential *family distribution such that its sufficient statistic $\mathbf{T}$ is differentiable almost surely. Then $T_i' \neq 0$ almost everywhere on $\mathbb{R}$ for all $1 \leq i \leq k$.*

**Lemma 12.** *Consider a strongly exponential distribution of size $k \geq 2$ with sufficient statistic $\mathbf{T}(x) = (T_1(x), \ldots, T_k(x))$. Further assume that $\mathbf{T}$ is differentiable almost everywhere. Then there exist $k$ distinct values $x_1$ to $x_k$ such that $(\mathbf{T}'(x_1), \ldots, \mathbf{T}'(x_k))$ are linearly independent in $\mathbb{R}^k$.*

**Lemma 13.** *Consider a strongly exponential distribution of size $k \geq 2$ with sufficient statistic $\mathbf{T}$. Further assume that $\mathbf{T}$ is twice differentiable almost everywhere. Then*

$$\dim\left(\text{span}\left((T_i'(x), T_i''(x))^T, 1 \leq i \leq k\right)\right) \geq 2 \tag{74}$$

*almost everywhere on $\mathbb{R}$.*

**Lemma 14.** *Consider $n$ strongly exponential distributions of size $k \geq 2$ with respective sufficient statistics $\mathbf{T}_j = (T_{j,1}, \ldots T_{j,k})$, $1 \leq j \leq n$. Further consider that the sufficient statistics are twice differentiable. Define the vectors $\mathbf{e}^{(j,i)} \in \mathbb{R}^{2n}$, such that $\mathbf{e}^{(j,i)} = (0, \ldots, 0, T_{j,i}', T_{j,i}'', 0, \ldots, 0)$, where the non-zero entries are at indices $(2j, 2j+1)$. Let $\mathbf{x} := (x_1, \ldots, x_n) \in \mathbb{R}^n$. Then the matrix $\bar{\mathbf{e}}(\mathbf{x}) := (\mathbf{e}^{(1,1)}(x_1), \ldots, \mathbf{e}^{(1,k)}(x_1), \ldots \mathbf{e}^{(n,1)}(x_n), \ldots, \mathbf{e}^{(n,k)}(x_n))$ of size $(2n \times nk)$ has rank $2n$ almost everywhere on $\mathbb{R}^n$.*

## D.3   Proofs

As mentioned above, we decompose Theorem 4 into two smaller results, summarized in what follows by Theorems 4a and 4b.

**Theorem 4a.** *Assume assumptions (i)-(v) hold. Then, after convergence of our model $p_{\boldsymbol{\theta}}(\mathbf{x}|\mathbf{y})$ to the true density $p(\mathbf{x}|\mathbf{y})$, we can recover the latent variables up to an invertible linear transformation and point-wise nonlinearities, i.e.*

$$\mathbf{H}(\mathbf{f}(\mathbf{x})) = \mathbf{A}\mathbf{T}(\mathbf{z}) + \mathbf{b} \tag{75}$$

*where $\mathbf{A}$ is an invertible matrix.*

*Proof.* We fit our density model (73) to the conditional density $p(\mathbf{x}|\mathbf{y})$, setting the dimension of the feature extractor $\mathbf{f}$ to be equal to $d$, and the dimensions of the output nonlinearities $\mathbf{H}_l$ chosen from $(\dim(\mathbf{T}_1), \ldots, \dim(\mathbf{T}_d))$, as per assumption (v):

$$Z(\mathbf{y})^{-1} \exp \mathbf{H}(\mathbf{f}(\mathbf{x}))^T \mathbf{g}(\mathbf{y}) = p(\mathbf{x}|\mathbf{y}) \tag{76}$$

by doing the change of variable $\mathbf{x} = \mathbf{h}(\mathbf{z})$, taking the log on both sides, we get:

$$-\log Z(\mathbf{y}) + \mathbf{H}(\mathbf{f}(\mathbf{x}))^T \mathbf{g}(\mathbf{y}) = \log p(\mathbf{z}|\mathbf{y}) - \log|\det \mathbf{J}_{\mathbf{h}^{-1}}(\mathbf{x})| \tag{77}$$

$$= \log \mu(\mathbf{h}^{-1}(\mathbf{x})) + \mathbf{T}(\mathbf{z})^T \boldsymbol{\lambda}(\mathbf{y}) - \Gamma(\mathbf{y}) - \log|\det \mathbf{J}_{\mathbf{h}^{-1}}(\mathbf{x})| \tag{78}$$

Let $\mathbf{y}_0, \ldots, \mathbf{y}_k$ be the points provided by assumption (iv) of the theorem, where $k = \sum_i k_i$, and $k_i = \dim(\mathbf{T}_i)$. Define $\overline{\boldsymbol{\lambda}}(\mathbf{y}) = \boldsymbol{\lambda}(\mathbf{y}) - \boldsymbol{\lambda}(\mathbf{y}_0)$, $\overline{\Gamma}(\mathbf{y}) = \Gamma(\mathbf{y}) - \Gamma(\mathbf{y}_0)$, $\overline{\mathbf{g}}(\mathbf{y}) = \mathbf{g}(\mathbf{y}) - \mathbf{g}(\mathbf{y}_0)$ and $\overline{Z}(\mathbf{y}) = \log Z(\mathbf{y}) - \log Z(\mathbf{y}_0)$. We plug each of those $\mathbf{y}_l$ in (78) to obtain $k + 1$ such equations. We subtract the first equation for $\mathbf{y}_0$ from the remaining $k$ equations to get for $l = 1, \ldots, k$:

$$-\overline{Z}(\mathbf{y}_l) + \mathbf{H}(\mathbf{f}(\mathbf{x}))^T \overline{\mathbf{g}}(\mathbf{y}_l) = \mathbf{T}(\mathbf{z})^T \overline{\boldsymbol{\lambda}}(\mathbf{y}_l) - \overline{\Gamma}(\mathbf{y}_l) \tag{79}$$

The **crucial point** here is that the non factorial terms $\mu(\mathbf{g}(\mathbf{x}))$ and $\tilde{\mu}(\tilde{\mathbf{g}}(\mathbf{x}))$ cancel out when we take these differences. This is what allows us to generalize the identifiability results of nonlinear ICA to the context of IMCA.

Let $\mathbf{L}$ bet the matrix defined in assumption (iv), and $\tilde{\mathbf{L}} := (\ldots, \overline{\mathbf{g}}(\mathbf{y}_l), \ldots)$. Define $\mathbf{b} = (\ldots, \overline{Z}(\mathbf{y}_l) - \overline{\Gamma}(\mathbf{y}_l), \ldots)$. Expressing (79) for all points $\mathbf{y}_l$ in matrix form, we get:

$$\tilde{\mathbf{L}}^T \mathbf{H}(\mathbf{f}(\mathbf{x})) = \mathbf{L}^T \mathbf{T}(\mathbf{z}) + \mathbf{b} \tag{80}$$

By assumption (iv), $\mathbf{L}$ is invertible, and thus we can write

$$\mathbf{T}(\mathbf{z}) = \mathbf{A}\mathbf{H}(\mathbf{f}(\mathbf{x})) + \mathbf{c} \tag{81}$$

where $\mathbf{c} = \mathbf{L}^{-T}\mathbf{b}$ and $\mathbf{A} = \mathbf{L}^{-T}\tilde{\mathbf{L}}^T$.

To prove that $\mathbf{A}$ is invertible, we first take the gradient of equation (81) with respect to $\mathbf{z}$. The Jacobian $\mathbf{J_T}$ of $\mathbf{T}$ is a matrix of size $k \times d$. Its columns are independent because each $\mathbf{T}_i$ is only a function of $z_i$, and thus the non-zero entries of each column are in different rows. This means that its rank is $d$ (since $k = \sum_{i=1}^{d} k_i \geq d$). This is not enough to prove that $\mathbf{A}$ is invertible though. For that, we consider the functions $\mathbf{T}_i$ for which $k_i > 1$: for each of these functions, using Lemma 12, there exists points $z_i^{(1)}, \ldots, z_i^{(k_i)}$ such that $(\mathbf{T}_i'(z_i^{(1)}), \ldots, \mathbf{T}_i'(z_i^{(k_i)}))$ are independent. Collate these point into $k_{\max} := \max_i k_i$ vectors $\mathbf{z}^{(j)} := (z_1^{(j)}, \ldots z_d^{(j)})$, where for each $i$, $z_i^{(j)} = z_i^{(1)}$ if $j > k_i$, and $z_i^{(1)}$ is a point such that $T_i(z_i^{(1)}) \neq 0$ if $k_i = 1$. We plug these vectors into equation (81) after differentiating it, and collate the $dk_{\max}$ equations in vector form:

$$\mathbf{M} = \mathbf{A}\tilde{\mathbf{M}} \tag{82}$$

where $\mathbf{M} := (\ldots, \mathbf{J_T}(\mathbf{z}^{(j)}), \ldots)$ and $\tilde{\mathbf{M}} := (\ldots, \mathbf{J}_{\mathbf{H}\circ\mathbf{f}\circ\mathbf{h}}(\mathbf{z}^{(j)}), \ldots)$. Now the matrix $\mathbf{M}$ is of size $k \times dk_{\max}$, and it has exactly $k$ independent columns by definition of the points $\mathbf{z}^{(j)}$. This means that $\mathbf{M}$ is of rank $k$, which in turn implies that $\operatorname{rank}(\mathbf{A}) \geq k$. Since $\mathbf{A}$ is a $k \times k$ matrix, we conclude that $\mathbf{A}$ is invertible. $\qquad\square$

The theorem above shows a first step in identifiability which holds up to a linear transformation. This is similar to Hyvärinen et al. (2019), but here we allow for dependencies between components. We can further sharpen the result, in line with Khemakhem et al. (2020) even in this non-independent case as follows:

**Theorem 4b.** *Assume assumptions (i)-(v) hold. Further assume that either assumptions (vi)-(viii) or assumptions (vi')-(viii') hold. Then equation (75) can be reduced to the component level, i.e. for each $i \in [\![1, d]\!]$:*

$$\mathbf{H}_i(f_i(\mathbf{x})) = A_i \mathbf{T}_{\gamma(i)}(z_{\gamma(i)}) + \mathbf{b}_i \tag{83}$$

*where $\gamma$ is a permutation of $[\![1, d]\!]$ such that $\dim(\mathbf{H}_i) = \dim(\mathbf{T}_{\gamma(i)})$ and $A_i$ a square invertible matrix.*

*Proof.* We prove this theorem separately for both sets of assumptions.

**Multi-dimensional sufficient statistics: assumptions (vi)-(viii)** We suppose that $k_i \geq 2, \forall i$. The assumptions of Theorem 4a hold, and so we have

$$\mathbf{H}(\mathbf{f}(\mathbf{h}(\mathbf{z}))) = \mathbf{A}\mathbf{T}(\mathbf{z}) + \mathbf{c} \tag{84}$$

for an invertible $\mathbf{A} \in \mathbb{R}^{k \times k}$. We will index $\mathbf{A}$ by four indices $(i, l, a, b)$, where $1 \leq i \leq d, 1 \leq l \leq k_i$ refer to the rows and $1 \leq a \leq d, 1 \leq b \leq k_a$ to the columns.

Let $\mathbf{y} = \mathbf{f} \circ \mathbf{h}(\mathbf{z})$. Since both $\mathbf{f}$ and $\mathbf{h}$ are $\mathcal{D}^2$-diffeomorphisms (assumptions (vii), (viii)), we can invert this relation and write $\mathbf{z} = \mathbf{v}(\mathbf{y})$. We introduce the notations $v_i^s(\mathbf{y}) := \frac{\partial v_i}{\partial y_s}(\mathbf{y})$, $v_i^{st}(\mathbf{y}) := \frac{\partial^2 v_i}{\partial y_s \partial y_t}(\mathbf{y})$, $T'_{a,b}(z) = \frac{\mathrm{d} T_{a,b}}{\mathrm{d} z}(z), T''_{a,b}(z) = \frac{\mathrm{d}^2 T_{a,b}}{\mathrm{d} z}(z)$ and $H'_{a,b}(y) = \frac{\mathrm{d} H_{a,b}}{\mathrm{d} y}(y)$. Each line of equation (84) can be written as:

$$H_{i,l}(y_i) = \sum_{a=1}^{d} \sum_{b=1}^{k_i} A_{i,l,a,b} T_{a,b}(v_a(\mathbf{y})) + c_{a,b} \tag{85}$$

for $i \leq d$, $l \leq k_i$. The first step is to show that $v_i(\mathbf{y})$ is a function of only one $y_{j_i}$, for all $i \leq d$. by differentiating (85) with respect to $y_s$, $s \leq d$:

$$\delta_{is} H'_{i,l}(y_i) = \sum_{a=1}^{d} \sum_{b=1}^{k_i} A_{i,l,a,b} T'_{a,b}(v_a(\mathbf{y})) v_a^s(\mathbf{y}) \tag{86}$$

and by differentiating (86) with respect to $y_t, s < t \leq d$:

$$0 = \sum_{a,b} A_{i,l,a,b} \left( T'_{a,b}(v_a(\mathbf{y})) v_a^{s,t}(\mathbf{y}) + T''_{a,b}(v_a(\mathbf{y})) v_a^s(\mathbf{y}) v_a^t(\mathbf{y}) \right) \tag{87}$$

This equation is valid for all pairs $(s,t), t > s$. Define $\mathbf{B}_a(\mathbf{y}) := \left( v_a^{1,2}(\mathbf{y}), \ldots, v_a^{d-1,d}(\mathbf{y}) \right) \in \mathbb{R}^{\frac{d(d-1)}{2}}$, $\mathbf{C}_a(\mathbf{y}) := \left( v_a^1(\mathbf{y}) v_a^2(\mathbf{y}), \ldots, v_a^{d-1}(\mathbf{y}) v_a^d(\mathbf{y}) \right) \in \mathbb{R}^{\frac{d(d-1)}{2}}$, $\mathbf{M}(\mathbf{y}) := (\mathbf{B}_1(\mathbf{y}), \mathbf{C}_1(\mathbf{y}), \ldots, \mathbf{B}_d(\mathbf{y}), \mathbf{C}_d(\mathbf{y}))$, $\mathbf{e}^{(a,b)} := (0, \ldots, 0, T'_{a,b}, T''_{a,b}, 0, \ldots, 0) \in \mathbb{R}^{2d}$, such that the non-zero entries are at indices $(2a, 2a + 1)$ and $\overline{\mathbf{e}}(\mathbf{y}) := (\mathbf{e}^{(1,1)}(y_1), \ldots, \mathbf{e}^{(1,k_1)}(y_1), \ldots, \mathbf{e}^{(d,1)}(y_d), \ldots, \mathbf{e}^{(d,k_d)}(y_d)) \in \mathbb{R}^{2d \times k}$. Then by grouping equation (87) for all valid pairs $(s,t)$ and pairs $(i,l)$ and writing it in matrix form, we get:

$$\mathbf{M}(\mathbf{y})\overline{\mathbf{e}}(\mathbf{y})\mathbf{A} = 0 \tag{88}$$

Now by Lemma 14, we know that $\overline{\mathbf{e}}(\mathbf{y})$ has rank $2d$ almost surely on $\mathcal{Z}$. Since $\mathbf{A}$ is invertible, it is full rank, and thus $\mathrm{rank}(\overline{\mathbf{e}}(\mathbf{y})\mathbf{A}) = 2d$ almost surely on $\mathcal{Z}$. It suffices then to multiply by its pseudo-inverse from the right to get

$$\mathbf{M}(\mathbf{y}) = 0 \tag{89}$$

In particular, $C_a(\mathbf{y}) = 0$ for all $1 \leq a \leq d$. This means that the Jacobian of $\mathbf{v}$ at each $\mathbf{y}$ has at most one non-zero entry in each row. By invertibility and continuity of $J_{\mathbf{v}}$, we deduce that the location of the non-zero entries are fixed and do not change as a function of $\mathbf{y}$. We deduce that there exists a permutation $\sigma$ of $[\![1, d]\!]$ such that each of the $v_i(\mathbf{y}) = v_i(y_{\sigma(i)})$, and the same would apply to $\mathbf{v}^{-1}$. Without any loss of generality, we assume that $\sigma$ is the identity.

Now let $\overline{\mathbf{H}}(\mathbf{z}) = \mathbf{H} \circ \mathbf{v}^{-1}(\mathbf{y}) - \mathbf{c}$. This function is a pointwise function because $\mathbf{H}$ and $\mathbf{v}^{-1}$ are such functions. Plugging this back into equation (84) yields:

$$\overline{\mathbf{H}}(\mathbf{z}) = \mathbf{A}\mathbf{T}(\mathbf{z}) \tag{90}$$

The last equation is valid for every component:

$$\overline{H}_{i,l}(z_i) = \sum_{a,b} A_{i,l,a,b} T_{a,b}(z_a) \tag{91}$$

By differentiating both sides with respect to $z_s$ where $s \neq i$ we get

$$0 = \sum_{b} A_{i,l,s,b} T'_{s,b}(z_s) \tag{92}$$

By Lemma 10, we get $A_{i,l,s,b} = 0$ for all $1 \leq b \leq k$. Since (92) is valid for all $l$ and all $s \neq i$, we deduce that the matrix $\mathbf{A}$ has a block diagonal form:

$$\mathbf{A} = \begin{pmatrix} \mathbf{A}_1 & & \\ & \ddots & \\ & & \mathbf{A}_n \end{pmatrix} \tag{93}$$

which achieves the proof.

**One-dimensional sufficient statistics: assumptions (vi')-(viii')**  We now suppose that $k_i = 1$, $\forall i$. The proof of Khemakhem et al. (2020, Theorem 3) can be used here, where we define $\mathbf{v} = (\mathbf{f} \circ \mathbf{h})^{-1}$ and $h_{i,a} = D_{i,a} H_a(y_a) - D_{i,a} c_a$, where $\mathbf{D} = \mathbf{A}^{-1}$. We can then rewrite equation (85) for every component as:

$$T_i(v_i(\mathbf{z})) = \sum_{a=1}^{d} h_{i,a}(z_a) \tag{94}$$

which is the same as equation (45) of Khemakhem et al. (2020). All the assumptions required to prove their theorem are met in our case, and the rest of their proof would simply apply here to prove that $\mathbf{A}$ is a permutation matrix. $\qquad\square$

In practice, it is a natural desire to have the feature extractor reduce the dimension of the data, as it is usually very large. This has been achieved in nonlinear ICA before (Khemakhem et al., 2020; Hyvärinen and Morioka, 2016). It turns out that we can also incorporate dimensionality reduction in IMCA and its estimation by ICE-BeeM, under some assumptions.

**Theorem 6.** *Assume either of the following hold:*

- *Assumptions (i)-(x).*

- *Assumptions (i)- (v), (vi')- (viii'), and (ix)- (x).*

*Then $\mathbf{f}_1$ recovers only the modulated latent components as per Theorem 4b.*

*Proof.* The proof of Theorem 4a in this case is unchanged. Simply, we update the total dimension of matrix $\mathbf{L}$ here to $k = \sum_{i=1}^{n} \dim(\mathbf{T}_i)$. when we evaluate equation (78) on these points $\mathbf{y}_0, \ldots, \mathbf{y}_k$, the constant term $\mathbf{g}_2$ and the non-modulated components cancel out, and we are left with the equation

$$\tilde{\mathbf{L}}^T \mathbf{H}_{1:n}(\mathbf{f}_1(\mathbf{x})) = \mathbf{L}^T \mathbf{T}_{1:n}(\mathbf{z}) + \mathbf{b} \tag{95}$$

We then use similar arguments to the proof of Theorem 4a to conclude that

$$\mathbf{H}_{1:n}(\mathbf{f}(\mathbf{x})) = \mathbf{A} \mathbf{T}_{1:n}(\mathbf{z}) + \mathbf{c} \tag{96}$$

where $\mathbf{A} \in \mathbb{R}^n$ a square invertible matrix. At this point, we can make the same conclusion as Theorem 4a, while reducing the dimension of the latent space.

We now explain how we can extend Theorem 4b to the lower dimensional latent space case. Note that we still assume that $\mathbf{f} = (\mathbf{f}_1, \mathbf{f}_2)$ is a diffeomorphism per assumptions (viii) and (viii'). We can then still define $\mathbf{v} = (\mathbf{f} \circ \mathbf{h})^{-1}$.

We consider now two cases like in the proof of Theorem 4b.

**One-dimensional sufficient statistics**  Let $\mathbf{D} = \mathbf{A}^{-1}$ and $h_{i,a} = D_{i,a} H_a(y_a) - D_{i,a} c_a$. We can still write equation (96) like equation (94) as

$$T_i(v_i(\mathbf{z})) = \sum_{a=1}^{n} h_{i,a}(z_a) \tag{97}$$

for all $i \leq n$. The assumptions required for the proof are still met, despite reducing the dimension from $d$ to $n$. This interesting fact is also used for the proof of Theorem 2b as well, which achieves this part of the proof.

**Multi-dimensional sufficient statistics**  We rewrite equation (96)

$$H_{i,l}(y_i) = \sum_{a=1}^{n} \sum_{b=1}^{k_i} A_{i,l,a,b} T_{a,b}(v_a(\mathbf{y})) + c_{a,b} \tag{98}$$

for all $i \leq n, l \leq k_i$. We proceed similarly to the proof of Theorem 4b, replacing all mentions of $d$ by $n$ and keeping all differentiations to indices $t, s \leq n$, up to equation (89), after which we can conclude that $v_i^s v_i^t = 0$ for all $i \leq n$, and all $s, t \leq n$. This is not enough to conclude that each of the $v_i$ is only function of one $y_{j_i}$.

For that, we go back to equation (98) and differentiate it with respect to $y_s$, $s > n$:

$$0 = \sum_{a=1}^{d} \sum_{b=1}^{k_i} A_{i,l,a,b} T'_{a,b}(v_a(\mathbf{y})) v_a^s(\mathbf{y}) \tag{99}$$

which is valid for all $i \leq n, l \leq k_i$. Since $\mathbf{A}$ is invertible, we can conclude that $T'_{a,b}(v_a(\mathbf{y})) v_a^s(\mathbf{y}) = 0$ for all $a \leq n$ and $s > n$. Since we only consider strongly exponential distributions (assumption (iii)), and using proposition 11, we conclude that $T'_{a,b}(v_a(\mathbf{y})) \neq 0$ almost everywhere, and that $v_a^s(\mathbf{y}) = 0$, for all $s > n$. This, in addition to the fact that $v_i^s v_i^t = 0$ for all $i \leq n$, and all $s, t \leq n$ allows us to conclude that the first $n$ components of $\mathbf{v}$ are each only a function of one different $y_j$ because $\mathbf{v}$ is a diffeomorphism and its Jacobian is continuous. Finally, we can use this fact to deduce that $\mathbf{A}$ is a block permutation matrix, which achieves the proof. $\square$

# E   Independently modulated component analysis

As mentioned in section 3, linear latent variable models (Hyvärinen and Oja, 2000) and more recently nonlinear latent variable models may be identifiable provided some additional auxiliary variables (Khemakhem et al., 2020; Hyvärinen et al., 2019). The purpose of this auxiliary variable serves to introduce additional constraints over the distribution over latent variables, which are typically required to be conditionally independent given the auxiliary variable. This avenue of research has thus formalized the trade-off between expressivity of the mapping between latents to observations (from linear to nonlinear) and distributional assumptions over latent variables (from independent to conditionally independent given auxiliary variables).

We would like to relax the assumption of independence while maintaining identifiability, resulting in the framework of Independently Modulated Component Analysis (IMCA). In this section of the Appendix, we will give a detailed analysis of the IMCA model independently of any estimation method, drawing parallels to the identifiability results of the nonlinear ICA model presented in Khemakhem et al. (2020).

## E.1   Definition of the generative model

Assume we observe a random variable $\mathbf{x} \in \mathbb{R}^d$ as a result of a nonlinear transformation $\mathbf{h}$ of a latent variable $\mathbf{z} \in \mathbb{R}^d$ whose distribution is conditioned on an auxiliary variable $\mathbf{y}$ that is also observed:

$$\begin{aligned} \mathbf{z} &\sim p(\mathbf{z}|\mathbf{y}) \\ \mathbf{x} &= \mathbf{h}(\mathbf{z}) \end{aligned} \tag{100}$$

The main modelisation assumption we make is on the latent variable distribution, given by the following definition, where $\mathbf{u}$ is a dummy variable.

**Definition 5** (Exponentially factorial distributions). *We say that a multivariate exponential family distribution is **exponentially factorial** if its density $p(\mathbf{u})$ has the form*

$$p(\mathbf{y}) = \mu(\mathbf{y}) \prod_i e^{\mathbf{T}_i(y_i)^T \boldsymbol{\lambda}_i - \Gamma(\boldsymbol{\lambda})}$$

We assume that the latent variable in the IMCA model has a conditional exponentially factorial distribution, where the parameters of the exponential family are a function of the auxiliary variable $\mathbf{y}$:

$$p(\mathbf{z}|\mathbf{y}) = \mu(\mathbf{z}) e^{\sum_i \mathbf{T}_i(z_i)^T \boldsymbol{\lambda}_i(\mathbf{y}) - \Gamma(\mathbf{y})} \tag{101}$$

Equations (100) and (101) together define the nonparametric IMCA model with parameters $(\mathbf{h}, \mathbf{T}, \boldsymbol{\lambda}, \mu)$. Most importantly, we allow for an arbitrary base measure $\mu(\mathbf{z})$, *i.e.* the components of the latent variable must no longer be independent, as $\mu$ doesn't necessarily factorize across dimensions. The crucial assumption is that the components of the latent variables are independently modulated given the auxiliary variable $\mathbf{y}$, and that through the term $\exp(\sum_i \mathbf{T}_i(z_i)^T \boldsymbol{\lambda}_i(\mathbf{y}))$.

## E.2 Identifiability

The concept of identifiability is core to this work. As such, it is important to understand the different views one can have of this concept.

According to the conventional definition, a probabilistic model $\mathcal{P} = \{\mathcal{P}_{\boldsymbol{\theta}} : \boldsymbol{\theta} \in \Theta\}$ is identifiable *iif* the mapping $\boldsymbol{\theta} \mapsto \mathcal{P}_{\boldsymbol{\theta}}$ is bijective, *i.e.* $\mathcal{P}_{\boldsymbol{\theta}_1} = \mathcal{P}_{\boldsymbol{\theta}_2} \implies \boldsymbol{\theta}_1 = \boldsymbol{\theta}_2$. However, this definition is very restrictive and impractical.

Often, the identifiability form we can prove for a model is equality of the parameters *up to some indeterminacies*. This can be understood as an equivalence relation between parameters. Identifiability in this context implies that the equivalence class of the ground truth parameter can be uniquely recovered from observations. This is relevant only if the definition of the equivalence class is sufficiently narrow and specific to be able to make meaningful conclusions. One example of such equivalence relations can be found in linear ICA: the mixing matrix is uniquely recovered up to a scaled permutation. The permutation is irrelevant, and the scaling is circumvented by whitening the data. This is a good example of an equivalence class that doesn't restrict the practical utility of the ICA model.

An example of indeterminacy which is relevant to us here can be found in variational inference of latent variable models: two parameters are equivalent if they map to the same *inference* distribution (Khemakhem et al., 2020). This is the definition we will be using in this work. We will say that a generative model is identifiable if we can uniquely recover the latent variables, as given by the following definition.

**Definition 6.** *Consider two different sets of parameters* $(\mathbf{h}, \mathbf{T}, \boldsymbol{\lambda}, \mu)$ *and* $(\tilde{\mathbf{h}}, \tilde{\mathbf{T}}, \tilde{\boldsymbol{\lambda}}, \tilde{\mu})$*, defining two densities* $p$ *and* $p'$*. We say that the IMCA model is strongly identifiable if*

$$p(\mathbf{x}|\mathbf{y}) = \tilde{p}(\mathbf{x}|\mathbf{y}) \implies \forall i, \mathbf{T}_i(z_i) = \mathbf{A}_i \tilde{\mathbf{T}}_{\gamma(i)}(\tilde{z}_{\gamma(i)}) + \mathbf{b}_i \tag{102}$$

*where* $\gamma$ *is a permutation,* $\mathbf{A}_i$ *is an invertible matrix, and* $\mathbf{b}_i$ *a vector,* $\forall i \in [\![1, d]\!]$*.*
*We say that it is weakly identifiable if*

$$p(\mathbf{x}|\mathbf{y}) = \tilde{p}(\mathbf{x}|\mathbf{y}) \implies \mathbf{T}(\mathbf{z}) = \mathbf{A}\tilde{\mathbf{T}}(\tilde{\mathbf{z}}) + \mathbf{b} \tag{103}$$

*where* $\mathbf{A}$ *is an invertible matrix, and* $\mathbf{b}$ *a vector.*

## E.3 Theoretical analysis

In this section, we develop the theory of IMCA. We will give sufficient conditions that guarantee a strong identifiability of the latent components, and discuss a degenerate case where we only obtain a weaker form of identifiability.

### E.3.1 Definitions

We will first define some sets of distributions which are subsets of the exponential family distribution. We will use $u$ as a dummy variable, and introduce the definitions for the unconditional case. Note that all these definitions apply to the conditional case, when the parameters of the exponential family are a function of an auxiliary variable $\mathbf{y}$. For completeness, we restate here Definition 4.

**Definition 7** (Strongly exponential distributions). *We say that a univariate exponential family distribution with density* $p(u) = \mu(u)e^{\mathbf{T}(u)^T \boldsymbol{\theta} - \Gamma(\boldsymbol{\theta})}$ *is **strongly exponential** if for any subset* $\mathcal{U}$ *of* $\mathbb{R}$ *the following is true:*

$$\left(\exists \boldsymbol{\theta} \in \mathbb{R}^k \,|\, \forall u \in \mathcal{U}, \langle \mathbf{T}(u), \boldsymbol{\theta} \rangle = const\right) \implies (\Lambda(\mathcal{U}) = 0 \text{ or } \boldsymbol{\theta} = 0) \tag{104}$$

*where* $\Lambda$ *is the Lebesgue measure.*

*We say that that a multivariate distribution is strongly exponential if all its univariate marginals are.*

In other words, the density of a strongly exponential distribution has almost surely the exponential component in its expression and can only be reduced to the base measure on a set of measure zero. This definition is very general, and is satisfied by all the usual exponential family distributions like the Gaussian, Laplace, Pareto, Chi-squared, Gamma, Beta, *etc.* We will only prove identifiability

results for strongly exponential families. The non-strongly exponential case will be explored in future work.

There is a certain class of exponential families for which we can only prove a weak form of identifiability. Loosely speaking, this is because this class doesn't constrain the latent space enough.

**Definition 8** (Quasi-location exponential distributions). *We say that a univariate exponential family distribution with density $p(u) = \mu(u)e^{\mathbf{T}(u)^T \boldsymbol{\theta} - \Gamma(\boldsymbol{\theta})}$ is in the **quasi-location** family if:*

  *(i)* $\dim(\mathbf{T}) = 1$

  *(ii)* $\mathbf{T}$ *is monotonic (either non-decreasing or non-increasing)*

*We say that that a multivariate distribution is quasi-location exponential if all its univariate marginals are.*

As a simple illustration, the Gaussian family with fixed variance is a quasi-location family, but with fixed mean it is not. This is because in the first case, the sufficient statistic is $T(u) = u$ which is a monotonic scalar function, while in the second case it is $T(u) = u^2$, a non-monotonic scalar function.

### E.3.2 Identifiability of the general case

As mentioned in section 3, the IMCA model described by equations (100) and (101) generalizes previous nonlinear ICA models by relaxing the independence assumption required for the latent variables. We propose here to extend the identifiability theory of nonlinear ICA developed in Hyvärinen et al. (2019); Khemakhem et al. (2020) to this new framework.

We start by providing a weaker form of identifiability guarantee that applies to the general case, including quasi-location families.

**Theorem 7.** *Assume the following:*

  *(I) The observed data follows the exponential IMCA model of equations (100)-(101).*

  *(II) The mixing function $\mathbf{h} : \mathbb{R}^d \to \mathbb{R}^d$ is invertible.*

  *(III) The conditional latent distribution $p(\mathbf{z}|\mathbf{y})$ is strongly exponential (definition 7), and its sufficient statistic is differentiable.*

  *(IV) There exist $k + 1$ distinct points $\mathbf{y}^0, \ldots, \mathbf{y}^k$ such that the matrix*
$$\mathbf{L} = (\boldsymbol{\lambda}(\mathbf{y}_1) - \boldsymbol{\lambda}(\mathbf{y}_0), \ldots, \boldsymbol{\lambda}(\mathbf{y}_k) - \boldsymbol{\lambda}(\mathbf{y}_0))$$
  *of size $k \times k$ is invertible, where $k = \sum_{i=1}^d \dim(\mathbf{T}_i)$.*

*Then the IMCA model is weakly identifiable.*

This theorem extends the basic identifiability result of Khemakhem et al. (2020, Theorem 1). It is fundamental as it proves a general identifiability results without the restriction of having independent latent variables. This was previously not considered to be possible and could only be demonstrated in very specific circumstances and under very restrictive additional assumptions (*e.g.*, Monti and Hyvärinen (2018) require *both* non-negativity and orthonormality of a mixing matrix in the *linear* case). In the nonlinear case, to prove Theorem 7, we still require that the latent variables are only dependent through the base measure, while still being independently modulated through the auxiliary variable $\mathbf{y}$. This (and the necessity of having an auxiliary variable) is the price to pay for obtaining identifiability in a nonlinear setting.

### E.3.3 Identifiability of the non quasi-location family

The identifiability result of Theorem 7 is weak because of the presence of the linear transformation $\mathbf{A}$ in equation (103). It turns out that by excluding the quasi-location family (definition 8), we can remove this matrix and achieve a stronger form of identifiability. The main technical result of this paper is the following.

**Theorem 8.** *Assume that the assumptions of Theorem 7 hold. Further assume one of the two following sets of assumptions:*

*(V)* *The sufficient statistic in* (101) *is twice differentiable and* $\dim(\mathbf{T}_l) \geq 2$, $\forall l$.

*(VI)* *The mixing function* $\mathbf{h}$ *is a* $\mathcal{D}^2$*-diffeomorphism*[6].

*or*

*(V)'* $\dim(\mathbf{T}_l) = 1$ *and* $\mathbf{T}_l$ *is non-monotonic* $\forall l$.

*(VI)'* *The mixing function* $\mathbf{h}$ *is a* $\mathcal{C}^1$*-diffeomorphism*[7].

*Then the IMCA model is strongly identifiable.*

This form of identifiability mirrors the strongest results proven in the nonlinear ICA (Khemakhem et al., 2020, Theorems 2,3), without requiring that the latent components be independent. As far as we know, this is the first proof of the kind for nonlinear representation learning. We further note that this theorem generalizes even existing identifiability theory of the linear case. The mixed case where we have both cases where some sufficient statistics are of dimension greater than 2 and some are univariate and non-monotonic will be studied in future work.

### E.4 Estimation of IMCA by self-supervised learning

A recent development in nonlinear ICA is given by Hyvärinen et al. (2019) where the authors assume they observe data $\mathbf{x} = \mathbf{h}(\mathbf{z})$ following a noiseless conditional nonlinear ICA model $p(\mathbf{z}|\mathbf{y}) = \prod_i p_i(z_i|\mathbf{y})$ For estimation, they rely on a self-supervised binary discrimination task based on randomization to learn the unmixing function. More specifically, from a dataset of observations and auxiliary variables pairs $\mathcal{D} = \{\mathbf{x}^{(i)}, \mathbf{y}^{(i)}\}$, they construct a randomized dataset $\mathcal{D}^* = \{\mathbf{x}^{(i)}, \mathbf{y}^*\}$ where $\mathbf{y}^*$ is randomly drawn from the observed distribution of $\mathbf{y}$. To distinguish between both datasets, a deep logistic regression is used. The last hidden layer of the neural network is a feature extractor whose purpose is to extract the relevant features which will allow to distinguish between the two datasets. Surprisingly, this estimation technique works for IMCA, and is summarized by the following theorem.

**Theorem 9.** *Self-supervised nonlinear ICA estimation algorithms presented in Hyvärinen and Morioka (2016); Hyvärinen et al. (2019) work for the estimation of IMCA.*

### E.5 Proofs

#### E.5.1 Proof of Theorem 7

Consider two different sets of parameters $(\mathbf{h}, \mathbf{T}, \boldsymbol{\lambda}, \mu)$ and $(\tilde{\mathbf{h}}, \tilde{\mathbf{T}}, \tilde{\boldsymbol{\lambda}}, \tilde{\mu})$, defining two conditional latent densities $p(\mathbf{z}|\mathbf{y})$ and $\tilde{p}(\mathbf{z}|\mathbf{y})$. Suppose that the density of the observations arising from these two different models are equal:

$$p(\mathbf{x}|\mathbf{y}) = \tilde{p}(\mathbf{x}|\mathbf{y}) \tag{105}$$

$$\log p(\mathbf{g}(\mathbf{x})|\mathbf{y}) - \log\left|\det \mathbf{J}_{\mathbf{h}}^{-1}(\mathbf{x})\right| = \log p(\tilde{\mathbf{g}}(\mathbf{x})|\mathbf{y}) - \log\left|\det \mathbf{J}_{\tilde{\mathbf{g}}}(\mathbf{x})\right| \tag{106}$$

$$\log \mu(\mathbf{g}(\mathbf{x})) + \mathbf{T}(\mathbf{g}(\mathbf{z}))^T \boldsymbol{\lambda}(\mathbf{y}) - \Gamma(\mathbf{y}) - \log|\det \mathbf{J}_{\mathbf{g}}(\mathbf{x})| =$$
$$\log \tilde{\mu}(\tilde{\mathbf{g}}(\mathbf{x})) + \tilde{\mathbf{T}}(\tilde{\mathbf{g}}(\mathbf{z}))^T \tilde{\boldsymbol{\lambda}}(\mathbf{y}) - \tilde{\Gamma}(\mathbf{y}) - \log|\det \mathbf{J}_{\tilde{\mathbf{g}}}(\mathbf{x})| \tag{107}$$

Let $\mathbf{y}_0, \ldots, \mathbf{y}_k$ be the points provided by assumption (IV) of the theorem for $\mathbf{T}$, where $k = \sum_i k_i$, and $k_i = \dim(\mathbf{T}_i)$. We plug each of those $\mathbf{y}_l$ in (107) to obtain $k + 1$ such equations. Then, we subtract the first equation for $\mathbf{y}_0$ from the remaining $k$ equations to get for $l = 1, \ldots, k$:

$$\mathbf{T}(\mathbf{z})^T(\boldsymbol{\lambda}(\mathbf{y}_l) - \boldsymbol{\lambda}(\mathbf{y}_0)) - G(\mathbf{y}_l) = \tilde{\mathbf{T}}(\mathbf{z})^T(\tilde{\boldsymbol{\lambda}}(\mathbf{y}_l) - \tilde{\boldsymbol{\lambda}}(\mathbf{y}_0)) - \tilde{G}(\mathbf{y}_l) \tag{108}$$

where we grouped terms that are only a function of $\mathbf{y}_l$ in $G$ and $\tilde{G}$.

Most importantly, both base measure terms disappear after taking the differences, which is the key enabler of identifiability in the IMCA framework.

The rest of the proof is similar to the proof of Khemakhem et al. (2020, Theorem 1). The only difference is that we don't restrict the sufficient statistics to have equal dimensions, and so we can't use the proof technique from Khemakhem et al. (2020, Theorem 1) without any modification. We present an alternative technique in the proof of Theorem 4, which we refer too for more details. We then conclude that

$$\mathbf{T}(\mathbf{h}^{-1}(\mathbf{x})) = \mathbf{A}\tilde{\mathbf{T}}(\tilde{\mathbf{h}}^{-1}(\mathbf{x})) + \mathbf{b} \tag{109}$$

which implies that the model is weakly identifiable. $\qquad\square$

### E.5.2   Proof of Theorem 8

The conclusion of Theorem 7 is the same as the conclusion of Khemakhem et al. (2020, Theorem 1). Since we make the same assumptions as Khemakhem et al. (2020, Theorems 2,3), the proof to Theorem 8 is similar to the proof of these theorems, which we refer too for more details. The IMCA model is strongly identifiable under the assumptions of Theorem 8. $\qquad\square$

### E.5.3   Proof of Theorem 9

We will first quickly summarize the method proposed in Hyvärinen et al. (2019), and then show how it works for IMCA.

We consider that we observe data $(\mathbf{x}, \mathbf{y})$ that follows the exponential IMCA model of equations (4)-(5). Following Hyvärinen et al. (2019) we start by constructing new data from the observations $\mathbf{x}$ and $\mathbf{y}$ to obtain two datasets

$$\tilde{\mathbf{x}} = (\mathbf{x}, \mathbf{y}) \tag{110}$$
$$\tilde{\mathbf{x}}^* = (\mathbf{x}, \mathbf{y}^*) \tag{111}$$

where $\mathbf{y}^*$ is a random value from the distribution of $\mathbf{y}$ and independent of $\mathbf{x}$. We then proceed by defining a multinomial classification task, where we consider the set of all $\{\tilde{\mathbf{x}}, \tilde{\mathbf{x}}^*\}$ as data points to be classified, and whether they come from the randomized dataset or not as labels. In particular, we train a deep neural network using multinomial logistic regression to perform this classification task. The last hidden layer of the neural network is a feature extractor denoted $\mathbf{s}(\mathbf{x})$. The purpose of the feature extractor is therefore to extract the relevant features which will allow to distinguish between the true dataset $\tilde{\mathbf{x}}$ and the randomized dataset $\tilde{\mathbf{x}}^*$. The final layer of the network is simply linear, and the regression function takes the form

$$r(\mathbf{x}, \mathbf{y}) = \mathbf{s}(\mathbf{x})^T \mathbf{v}(\mathbf{y}) + \mathbf{a}(x) + \mathbf{b}(u) \tag{112}$$

We state now the main result.

**Theorem 9** (Hyvärinen et al. (2019), adapted). *Assume that the assumptions of Theorem 7, and the assumptions (V)-(VI) of Theorem 8 hold. Further assume that we train a nonlinear logistic regression with universal approximation capability to discriminate between $\tilde{\mathbf{x}}$ in (110) and $\tilde{\mathbf{x}}^*$ in (111) with the regression function in (112), where the feature extractor has dimension $d$.*

*Then in the limit of infinite data, the components $s_i(\mathbf{x})$ of the regression function give the latent components up to pointwise nonlinearities.*

*Proof.* The proof of this theorem is inspired by Hyvärinen et al. (2019). By well known theory, after convergence of logistic regression, the regression function equals the difference of the log-densities of the two classes:

$$\sum_{i=1}^{d} s_i(\mathbf{x})v_i(\mathbf{y}) + a(x) + b(u) = \log p_{\tilde{\mathbf{x}}}(\mathbf{x}, \mathbf{y}) - \log p_{\tilde{\mathbf{x}}^*}(\mathbf{x}, \mathbf{y}^*)$$

$$= \log p(\mathbf{z}, \mathbf{y}) + \log\left|\det \mathbf{J}_{\mathbf{h}}^{-1}(\mathbf{x})\right| - \log p(\mathbf{z})p(\mathbf{y}) - \log\left|\det \mathbf{J}_{\mathbf{h}}^{-1}(\mathbf{x})\right|$$

$$= \log p(\mathbf{z}|\mathbf{y}) - \log p(\mathbf{z})$$

$$= \log \mu(\mathbf{z}) - \log Z(\mathbf{y}) + \sum_{i=1}^{d} \mathbf{T}_i(z_i)^T \boldsymbol{\lambda}_i(\mathbf{y}) - \log p(\mathbf{z})$$

$$\tag{113}$$

where $\mathbf{J_h^{-1}}(\mathbf{x})$ is the Jacobian matrix of $\mathbf{h}^{-1}$ at point $\mathbf{x}$. Let $\mathbf{y}_0, \dots, \mathbf{y}_k$ be the point provided by assumption (iv). We plug each of those $\mathbf{y}_k$ in (113) to obtain $k+1$ such equations. We subtract the first equation for $\mathbf{y}_0$ from the remaining $k$ equations to get for $l = 1, \dots, k$:

$$\sum_{i=1}^{d} s_i(\mathbf{x})(v_i(\mathbf{y}_l) - v_i(\mathbf{y}_0)) + (\mathbf{b}(\mathbf{y}_l) - \mathbf{b}(\mathbf{y}_0)) - \log \frac{Z(\mathbf{y}_l)}{Z(\mathbf{y}_0)} = \sum_{i=1}^{d} \mathbf{T}_i(z_i)^T (\boldsymbol{\lambda}_i(\mathbf{y}_l) - \boldsymbol{\lambda}_i(\mathbf{y}_0)) \quad (114)$$

Interestingly, the term $\log \mu(\mathbf{z})$ cancels out. The rest of the proof is similar to Theorems 4a and 4b. The only minor difference is that the matrix $\mathbf{A}$ will not be square, but it is still full rank, and can be used to prove that $\mathbf{s} \circ \mathbf{h}$ is a point-wise nonlinearity. $\qquad \square$

[Supplementary Material 2 · 1616-supp.pdf]

*Appendix for*

# ICE-BeeM: Identifiable Conditional Energy-Based Deep Models Based on Nonlinear ICA

We divide the Appendix into 5 main sections:

- Section A: we give extensive details on the experimental setup, as well as additional experiments;
- Section B: we discuss the estimation algorithms we used with ICE-BeeM and how they can be extended to the conditional setting;
- Section C: we prove the identifiability of ICE-BeeM and its universal approximation capability;
- Section D: we show how ICE-BeeM estimates IMCA;
- Section E: we provide a thorough theoretical analysis of the IMCA framework and draw parallels to the identifiability results in nonlinear ICA.

## A Experimental protocol

### A.1 Model architecture details

In this section, we describe the neural network architectures used for the experiments of Section 5.1, on the image datasets (MNIST, FashionMNIST, CIFAR10 and CIFAR100). Code to reproduce these experiments can be found in the supplementary material.

We can distinguish three different types of configurations:

1. A series of fully connected layers — denoted *MLP*. This configuration satisfies the assumptions of Section 2.3.
2. A mix of convolutional and fully connected layers — denoted *ConvMLP*. We expect this configuration to work better than an MLP for images.
3. A variant of a RefineNet (Lin et al., 2017), following Song and Ermon (2019), which implements skip connections to help low level information reach the top layers — denoted for simplicity *Unet* (RefineNets are modern variants of U-net architectures). This configuration is very advanced and complicated, and serves to test if identifiable representations can be learnt for modern architectures.

The detailed architectures are in Table [2].

After choosing one of the configurations, we can further chose to reduce the dimensionality of the features ($d_z < d_x$), to use it in conjunction with positive features (condition 3 of Theorem 2) or with augmented features (condition 4 of Theorem 2). This results in the following nomenclature, where we will take as an example a *ConvMLP* network:

- If we reduce the dimension of the latent space ($d_z < d_x$)—for example $d_z = 50$, we denote the configuration by *ConvMLP-50*.
- If we used positive features, we denote the configuration by *ConvMLP-p*.
- If we used augmented features, we denote the configuration by *ConvMLP-a*.
- We can also have a mix of the above, for examples *ConvMLP-50p*.
- We can also have non of the above, in which case we simply write *ConvMLP*—implying that $d_z = d_x$.

We summarize the configurations used for the different experiments of Section 5.1 in Table [3].

For all the experiments, we used the Adam optimizer (Kingma and Ba, 2014) to update the parameters of the networks. We used a learning rate of 0.001, and $(\beta_1, \beta_2) = (0.9, 0.999)$; `amsgrad` was turned off, as well as weight decay. Data was fed to the networks in mini-batches of size 63, and the

Table 2: Architecture detail

| Configuration | Architecture | Comment |
|---|---|---|
| | Input: $d_x = w \times w \times n_c$ | $n_c$: channels, $w$: width/height |
| | | MNIST: $n_c = 1$, $w = 28$ |
| | | FashionMNIST: $n_c = 1$, $w = 28$ |
| | | CIFAR10: $n_c = 3$, $w = 32$ |
| | | CIFAR100: $n_c = 3$, $w = 32$ |
| | Output: $d_z$ | |
| *MLP* | Input: $d_x$ | |
| | FC 512, LeakyReLU(0.1) | |
| | FC 384, LeakyReLU(0.1) | |
| | Dropout(0.1) | |
| | FC 256, LeakyReLU(0.1) | |
| | FC 256, LeakyReLU(0.1) | |
| | FC $d_z$ | |
| *ConvMLP* | Input: $d_x = w \times w \times c$ | stride 1 for all conv. layers |
| | Conv $w \times w \times 32$, BatchNorm, ReLU | padding 1, filter size 3 |
| | Conv $w \times w \times 64$, BatchNorm, ReLU | padding 1, filter size 3 |
| | MaxPool $\frac{w}{2} \times \frac{w}{2} \times 64$ | |
| | Conv $\frac{w}{2} \times \frac{w}{2} \times 128$, BatchNorm, ReLU | padding 1, filter size 3 |
| | Conv $\frac{w}{2} \times \frac{w}{2} \times 256$, BatchNorm, ReLU | padding 1, filter size 3 |
| | MaxPool $\frac{w}{4} \times \frac{w}{4} \times 256$ | |
| | Conv $1 \times 1 \times 256$ | padding 0, filter size $\frac{w}{4}$ |
| | Dropout(0.1) | |
| | FC 256, LeakyReLU(0.1) | |
| | FC $d_z$ | |
| *Unet* | Input: $d_x = w \times w \times n_c$ | stride 1 for all conv. layers |
| | Conv $w \times w \times 64$ | padding 1, filter size 3 |
| | 4-cascaded RefinNet | see Song and Ermon (2019) |
| | &#124;   activation: ELU | exponential LU |
| | &#124;   normalization: InstanceNorm+ | see Song and Ermon (2019) |
| | InstanceNorm+, ELU | |
| | Conv $w \times w \times n_c$ | padding 1, filter size 3 |
| | FC $d_z$ | only if $d_z < d_x$ |

Table 3: Architectures used in the experiments

| Fig./Tab. | Dataset | Description | Configuration |
|---|---|---|---|
| Fig. [1a] | MNIST | Quantifying quality of representations | *Unet-a* |
| Fig. [1b] | CIFAR10 | Quantifying quality of representations | *Unet* |
| Fig. [1b] | CIFAR100 | Quantifying quality of representations | *Unet* |
| Fig. [1c] | MNIST | Transfer learning | *ConvMLP-50* |
| Fig. [1d] | CIFAR10 | Transfer learning | *ConvMLP-90* |
| Tab. [1a] | MNIST | Transfer learning | *ConvMLP-50* |
| Tab. [1a] | CIFAR10 | Transfer learning | *ConvMLP-90* |
| Tab. [1b] | FashionMNIST | Semi-supervised learning | *ConvMLP-50* |
| Tab. [1b] | CIFAR10 | Semi-supervised learning | *ConvMLP-50p* |

training was done for 5000 iterations (no visible improvements in the results were observed after this many iterations). For CIFAR10 and CIFAR100 experiments, we introduced a random horizontal flip to the data, with probability $0.5$.

We used conditional denoising score matching (CDSM, Appendix B.1) to train the energy models. The noise parameter used is $\sigma = 0.01$.

## A.2 The MCC metric

To quantify identifiability, we use the mean correlation coefficient (MCC) metric, which computes the maximum linear correlations up to permutation of components. To obtain the value of this metric between two vectors $\mathbf{x}$ and $\mathbf{y}$, we first calculate all pairs of correlation coefficients between the components $x_i$ of $\mathbf{x}$, and the components $y_j$ of $\mathbf{y}$. Since the order of the components in each vector can be arbitrary, we have to account for possible permutations between the indices $i$ and $j$. This is done by solving a linear sum assignment problem (for instance, using the auction algorithm). We finally average over all correlation coefficients (after finding the right permutation). This makes the MCC metric invariant by permutation and component-wise transformations (as a consequence of the transformation invariance of the correlation coefficient).

To better understand this metric, let's consider the following example. Let $\mathbf{x} \in \mathbb{R}^2$ be a bivariate random variable such that $x_1 \perp\!\!\!\perp x_2$, and let $\mathbf{y} = (x_2^2, x_1^2)$. If we don't account for any permutations, then the average correlation is equal to $\frac{1}{2} \sum_i \mathrm{corr}(x_i, y_i) = 0$ because $x_1 \perp\!\!\!\perp x_2$. In reality, though, $\mathbf{y}$ and $\mathbf{x}$ are perfectly correlated, since the value of $\mathbf{x}$ completely determines that of $\mathbf{y}$. Thus, we have to find the optimal permutation of the elements of $\mathbf{y}$ in order to maximize the average correlation. The MCC does this by computing all pair-wise correlations, and finding the assignment that maximizes the average correlation.

When the latent ground truth is known (Section 5.2—IMCA and nonlinear ICA simulations, for instance), we can test for identifiability of the components by comparing the recovered latents to this ground truth. A high MCC means that we recovered the true latents.

When the ground truth is unknown (Section 5.1—real image datasets), we compare pairs of learnt representations, each from a different random initialization. A consistently high MCC means that changing the random state of the model doesn't drastically change the learnt representations.

## A.3 Quality of representations

We argued that conditioning enables EBMs to learn identifiable representations. The results in Section 5.1 validate this. The plots presented in Figures [1a] and [1b] were produced using the *Unet* configuration, described in Table [3]. This architecture is complex and deep, and involves multiple layers for which a thorough theoretical analysis is very difficult, unlike MLPs for instance. In addition, the dimension of the latent space was chosen to be equal to that of the input space. Intuitively, we would expect that the chance of learning arbitrary representations increases as we increase the number of features because this increases the entropy of the system.

This allows us to challenge the capabilities of ICE-BeeM, and test its limits. We concluded from the results that the theory presented here does benefit modern deep learning architectures. This experiment serves to empirically validate our theoretical result, and is the first of its kind in recent identifiability literature, which focused on validating the theory on simulated data with well know ground truth.

The matrix $\mathbf{A}$ in equation (2) and the permutation $\sigma$ in equation (3) were learnt from the first half of the test partition for each dataset. The evaluation of the MCCs was done on the remaining half of the test dataset.

We present further plots detailing the quality of the learnt representations on MNIST, FashionMNIST, CIFAR10 and CIFAR100 for a variety of different configurations in Figures [2] and [3].

(a) MNIST - *ConvMLP-50/50p*   (b) MNIST - *ConvMLP-90/90p*   (c) MNIST - *ConvMLP-200/200p*

(d) FMNIST - *ConvMLP-50/50p*   (e) FMNIST - *ConvMLP-90/90p*   (f) FMNIST - *ConvMLP-200/200p*

(g) C10 - *ConvMLP-50/50p*   (h) C10 - *ConvMLP-90/90p*   (i) C10 - *ConvMLP-200/200p*

(j) C100 - *ConvMLP-50/50p*   (k) C100 - *ConvMLP-90/90p*   (l) C100 - *ConvMLP-200/200p*

Figure 2: Further experiments on the *strong* identifiability of learnt representations using the *ConvMLP* architecture on image datasets — C10/100 stands for CIFAR10 and CIFAR100, respectively.

|                | (a) MNIST - *Unet* | (b) CIFAR10 - *Unet* | (c) CIFAR100 - *Unet* |
|----------------|--------------------|----------------------|-----------------------|

Figure 3: Further experiments on the identifiability of representations using the *Unet* architecture on image datasets.

## A.4 Transfer learning experiments

### A.4.1 Intuition

As a practical application of our framework where identifiability is important, we consider meta-learning, in particular multi-task and transfer learning. Assume we have $N$ datasets, which could be, *e.g.*, different subjects in biomedical settings, or different image datasets. This fits well with our framework, where $y = 1, \ldots, N$ is now the index of the dataset, or "task". The key question in such a setting is how we can leverage all the observations to better model each single dataset, and especially transfer knowledge of existing models to a new dataset.

To this end, we propose an intuitively appealing approach, where we approximate the unnormalized log-pdf in $y$-th dataset $p(\mathbf{x}; y)$ by a linear combination of a learned "basis" functions $f_{i,\boldsymbol{\theta}}$ as

$$\log p(\mathbf{x}; y) + \log Z(\boldsymbol{\theta}) \approx \sum_{i=1}^{k} g_i(y) f_{i,\boldsymbol{\theta}}(\mathbf{x}) \tag{6}$$

where the $g_i(y)$ are scalar parameters as a function of $y$, which act as coefficients in the basis $(f_{i,\boldsymbol{\theta}})$. This linear approximation is nothing else than a special case of ICE-BeeM, but here, we interpret such an approximation as a linear approximation in log-pdf space. In fact, what we are doing is a kind of PCA in the set of probability distributions $p(\mathbf{x}; y)$. Such "probability space" PCA allows the models for the different datasets to learn from each other, as in the classical idea of denoising by projection onto the PCA subspace.

In transfer learning, we observe a new dataset, with distribution $p(\mathbf{x}; y_{\text{new}})$ for $y_{\text{new}} = N + 1$. Based on our decomposition, we approximate $p(\mathbf{x}; y_{\text{new}})$ as in (6). This leads to a drastic simplification: we can learn the basis functions $f_{i,\boldsymbol{\theta}}$ from the first $N$ datasets, then we only need to estimate the $k$ scalar parameters $g_i(y_{\text{new}})$ for the new dataset. The coefficients are likely to be sparse as well, which provides an additional penalty.

Reducing the transfer learning to estimation of the $g_i(y_{\text{new}})$ clearly requires that we have estimated the true $f_i$ up to a linear transformation, which is the weaker form of identifiability in Theorem 1. Moreover, using a sparsity penalty is only meaningful if we have the true $f_i$ without any linear mixing, which requires the stronger identifiability in Theorem 2.

Training can be done by any method for EBM estimation. In particular, it is very easy by score matching because equation (6) is an exponential family for fixed $f_i$ (Hyvärinen, 2007).

### A.4.2 Further experiments

The pre-training was done on labels 0-7 from the train partition for MNIST, FashionMNIST and CIFAR10, and on labels 0-84 from the train partition for CIFAR100. The second (transfer) step was done on labels 8-9 from the train partition for MNIST, FashionMNIST and CIFAR10, and on and labels 85-99 the train partition for CIFAR100.

(a) Transfer learning, $\mathbf{f}_\theta$ fixed        (b) Baseline, both $\mathbf{f}_\theta$ and $\mathbf{g}_\theta$ estimated

Figure 4: Further results for transfer learning experiments on MNIST. In the case of transfer learning 99 out of a hundred returned digits are class 8 compared to only 58 in the baseline.

We considered a subset of size 6000 to produce the values in Table [1a]. This table should be read in conjunction with Figures [1c]-[1d] for a proper evaluation of performance.

We present further plots and results of transfer learning experiments in Figures [5]-[6] and Table [4] ran on MNIST, FashionMNIST, CIFAR10 and CIFAR100 for a variety of different configurations. for different configurations and datasets. We considered a subset of size 6000 to produce the values in Table 4. We expect the baseline where we don't perform transfer learning to perform comparatively for such a subset size: transfer learning is mostly important when data is scarce. For the complete picture, this table should be read in conjunction with Figures [5]-[6].

As an additional way to visualize the results, Figure [4a] shows unseen MNIST samples (taken across all possible classes) which are assigned high confidence of belonging to the "new" class 8 after transfer learning, indicating that the ICE-BeeM model has learnt a reasonable distribution over unseen classes. By comparison the case where no transfer learning is employed (Figure [4b]), incorrectly assigns high confidences to other digits.

## A.5 Semi-supervised learning

In this experiment, we train both an identifiable ICE-BeeM model and an unconditional (non-identifiable) EBM on classes 0-7. The purpose of this step is to learn a feature extractor $\mathbf{f}_\theta$ that is able of learning meaningful features from the images. To test the quality of the features learnt by both models (the ICE-BeeM, and the unconditional EBM), we use the feature map $\mathbf{f}_\theta$ to classify unseen samples from classes 8-9. Results show that ICE-BeeM outperforms the unconditional baseline in this classification task. We attribute this to the identifiability of ICE-BeeM: our model seems to be performing a principled form of disentanglement by learning features that are faithful to the unknown factors of variation in the data.

Training was done on labels 0-7, using the train partition for MNIST, FashionMNIST and CIFAR10. Evaluation was done on labels 8-9, using the test partition for all three datasets. This data was in turn partitioned for the classification into a train and test split. The split proportion is $15\%$ for MNIST and FashionMNIST, and $33\%$ for CIFAR10 and CIFAR100.

We present further results for the semi-supervised learning experiments in Table [5], ran on MNIST, FashionMNIST, CIFAR10 for a variety of different configurations.

## A.6 IMCA and nonlinear ICA simulations

We give here more detail on the data generation process for the simulations in Section 5.2, as well as the architectures used.

(a) MNIST - *ConvMLP-50*

(b) MNIST - *ConvMLP-200*

(c) FMNIST - *ConvMLP-90*

(d) FMNIST - *ConvMLP-90p*

(e) CIFAR10 - *ConvMLP-200*

(f) CIFAR10 - *ConvMLP-200p*

(g) CIFAR100 - *ConvMLP-50*

(h) CIFAR100 - *ConvMLP-50p*

Figure 5: Further transfer learning — the dataset/configuration combo are reported in the captions.

(a) MNIST - *Unet*

(b) MNIST - *Unet-a*

(c) FMNIST - *Unet*

(d) FMNIST - *Unet-a*

(e) CIFAR10 - *Unet*

(f) CIFAR10 - *Unet-a*

(g) CIFAR100 - *Unet*

(h) CIFAR100 - *Unet-a*

Figure 6: Further transfer learning — the dataset/configuration combo are reported in the captions.

Table 4: Transfer learning — CDSM score (lower is better)

| Dataset | Configuration | $\mathbf{f} \cdot \mathbf{g}_\theta$ | $\mathbf{f} \cdot \mathbf{1}$ | $\mathbf{f}_\theta \cdot \mathbf{g}_\theta$ | $\mathbf{f}_\theta \cdot \mathbf{1}$ |
|---|---|---|---|---|---|
| MNIST | *ConvMLP-50* | $2.95 \pm 0.02$ | $23.43 \pm 0.04$ | $4.22 \pm 0.15$ | $3.64 \pm 0.10$ |
| | *ConvMLP-50p* | $2.79 \pm 0.00$ | $796.99 \pm 0.86$ | $10.13 \pm 4.74$ | $3.63 \pm 0.09$ |
| | *ConvMLP-90* | $2.94 \pm 0.01$ | $12.18 \pm 0.03$ | $4.29 \pm 0.13$ | $3.67 \pm 0.12$ |
| | *ConvMLP-90p* | $3.03 \pm 0.01$ | $694.94 \pm 1.03$ | $10.22 \pm 4.63$ | $3.70 \pm 0.12$ |
| | *ConvMLP-200* | $2.91 \pm 0.01$ | $27.70 \pm 0.02$ | $4.29 \pm 0.12$ | $3.74 \pm 0.09$ |
| | *ConvMLP-200p* | $2.95 \pm 0.01$ | $805.45 \pm 3.56$ | $12.08 \pm 3.79$ | $3.71 \pm 0.13$ |
| | *Unet* | $2.23 \pm 0.01$ | $10.04 \pm 0.01$ | $3.44 \pm 0.03$ | $2.97 \pm 0.25$ |
| | *Unet-a* | $2.29 \pm 0.01$ | $6.18 \pm 0.00$ | $3.44 \pm 0.02$ | $6.27 \pm 4.21$ |
| | *Unet-p* | $14.00 \pm 0.01$ | $14.08 \pm 0.00$ | $11.97 \pm 4.01$ | $6.14 \pm 4.17$ |
| | *Unet-50a* | $2.61 \pm 0.02$ | $14.24 \pm 0.01$ | $3.79 \pm 0.56$ | $2.92 \pm 0.20$ |
| | *MLP-50* | $13.99 \pm 0.01$ | $13.99 \pm 0.01$ | $14.00 \pm 0.01$ | $14.00 \pm 0.01$ |
| | *MLP-50p* | $13.99 \pm 0.01$ | $14.00 \pm 0.01$ | $14.00 \pm 0.01$ | $14.00 \pm 0.01$ |
| | *MLP-90* | $14.00 \pm 0.01$ | $14.00 \pm 0.01$ | $14.00 \pm 0.01$ | $13.99 \pm 0.01$ |
| | *MLP-90p* | $13.99 \pm 0.01$ | $14.00 \pm 0.01$ | $14.00 \pm 0.01$ | $14.00 \pm 0.01$ |
| | *MLP-200* | $13.99 \pm 0.01$ | $14.00 \pm 0.01$ | $14.00 \pm 0.01$ | $14.00 \pm 0.01$ |
| | *MLP-200p* | $13.99 \pm 0.01$ | $13.99 \pm 0.01$ | $14.00 \pm 0.01$ | $14.00 \pm 0.01$ |
| FMNIST | *ConvMLP-50* | $7.88 \pm 0.01$ | $9.82 \pm 0.03$ | $7.88 \pm 0.07$ | $7.18 \pm 0.25$ |
| | *ConvMLP-50p* | $8.00 \pm 0.02$ | $197.84 \pm 2.27$ | $7.92 \pm 0.18$ | $7.10 \pm 0.24$ |
| | *ConvMLP-90* | $8.09 \pm 0.02$ | $10.86 \pm 0.04$ | $7.88 \pm 0.05$ | $7.14 \pm 0.24$ |
| | *ConvMLP-90p* | $7.94 \pm 0.01$ | $197.93 \pm 2.33$ | $7.87 \pm 0.13$ | $7.13 \pm 0.20$ |
| | *ConvMLP-200* | $7.98 \pm 0.00$ | $15.86 \pm 0.01$ | $7.91 \pm 0.16$ | $7.17 \pm 0.21$ |
| | *ConvMLP-200p* | $7.86 \pm 0.01$ | $196.14 \pm 2.07$ | $7.81 \pm 0.15$ | $7.11 \pm 0.15$ |
| | *Unet* | $6.47 \pm 0.02$ | $277.56 \pm 1.06$ | $6.52 \pm 0.03$ | $6.46 \pm 0.07$ |
| | *Unet-a* | $6.60 \pm 0.02$ | $24.62 \pm 0.02$ | $6.52 \pm 0.02$ | $6.41 \pm 0.01$ |
| | *MLP-50* | $13.99 \pm 0.01$ | $14.00 \pm 0.01$ | $13.99 \pm 0.01$ | $14.00 \pm 0.01$ |
| | *MLP-200* | $13.99 \pm 0.01$ | $14.00 \pm 0.01$ | $13.99 \pm 0.01$ | $14.00 \pm 0.01$ |
| CIFAR10 | *ConvMLP-50* | $8.02 \pm 0.01$ | $32.09 \pm 0.07$ | $8.36 \pm 0.03$ | $8.15 \pm 0.03$ |
| | *ConvMLP-50p* | $8.04 \pm 0.02$ | $412.15 \pm 2.54$ | $8.35 \pm 0.04$ | $8.17 \pm 0.01$ |
| | *ConvMLP-90* | $8.03 \pm 0.01$ | $23.08 \pm 0.04$ | $8.37 \pm 0.02$ | $8.16 \pm 0.05$ |
| | *ConvMLP-90p* | $8.05 \pm 0.01$ | $408.51 \pm 2.30$ | $8.37 \pm 0.04$ | $8.16 \pm 0.01$ |
| | *ConvMLP-200* | $8.02 \pm 0.02$ | $13.35 \pm 0.01$ | $8.41 \pm 0.07$ | $8.13 \pm 0.03$ |
| | *ConvMLP-200p* | $8.06 \pm 0.01$ | $509.09 \pm 2.31$ | $8.35 \pm 0.02$ | $8.11 \pm 0.03$ |
| | *Unet* | $7.29 \pm 0.01$ | $118.93 \pm 0.34$ | $7.51 \pm 0.05$ | $9.21 \pm 3.43$ |
| | *Unet-a* | $7.18 \pm 0.01$ | $18.73 \pm 0.01$ | $7.48 \pm 0.09$ | $7.47 \pm 0.13$ |
| | *Unet-50a* | $7.30 \pm 0.05$ | $16.41 \pm 0.00$ | $7.64 \pm 0.26$ | $7.27 \pm 0.03$ |
| | *MLP-50* | $16.00 \pm 0.00$ | $16.00 \pm 0.00$ | $16.00 \pm 0.00$ | $16.00 \pm 0.00$ |
| | *MLP-200* | $16.00 \pm 0.01$ | $16.00 \pm 0.00$ | $16.00 \pm 0.01$ | $16.00 \pm 0.00$ |
| CIFAR100 | *ConvMLP-50* | $8.25 \pm 0.01$ | $45.19 \pm 0.15$ | $8.69 \pm 0.04$ | $8.59 \pm 0.02$ |
| | *ConvMLP-50p* | $8.24 \pm 0.01$ | $2560.77 \pm 7.15$ | $8.68 \pm 0.04$ | $8.61 \pm 0.04$ |
| | *ConvMLP-90* | $8.23 \pm 0.01$ | $8.74 \pm 0.01$ | $8.68 \pm 0.05$ | $8.61 \pm 0.03$ |
| | *ConvMLP-90p* | $8.25 \pm 0.01$ | $3018.50 \pm 7.27$ | $8.65 \pm 0.02$ | $8.58 \pm 0.03$ |
| | *ConvMLP-200* | $8.26 \pm 0.01$ | $42.80 \pm 0.09$ | $8.69 \pm 0.06$ | $8.59 \pm 0.03$ |
| | *ConvMLP-200p* | $8.18 \pm 0.01$ | $3827.36 \pm 16.14$ | $8.65 \pm 0.07$ | $8.63 \pm 0.05$ |
| | *Unet* | $7.41 \pm 0.02$ | $106.28 \pm 0.75$ | $7.77 \pm 0.05$ | $8.38 \pm 0.55$ |
| | *Unet-a* | $7.39 \pm 0.02$ | $11.15 \pm 0.01$ | $7.82 \pm 0.42$ | $9.35 \pm 3.33$ |
| | *Unet-50a* | $7.54 \pm 0.01$ | $15.95 \pm 0.00$ | $7.97 \pm 0.13$ | $7.60 \pm 0.05$ |
| | *MLP-50p* | $16.00 \pm 0.01$ | $16.00 \pm 0.00$ | $16.00 \pm 0.00$ | $16.00 \pm 0.00$ |
| | *MLP-200p* | $16.00 \pm 0.01$ | $16.00 \pm 0.00$ | $16.00 \pm 0.00$ | $16.00 \pm 0.00$ |

Table 5: Semi-supervised learning — classification accuracy (higher is better)

| Dataset | Configuration | ICE-BeeM | Unconditional EBM |
|---|---|---|---|
| MNIST | *ConvMLP-50* | $76.98 \pm 1.61$ | $62.82 \pm 1.48$ |
| | *ConvMLP-50p* | $88.46 \pm 1.14$ | $66.58 \pm 2.64$ |
| | *ConvMLP-90* | $78.93 \pm 1.51$ | $71.61 \pm 1.71$ |
| | *ConvMLP-90p* | $78.66 \pm 1.91$ | $69.13 \pm 1.49$ |
| | *ConvMLP-200* | $81.21 \pm 2.6$ | $71.48 \pm 2.23$ |
| | *ConvMLP-200p* | $77.38 \pm 1.32$ | $68.99 \pm 1.68$ |
| | *MLP-50* | $91.74 \pm 1.72$ | $85.77 \pm 1.14$ |
| | *MLP-50p* | $92.21 \pm 1.74$ | $84.56 \pm 1.1$ |
| | *MLP-90* | $95.17 \pm 0.46$ | $85.91 \pm 2.07$ |
| | *MLP-90p* | $94.97 \pm 0.7$ | $85.97 \pm 1.61$ |
| | *MLP-200* | $94.36 \pm 1.28$ | $89.26 \pm 1.7$ |
| | *MLP-200p* | $91.81 \pm 2.33$ | $90.87 \pm 1.05$ |
| | *Unet* | $97.79 \pm 0.34$ | $98.39 \pm 0.68$ |
| | *Unet-a* | $97.18 \pm 0.5$ | $97.79 \pm 0.78$ |
| | *Unet-50a* | $97.52 \pm 0.4$ | $97.92 \pm 0.49$ |
| | *Unet-20a* | $95.64 \pm 0.7$ | $92.08 \pm 1.71$ |
| FMNIST | *ConvMLP-50* | $77.07 \pm 1.39$ | $56.33 \pm 3.18$ |
| | *ConvMLP-50p* | $71.67 \pm 1.85$ | $57.6 \pm 2.24$ |
| | *ConvMLP-90* | $74.13 \pm 1.86$ | $57.73 \pm 3.12$ |
| | *ConvMLP-90p* | $70.87 \pm 1.13$ | $60.07 \pm 2.9$ |
| | *ConvMLP-200* | $81.4 \pm 1.93$ | $68.27 \pm 2.78$ |
| | *ConvMLP-200p* | $78.47 \pm 0.96$ | $57.47 \pm 2.62$ |
| | *MLP-50* | $98.07 \pm 1.06$ | $90.47 \pm 1.56$ |
| | *MLP-50p* | $97.6 \pm 0.53$ | $90.47 \pm 1.56$ |
| | *MLP-90* | $97.8 \pm 0.34$ | $94.4 \pm 0.53$ |
| | *MLP-90p* | $97.8 \pm 0.34$ | $94.4 \pm 0.53$ |
| | *MLP-200* | $98.6 \pm 0.49$ | $94.87 \pm 0.96$ |
| | *MLP-200p* | $98.6 \pm 0.65$ | $95.33 \pm 1.05$ |
| | *Unet* | $99.67 \pm 0.3$ | $99.93 \pm 0.13$ |
| | *Unet-a* | $99.53 \pm 0.16$ | $99.87 \pm 0.16$ |
| CIFAR10 | *ConvMLP-50* | $69.36 \pm 2.23$ | $56.39 \pm 1.0$ |
| | *ConvMLP-50p* | $64.42 \pm 1.09$ | $51.88 \pm 1.33$ |
| | *ConvMLP-90* | $68.24 \pm 2.0$ | $52.82 \pm 0.95$ |
| | *ConvMLP-90p* | $66.18 \pm 1.01$ | $52.33 \pm 1.73$ |
| | *ConvMLP-200* | $64.73 \pm 1.36$ | $54.18 \pm 1.09$ |
| | *ConvMLP-200p* | $66.3 \pm 0.99$ | $54.48 \pm 1.28$ |
| | *MLP-50* | $68.73 \pm 1.35$ | $70.27 \pm 2.67$ |
| | *MLP-50p* | $69.82 \pm 1.78$ | $69.36 \pm 2.3$ |
| | *MLP-90* | $71.58 \pm 1.21$ | $72.85 \pm 1.16$ |
| | *MLP-90p* | $71.12 \pm 1.64$ | $72.85 \pm 1.16$ |
| | *MLP-200* | $72.39 \pm 1.92$ | $72.97 \pm 1.75$ |
| | *MLP-200p* | $70.94 \pm 1.25$ | $71.97 \pm 2.29$ |
| | *Unet* | $80.27 \pm 4.0$ | $80.58 \pm 0.9$ |
| | *Unet-a* | $80.48 \pm 1.45$ | $80.48 \pm 1.45$ |
| | *Unet-50a* | $77.64 \pm 1.02$ | $73.79 \pm 0.81$ |
| | *Unet-20a* | $74.21 \pm 0.73$ | $68.82 \pm 0.67$ |

**Data generation**    We generate 5-dimensional synthetic datasets following the nonlinear ICA model which is a special case of equation (4) where the base measure, $\mu(\mathbf{z})$, is factorial. In particular, we set it to $\mu(\mathbf{z}) = 1$. As such, latent variables are conditionally independent given segment labels. The sources are divided into $M = 8$ segments, and the conditioning variable $\mathbf{y}$ is defined to be the segment index, uniformly drawn from the integer set $[\![1, M]\!]$. Following Hyvärinen and Morioka (2016), the $\mathbf{z}$ are generated according to isotropic Gaussian distributions with distinct precisions $\boldsymbol{\lambda}(\mathbf{y})$ determined by the segment index. Second, we perform the same experiment but on data generated from an IMCA model where the base measure $\mu(\mathbf{z})$ is *not factorial*. More specifically, we randomly generate an invertible and symmetric matrix $\boldsymbol{\Sigma}_0 \in \mathbb{R}^{d \times d}$, such that $\mu(\mathbf{z}) \propto e^{-0.5\mathbf{z}^T \Sigma_0^{-1} \mathbf{z}}$. As before, we define $\boldsymbol{\lambda}(\mathbf{y})$ to be the distinct conditional precisions. The precision matrix of each segment is now equal to $\Sigma(\mathbf{y})^{-1} = \Sigma_0^{-1} + \text{diag}(\boldsymbol{\lambda}(\mathbf{y}))^{-1}$, meaning the latent variables are no longer conditionally independent.

For both nonlinear ICA and IMCA data, a randomly initialized neural network with varying number of layers, $L \in \{2, 4\}$, was employed to generate the nonlinear mixing function $\mathbf{h}$. Leaky ReLU with negative slope equal to $0.1$ was employed as the activation function in order to ensure the network was invertible. The hidden dimensions of the mixing network are equal to the latent dimension $d_x$, and the output dimension is $d_x = d_z$.

**Baseline methods**    The first baseline we compare to is TCL (Hyvärinen and Morioka, 2016), which is a self-supervised method for nonlinear ICA based on the nonstationarity of the sources. TCL learns to invert the mixing function $\mathbf{h}$, by performing a surrogate classification task, where the goal is to classify original observations against their segment indices in a multinomial classification task. Its theory is premised on the fact that the feature extractor used for the classification has to extract meaningful latents in order to perform well in the classification task.

The second baseline is iVAE (Khemakhem et al., 2020), a nonlinear ICA method which uses an identifiable VAE to recover the independent sources. Its theory is premised on the consistency of maximum likelihood training, and on the flexibility of VAEs in approximating densities. They show that given enough data, the variational posterior learns to approximate the true posterior distribution, and can thus be used to invert the mixing function. The iVAE, like a regular VAE, is trained by maximizing the ELBO (Kingma and Welling, 2013).

**Training of ICE-BeeM via flow contrastive estimation**    To demonstrate that ICE-BeeM can be trained by any method for training EBMs, we switched from denoising score matching to flow contrastive estimation (FCE, Appendix B.2). As a contrastive flow, we used a normalizing flow model (Rezende and Mohamed, 2015), with an isotropic and tractable base distribution. It is then transformed by a 10-layer flow, where each layer is made of a succession of a neural spline flow (Durkan et al., 2019), an invertible $1 \times 1$ convolution (Kingma and Dhariwal, 2018), and an ActNorm layer (Kingma and Dhariwal, 2018). The flow parameters are updated by and Adam optimizer, with a learning rate of $10^{-5}$.

**Used architectures**    The architectures used to produce Figures [1e] and [1f] are summarized by Table [6].

# B    Estimation algorithms

It is important to note that the identifiability results presented above apply to conditional EBMs in general. As such, we may employ any of the wide variety of methods which have been proposed for the estimation of unnormalized EBMs. In this work we used two different options with good results for both: flow contrastive estimation (Gao et al., 2019) and denoising score matching (Vincent, 2011). Both methods can also be extended to the conditional case in a straightforward fashion.

Flow-contrastive estimation (FCE) can be seen as an extension of noise-contrastive estimation (Gutmann and Hyvärinen, 2012, NCE), which seeks to learn unnormalized EBMs by solving a surrogate classification task. The proposed classification task seeks to discriminate between the true data and some synthetic noise data based on the log-odds ratio of the EBM and the noise distribution. However, a limitation of NCE is the need to specify a noise distribution which can be sampled from and whose log-density can be evaluated pointwise but which also shares some of the empirical

Table 6: Architectures used in the simulations

| Model | Optimizer | Architecture | |
|---|---|---|---|
| | | Input | $d_x = 5$ |
| | | Condition | one hot encoded $d_y = M = 8$ |
| | | Latent | $d_z = d_x = 5$ |
| | | Num. layers | $L \in \{2, 4\}$ |
| ICE-BeeM | Adam lr $3.10^{-4}$ | $\mathbf{f}_{\boldsymbol{\theta}}$ | $(L+1)$-layer MLP batch norm after each FC layer hidden dim 32, LeakyReLU(0.1) act |
| | | $\mathbf{g}_{\boldsymbol{\theta}}$ | $(d_z \times d_y)$ learnable matrix |
| iVAE | Adam lr $10^{-3}$ | Encoder | $p(\mathbf{z}|\mathbf{x})$ Normal 3-layer MLP hidden dim $2d_x$, LeakyReLU(0.1) act |
| | | Decoder | $p(\mathbf{x}|\mathbf{z}, \mathbf{y})$ Normal 3-layer MLP hidden dim $2d_x$, LeakyReLU(0.1) act |
| | | Prior | $p(\mathbf{z}|\mathbf{y})$ Normal 3-layer MLP hidden dim $2d_x$, LeakyReLU(0.1) act |
| TCL | Momentum 0.9 lr 0.01 exp decay 0.1 | | $L$-layer MLP FC $2d_x$, maxout(2) $(L-2)\times$ [FC $d_x$, maxout(2)] FC $d_x$, absolute value |

properties of the observed data. To address this concern Gao et al. (2019) propose to employ a flow model as the contrast noise distribution. FCE seeks to simultaneously learn both an unnormalized EBM as well as a flow model for the contrast noise in an alternating fashion. We naturally get a conditional version for FCE by learning a conditional EBM (Gao et al., 2019, eq. 12).

Score matching is another well-known method for learning unnormalized models (Hyvärinen, 2005). However, its computational implementation in deep networks is problematic, which is why Vincent (2011) proposed a stochastic approximation which can be interpreted as denoising the data, and which works efficiently in deep networks (Saremi et al., 2018; Song and Ermon, 2019).

### B.1 Conditional denoising score matching

We extend the original score matching objective to the conditional setting in a natural way: for a fixed $\mathbf{y}$, we compute the unconditional score matching objective: $J(\boldsymbol{\theta}, \mathbf{y}) = \mathbb{E}_{p(\mathbf{x}|\mathbf{y})} \|\nabla_{\mathbf{x}} \log p_{\boldsymbol{\theta}}(\mathbf{x}|\mathbf{y}) - \nabla_{\mathbf{x}} \log p(\mathbf{x}|\mathbf{y})\|^2$, and then average over all values of $\mathbf{y}$. The expression of the conditional score matching objective is then:

$$\mathcal{J}_{\text{CSM}}(\boldsymbol{\theta}) = \mathbb{E}_{p(\mathbf{x}, \mathbf{y})} \|\nabla_{\mathbf{x}} \log p_{\boldsymbol{\theta}}(\mathbf{x}|\mathbf{y}) - \nabla_{\mathbf{x}} \log p(\mathbf{x}|\mathbf{y})\|^2 \tag{7}$$

We build on the recent developments by Vincent (2011), and introduce a conditional denoising score matching objective by replacing the unknown density by a kernel density estimator. Formally, given observations $\mathcal{D} = \left\{ \left(\mathbf{x}^{(1)}, \mathbf{y}^{(1)}\right), \dots, \left(\mathbf{x}^{(N)}, \mathbf{y}^{(N)}\right) \right\}$, we first derive nonparamteric kernel density estimates of $p(\mathbf{x}, \mathbf{y})$ and $p(\mathbf{y})$, which we then use to derive the estimate for $p(\mathbf{x}|\mathbf{y})$ using the product rule. These estimates have the forms:

$$q_b(\mathbf{y}) = \mathbb{E}_{\mathbf{y}' \sim q_{\mathcal{D}}} \left[ l_b(\mathbf{y}|\mathbf{y}') \right] \tag{8}$$

$$q_{ab}(\mathbf{x}, \mathbf{y}) = \mathbb{E}_{(\mathbf{x}', \mathbf{y}') \sim q_{\mathcal{D}}} \left[ k_a(\mathbf{x}|\mathbf{x}') l_b(\mathbf{y}|\mathbf{y}') \right] \tag{9}$$

$$q_{ab}(\mathbf{x}|\mathbf{y}) = \frac{q_{ab}(\mathbf{x}, \mathbf{y})}{q_b(\mathbf{y})} \tag{10}$$

where $k_a$ and $l_b$ are bounded kernel functions defined on $\mathcal{X}$ and $\mathcal{Y}$ and with bandwidths[5] $a$ and $b$, respectively. In the following, we assume that the bandwidth sequences are equal ($a = b = \sigma$).

We replace $p(\mathbf{x}, \mathbf{y})$ and $p(\mathbf{x}|\mathbf{y})$ in (7) by their estimates $q_\sigma(\mathbf{x}, \mathbf{y})$ and $q_\sigma(\mathbf{x}|\mathbf{y})$, to arrive at the new objective

$$\mathcal{J}_{\mathrm{CSM}_\sigma}(\boldsymbol{\theta}) = \mathbb{E}_{q_\sigma(\mathbf{x},\mathbf{y})} \|\nabla_\mathbf{x} \log p_{\boldsymbol{\theta}}(\mathbf{x}|\mathbf{y}) - \nabla_\mathbf{x} \log q_\sigma(\mathbf{x}|\mathbf{y})\|^2 \tag{11}$$

which is the conditional score matching objective when applied to the nonparametric estimates of the unknown target density. We will show below that it is equivalent to a simpler objective, in which we only need to compute gradients of the conditioning kernel $k_\sigma(\mathbf{x}|\mathbf{y})$:

$$\mathcal{J}_{\mathrm{CDSM}_\sigma}(\boldsymbol{\theta}) = \mathbb{E}\|\nabla_\mathbf{x} \log p_{\boldsymbol{\theta}}(\mathbf{x}|\mathbf{y}) - \nabla_\mathbf{x} \log k_\sigma(\mathbf{x}|\mathbf{x}')\|^2 \tag{12}$$

where the expectation is taken with respect to $p_\mathcal{D}(\mathbf{x}', \mathbf{y}')k_\sigma(\mathbf{x}|\mathbf{x}')l_\sigma(\mathbf{y}|\mathbf{y}')$. We call this objective conditional denoising score matching. Its extrema landscape is the same as $\mathcal{J}_{\mathrm{CSM}_\sigma}$, but it has the advantage of being simpler to evaluate and interpret.

Above, we presented this objective when $k_\sigma$ is the Gaussian kernel, and $l_\sigma$ is simply the identity kernel.

**From CSM to CDSM** We will show here that the stochastic approximation used in denoising score matching can also be used for the conditional case to get to the CDSM objective (12) from the CSM objective (11):

$$\mathcal{J}_{\mathrm{CSM}_\sigma}(\boldsymbol{\theta}) = \mathbb{E}_{q_\sigma(\mathbf{x},\mathbf{y})} \left\| \nabla_\mathbf{x} \log \frac{p_{\boldsymbol{\theta}}(\mathbf{x}|\mathbf{y})}{q_\sigma(\mathbf{x}|\mathbf{y})} \right\|^2 = \mathbb{E}_{q_\sigma(\mathbf{x},\mathbf{y})} \|\nabla_\mathbf{x} \log p_{\boldsymbol{\theta}}(\mathbf{x}|\mathbf{y})\|^2 - S(\boldsymbol{\theta}) + C_1 \tag{13}$$

where $C_1$ is a constant term that only depends on $q_\sigma(\mathbf{x}|\mathbf{y})$, and

$$\begin{aligned}
S(\boldsymbol{\theta}) &= \mathbb{E}_{q_\sigma(\mathbf{x},\mathbf{y})} \langle \nabla_\mathbf{x} \log p_{\boldsymbol{\theta}}(\mathbf{x}|\mathbf{y}), \nabla_\mathbf{x} \log q_\sigma(\mathbf{x}|\mathbf{y}) \rangle \\
&= \int q_\sigma(\mathbf{x}, \mathbf{y}) \langle \nabla_\mathbf{x} \log p_{\boldsymbol{\theta}}(\mathbf{x}|\mathbf{y}), \frac{\nabla_\mathbf{x} q_\sigma(\mathbf{x}|\mathbf{y})}{q_\sigma(\mathbf{x}|\mathbf{y})} \rangle \mathrm{d}\mathbf{x}\mathrm{d}\mathbf{y} \\
&= \int q_\sigma(\mathbf{y}) \langle \nabla_\mathbf{x} \log p_{\boldsymbol{\theta}}(\mathbf{x}|\mathbf{y}), \nabla_\mathbf{x} q_\sigma(\mathbf{x}|\mathbf{y}) \rangle \mathrm{d}\mathbf{x}\mathrm{d}\mathbf{y} \\
&= \int q_\sigma(\mathbf{y}) \langle \nabla_\mathbf{x} \log p_{\boldsymbol{\theta}}(\mathbf{x}|\mathbf{y}), \nabla_\mathbf{x} \frac{\int p_\mathcal{D}(\mathbf{x}', \mathbf{y}')k_\sigma(\mathbf{x}|\mathbf{x}')l_\sigma(\mathbf{y}|\mathbf{y}')\mathrm{d}\mathbf{x}'\mathrm{d}\mathbf{y}'}{q_\sigma(\mathbf{y})} \rangle \mathrm{d}\mathbf{x}\mathrm{d}\mathbf{y} \\
&= \int \int p_\mathcal{D}(\mathbf{x}', \mathbf{y}')l_\sigma(\mathbf{y}|\mathbf{y}')k_\sigma(\mathbf{x}|\mathbf{x}') \langle \nabla_\mathbf{x} \log p_{\boldsymbol{\theta}}(\mathbf{x}|\mathbf{y}), \nabla_\mathbf{x} \log k_\sigma(\mathbf{x}|\mathbf{x}') \rangle \mathrm{d}\mathbf{x}'\mathrm{d}\mathbf{y}'\mathrm{d}\mathbf{x}\mathrm{d}\mathbf{y} \\
&= \mathbb{E}_{p_\mathcal{D}(\mathbf{x}',\mathbf{y}')k_\sigma(\mathbf{x}|\mathbf{x}')l_\sigma(\mathbf{y}|\mathbf{y}')} \langle \nabla_\mathbf{x} \log p_{\boldsymbol{\theta}}(\mathbf{x}|\mathbf{y}), \nabla_\mathbf{x} \log k_\sigma(\mathbf{x}|\mathbf{x}') \rangle
\end{aligned}$$

Plugging this back into equation (13), we find that

$$\begin{aligned}
\mathcal{J}_{\mathrm{CSM}_\sigma}(\boldsymbol{\theta}) &= \mathbb{E}\|\nabla_\mathbf{x} \log p_{\boldsymbol{\theta}}(\mathbf{x}|\mathbf{y}) - \nabla_\mathbf{x} \log k_\sigma(\mathbf{x}|\mathbf{x}')\|^2 + C_1 - C_2 \\
&= \mathcal{J}_{\mathrm{CDSM}_\sigma}(\boldsymbol{\theta}) + C_1 - C_2
\end{aligned}$$

where the expectation is with respect to $p_\mathcal{D}(\mathbf{x}', \mathbf{y}')k_\sigma(\mathbf{x}|\mathbf{x}')l_\sigma(\mathbf{y}|\mathbf{y}')$ and $C_2$ is another constant that is only a function of $k_\sigma(\mathbf{x}|\mathbf{x}')$. $\qquad\square$

### B.2 Conditional flow contrastive estimation

As described above, FCE learns the parameter for the density $p_{\boldsymbol{\theta}}$ of an EBM by performing a surrogate classification task: noise is generated from a noise distribution $q_{\boldsymbol{\alpha}}$ which is parameterized as a flow model, and a logistic regression is performed to classify observation into real data samples or noise samples. The objective function is simply the log-odds:

$$\mathcal{J}_{\mathrm{FCE}}(\boldsymbol{\theta}, \boldsymbol{\alpha}) = \mathbb{E}_{p_{\mathrm{data}}(\mathbf{x})} \log \frac{p_{\boldsymbol{\theta}}(\mathbf{x})}{q_{\boldsymbol{\alpha}}(\mathbf{x}) + p_{\boldsymbol{\theta}}(\mathbf{x})} + \mathbb{E}_{q_{\boldsymbol{\alpha}}(\mathbf{x})} \log \frac{q_{\boldsymbol{\alpha}}(\mathbf{x})}{q_{\boldsymbol{\alpha}}(\mathbf{x}) + p_{\boldsymbol{\theta}}(\mathbf{x})} \tag{14}$$

This objective is minimized with respect to $\boldsymbol{\theta}$ and maximized with respect to $\boldsymbol{\alpha}$: the EBM and the flow model are playing a min-max game. This objective can be extended to the conditional case

naturally: we replace the model density by the conditional density $p_{\boldsymbol{\theta}}(\mathbf{x}|\mathbf{y})$. In the conditional case, it follows that noise samples should also be associated with a conditioning variable, $\mathbf{y}$. One way this can be achieved is by also considering a conditional flow. This also has the additional benefit that an improved flow should lead to better estimation of EBM. Alternatively, a standard (non-conditional) flow could be employed. This would require marginalizing over the conditioning variable, $\mathbf{y}$. The objective simply becomes:

$$\mathcal{J}_{\text{CFCE}}(\boldsymbol{\theta}, \boldsymbol{\alpha}) = \mathbb{E}_{p_{\text{data}(\mathbf{x}, \mathbf{y})}} \log \frac{p_{\boldsymbol{\theta}}(\mathbf{x}|\mathbf{y})}{q_{\boldsymbol{\alpha}}(\mathbf{x}, \mathbf{y}) + p_{\boldsymbol{\theta}}(\mathbf{x}|\mathbf{y})} + \mathbb{E}_{q_{\boldsymbol{\alpha}}(\mathbf{x}, \mathbf{y})} \log \frac{q_{\boldsymbol{\alpha}}(\mathbf{x}, \mathbf{y})}{q_{\boldsymbol{\alpha}}(\mathbf{x}, \mathbf{y}) + p_{\boldsymbol{\theta}}(\mathbf{x}|\mathbf{y})} \quad (15)$$

We can write the flow density as $q_{\boldsymbol{\alpha}}(\mathbf{x}, \mathbf{y}) = p(\mathbf{y}) q_{\boldsymbol{\alpha}}(\mathbf{x}|\mathbf{y})$. This is particularly useful when the conditioning variable $\mathbf{y}$ is discrete, like for instance the index of a dataset or a segment, as we can sample draw a index from a uniform distribution, and use the conditional flow to sample an observation.

## C  Identifiability of the conditional energy-based model

Recall the form of our conditional energy model

$$p_{\boldsymbol{\theta}}(\mathbf{x}|\mathbf{y}) = Z(\mathbf{y}; \boldsymbol{\theta})^{-1} \exp\left(-\mathbf{f}_{\boldsymbol{\theta}}(\mathbf{x})^T \mathbf{g}_{\boldsymbol{\theta}}(\mathbf{y})\right) \quad (16)$$

We present in this section the proofs for the different forms of identifiability that is guaranteed for the feature extractors $\mathbf{f}$ and $\mathbf{g}$. We will focus on the proofs for the feature extractor $\mathbf{f}$, as the proofs for the feature extractor $\mathbf{g}$ are very similar. For the rest of the Appendix, we will denote by $d = d_x$, $m = d_y$ and $n = d_z$.

### C.1  More on the equivalence relations

The relation $\sim_w^{\mathbf{f}}$ in equation (2) is an equivalence relation in the strict term only if $\mathbf{A}$ is full rank. If $\mathbf{A}$ is not full rank (which is only possible if $d_z > d_x$, given the rest of assumptions), then it is not necessarily symmetric. This is not a real problem, and can be fixed by changing the definition to: there exists $\mathbf{A}_1, \mathbf{A}_2$ such that $\mathbf{f}_{\boldsymbol{\theta}} = \mathbf{A}_1 \mathbf{f}_{\boldsymbol{\theta}'} + \mathbf{c}_1$ and $\mathbf{f}_{\boldsymbol{\theta}'} = \mathbf{A}_2 \mathbf{f}_{\boldsymbol{\theta}} + \mathbf{c}_2$. We present the simpler version in the paper for clarity.

### C.2  Proof of Theorem 1

We start by proving the main theoretical result of this paper, which applies to all dimensions of the feature extractor. Alternative and weaker assumptions are discussed after the proof.

**Theorem 1** (Identifiable conditional EBMs). *Assume:*

1. *The feature extractor $\mathbf{f}$ is differentiable, and its Jacobian $\mathbf{J_f}$ is full rank.*

2. *There exist $n + 1$ points $\mathbf{y}^0, \dots, \mathbf{y}^n$ such that the matrix*
$$\mathbf{R} = \left(\mathbf{g}(\mathbf{y}^1) - \mathbf{g}(\mathbf{y}^0), \dots, \mathbf{g}(\mathbf{y}^n) - \mathbf{g}(\mathbf{y}^0)\right) \quad (17)$$
   *of size $n \times n$ is invertible.*

*then*
$$p_{\boldsymbol{\theta}}(\mathbf{x}|\mathbf{y}) = p_{\boldsymbol{\theta}'}(\mathbf{x}|\mathbf{y}) \implies \boldsymbol{\theta} \sim_w^{\mathbf{f}} \boldsymbol{\theta}'$$
*where $\sim_w^{\mathbf{f}}$ is defined as follows:*
$$\boldsymbol{\theta} \sim_w^{\mathbf{f}} \boldsymbol{\theta}' \Leftrightarrow \mathbf{f}_{\boldsymbol{\theta}}(\mathbf{y}) = \mathbf{A} \mathbf{f}_{\boldsymbol{\theta}'}(\mathbf{y}) + \mathbf{c} \quad (18)$$
$\mathbf{A}$ *is a $(d_z \times d_z)$-matrix of rank at least $\min(d_z, d_x)$.*

*If, instead or in addition, we assume that:*

3. *The feature extractor $\mathbf{g}$ is differentiable, and its Jacobian $\mathbf{J_g}$ is full rank.*

4. *There exist $n + 1$ points $\mathbf{x}^0, \dots, \mathbf{x}^n$ such that the matrix*
$$\mathbf{Q} = \left(\mathbf{f}(\mathbf{x}^1) - \mathbf{f}(\mathbf{x}^0), \dots, \mathbf{f}(\mathbf{x}^n) - \mathbf{f}(\mathbf{x}^0)\right)$$
   *of size $n \times n$ is invertible.*

*then*

$$p_{\boldsymbol{\theta}}(\mathbf{x}|\mathbf{y}) = p_{\boldsymbol{\theta}'}(\mathbf{x}|\mathbf{y}) \implies \boldsymbol{\theta} \sim_w^{\mathbf{g}} \boldsymbol{\theta}'$$

*where $\sim_w^{\mathbf{g}}$ is defined as follows:*

$$\boldsymbol{\theta} \sim_w^{\mathbf{g}} \boldsymbol{\theta}' \Leftrightarrow \mathbf{g}_{\boldsymbol{\theta}}(\mathbf{y}) = \mathbf{B}\mathbf{g}_{\boldsymbol{\theta}'}(\mathbf{y}) + \mathbf{e} \tag{19}$$

$\mathbf{B}$ *is a $(d_z \times d_z)$-matrix of rank at least $\min(d_z, d_x)$.*

*Finally, if $d_z \geq \max(d_x, d_y)$ and all assumptions 1- 4 hold, then the matrices $\mathbf{A}$ and $\mathbf{B}$ have full rank (equal to $d_z$).*

*Proof.* We will only prove this theorem for the feature extractor $\mathbf{f}$. The proof for $\mathbf{g}$ is very similar. Suppose assumptions 1 and 2 hold.

Consider two parameters $\boldsymbol{\theta}$ and $\tilde{\boldsymbol{\theta}}$ such that

$$p_{\boldsymbol{\theta}}(\mathbf{x}|\mathbf{y}) = p_{\tilde{\boldsymbol{\theta}}}(\mathbf{x}|\mathbf{y}) \tag{20}$$

Then, by applying the logarithm to both sides, we get:

$$\log Z(\mathbf{y}; \boldsymbol{\theta}) - \mathbf{f}_{\boldsymbol{\theta}}(\mathbf{x})^T \mathbf{g}_{\boldsymbol{\theta}}(\mathbf{y}) = \log Z(\mathbf{y}; \tilde{\boldsymbol{\theta}}) - \mathbf{f}_{\tilde{\boldsymbol{\theta}}}(\mathbf{x})^T \mathbf{g}_{\tilde{\boldsymbol{\theta}}}(\mathbf{y}) \tag{21}$$

Consider the points $\mathbf{y}^0, \dots, \mathbf{y}^n$ provided by assumption 2 for $\mathbf{g}_{\boldsymbol{\theta}}$. We plug each of these points in (21) to obtain $n + 1$ such equations. We subtract the first equation for $\mathbf{y}^0$ from the remaining $n$ equations, and write the resulting equations in matrix form:

$$\mathbf{R}\mathbf{f}_{\boldsymbol{\theta}}(\mathbf{x}) = \tilde{\mathbf{R}}\mathbf{f}_{\tilde{\boldsymbol{\theta}}}(\mathbf{x}) + \mathbf{b} \tag{22}$$

where $\mathbf{R} = (\dots, \mathbf{g}_{\boldsymbol{\theta}}(\mathbf{y}^l) - \mathbf{g}_{\boldsymbol{\theta}}(\mathbf{y}^0), \dots)$, $\tilde{\mathbf{R}} = (\dots, \mathbf{g}_{\tilde{\boldsymbol{\theta}}}(\mathbf{y}^l) - \mathbf{g}_{\tilde{\boldsymbol{\theta}}}(\mathbf{y}^0), \dots)$, and $\mathbf{b} = (\dots, \log \frac{Z(\mathbf{y}^l; \boldsymbol{\theta})}{Z(\mathbf{y}^l; \tilde{\boldsymbol{\theta}})} - \log \frac{Z(\mathbf{y}^0; \boldsymbol{\theta})}{Z(\mathbf{y}^0; \tilde{\boldsymbol{\theta}})}, \dots)$. Since $\mathbf{R}$ is invertible (by assumption 2), we multiply by its inverse from the left to get:

$$\mathbf{f}_{\boldsymbol{\theta}}(\mathbf{x}) = \mathbf{A}\mathbf{f}_{\tilde{\boldsymbol{\theta}}}(\mathbf{x}) + \mathbf{c} \tag{23}$$

where $\mathbf{A} = \mathbf{R}^{-1}\tilde{\mathbf{R}}$ and $\mathbf{c} = \mathbf{R}^{-1}\mathbf{b}$. Now since $\mathbf{f}_{\boldsymbol{\theta}}$ is differentiable and its Jacobian is full rank (assumption 1), by differentiating the last equation we deduce that $\text{rank}(\mathbf{A}) \geq \min(n, d)$, which in turn proves that $\boldsymbol{\theta} \sim_w^{\mathbf{f}} \tilde{\boldsymbol{\theta}}$.

Finally, suppose that in addition, assumptions 4 holds. Then there exists $\mathbf{x}^0, \dots \mathbf{x}^n$ such that $\mathbf{Q} := (\dots, \mathbf{f}_{\boldsymbol{\theta}}(\mathbf{x}^i) - \mathbf{f}_{\boldsymbol{\theta}}(\mathbf{x}^0), \dots)$. Plugging these $n + 1$ points into equation (23), and subtracting the first equation for $\mathbf{x}^0$ from the remaining $n$ equations, we get

$$\mathbf{Q} = \mathbf{A}(\dots, \mathbf{f}_{\tilde{\boldsymbol{\theta}}}(\mathbf{x}^i) - \mathbf{f}_{\tilde{\boldsymbol{\theta}}}(\mathbf{x}^0), \dots) \tag{24}$$

Since $\mathbf{Q}$ is an $n \times n$ invertible matrix, we conclude that $\mathbf{A}$ is also invertible, which concludes the proof. $\square$

**Intuition behind assumption 2** Assumption 2 requires that the conditioning feature extractor $\mathbf{g}$ has an image that is rich enough. Intuitively, this relaxes the amount of flexibility the main feature extractor $\mathbf{f}$ would need to have if $\mathbf{g}$ were to be very simple. It implies that the search for $\mathbf{f}$ will be naturally restricted to a smaller space, for which we can prove identifiability.

**Proof under weaker assumptions** Assumption 1 of full rank Jacobian can be weakened without changing the conclusion of Theorem 1. In fact, this assumption is only used right after equation (23) to prove that the matrix $\mathbf{A}$ has a rank that is at least equal to $\min(n, d)$. Suppose instead that

    1.' There exists a point $\mathbf{x}^0 \in \mathbb{R}^d$ where the Jacobian $\mathbf{J}_{\mathbf{f}_{\boldsymbol{\theta}}}$ of $\mathbf{f}_{\boldsymbol{\theta}}$ exists and is invertible

Then by computing the differential of equation (23) at $\mathbf{x}^0$ (assuming that $\mathbf{J}_{\mathbf{f}_{\tilde{\boldsymbol{\theta}}}}(\mathbf{x}^0)$ exists), we can make the same conclusion on the rank of $\mathbf{A}$.

In fact, this condition can be scrapped altogether if we relax the definition of the equivalence class in Appendix C.1 to have no conditions on the ranks of matrices $\mathbf{A}_1$ and $\mathbf{A}_2$. This however comes at the expense of a relatively weak, and potentially meaningless, equivalence class.

Finally, assumption 2 of Theorem 1 can be replaced by requiring the Jacobian of $\mathbf{g}_{\theta}$ to be differentiable and full rank in at least one point, but this requires the conditioning variable to be continuous.

### C.3 Proof of Proposition 1

**Proposition 1.** *Consider an MLP with $L$ layers, where each layer consists of a linear mapping with weight matrix $\mathbf{W}_l \in \mathbb{R}^{d_l \times d_{l-1}}$ and bias $\mathbf{b}_l$, followed by an activation function. Assume*

    *a. All activation functions are LeakyReLUs.*

    *b. All weight matrices $\mathbf{W}_l$ are full rank.*

    *c. The row dimension of the weight matrices are either monotonically increasing or decreasing: $d_l \geq d_{l+1}, \forall l \in [\![0, L-1]\!]$ or $d_l \leq d_{l+1}, \forall l \in [\![0, L-1]\!]$.*

*Then the MLP has a full rank Jacobian almost everywhere. If in addition, $d_L \leq d_0$, then the MLP is surjective.*

*Proof.* Denote by $\mathbf{x}$ the input to the MLP, and by $\mathbf{x}^l$ the output of layer $l$,

$$\mathbf{x}^0 = \mathbf{x} \tag{25}$$

$$\overline{\mathbf{x}}^l = \mathbf{W}_l \mathbf{x}^{l-1} + \mathbf{b}_l \tag{26}$$

$$\mathbf{x}^l = h(\mathbf{W}_l \mathbf{x}^{l-1} + \mathbf{b}_l) = h(\overline{\mathbf{x}}^l) \tag{27}$$

$$h(y) = \alpha y \mathbf{1}_{y<0} + y \mathbf{1}_{y>0} \tag{28}$$

with $h$ in equation (27) is applied to each element of its input, and $\alpha \in (0, 1)$.

Denote by $\mathbf{v}^l \in \mathbb{R}^{d_l}$ the vector whose elements are

$$v_k^l = h'(\overline{x}_k^l) = \begin{cases} 1 \text{ if } \overline{x}_k^l > 0 \\ \alpha \text{ if } \overline{x}_k^l < 0 \end{cases} \tag{29}$$

which is undefined if $\overline{x}_k^l = 0$, and by $\mathbf{V}_l = \mathrm{diag}(\mathbf{v}^l)$. Note that $\mathbf{V}_l$ is a function of its input, and thus of $\mathbf{x}$, but we keep this implicit for simplicity. Using these notations, and the fact that $h$ is piece-wise linear, we can write,

$$\mathbf{x}^L = h(\overline{\mathbf{x}}^L) = \mathbf{V}_L \overline{\mathbf{x}}^L = \mathbf{V}_L \mathbf{W}_L \mathbf{x}^{L-1} + \mathbf{V}_L \mathbf{b}_{L-1} = \cdots = \overline{\mathbf{V}}^L \mathbf{x} + \overline{\mathbf{b}}^L \tag{30}$$

where $\overline{\mathbf{V}}^l = \mathbf{V}_l \mathbf{W}_l \mathbf{V}_{l-1} \mathbf{W}_{l-1} \ldots \mathbf{V}_1 \mathbf{W}_1$, $\overline{\mathbf{b}}^0 = 0$ and $\overline{\mathbf{b}}^l = \mathbf{V}_l \mathbf{b}_l + \mathbf{V}_l \mathbf{W}_l \overline{\mathbf{b}}^{l-1}$. This is of course only possible if $\overline{x}_k^l \neq 0$ for all $l \in [\![1, L]\!]$ and all $k \in [\![1, d_l]\!]$. As such, define the set

$$\mathcal{N} = \bigcup_{l=1}^{L} \bigcup_{k=1}^{d_l} \left\{ \mathbf{x} \in \mathbb{R}^d | \overline{x}_k^l = 0 \right\} = \bigcup_{l=1}^{L} \bigcup_{k=1}^{d_l} \left\{ \mathbf{x} \in \mathbb{R}^d | (\overline{\mathbf{v}}_k^l)^T \mathbf{x} + \overline{b}_k^l = 0 \right\} \tag{31}$$

where $\overline{\mathbf{v}}_k^l$ is the $k$-th row of $\overline{\mathbf{V}}^l$. For each $\mathbf{x} \notin \mathcal{N}$, we have that $\mathbf{V}_l$ is full rank, and, using Lemma 2, $\overline{\mathbf{V}}^l$ is also a full rank matrix.

While it is true that $\overline{b}_k^l$ and $\overline{\mathbf{v}}_k^l$ are functions of $\mathbf{x}$, yet they only take a finite number of values. Thus, the set $\left\{ \mathbf{x} \in \mathbb{R}^d | (\overline{\mathbf{v}}_k^l)^T \mathbf{x} + \overline{b}_k^l = 0 \right\}$ is included in the union over all the values taken by $\overline{b}_k^j$ and $\overline{\mathbf{v}}_k^j$ up to layer $l$. For each of these values, the set becomes a dot product between a row of $\overline{\mathbf{V}}^j$ which is independent of the input $\mathbf{x}$, and is nonzero because $\overline{\mathbf{V}}^j$ is full rank; such set has measure zero in $\mathbb{R}^d$. Thus, $\mathcal{N}$ is included in a finite union of sets of measure zero, which implies that it also has measure zero.

Now, for all $\mathbf{x} \notin \mathcal{N}$, $\frac{\partial \mathbf{x}^L}{\partial \mathbf{x}}$ exists, and can be computed using the chain rule:

$$\frac{\partial \mathbf{x}^L}{\partial \mathbf{x}} = \prod_{l=L}^{1} \frac{\partial \mathbf{x}^l}{\partial \mathbf{x}^{l-1}} = \prod_{l=L}^{1} \frac{\partial \mathbf{x}^l}{\partial \overline{\mathbf{x}}^l} \frac{\partial \overline{\mathbf{x}}^l}{\partial \mathbf{x}^{l-1}} = \prod_{l=L}^{1} \mathbf{V}_l \mathbf{W}_l = \overline{\mathbf{V}}^L \tag{32}$$

which is full rank. Thus, the MLP has a full rank Jacobian almost everywhere.

The surjectivity is easy to prove since $h$ is surjective and so is $\overline{\mathbf{x}}^l$ as a function of $\mathbf{x}^{l-1}$ if $d_{l-1} \geq d_l$ and $\mathrm{rank}(\mathbf{W}_l) = d_l$. $\qquad\square$

**Lemma 1.** *Denote by $\sigma_{min}(\mathbf{A})$ the smallest singular value of a matrix $\mathbf{A}$. Let $\mathbf{M}$ be an $m \times n$ matrix, and $\mathbf{N}$ be an $n \times p$ matrix, such that $m \leq n \leq p$ or $m \geq n \geq p$. Then $\sigma_{min}(\mathbf{MN}) \geq \sigma_{min}(\mathbf{M})\sigma_{min}(\mathbf{N})$.*

*Proof.* The proof in the case $m \geq n \geq p$ can be found in (Arbel et al., 2018, Lemma 10), but we provide a proof here for completeness, and for the other case $m \leq n \leq p$.

Let $\mathbb{R}^n_* := \mathbb{R}^n \setminus \{0\}$, and $\lambda_{min}(\mathbf{A})$ the smallest eigenvalue of $\mathbf{A}$. Recall that for a matrix $\mathbf{A} \in \mathbb{R}^{n \times m}$, with $m \geq n$,

$$\sigma_{\min}(\mathbf{A}) = \sqrt{\lambda_{\min}(\mathbf{A}^T\mathbf{A})} = \sqrt{\inf_{\mathbf{x} \in \mathbb{R}^n_*} \frac{\mathbf{x}^T\mathbf{A}^T\mathbf{A}\mathbf{x}}{\mathbf{x}^T\mathbf{x}}} = \inf_{\mathbf{x} \in \mathbb{R}^n_*} \frac{\|\mathbf{A}\mathbf{x}\|}{\|\mathbf{x}\|} \tag{33}$$

Thus, if the null space of $\mathbf{N}$ is non trivial, then $\sigma_{\min}(\mathbf{N}) = 0$, and the inequality is satisfied. Otherwise, we have $\mathbf{Nx} \neq 0$, $\forall \mathbf{x} \in \mathbb{R}^n_*$,

$$\begin{aligned}
\sigma_{\min}(\mathbf{MN}) &= \inf_{\mathbf{x} \in \mathbb{R}^p_*} \frac{\|\mathbf{MNx}\|}{\|\mathbf{x}\|} \\
&= \inf_{\mathbf{x} \in \mathbb{R}^p_*} \frac{\|\mathbf{MNx}\| \, \|\mathbf{Nx}\|}{\|\mathbf{Nx}\| \, \|\mathbf{x}\|} \\
&\geq \left( \inf_{\mathbf{x} \in \mathbb{R}^p_*} \frac{\|\mathbf{MNx}\|}{\|\mathbf{Nx}\|} \right) \left( \inf_{\mathbf{x} \in \mathbb{R}^p_*} \frac{\|\mathbf{Nx}\|}{\|\mathbf{x}\|} \right) \\
&\geq \left( \inf_{\mathbf{x} \in \mathbb{R}^n_*} \frac{\|\mathbf{Mx}\|}{\|\mathbf{x}\|} \right) \left( \inf_{\mathbf{x} \in \mathbb{R}^p_*} \frac{\|\mathbf{Nx}\|}{\|\mathbf{x}\|} \right) \\
&= \sigma_{\min}(\mathbf{M})\sigma_{\min}(\mathbf{N})
\end{aligned}$$

If, instead, $\mathbf{A} \in \mathbb{R}^{m \times n}$ with $m \leq n$, then

$$\sigma_{\min}(\mathbf{A}) = \sqrt{\lambda_{\min}(\mathbf{A}\mathbf{A}^T)} = \sqrt{\inf_{\mathbf{x} \in \mathbb{R}^m_*} \frac{\mathbf{x}^T\mathbf{A}\mathbf{A}^T\mathbf{x}}{\mathbf{x}^T\mathbf{x}}} = \inf_{\mathbf{x} \in \mathbb{R}^m_*} \frac{\|\mathbf{A}^T\mathbf{x}\|}{\|\mathbf{x}\|} \tag{34}$$

Similarly, if the null space of $\mathbf{M}^T$ is non trivial, then $\sigma_{\min}(\mathbf{M}^T) = \sigma_{\min}(\mathbf{M}) = 0$, and the inequality holds. Otherwise, we have $\mathbf{M}^T\mathbf{x} \neq 0$, $\forall \mathbf{x} \in \mathbb{R}^m_*$,

$$\begin{aligned}
\sigma_{\min}(\mathbf{MN}) &= \inf_{\mathbf{x} \in \mathbb{R}^m_*} \frac{\|\mathbf{N}^T\mathbf{M}^T\mathbf{x}\|}{\|\mathbf{x}\|} \\
&= \inf_{\mathbf{x} \in \mathbb{R}^m_*} \frac{\|\mathbf{N}^T\mathbf{M}^T\mathbf{x}\| \, \|\mathbf{M}^T\mathbf{x}\|}{\|\mathbf{M}^T\mathbf{x}\| \, \|\mathbf{x}\|} \\
&\geq \left( \inf_{\mathbf{x} \in \mathbb{R}^m_*} \frac{\|\mathbf{N}^T\mathbf{M}^T\mathbf{x}\|}{\|\mathbf{M}^T\mathbf{x}\|} \right) \left( \inf_{\mathbf{x} \in \mathbb{R}^m_*} \frac{\|\mathbf{M}^T\mathbf{x}\|}{\|\mathbf{x}\|} \right) \\
&\geq \left( \inf_{\mathbf{x} \in \mathbb{R}^n_*} \frac{\|\mathbf{N}^T\mathbf{x}\|}{\|\mathbf{x}\|} \right) \left( \inf_{\mathbf{x} \in \mathbb{R}^m_*} \frac{\|\mathbf{M}^T\mathbf{x}\|}{\|\mathbf{x}\|} \right) \\
&= \sigma_{\min}(\mathbf{N})\sigma_{\min}(\mathbf{M})
\end{aligned}$$

which concludes the proof. □

**Lemma 2.** *Consider a finite sequence of matrices $(\mathbf{M}_i)_{1 \leq i \leq p}$, with $\mathbf{M}_i \in \mathbb{R}^{n_{i-1} \times n_i}$. If $\mathbf{M}_i$ is full rank for all $i \in [\![1, p]\!]$, and either $n_0 \leq n_1 \leq \ldots \leq n_p$ or $n_0 \geq n_1 \geq \ldots \geq n_p$, then the product $\mathbf{M}_1\mathbf{M}_2 \ldots \mathbf{M}_p$ is also full rank.*

*Proof.* If two matrices $\mathbf{M}_1$ and $\mathbf{M}_2$ with ordered dimensions are full rank, then $\sigma_{\min}(\mathbf{M}_1) > 0$ and $\sigma_{\min}(\mathbf{M}_2) > 0$. According to Lemma 1, this implies that $\sigma_{\min}(\mathbf{M}_1\mathbf{M}_2) > 0$, and that $\mathbf{M}_1\mathbf{M}_2$ is full rank. The proof for $p \geq 3$ is done by induction on $p$. □

## C.4 Proof of Proposition 2

**Linear MLPs** The particular case of linear feature extractors is quite interesting. If $d_z \leq d_y$ and the feature extractor $\mathbf{g}$ satisfies the assumptions of Proposition 1, then assumption 2 is trivially satisfied.

On the other hand, if $d_z > d_y$, then assumption 2 can't hold when the network is linear. This signals that it is important to use *deep* nonlinear networks to parameterize the feature extractors, at least in the overcomplete case.

**Proposition 2.** *Consider an MLP* $\mathbf{g}$ *with $L$ layers, where each layer consists of a linear mapping with weight matrix $\mathbf{W}_l \in \mathbb{R}^{d_l \times d_{l-1}}$ and bias $\mathbf{b}_l$, followed by an activation function. Assume*

    *a. All activation functions are LeakyReLUs.*

    *b. All weight matrices $\mathbf{W}_l$ are full rank.*

    *c. All submatrices of $\mathbf{W}_l$ of size $d_l \times d_l$ are invertible if $d_l < d_{l+1}$.*

*Then there exist $d_L + 1$ points $\mathbf{y}^0, \ldots, \mathbf{y}^{d_L}$ such that the matrix $\mathbf{R} = \left(\mathbf{g}(\mathbf{y}^1) - \mathbf{g}(\mathbf{y}^0), \ldots, \mathbf{g}(\mathbf{y}^{d_L}) - \mathbf{g}(\mathbf{y}^0)\right)$ is invertible.*

*Proof.* Let $\mathbf{y}^0$ be an arbitrary point in $\mathbb{R}^{d_0}$. Without loss of generality, suppose that $\mathbf{g}(\mathbf{y}^0) = 0$. This is because $\mathbf{y} \mapsto \mathbf{g}(\mathbf{y}) - \mathbf{g}(\mathbf{y}^0)$ is still an MLP that satisfies all the assumptions above. If for any choice of points $\mathbf{y}^1$ to $\mathbf{y}^{d_L}$, the matrix $\mathbf{R}$ defined above isn't invertible, then this means that $\mathbf{g}(\mathbb{R}^{d_0})$ is necessarily included in a subspace of $\mathbb{R}^{d_L}$ of dimension at most $d_L - 1$. In other words, this would imply that the functions $g_1, \ldots, g_{d_L}$ are not linearly independent. However, this is in contradiction with the result of Lemma 8, which stipulates that $g_1, \ldots, g_{d_L}$ are linearly independent, provided all weight matrices satisfy the assumptions of the lemma (which are the same as the assumptions made in this proposition).

Thus, we can conclude that there exist $d_L + 1$ points $\mathbf{y}^0, \ldots, \mathbf{y}^{d_L}$ such that the matrix $\mathbf{R} = \left(\mathbf{g}(\mathbf{y}^1) - \mathbf{g}(\mathbf{y}^0), \ldots, \mathbf{g}(\mathbf{y}^{d_L}) - \mathbf{g}(\mathbf{y}^0)\right)$ is invertible. $\qquad\square$

**Proof under weaker conditions**    Note that the proof argument used for the overcomplete case can be used for the undercomplete as well. This same argument can be proved for ReLU as the nonlinearity instead of LeakyReLU. We chose to give the proof for, and suggest to use the latter because it is needed for Proposition 1.

**Lemma 3.** *Let $\mathbf{A}$ be an $n \times n$ invertible matrix. Denote by $\mathbf{a}_n$ the $n$-th row of $\mathbf{A}$. Then the matrix $\mathbf{B} \in \mathbb{R}^{n+1, n+1}$ such that*

$$\mathbf{B} = \left(\begin{array}{c|c} \mathbf{A} & \begin{array}{c} \gamma_1 \\ \vdots \\ \gamma_{n-1} \\ \lambda \end{array} \\ \hline \mathbf{a}_n & 1 \end{array}\right) \tag{35}$$

*is invertible for any choice of $\gamma_1, \ldots, \gamma_{n-1}$, and for $\lambda \neq 1$.*

*Proof.* Denote by $\mathbf{b}_i$ the $i$-th row of $\mathbf{B}$. Let $\alpha_1, \ldots, \alpha_{n+1}$ such that

$$\sum_{i=1}^{n+1} \alpha_i \mathbf{b}_i = 0 \tag{36}$$

Then in particular, by looking at the first $n$ lines of this vectorial equation, we have that $\sum_{i=1}^{n-1} \alpha_i \mathbf{a}_i + (\alpha_n + \alpha_{n+1})\mathbf{a}_n = 0$. Since $\mathbf{A}$ is invertible, its rows are linearly independent, and thus $\alpha_n = -\alpha_{n+1}$ and $\alpha_i = 0, \forall i < n$. Plugging this back into equation (36), and looking closely at the last equation, we have that $(1 - \lambda)\alpha_n = 0$, and we conclude that $\alpha_{n+1} = \alpha_n = 0$ (because $\lambda \neq 1$), and that $\mathbf{B}$ is invertible. $\qquad\square$

**Lemma 4.** *Consider $n$ affine functions $f_i : \mathbf{x} \in \mathbb{R}^d \mapsto \mathbf{a}_i^T \mathbf{x} + b_i$, such that the matrix $\mathbf{A} \in \mathbb{R}^{n \times d}$ whose rows are the $\mathbf{a}_i$ is full column rank, and all its submatrices of size $d \times d$ are invertible if $d < n$. Then there exist $n$ non-empty regions $\mathcal{H}_1, \ldots, \mathcal{H}_n$ of $\mathbb{R}^d$ defined by the signs of the functions $f_i$ (for instance, $\mathcal{H} = \{\mathbf{x} \in \mathbb{R}^n | \forall i, f_i(\mathbf{x}) > 0\}$) such that the matrix $\mathbf{S}^n \in \mathbb{R}^{n \times n}$ defined as $S_{i,j}^n = \text{sign}_{\mathbf{x} \in \mathcal{H}_i}(f_j(\mathbf{x}))$ is invertible.*

*Proof.* We will prove this Lemma by induction on $n$ the number of functions $f_i$. Denote by $V_i = \{\mathbf{x} \in \mathbb{R}^d | f_i(\mathbf{x}) = 0\}$. The sign of $f_i$ changes if we cross the hyperplan $V_i$.

First, suppose that $n = 2$. By assumption, we now that $\mathbf{a}_1 \not\propto \mathbf{a}_2$, and thus the hyperplans $V_1$ and $V_2$ are not parallel and divide $\mathbb{R}^d$ into 4 regions. This implies that the regions $\mathcal{H}_1 = \{\mathbf{x} \in \mathbb{R}^d | \mathbf{a}_1^T \mathbf{x} + b_1 > 0, \mathbf{a}_2^T \mathbf{x} + b_2 > 0\}$ and $\mathcal{H}_2 = \{\mathbf{x} \in \mathbb{R}^d | \mathbf{a}_1^T \mathbf{x} + b_1 > 0, \mathbf{a}_2^T \mathbf{x} + b_2 < 0\}$ are not empty.

Second, suppose that there exists $n$ regions $\mathcal{H}_1, \dots \mathcal{H}_n$ such that the the matrix $\mathbf{S}^n$ is invertible. Consider the affine function $f_{n+1} = \mathbf{a}_{n+1}^T \mathbf{x} + b_{n+1}$. The hyperplan $V_{n+1} = \{\mathbf{x} \in \mathbb{R}^d | f_{n+1}(\mathbf{x}) = 0\}$ intersects at least one of the regions $\mathcal{H}_1, \dots \mathcal{H}_n$. This is because $(\dots, \mathbf{a}_i, \dots)_{i \in J}$ are linearly independent for any $J$ of size $\min(d, n+1)$ such that $n + 1 \in J$, and thus there exists $i_0$ such that $\mathbf{a}_{n+1} \not\propto \mathbf{a}_{i_0}$. Suppose without loss of generality that this region is $\mathcal{H}_n$. Denote by $\tilde{\mathcal{H}}_n = \{\mathbf{x} \in \mathbb{R}^n | \mathbf{x} \in \mathcal{H}_n, f_{n+1}(\mathbf{x}) < 0\} \subset \mathcal{H}_n$. Now consider the matrix $\tilde{\mathbf{S}}^n$ such that $\tilde{S}_{n,j}^n = \mathrm{sign}_{\mathbf{x} \in \tilde{\mathcal{H}}_n}(f_j(\mathbf{x}))$ and $\tilde{S}_{i,j}^n = S_{i,j}^n$. Because $\tilde{\mathcal{H}}_n \subset \mathcal{H}_n$, we have that $\mathrm{sign}_{\mathbf{x} \in \mathcal{H}_n}(f_j(\mathbf{x})) = \mathrm{sign}_{\mathbf{x} \in \tilde{\mathcal{H}}_n}(f_j(\mathbf{x}))$ and thus $\tilde{\mathbf{S}}^n = \mathbf{S}^n$, which implies that $\tilde{\mathbf{S}}^n$ is also invertible. Now define $\mathcal{H}_{n+1} = \{\mathbf{x} \in \mathbb{R}^n | \mathbf{x} \in \mathcal{H}_n, f_{n+1}(\mathbf{x}) > 0\} \subset \mathcal{H}_n$. Again, the inclusion implies that $\mathrm{sign}_{\mathbf{x} \in \mathcal{H}_n}(f_j(\mathbf{x})) = \mathrm{sign}_{\mathbf{x} \in \tilde{\mathcal{H}}_n}(f_j(\mathbf{x}))$. Finally, consider the regions $\mathcal{H}_1, \dots, \mathcal{H}_{n-1}, \tilde{\mathcal{H}}_n, \mathcal{H}_{n+1}$, and the matrix $\mathbf{S}^{n+1}$ defined on those regions. Then

$$\mathbf{S}^{n+1} = \begin{pmatrix} & & & u_1 \\ & \mathbf{S}^n & & \vdots \\ & & & u_{n-1} \\ & & & -1 \\ \hline & \mathbf{s}_n^n & & 1 \end{pmatrix} \tag{37}$$

where $u_i = \mathrm{sign}_{\mathbf{x} \in \mathcal{H}_i} f_{n+1}(\mathbf{x})$ and $\mathbf{s}_n^n$ is the $n$-th line of $\mathbf{S}^n$. According to Lemma 3, $\mathbf{S}^{n+1}$ is invertible, which achieves the proof. □

**Lemma 5.** *Let $h$ denote a LeakyReLU activation function with slope $\lambda \in [0, 1)$ (if $\lambda = 0$, then $h$ is simply a ReLU). Consider $n$ piece-wise affine functions $g_i : \mathbf{x} \in \mathbb{R}^d \mapsto h(\mathbf{a}_i^T \mathbf{x} + b_i)$, such that the matrix $\mathbf{A} \in \mathbb{R}^{n \times d}$ whose rows are the $\mathbf{a}_i$ is full column rank, and all its submatrices of size $d \times d$ are invertible if $d < n$. Then the functions $g_1, \dots, g_n$ are linearly independent, and their generalized slopes (as piece-wise affine functions) are also linearly independent.*

*Proof.* Let $f_i = \mathbf{a}_i^T \mathbf{x} + b_i$ such that $g_i = h(f_i) = \mathbf{1}_{f_i \geq 0} f_i + \mathbf{1}_{f_i < 0} \lambda f_i$.

The assumptions of Lemma 4 are met for the function $f_1, \dots, f_n$, and we conclude that there exists n regions $\mathcal{H}_1, \dots, \mathcal{H}_n$ such that $\mathbf{S}^n = \left( \mathrm{sign}_{\mathbf{x} \in \mathcal{H}_i}(f_j(\mathbf{x})) \right)_{i,j}$ is invertible. Define the matrix $\tilde{\mathbf{S}}$ where we replace all entries of $\mathbf{S}^n$ by $\lambda$ if they are equal to $-1$. Then $\tilde{\mathbf{S}}$ is invertible (in fact, to see this, consider the proof of the previous lemma with the slightly unconventional choice of sign function $\mathrm{sign}(x) = \lambda$ if $x < 0$).

Now consider $\alpha_1, \dots, \alpha_n$ such that

$$\sum_{i=1}^n \alpha_i g_i = 0 \tag{38}$$

Let $k \in [\![1, n]\!]$, and evaluate this equation at $\mathbf{x} \in \mathcal{H}_k$. After taking the gradient with respect to $\mathbf{x}$, we get

$$\sum_i (\mathbf{1}_{\mathbf{x} \in \mathcal{H}_k, f_i(\mathbf{x}) \geq 0} + \lambda \mathbf{1}_{\mathbf{x} \in \mathcal{H}_k, f_i(\mathbf{x}) < 0}) \alpha_i \mathbf{a}_i = 0 \tag{39}$$

Denote by $\tilde{\mathbf{s}}_k$ the $k$-th line of the matrix $\tilde{\mathbf{S}}$, and define $\mathbf{e}_l = (\alpha_1 a_{1,l}, \dots, \alpha_n a_{n,l}) \in \mathbb{R}^n$. We can write the $l$-th line of equation (39) as:

$$\tilde{\mathbf{s}}_k^T \mathbf{e}_l = 0 \tag{40}$$

Collating these equations for a fixed $l$ and $k \in [\![1, n]\!]$, we get

$$\tilde{\mathbf{S}} \mathbf{e}_l = 0 \tag{41}$$

which implies that $\mathbf{e}_l = 0$ because $\mathbf{S}$ is invertible. In particular, $\alpha_i a_{i,l} = 0$ for all $i \in [\![1, n]\!]$ and $l \in [\![1, d]\!]$. This implies that $\mathbf{A}_J^T \boldsymbol{\alpha}_J = 0$, where $J \subset [\![1, n]\!]$ of size $\min(n, d)$, $\mathbf{A}_J = (a_{i,l})_{i \in J, l \in [\![1, d]\!]} \in \mathbb{R}^{d \times d}$ is a submatrix of $\mathbf{A}$ and $\boldsymbol{\alpha}_J = (\alpha_i)_{i \in J} \in \mathbb{R}^d$. Since we know, by assumption, that $\mathbf{A}_J$ is invertible for any choice of set of indices $J$ (relevant when $n > d$), we conclude that $\boldsymbol{\alpha} = 0$ and that the functions $g_1, \dots, g_n$ are linearly independent.

Each function $g_i$ is a piece-wise affine function, with a "generalized slope" equal to $\tilde{\mathbf{a}}_i(\mathbf{x}) = (\mathbf{1}_{f_i \geq 0}(\mathbf{x}) + \lambda \mathbf{1}_{f_i < 0}(\mathbf{x}))\mathbf{a}_i$. As a corollary of the independence of $g_1, \ldots g_n$, we can conclude that the slopes $\tilde{\mathbf{a}}_1(\mathbf{x}), \ldots, \tilde{\mathbf{a}}_n(\mathbf{x})$ are also independent. $\qquad\square$

**Lemma 6.** *Let* $\mathbf{f} = (f_1, \ldots, f_n)$ *be a vector-valued function defined on* $\mathbb{R}^d$. *We suppose that* $f_1, \ldots, f_n$ *are linearly independent piece-wise affine functions, and that their generalized slopes* $\mathbf{a}_1(\mathbf{x}), \ldots, \mathbf{a}_n(\mathbf{x})$ *are also linearly independent. Consider* $m$ *piece-wise affine functions* $g_i : \mathbf{x} \in \mathbb{R}^d \mapsto \mathbf{c}_i^T \mathbf{f}(\mathbf{x}) + d_i$, *such that the matrix* $\mathbf{C} \in \mathbb{R}^{m \times n}$ *whose rows are the* $\mathbf{c}_i$ *is full column rank, and all its submatrices of size* $n \times n$ *are invertible if* $n < m$. *Then there exist* $m$ *non-empty regions* $\mathcal{K}_1, \ldots, \mathcal{K}_m$ *of* $\mathbb{R}^d$ *defined by the signs of the functions* $g_i$ *such that the matrix* $\mathbf{T}^m \in \mathbb{R}^{m \times m}$ *defined as* $T_{i,j}^m = \mathrm{sign}_{\mathbf{x} \in \mathcal{K}_i}(g_j(\mathbf{x}))$ *is invertible.*

*Proof.* Denote by $\tilde{\mathbf{c}}_i(\mathbf{x})$ the generalized slope of the p.w. affine function $g_i$: $\tilde{\mathbf{c}}_i(\mathbf{x}) = \sum_j c_{i,j} \mathbf{a}_j(\mathbf{x})$. The key is to show than under the assumptions made here, the slopes $(\ldots, \tilde{\mathbf{c}}_i(\mathbf{x}), \ldots)_{i \in J}$ are linearly independent for any choice of subset $J \subset [\![1, m]\!]$ of size $\min(m, n)$.

If $m > n$, chose a subset $J \in [\![1, m]\!]$ of size $n$, and let $(\alpha_i)_{i \in J}$ such that $\sum_{i \in J} \alpha_i \tilde{\mathbf{c}}_i(\mathbf{x}) = 0$. By replacing $\tilde{\mathbf{c}}_i$ by its expression, we get: $\sum_j (\sum_i \alpha_i c_{i,j}) \mathbf{a}_j(\mathbf{x}) = 0$. Since $\mathbf{a}_1, \ldots, \mathbf{a}_n$ are linearly independent, we conclude that $\sum_{i \in J} \alpha_i c_{i,j} = 0$ for all $j \in [\![1, n]\!]$. This, along with the full rank assumption on $\mathbf{C}$ prove that $(\alpha_i)_{i \in J} = 0$ and that $(\ldots, \tilde{\mathbf{c}}_i(\mathbf{x}), \ldots)_{i \in J}$ are linearly independent. We can use the same argument if, instead, $m \leq n$, where $J = [\![1, m]\!]$, and conclude.

The rest of the proof follows the same argument of the proof of Lemma 4: we proceed by induction on $m$. For $m = 2$, we know that $\tilde{\mathbf{c}}_1 \not\propto \tilde{\mathbf{c}}_2$, and so the "generalized hyperplans" defined by these two vectors divide $\mathbb{R}^d$ into at least 3 different regions, 2 of which yield a matrix $\mathbf{T}^2$ that is invertible. Then, if the result hold for $m$, then the hyperplan defined by the generalized slope of the $(m+1)$-th p.w. affine function $g_{m+1}$ necessarily intersects one of the regions $\mathcal{K}_1, \ldots, \mathcal{K}_m$ since for any subset $J$ of size $\min(m+1, n)$ s.t. $(m+1) \in J$, the generalized slopes $(\ldots, \tilde{\mathbf{c}}_i(\mathbf{x}), \ldots)_{i \in J}$ are linearly independent. The rest is identical to Lemma 4. $\qquad\square$

**Lemma 7.** *Let* $h$ *denote a LeakyReLU activation function with slope* $\lambda \in [0, 1)$ *(if* $\lambda = 0$, *then* $h$ *is simply a ReLU), and* $\mathbf{f} = (f_1, \ldots, f_n)$ *be a vector-valued function defined on* $\mathbb{R}^d$. *We suppose that* $f_1, \ldots, f_n$ *are linearly independent piece-wise affine functions, and that their generalized slopes* $\mathbf{a}_1(\mathbf{x}), \ldots, \mathbf{a}_n(\mathbf{x})$ *are also linearly independent. Consider* $m$ *piece-wise affine functions* $g_i : \mathbf{x} \in \mathbb{R}^d \mapsto h(\mathbf{c}_i^T \mathbf{f}(\mathbf{x}) + d_i)$, *such that the matrix* $\mathbf{C} \in \mathbb{R}^{m \times n}$ *whose rows are the* $\mathbf{c}_i$ *is full column rank, and all its submatrices of size* $n \times n$ *are invertible if* $n < m$. *Then the functions* $g_1, \ldots, g_m$ *are linearly independent, and their generalized slopes are also linearly independent.*

*Proof.* Let $\tilde{g}_i = \mathbf{c}_i^T \mathbf{f} + d_i$ such that $g_i = h(\tilde{g}_i)$. The assumptions of Lemma 6 are met for the functions $\tilde{g}_1, \ldots, \tilde{g}_m$, and we conclude that there exists m regions $\mathcal{K}_1, \ldots, \mathcal{K}_m$ such that $\mathbf{T}^m = \left(\mathrm{sign}_{\mathbf{x} \in \mathcal{K}_i}(\tilde{g}_j(\mathbf{x}))\right)_{i,j}$ is invertible. Let $\tilde{\mathbf{T}}$ the invertible matrix equal to $\mathbf{T}^m$ after substituting $-1$ for $\lambda$.

Now consider $\alpha_1, \ldots, \alpha_m$ such that $\sum_{i=1}^m \alpha_i g_i = 0$ After taking the gradient with respect to $\mathbf{x}$, we get:

$$\sum_j (\sum_i \alpha_i (\mathbf{1}_{\tilde{g}_i \geq 0}(\mathbf{x}) + \lambda \mathbf{1}_{\tilde{g}_i < 0}(\mathbf{x})) c_{i,j}) \alpha_j(\mathbf{x}) = 0 \tag{42}$$

Since $\mathbf{a}_1, \ldots, \mathbf{a}_n$ are independent, we conclude that $\sum_i \alpha_i (\mathbf{1}_{\tilde{g}_i \geq 0}(\mathbf{x}) + \lambda \mathbf{1}_{\tilde{g}_i < 0}(\mathbf{x})) c_{i,j}$ for all $j \in [\![1, m]\!]$. This in turn implies that

$$\sum_i \alpha_i (\mathbf{1}_{\tilde{g}_i \geq 0}(\mathbf{x}) + \lambda \mathbf{1}_{\tilde{g}_i < 0}(\mathbf{x})) \mathbf{c}_i = 0 \tag{43}$$

Let $k \in [\![1, m]\!]$, and evaluate the last equation at $\mathbf{x} \in \mathcal{K}_k$:

$$\sum_i (\mathbf{1}_{\mathbf{x} \in \mathcal{H}_k, f_i(\mathbf{x}) \geq 0} + \lambda \mathbf{1}_{\mathbf{x} \in \mathcal{H}_k, f_i(\mathbf{x}) < 0}) \alpha_i \mathbf{c}_i = 0 \tag{44}$$

This last equation is similar to equation (39), and we can use the same argument used for the proof of Lemma 5 here (using $\tilde{\mathbf{T}}$ instead of $\tilde{\mathbf{S}}$) and deduce that $\alpha_i = 0$ for all $i$.

We conclude that $g_1, \ldots, g_m$ are linearly independent, and so are their generalized slopes as a consequence. $\qquad\square$

**Lemma 8.** *Let $\mathbf{f}^L = (f_1^L, \ldots, f_{d_L}^L)$ be the output of an L-layer MLP (we assume that $L \geq 2$: there is at least one nonlinearity) that satisfies:*

(a.) *All activation functions are LeakyReLUs with slope $\lambda \in [0, 1)$ (if $\lambda = 0$, then the activation function is simply a ReLU).*

(b.) *All weight matrices $\mathbf{W}_l \in \mathbb{R}^{d_{l+1} \times d_l}$ are full rank, and all submatrices of $\mathbf{W}_l$ of size $d_l \times d_l$ are invertible if $d_l < d_{l+1}$.*

*Then $f_1^L, \ldots, f_{d_L}^L$ are linearly independent. In addition, all the intermediate features $(f_1^l, \ldots, f_{d_l}^l)$ are also linearly independent.*

*Proof.* We prove the Lemma by induction on the number of layers $L \geq 2$. If $L = 2$, then by Lemma 5, we conclude that $f_1, \ldots, f_n$ are independent. If we suppose the result hold for $L \geq 2$, we can use Lemma 7 to prove that it also holds for $L + 1$. Finally, since all layers satisfy the same conditions, the conclusion also applies to intermediate layers. $\square$

## C.5 Proof of Theorem 2

We will decompose Theorem 2 into two sub-theorems, which will make the proof easier to understand, but also more adaptable into future work. Each of these sub-theorems corresponds to one of the assumptions.

### C.5.1 Positive features

We will prove here a more general version where we assume that each component $f_i$ of the feature extractor $\mathbf{f}$ has a global minimum that is reached, instead of being necessarily non-negative.

**Theorem 2a.** *Assume the assumptions of Theorem 1 hold. Further assume that $n \leq d$, and that each $f_i$ has a global minimum that is reached at least in the limit, and the feature extractor $\mathbf{f} = (f_1, \ldots, f_n)$ is surjective onto the set that is defined by the lower bounds of the $f_i$. Then*

$$p_{\boldsymbol{\theta}}(\mathbf{x}|\mathbf{y}) = p_{\boldsymbol{\theta}'}(\mathbf{x}|\mathbf{y}) \implies \boldsymbol{\theta} \sim_s \boldsymbol{\theta}'$$

*where $\sim_s$ is defined as follows:*

$$\boldsymbol{\theta} \sim_s \boldsymbol{\theta}' \Leftrightarrow \forall i, f_{i,\boldsymbol{\theta}}(\mathbf{x}) = a_i f_{\sigma(i),\boldsymbol{\theta}'}(\mathbf{x}) + b_i \tag{45}$$

*where $\sigma$ is a permutation of $[\![1, n]\!]$, $a_i$ is a non zero scalar and $b_i$ is a scalar.*

*Proof.* Consider two different parameters $\boldsymbol{\theta}$ and $\tilde{\boldsymbol{\theta}}$ such that:

$$p_{\boldsymbol{\theta}}(\mathbf{x}|\mathbf{y}) = p_{\tilde{\boldsymbol{\theta}}}(\mathbf{x}|\mathbf{y}) \tag{46}$$

To simplify notations, denote by $\mathbf{f} = \mathbf{f}_{\boldsymbol{\theta}}$ and $\tilde{\mathbf{f}} = \mathbf{f}_{\tilde{\boldsymbol{\theta}}}$. We start the proof from the conclusion of Theorem 1, since its assumptions hold:

$$\mathbf{f}(\mathbf{x}) = \mathbf{A}\tilde{\mathbf{f}}(\mathbf{x}) + \mathbf{c} \tag{47}$$

where $\mathbf{A}$ is an invertible $n \times n$ matrix and $\mathbf{c}$ a constant vector. Without loss of generality, we can suppose that $f_i$ has an infimum equal to zero, simply by subtracting $\inf f_i$, and including in $\mathbf{c}$, and similarly for $\tilde{\mathbf{f}}$. We will also suppose that the infima are reached, as the next argument would hold if we change exact minima by limits.

Now since $\mathbf{f} \geq 0$ and is surjective, then there exists $\mathbf{x}_0 \in \mathbb{R}^d$ such that $\mathbf{f}(\mathbf{x}_0) = 0$. This implies that $\mathbf{c} = -\mathbf{A}\tilde{\mathbf{f}}(\mathbf{x}_0)$, and that $\mathbf{f}(\mathbf{x}) = \mathbf{A}(\tilde{\mathbf{f}}(\mathbf{x}) - \tilde{\mathbf{f}}(\mathbf{x}_0))$. Define $\mathbf{h}(\mathbf{x}) = \tilde{\mathbf{f}}(\mathbf{x}) - \tilde{\mathbf{f}}(\mathbf{x}_0)$. We know that $\tilde{\mathbf{f}} \geq 0$ and is surjective, and so $\mathbf{h}$ is also surjective, and its image includes $\mathbb{R}_+^n$. Let $\mathbf{I} = (\mathbf{e}_1, \ldots, \mathbf{e}_n)$ be the matrix of canonical basis vectors, or positive scalar multiples of the canonical basis vectors $\mathbf{e}_i$. These must be mapped to the non-negative quadrant, so $\mathbf{A}\mathbf{I}$ must be non-negative, which implies that $\mathbf{A}$ must be non-negative.

Denote by $\mathbf{B} = \mathbf{A}^{-1}$. $\mathbf{B}$ is also non-negative for the same reasons described above. Denote the **rows** of $\mathbf{A}$ by $\mathbf{a}_i$ and the **columns** of $\mathbf{B}$ by $\mathbf{b}_j$. We have by definition of inverse:

$$\mathbf{a}_i^T \mathbf{b}_j = \delta_{ij} \tag{48}$$

where if $i = j$ then $\delta_{ij} = 1$, else $\delta_{ij} = 0$. Now, assume there is a row $\mathbf{a}_k$ which has at least two non-zero entries. By the property above, $d - 1$ of the vectors $\mathbf{b}_j$ must have zero dot-product with that vector. By non-negativity of $\mathbf{B}$ and $\mathbf{A}$, those $d - 1$ vectors must have zeros in the at least two indices corresponding to the non-zeros of $\mathbf{a}_k$. But that means they can only span a $d - 2$-dimensional subspace, and all the $\mathbf{b}_j$ together can only span a $d - 1$-dimensional subspace. This is in contradiction of the invertibility of $\mathbf{B}$. Thus, each $\mathbf{a}_i$ can have only one non-zero entry, which, together with the invertibility of $\mathbf{A}$, proves it is a scaled permutation matrix.

Thus, there exists a permutation $\sigma$ of $[\![1, n]\!]$, such that $f_i(\mathbf{x}) = a_{i,\sigma(i)} \tilde{f}_{\sigma(i)}(\mathbf{x}) + c_i$, which concludes the proof. $\qquad\square$

### C.5.2 Augmented features

**Theorem 2b.** *Assume that $n \leq d$, and that:*

1. *The feature extractor $\mathbf{f}$ is differentiable and surjective, and its Jacobian $\mathbf{J_f}$ is full rank.*

2. *There exist $2n + 1$ points $\mathbf{y}^0, \ldots, \mathbf{y}^{2n}$ such that the matrix*
$$\tilde{\mathbf{R}} = \left( \tilde{\mathbf{g}}(\mathbf{y}^1) - \tilde{\mathbf{g}}(\mathbf{y}^0), \ldots, \tilde{\mathbf{g}}(\mathbf{y}^{2n}) - \tilde{\mathbf{g}}(\mathbf{y}^0) \right) \tag{49}$$
*of size $2n \times 2n$ is invertible.*

*Then*
$$p_{\boldsymbol{\theta}}(\mathbf{x}|\mathbf{y}) = p_{\boldsymbol{\theta}'}(\mathbf{x}|\mathbf{y}) \implies \boldsymbol{\theta} \sim_s \boldsymbol{\theta}'$$
*where $\sim_s$ is defined in* (45).

*Proof.* Similarly to the proof of Theorem 2a, we pass the features $f_i$ through the nonlinear function $\mathbf{H}_i(f_i) = (f_i, f_i^2)$ which produces the augmented features $\tilde{\mathbf{f}}$ introduced in section 2.2.2.

Consider two different parameters $\boldsymbol{\theta}$ and $\tilde{\boldsymbol{\theta}}$ such that:
$$p_{\boldsymbol{\theta}}(\mathbf{x}|\mathbf{y}) = p_{\tilde{\boldsymbol{\theta}}}(\mathbf{x}|\mathbf{y}) \tag{50}$$
Since we have similar assumptions to Theorem 1, we will skip the first part of the proof and make the same conclusion, where the equivalence up to linear transformation here applies to $\mathbf{H}(\mathbf{f}_{\boldsymbol{\theta}})$ and $\mathbf{H}(\mathbf{f}_{\tilde{\boldsymbol{\theta}}})$:

$$\mathbf{H}(\mathbf{f}_{\boldsymbol{\theta}}(\mathbf{x})) = \mathbf{A}\mathbf{H}(\mathbf{f}_{\tilde{\boldsymbol{\theta}}}(\mathbf{x})) + \mathbf{c} \tag{51}$$
where $\mathbf{A}$ is a $2n \times 2n$ matrix of rank at least $n$ because $\mathbf{J_f}$ and $\mathbf{J_H}$ are full rank ($\mathbf{A}$ is not necessarily invertible yet, but this will be proven later) and $\mathbf{c}$ a constant vector. By replacing $\mathbf{H}$ by its expression, we get:
$$\begin{pmatrix} \mathbf{f}_{\boldsymbol{\theta}}(\mathbf{x}) \\ \mathbf{f}_{\boldsymbol{\theta}}^2(\mathbf{x}) \end{pmatrix} = \begin{pmatrix} \mathbf{A}^{(1)} & \mathbf{A}^{(2)} \\ \mathbf{A}^{(3)} & \mathbf{A}^{(4)} \end{pmatrix} \begin{pmatrix} \mathbf{f}_{\tilde{\boldsymbol{\theta}}}(\mathbf{x}) \\ \mathbf{f}_{\tilde{\boldsymbol{\theta}}}^2(\mathbf{x}) \end{pmatrix} + \begin{pmatrix} \boldsymbol{\alpha} \\ \boldsymbol{\beta} \end{pmatrix} \tag{52}$$
where each $\mathbf{A}^{(i)}$ is an $n \times n$ matrix, and $\mathbf{c} = (\boldsymbol{\alpha}, \boldsymbol{\beta})$. To simplify notations, denote by $\mathbf{h} = \mathbf{f}_{\tilde{\boldsymbol{\theta}}}$. We will also drop reference to $\boldsymbol{\theta}$ and $\tilde{\boldsymbol{\theta}}$. The first $n$ lines in the previous equation are:
$$f_i(\mathbf{x}) = \sum_{j=1}^{n} A_{ij}^{(1)} h_j(\mathbf{x}) + A_{ij}^{(2)} h_j^2(\mathbf{x}) + \alpha_i \tag{53}$$
and the last $n$ lines are:
$$f_i^2(\mathbf{x}) = \sum_{j=1}^{n} A_{ij}^{(3)} h_j(\mathbf{x}) + A_{ij}^{(4)} h_j^2(\mathbf{x}) + \beta_i \tag{54}$$

Fix an index $i$ in equations (53) and (54). To alleviate notations and reduce the number of subscripts and superscripts, we introduce $a_j = A_{ij}^{(1)}$, $b_j = A_{ij}^{(2)}$, $c_j = A_{ij}^{(3)}$, $d_j = A_{ij}^{(4)}$, $\alpha = \alpha_i$ and $\beta = \beta_i$. This proof is done in 5 steps. Note that the surjectivity assumption is key for the rest of the proof, and it requires that we set the dimension of the feature extractor to be lower than the dimension of the observations.

By equating equations (54) and (53) after squaring, we get, using our new notations:
$$\left( \sum_j a_j h_j(\mathbf{x}) + b_j h_j^2(\mathbf{x}) + \alpha \right)^2 = \sum_j c_j h_j(\mathbf{x}) + d_j h_j^2(\mathbf{x}) + \beta \tag{55}$$

**Step 1**  First, since $\mathbf{h}$ is surjective, there exists a point where it is equal to zero. Evaluating equation (55) at this point shows that $\beta = \alpha^2$.

**Step 2**  Second, the left hand side of equation (55) has terms raised to the power 4. These terms grow to infinity much faster than the rest of the terms of the rhs and the lhs. It is thus equal to zero. More rigorously, consider the vectors $\mathbf{e}_l(y) = (0, \ldots, y, \ldots, 0) \in \mathbb{R}^n$ where the only non zero entry is $y$ at the $l$-th position. Each of these vectors has a preimage by $\mathbf{h}$ (since it is surjective), which we denote by $\mathbf{x}_l(y)$. By evaluating equation (55) at each of these points, we get

$$(a_l y + b_l y^2 + \alpha)^2 = c_l y + d_l y^2 + \beta \tag{56}$$

Divide both sides of this equation by $y^4$, then take the limit $y \to \infty$. The right hand side will converge to 0, while the left hand side will converge to $b_l$, which shows that $b_l = 0$. By doing this process for all $l \in [\![1, n]\!]$, we can show that $\mathbf{b} = 0$.

**Step 3**  So far, we've shown that (55) becomes, after expanding the square in the lhs, and writing $\sum_j a_j h_j(\mathbf{x}) = \mathbf{a}^T \mathbf{h}(\mathbf{x})$:

$$(\mathbf{a}^T \mathbf{h}(\mathbf{x}))^2 + 2\alpha \mathbf{a}^T \mathbf{h}(\mathbf{x}) + \alpha^2 = \sum_j c_j h_j(\mathbf{x}) + d_j h_j^2(\mathbf{x}) + \alpha^2 \tag{57}$$

Let's again consider the vectors $\mathbf{e}_l(y)$ from earlier, and their preimages $\mathbf{x}_l(y)$. By evaluating (57) at the points $\mathbf{x}_l(y)$, we get

$$a_l^2 y^2 + 2\alpha a_l y + \alpha^2 = c_l y + d_l y^2 + \alpha^2 \tag{58}$$

Divide both sides by $y$, and take the limit $y \to 0$. The lhs converges to $2\alpha a_l$, while the rhs converges to $c_l$. Since this is valid for all $l \in [\![1, n]\!]$, we conclude that $\mathbf{c} = 2\alpha \mathbf{a}$. It also follows that $\mathbf{d} = \mathbf{a}^2$.

**Step 4**  Injecting this back into equation (57), and writing $\sum_j d_j h_j^2(\mathbf{x}) = \mathbf{h}(\mathbf{x})^T \operatorname{diag}(\mathbf{d}) \mathbf{h}(\mathbf{x})$, we are left with:

$$(\mathbf{a}^T \mathbf{h}(\mathbf{x}))^2 = \mathbf{h}(\mathbf{x})^T \operatorname{diag}(\mathbf{d}) \mathbf{h}(\mathbf{x}) \tag{59}$$

By applying the trace operator to both sides of this equation, and rearranging terms, we get

$$\operatorname{trace}\left( \left( \mathbf{a}\mathbf{a}^T - \operatorname{diag}(\mathbf{d}) \right) \mathbf{h}(\mathbf{x}) \mathbf{h}(\mathbf{x})^T \right) = 0 \tag{60}$$

which is of the form $\operatorname{trace}(\mathbf{C}^T \mathbf{B}(\mathbf{x})) = 0$. This is a dot product on the space $\mathcal{S}_n$ of $n \times n$ symmetric matrices (both $\mathbf{C}$ and $\mathbf{B}(\mathbf{x})$ are symmetric!), which is a vector space of dimension $\frac{n(n+1)}{2}$. If we can show that the matrix $\mathbf{C}$ is orthogonal to a basis of $\mathcal{S}_n$, then we can conclude that $\mathbf{C} = 0$.

For this, let $(\mathbf{e}_j)_{1 \leq j \leq n}$ be the Euclidean basis of $\mathbb{R}^n$, where each vector $\mathbf{e}_j$ has one non-zero entry equal to 1 at index $j$, and let $(\mathbf{E}_{ij})_{1 \leq i \leq n, 1 \leq j \leq n}$ be the Euclidean basis of $\mathbb{R}^{n \times n}$, where each matrix $\mathbf{E}_{ij}$ has only one non-zero entry equal to 1 at row $i$ and column $j$.

Now since $\mathbf{h}$ is surjective, there exists $\mathbf{x}_j$ such that $\mathbf{h}(\mathbf{x}_j) = \mathbf{e}_j$, and $\mathbf{h}(\mathbf{x}_j) \mathbf{h}(\mathbf{x}_j)^T = \mathbf{e}_j \mathbf{e}_j^T = \mathbf{E}_{jj}$. The $n$ different $\mathbf{x}_j$ give us our first $n$ matrices we will use to construct a basis of $\mathcal{S}_n$. We now need to find $\frac{n(n-1)}{2}$ remaining basis matrices. For this, consider the sums $(\mathbf{e}_j + \mathbf{e}_l)_{1 \leq j < l \leq n}$, of which there is exactly $\frac{n(n-1)}{2}$. Each of these sums of vectors have a preimage $\mathbf{x}_{j,l}$ by $\mathbf{h}$, and $\mathbf{h}(\mathbf{x}_{j,l}) \mathbf{h}(\mathbf{x}_{j,l})^T = (\mathbf{e}_j + \mathbf{e}_l)(\mathbf{e}_j + \mathbf{e}_l)^T = \mathbf{E}_{jj} + \mathbf{E}_{ll} + (\mathbf{E}_{il} + \mathbf{E}_{li})$, which is a matrix in $\mathcal{S}_n$ that is linearly independent of all $\mathbf{E}_{jj}$, and all other $(\mathbf{e}_s + \mathbf{e}_t)(\mathbf{e}_s + \mathbf{e}_t)^T$ where $(s, t) \neq (j, l)$ because they have non-zero entries at different rows and columns.

We have then found a total of $\frac{n(n+1)}{2}$ different vectors $(\mathbf{x}_1, \ldots, \mathbf{x}_n, \mathbf{x}_{1,2}, \ldots, \mathbf{x}_{n-1,n})$ such that their images by $\mathbf{h}\mathbf{h}^T$ form a basis of $\mathcal{S}_n$. If we now evaluate equation (60) at each of these points, we find that the matrix $\mathbf{a}\mathbf{a}^T - \operatorname{diag}(\mathbf{d})$ is orthogonal to a basis of $\mathcal{S}_n$, which implies that it is necessarily equal to 0. This in turn implies that $\mathbf{a}\mathbf{a}^T$ is a diagonal matrix, and that $a_j a_l = 0$ for all $j \neq l$, which implies that at most one $a_j$ is non-zero.

**Step 5**  So far, we have proven that, among other things, $A_{i,j}^{(2)} = 0$ for all $i, j$. We now go back to equation (53), which we can write as:

$$\mathbf{f}(\mathbf{x}) = \mathbf{A}^{(1)} \mathbf{h}(\mathbf{x}) + \boldsymbol{\alpha} \tag{61}$$

Both $\mathbf{f}$ and $\mathbf{h}$ are differentiable, and according to assumption 2, $J_{\mathbf{f}}$ has rank $n$ (it is full rank and $n \leq d$). Thus, by differentiating the last equation, we conclude that $\mathbf{A}^{(1)}$ has rank $n$, and is thus invertible.

**Conclusion**   We've shown that $f_i(\mathbf{x}) = a_j h_j(\mathbf{x}) + \alpha_i$, where $a_j = A^{(1)}_{ij}$. This is valid for all $i \in [\![1, n]\!]$. Now since $\mathbf{A}^{(1)}$ is invertible, the non-zero entry $A^{(1)}_{ij}$ has to be in a different column for each row, otherwise some rows will be linearly dependent. Thus, there exists a permutation $\sigma$ of $[\![1, n]\!]$, such that $A^{(1)}_{i\sigma(i)} \neq 0$, and we deduce that

$$f_i(\mathbf{x}) = a_{\sigma(i)} h_{\sigma(i)}(\mathbf{x}) + \alpha_i \tag{62}$$

which concludes the proof.

From the second conclusion of step 3, we have that $\mathbf{d} = \mathbf{a}^2$. Combined with the fact that exactly one element of $\mathbf{a}$ is nonzero such that $\mathbf{A}^{(1)}$ is full rank, this implies that $\mathbf{A}^{(4)}$ is also full rank, which in turn means that $\mathbf{A}$ is full rank. $\qquad\square$

### C.6   Proof of Theorem 3

**Theorem 3.** *Let $p(\mathbf{x}|\mathbf{y})$ be a conditional probability density. Assume that $\mathcal{X}$ and $\mathcal{Y}$ are compact Hausdorff spaces, and that $p(\mathbf{x}|\mathbf{y}) > 0$ almost surely $\forall (\mathbf{x}, \mathbf{y}) \in \mathcal{X} \times \mathcal{Y}$. Then for each $\varepsilon > 0$, there exists $(\boldsymbol{\theta}, n) \in \Theta \times \mathbb{N}$, where $n$ is the dimension of the feature extractor, such that $\sup_{\mathbf{x}, \mathbf{y}} |p_{\boldsymbol{\theta}}(\mathbf{x}|\mathbf{y}) - p(\mathbf{x}|\mathbf{y})| < \varepsilon$.*

*Proof.* We consider here two cases.

**Continuous auxiliary variable**   Recall the form of our model:

$$\log p_{\boldsymbol{\theta}}(\mathbf{x}|\mathbf{y}) = -\log Z(\mathbf{y}) - \mathbf{f}(\mathbf{x})^T \mathbf{g}(\mathbf{y}) \tag{63}$$

By parameterizing each of $f_i, g_i$ as neural networks, these functions can approximate continuous function on their respective domains arbitrarily well. According to Lemma 9, this implies that any continuous function on $\mathcal{X} \times \mathcal{Y}$ can be approximated arbitrarily well by a term of the form $-\mathbf{f}(\mathbf{x})^T \mathbf{g}(\mathbf{y})$.

Thus, any continuous function can be approximated by $\log p_{\boldsymbol{\theta}}(\mathbf{x}|\mathbf{y}) + \log Z(\mathbf{y})$ for some $\boldsymbol{\theta}$, where $Z(\mathbf{y})$ captures the difference in scale between the function in question and the normalized density $p_{\boldsymbol{\theta}}(\mathbf{x}|\mathbf{y})$. We apply this result to $\log p(\mathbf{x}|\mathbf{y})$: for any $\varepsilon > 0$, there exists $(\boldsymbol{\theta}, n) \in \Theta \times \mathbb{N}$ such that:

$$\sup_{\mathbf{x}, \mathbf{y}} \left| \log p(\mathbf{x}|\mathbf{y}) + \sum_{i=1}^{n} f_i(\mathbf{x}; \boldsymbol{\theta}) g_i(\mathbf{y}; \boldsymbol{\theta}) \right| < \varepsilon \tag{64}$$

Since $p(\mathbf{x}|\mathbf{y}) > 0$ a.s. on $\mathcal{X} \times \mathcal{Y}$, $\log p(\mathbf{x}|\mathbf{y})$ is finite and bounded. So is the term $-\sum_{i=1}^{n} f_i(\mathbf{x}; \boldsymbol{\theta}) g_i(\mathbf{y}; \boldsymbol{\theta})$. We can then use the fact that $\exp$ is Lipschitz on compacts to conclude for $p(\mathbf{x}|\mathbf{y})$, to conclude that:

$$\sup_{\mathbf{x}, \mathbf{y}} |p(\mathbf{x}|\mathbf{y}) - p_{\boldsymbol{\theta}}(\mathbf{x}|\mathbf{y})| < K\varepsilon \tag{65}$$

where $K$ is the Lipschitz constant of $\exp$, which concludes the proof.

**Discrete auxiliary variable**   If $\mathbf{y}$ is discrete and $\mathcal{Y}$ is compact, then $\mathbf{y}$ only takes finitely many values. In this case, we do not need Lemma 9 for the proof. $\mathbf{g}(\mathbf{y})$ can simply be a lookup table, and we learn different approximations for each fixed value of $\mathbf{y}$, since $\mathbf{f}$ has the universal approximation capability, which concludes the proof. $\qquad\square$

Denote by $\mathcal{C}(\mathcal{X})$ (respectively $\mathcal{C}(\mathcal{Y})$ and $\mathcal{C}(\mathcal{X} \times \mathcal{Y})$) the Banach algebra of continuous functions from $\mathcal{X}$ (respectively $\mathcal{Y}$ and $\mathcal{X} \times \mathcal{Y}$) to $\mathbb{R}$. For any subsets of functions $\mathcal{F}_{\mathcal{X}} \subset \mathcal{C}(\mathcal{X})$ and $\mathcal{F}_{\mathcal{Y}} \subset \mathcal{C}(\mathcal{Y})$, let $\mathcal{F}_{\mathcal{X}} \otimes \mathcal{F}_{\mathcal{Y}} := \{\sum_{i=1}^{n} f_i g_i | n \in \mathbb{N}, f_i \in \mathcal{F}_{\mathcal{X}}, g_i \in \mathcal{F}_{\mathcal{Y}}\}$ be the set of *all linear combinations* of products of functions from $\mathcal{F}_{\mathcal{X}}$ and $\mathcal{F}_{\mathcal{Y}}$ to $\mathbb{R}$. The energy function defining our model belongs to this last set. Finally, universal approximation is expressed in terms of density: for instance, the set of functions $\mathcal{F}_{\mathcal{X}}$ have universal approximation of $\mathcal{C}(\mathcal{X})$ if it is dense in it, *i.e.* for any function in $\mathcal{C}(\mathcal{X})$, we can always find a limit of a sequence of functions of $\mathcal{F}_{\mathcal{X}}$ that converges to it. We mathematically express density by writing $\overline{\mathcal{F}_{\mathcal{X}}} = \mathcal{C}(\mathcal{X})$.

Let $\mathcal{F}_{\mathcal{X}}$ (respectively $\mathcal{F}_{\mathcal{Y}}$) be the set of deep neural networks with input in $\mathcal{X}$ (respectively in $\mathcal{Y}$). The universal approximation capability is summarised in the following Lemma.

**Lemma 9** (Universal approximation capability)**.** *Suppose the following:*

*(i) $\mathcal{X}$ and $\mathcal{Y}$ are compact Hausdorff spaces.*

*(ii) $\overline{\mathcal{F}_{\mathcal{X}}} = \mathcal{C}(\mathcal{X})$ and $\overline{\mathcal{F}_{\mathcal{Y}}} = \mathcal{C}(\mathcal{Y})$*

*then $\overline{\mathcal{F}_{\mathcal{X}} \otimes \mathcal{F}_{\mathcal{Y}}} = \mathcal{C}(\mathcal{X} \times \mathcal{Y})$. All completions here are with respect to the infinity norm.*

*Proof.* We prove this theorem in two steps:

1. We first prove that $\mathcal{F}_{\mathcal{X}} \otimes \mathcal{F}_{\mathcal{Y}}$ is dense in $\mathcal{C}(\mathcal{X}) \otimes \mathcal{C}(\mathcal{Y})$ using the hypotheses of Theorem 3.
2. we prove that $\mathcal{C}(\mathcal{X}) \otimes \mathcal{C}(\mathcal{Y})$ is dense in $\mathcal{C}(\mathcal{X} \times \mathcal{Y})$ using Theorem 5.

**Step 1**     Let $\varepsilon > 0$. Let $h \in \mathcal{C}(\mathcal{X}) \otimes \mathcal{C}(\mathcal{Y})$. Then there exists $k \in \mathbb{N}$ and functions $f_i \in \mathcal{C}(\mathcal{X})$ and $g_i \in \mathcal{C}(\mathcal{Y})$ such that $h = \sum_{i=1}^{k} f_i g_i$. For each $i$, since $\mathcal{F}_{\mathcal{Y}}$ dense in $\mathcal{C}(\mathcal{Y})$, there exists $\tilde{g}_i \in \mathcal{F}_{\mathcal{Y}}$ such that $\|g_i - \tilde{g}_i\|_\infty < \frac{\varepsilon}{2k\|f_i\|_\infty}$. From $\mathcal{F}_{\mathcal{X}}$ dense in $\mathcal{C}(\mathcal{X})$, there exists $\tilde{f}_i \in \mathcal{F}_{\mathcal{X}}$ such that $\|f_i - \tilde{f}_i\|_\infty < \frac{\varepsilon}{2k\|\tilde{g}_i\|_\infty}$. We then have

$$\|f_i g_i - \tilde{f}_i \tilde{g}_i\|_\infty = \|f_i g_i - f_i \tilde{g}_i + f_i \tilde{g}_i - \tilde{f}_i \tilde{g}_i\|_\infty \leq \|f_i\|_\infty \|g_i - \tilde{g}_i\|_\infty + \|\tilde{g}_i\|_\infty \|f_i - \tilde{f}_i\|_\infty < \frac{\varepsilon}{k} \quad (66)$$

Using this, we conclude that

$$\|h - \sum_{i=1}^{k} \tilde{f}_i \tilde{g}_i\|_\infty \leq \sum_{i=1}^{k} \|f_i g_i - \tilde{f}_i \tilde{g}_i\|_\infty < \varepsilon \qquad (67)$$

which proves that $\mathcal{F}_{\mathcal{X}} \otimes \mathcal{F}_{\mathcal{Y}}$ is dense in $\mathcal{C}(\mathcal{X}) \otimes \mathcal{C}(\mathcal{Y})$.

**Step 2**     We will use the Stone-Weirstrass theorem for this step. It is enough to show that:

(i) $\mathcal{X} \times \mathcal{Y}$ is a compact Hausdorff space.
(ii) $\mathcal{C}(\mathcal{X}) \otimes \mathcal{C}(\mathcal{Y}) \subset \mathcal{C}(\mathcal{X} \times \mathcal{Y})$.
(iii) $\mathcal{C}(\mathcal{X}) \otimes \mathcal{C}(\mathcal{Y})$ is a unital sub-algebra of $\mathcal{C}(\mathcal{X} \times \mathcal{Y})$ (see Definition 3).
(iv) $\mathcal{C}(\mathcal{X}) \otimes \mathcal{C}(\mathcal{Y})$ separates points in $\mathcal{X} \times \mathcal{Y}$ (see Definition 3).

To prove $(i)$, we use the fact that every finite product of compact spaces is a compact space, and every finite product of Hausdorff spaces is a Hausdorff space. Points $(ii)$ and $(iii)$ are easy to verify. To prove $(iv)$, let $(\mathbf{x}, \mathbf{y})$ and $(\mathbf{x}', \mathbf{y}')$ be distinct points in $\mathcal{X} \times \mathcal{Y}$. Assume that $\mathbf{x} \neq \mathbf{x}'$ (we proceed similarly if $\mathbf{y} \neq \mathbf{y}'$). Define the continuous function $f \in \mathcal{C}(\mathcal{X})$ such that $f(\mathbf{x}) \neq 0$ and $f(\mathbf{x}') = 0$. Then for $g = 1 \in \mathcal{C}(\mathcal{Y})$, we have $f(\mathbf{x})g(\mathbf{y}) = f(\mathbf{x}) \neq 0 = f(\mathbf{x}')g(\mathbf{y}')$.

All the conditions required to use the Stone-Weirestrass Theorem are verified, and we can conclude that $\mathcal{C}(\mathcal{X}) \otimes \mathcal{C}(\mathcal{Y})$ is dense in $\mathcal{C}(\mathcal{X} \times \mathcal{Y})$

**Conclusion**     Combining the results of steps 1 and 2, we conclude that $\mathcal{F}_{\mathcal{X}} \otimes \mathcal{F}_{\mathcal{Y}}$ is dense in $\mathcal{C}(\mathcal{X} \times \mathcal{Y})$.
□

**Definition 3.** *Let $K$ be a compact Hausdorff space. Consider the Banach algebra $\mathcal{C}(K)$ equipped with the supremum norm $\|f\|_\infty = \sup_{t \in K} |f(t)|$. Then:*

1. *$\mathcal{A} \in \mathcal{C}(K)$ is a unital sub-algebra if:*

   *(i) $1 \subset \mathcal{A}$.*
   *(ii) for all $f, g \in \mathcal{A}$ and $\alpha, \beta \in \mathbb{R}$, we have $\alpha f + \beta g \in \mathcal{A}$ and $fg \in \mathcal{A}$.*

2. *$\mathcal{A} \subset \mathcal{C}(K)$ separates points of $K$ if $\forall s, t \in K$ such that $s \neq t$, $\exists f \in \mathcal{A}$ s.t. $f(s) \neq f(t)$.*

**Theorem 5** (Stone-Weirstrass)**.** *Let $K$ be a compact Hausdorff space, and $\mathcal{A}$ a unital sub-algebra of $\mathcal{C}(K)$ which separates points of $K$. Then $\mathcal{A}$ is dense in $\mathcal{C}(K)$.*

*Proof.* A proof to this theorem can be found in many references, for instance Brosowski and Deutsch (1981). □

# D   Latent variable estimation in generative models

Recall the generative model of IMCA: we observe a random variable $\mathbf{x} \in \mathbb{R}^d$ as a result of a nonlinear transformation $\mathbf{h}$ of a latent variable $\mathbf{z} \in \mathbb{R}^d$ whose distribution is conditioned on an auxiliary variable $\mathbf{y}$ that is also observed:

$$\mathbf{z} \sim p(\mathbf{z}|\mathbf{y}) \tag{68}$$
$$\mathbf{x} = \mathbf{h}(\mathbf{z}) \tag{69}$$

We assume the latent variable in the IMCA model has a density of the form

$$p(\mathbf{z}|\mathbf{y}) = \mu(\mathbf{z})e^{\sum_i \mathbf{T}_i(z_i)^T \boldsymbol{\lambda}_i(\mathbf{y}) - \Gamma(\mathbf{y})} \tag{70}$$

where $\mu$ is not necessarily factorial.

Further, we will suppose that the density $p(\mathbf{z}|\mathbf{y})$ belongs to the following subclass of the exponential families, introduced by Khemakhem et al. (2020):

**Definition 4** (Strongly exponential). *We say that an exponential family distribution is* strongly exponential *if for any subset $\mathcal{X}$ of $\mathbb{R}$ the following is true:*

$$\left(\exists\, \boldsymbol{\theta} \in \mathbb{R}^k \,|\, \forall x \in \mathcal{X}, \langle\, \mathbf{T}(x), \boldsymbol{\theta}\, \rangle = const\right) \implies (\Lambda(\mathcal{X}) = 0 \text{ or } \boldsymbol{\theta} = 0) \tag{71}$$

*where $\Lambda$ is the Lebesgue measure.*

If we suppose that only $n$ out of $d$ components of the latent variable are modulated by the auxiliary variable $\mathbf{y}$ (equivalently, if we suppose that the parameters $\boldsymbol{\lambda}_{n+1:d}(\mathbf{y})$ are constant), then we can write its density as

$$p(\mathbf{z}|\mathbf{y}) = \mu(\mathbf{z})e^{\sum_{i=1}^n \mathbf{T}_i(z_i)^T \boldsymbol{\lambda}_i(\mathbf{y}) - \Gamma(\mathbf{y})} \tag{72}$$

The term $e^{\sum_{i=n+1}^d \mathbf{T}_i(z_i)^T \boldsymbol{\lambda}_i}$ is absorbed into $\mu(\mathbf{z})$. This last expression will be useful for dimensionality reduction.

To estimate the latent variables of the IMCA model, we fit an augmented version of our energy model

$$p_{\boldsymbol{\theta}}(\mathbf{x}|\mathbf{y}) = Z(\mathbf{y};\boldsymbol{\theta})^{-1} \exp\left(-\mathbf{H}(\mathbf{f}_{\boldsymbol{\theta}}(\mathbf{x}))^T \mathbf{g}_{\boldsymbol{\theta}}(\mathbf{y})\right) \tag{73}$$

where $\mathbf{H}(\mathbf{f}(\mathbf{x})) = (\mathbf{H}_1(f_1(\mathbf{x})), \ldots, \mathbf{H}_d(f_d(\mathbf{x})))$, and each $\mathbf{H}_l$ is a (nonlinear) output activation. An example of such map is $\mathbf{H}_l(x) = (x, x^2)$.

In this section, we present the proofs for the estimation of the Independently Modulated Component Analysis by an identifiable energy model. These proofs are based on similar ideas and techniques to previous proofs, but are different enough that we can't forgo them.

## D.1   Assumptions

We prove dimensionality reduction capability in Theorem 6. We will decompose Theorem 4 into two sub-theorems, which will make the proof easier to understand, but also more adaptable into future work. For the sake of clarity, we will separate its assumptions into smaller assumptions, and refer to them when needed in the proofs.

 (i) The observed data follows the exponential IMCA model of equations (68)-(70).

 (ii) The mixing function $\mathbf{h} : \mathbb{R}^d \to \mathbb{R}^d$ in (69) is invertible.

 (iii) The sufficient statistics $\mathbf{T}_i$ in (70) are differentiable, and the functions $T_{ij} \in \mathbf{T}_i$ are linearly independent on any subset of $\mathcal{X}$ of measure greater than zero.

 (iv) There exist $k+1$ distinct points $\mathbf{y}^0, \ldots, \mathbf{y}^k$ such that the matrix

$$\mathbf{L} = (\boldsymbol{\lambda}(\mathbf{y}_1) - \boldsymbol{\lambda}(\mathbf{y}_0), \ldots, \boldsymbol{\lambda}(\mathbf{y}_k) - \boldsymbol{\lambda}(\mathbf{y}_0))$$

of size $k \times k$ is invertible, where $k = \sum_{i=1}^d \dim(\mathbf{T}_i)$.

 (v) We fit the model (73) to the conditional density $p(\mathbf{x}|\mathbf{y})$, where we assume the feature extractor $\mathbf{f}(\mathbf{x})$ to be differentiable, $d$-dimensional, and the pointwise nonlinearitiy $\mathbf{H}$ to be differentiable and $k$-dimensional, and the dimension of its vector-valued components $\mathbf{H}_l$ to be chosen from $(\dim(\mathbf{T}_1), \ldots, \dim(\mathbf{T}_d))$ without replacement.

(vi) The sufficient statistic in (70) is twice differentiable and $\dim(\mathbf{T}_l) \geq 2$, $\forall l$.

(vii) The mixing function $\mathbf{h}$ is a $\mathcal{D}^2$-diffeomorphisms.

(viii) The feature extractor $\mathbf{f}$ in (73) is a $\mathcal{D}^2$-diffeomorphism.

(vi') $\dim(\mathbf{T}_l) = 1$ and $\mathbf{T}_l$ is non-monotonic $\forall l$.

(vii') The mixing function $\mathbf{h}$ is a $\mathcal{C}^1$-diffeomorphism.

(viii') The feature extractor $\mathbf{f}$ in (73) is a $\mathcal{C}^1$-diffeomorphism, and the nonlinearities $\mathbf{H}_l$ have a unique extremum.

(ix) Only $n \leq d$ components of the latent variable are modulated, and its density has the form (72).

(x) The feature extractor $\mathbf{f}$ has the form $\mathbf{f}(\mathbf{x}) = (\mathbf{f}_1(\mathbf{x}), \mathbf{f}_2(\mathbf{x}))$ where $\mathbf{f}_1(\mathbf{x}) \in \mathbb{R}^n$, and the auxiliary feature extractor $\mathbf{g}$ has the form $\mathbf{g}(\mathbf{y}) = (\mathbf{g}_1(\mathbf{y}), \mathbf{g}_2)$ where $\mathbf{g}_1(\mathbf{y}) \in \mathbb{R}^n$ and $\mathbf{g}_2$ is constant.

## D.2 Lemmas

We rely on the following Lemmas from Khemakhem et al. (2020), which we state below in the interest of completeness.

**Lemma 10.** *Consider an exponential family distribution with $k \geq 2$ components. Then the components of the sufficient statistic $\mathbf{T}$ are linearly independent.*

**Lemma 11.** *Consider a* strongly exponential *family distribution such that its sufficient statistic $\mathbf{T}$ is differentiable almost surely. Then $T_i' \neq 0$ almost everywhere on $\mathbb{R}$ for all $1 \leq i \leq k$.*

**Lemma 12.** *Consider a strongly exponential distribution of size $k \geq 2$ with sufficient statistic $\mathbf{T}(x) = (T_1(x), \ldots, T_k(x))$. Further assume that $\mathbf{T}$ is differentiable almost everywhere. Then there exist $k$ distinct values $x_1$ to $x_k$ such that $(\mathbf{T}'(x_1), \ldots, \mathbf{T}'(x_k))$ are linearly independent in $\mathbb{R}^k$.*

**Lemma 13.** *Consider a strongly exponential distribution of size $k \geq 2$ with sufficient statistic $\mathbf{T}$. Further assume that $\mathbf{T}$ is twice differentiable almost everywhere. Then*

$$\dim\left(\text{span}\left((T_i'(x), T_i''(x))^T, 1 \leq i \leq k\right)\right) \geq 2 \tag{74}$$

*almost everywhere on $\mathbb{R}$.*

**Lemma 14.** *Consider $n$ strongly exponential distributions of size $k \geq 2$ with respective sufficient statistics $\mathbf{T}_j = (T_{j,1}, \ldots T_{j,k})$, $1 \leq j \leq n$. Further consider that the sufficient statistics are twice differentiable. Define the vectors $\mathbf{e}^{(j,i)} \in \mathbb{R}^{2n}$, such that $\mathbf{e}^{(j,i)} = (0, \ldots, 0, T_{j,i}', T_{j,i}'', 0, \ldots, 0)$, where the non-zero entries are at indices $(2j, 2j+1)$. Let $\mathbf{x} := (x_1, \ldots, x_n) \in \mathbb{R}^n$. Then the matrix $\bar{\mathbf{e}}(\mathbf{x}) := (\mathbf{e}^{(1,1)}(x_1), \ldots, \mathbf{e}^{(1,k)}(x_1), \ldots \mathbf{e}^{(n,1)}(x_n), \ldots, \mathbf{e}^{(n,k)}(x_n))$ of size $(2n \times nk)$ has rank $2n$ almost everywhere on $\mathbb{R}^n$.*

## D.3 Proofs

As mentioned above, we decompose Theorem 4 into two smaller results, summarized in what follows by Theorems 4a and 4b.

**Theorem 4a.** *Assume assumptions (i)-(v) hold. Then, after convergence of our model $p_{\boldsymbol{\theta}}(\mathbf{x}|\mathbf{y})$ to the true density $p(\mathbf{x}|\mathbf{y})$, we can recover the latent variables up to an invertible linear transformation and point-wise nonlinearities, i.e.*

$$\mathbf{H}(\mathbf{f}(\mathbf{x})) = \mathbf{A}\mathbf{T}(\mathbf{z}) + \mathbf{b} \tag{75}$$

*where $\mathbf{A}$ is an invertible matrix.*

*Proof.* We fit our density model (73) to the conditional density $p(\mathbf{x}|\mathbf{y})$, setting the dimension of the feature extractor $\mathbf{f}$ to be equal to $d$, and the dimensions of the output nonlinearities $\mathbf{H}_l$ chosen from $(\dim(\mathbf{T}_1), \ldots, \dim(\mathbf{T}_d))$, as per assumption (v):

$$Z(\mathbf{y})^{-1} \exp \mathbf{H}(\mathbf{f}(\mathbf{x}))^T \mathbf{g}(\mathbf{y}) = p(\mathbf{x}|\mathbf{y}) \tag{76}$$

by doing the change of variable $\mathbf{x} = \mathbf{h}(\mathbf{z})$, taking the log on both sides, we get:

$$-\log Z(\mathbf{y}) + \mathbf{H}(\mathbf{f}(\mathbf{x}))^T \mathbf{g}(\mathbf{y}) = \log p(\mathbf{z}|\mathbf{y}) - \log|\det \mathbf{J}_{\mathbf{h}^{-1}}(\mathbf{x})| \tag{77}$$

$$= \log \mu(\mathbf{h}^{-1}(\mathbf{x})) + \mathbf{T}(\mathbf{z})^T \boldsymbol{\lambda}(\mathbf{y}) - \Gamma(\mathbf{y}) - \log|\det \mathbf{J}_{\mathbf{h}^{-1}}(\mathbf{x})| \tag{78}$$

Let $\mathbf{y}_0, \dots, \mathbf{y}_k$ be the points provided by assumption (iv) of the theorem, where $k = \sum_i k_i$, and $k_i = \dim(\mathbf{T}_i)$. Define $\overline{\boldsymbol{\lambda}}(\mathbf{y}) = \boldsymbol{\lambda}(\mathbf{y}) - \boldsymbol{\lambda}(\mathbf{y}_0)$, $\overline{\Gamma}(\mathbf{y}) = \Gamma(\mathbf{y}) - \Gamma(\mathbf{y}_0)$, $\overline{\mathbf{g}}(\mathbf{y}) = \mathbf{g}(\mathbf{y}) - \mathbf{g}(\mathbf{y}_0)$ and $\overline{Z}(\mathbf{y}) = \log Z(\mathbf{y}) - \log Z(\mathbf{y}_0)$. We plug each of those $\mathbf{y}_l$ in (78) to obtain $k+1$ such equations. We subtract the first equation for $\mathbf{y}_0$ from the remaining $k$ equations to get for $l = 1, \dots, k$:

$$-\overline{Z}(\mathbf{y}_l) + \mathbf{H}(\mathbf{f}(\mathbf{x}))^T \overline{\mathbf{g}}(\mathbf{y}_l) = \mathbf{T}(\mathbf{z})^T \overline{\boldsymbol{\lambda}}(\mathbf{y}_l) - \overline{\Gamma}(\mathbf{y}_l) \tag{79}$$

The **crucial point** here is that the non factorial terms $\mu(\mathbf{g}(\mathbf{x}))$ and $\tilde{\mu}(\tilde{\mathbf{g}}(\mathbf{x}))$ cancel out when we take these differences. This is what allows us to generalize the identifiability results of nonlinear ICA to the context of IMCA.

Let $\mathbf{L}$ bet the matrix defined in assumption (iv), and $\tilde{\mathbf{L}} := (\dots, \overline{\mathbf{g}}(\mathbf{y}_l), \dots)$. Define $\mathbf{b} = (\dots, \overline{Z}(\mathbf{y}_l) - \overline{\Gamma}(\mathbf{y}_l), \dots)$. Expressing (79) for all points $\mathbf{y}_l$ in matrix form, we get:

$$\tilde{\mathbf{L}}^T \mathbf{H}(\mathbf{f}(\mathbf{x})) = \mathbf{L}^T \mathbf{T}(\mathbf{z}) + \mathbf{b} \tag{80}$$

By assumption (iv), $\mathbf{L}$ is invertible, and thus we can write

$$\mathbf{T}(\mathbf{z}) = \mathbf{A}\mathbf{H}(\mathbf{f}(\mathbf{x})) + \mathbf{c} \tag{81}$$

where $\mathbf{c} = \mathbf{L}^{-T}\mathbf{b}$ and $\mathbf{A} = \mathbf{L}^{-T}\tilde{\mathbf{L}}^T$.

To prove that $\mathbf{A}$ is invertible, we first take the gradient of equation (81) with respect to $\mathbf{z}$. The Jacobian $\mathbf{J}_{\mathbf{T}}$ of $\mathbf{T}$ is a matrix of size $k \times d$. Its columns are independent because each $\mathbf{T}_i$ is only a function of $z_i$, and thus the non-zero entries of each column are in different rows. This means that its rank is $d$ (since $k = \sum_{i=1}^d k_i \geq d$). This is not enough to prove that $\mathbf{A}$ is invertible though. For that, we consider the functions $\mathbf{T}_i$ for which $k_i > 1$: for each of these functions, using Lemma 12, there exists points $z_i^{(1)}, \dots, z_i^{(k_i)}$ such that $(\mathbf{T}_i'(z_i^{(1)}), \dots, \mathbf{T}_i'(z_i^{(k_i)}))$ are independent. Collate these point into $k_{\max} := \max_i k_i$ vectors $\mathbf{z}^{(j)} := (z_1^{(j)}, \dots z_d^{(j)})$, where for each $i$, $z_i^{(j)} = z_i^{(1)}$ if $j > k_i$, and $z_i^{(1)}$ is a point such that $T_i(z_i^{(1)}) \neq 0$ if $k_i = 1$. We plug these vectors into equation (81) after differentiating it, and collate the $dk_{\max}$ equations in vector form:

$$\mathbf{M} = \mathbf{A}\tilde{\mathbf{M}} \tag{82}$$

where $\mathbf{M} := (\dots, \mathbf{J}_{\mathbf{T}}(\mathbf{z}^{(j)}), \dots)$ and $\tilde{\mathbf{M}} := (\dots, \mathbf{J}_{\mathbf{H}\circ\mathbf{f}\circ\mathbf{h}}(\mathbf{z}^{(j)}), \dots)$. Now the matrix $\mathbf{M}$ is of size $k \times dk_{\max}$, and it has exactly $k$ independent columns by definition of the points $\mathbf{z}^{(j)}$. This means that $\mathbf{M}$ is of rank $k$, which in turn implies that $\text{rank}(\mathbf{A}) \geq k$. Since $\mathbf{A}$ is a $k \times k$ matrix, we conclude that $\mathbf{A}$ is invertible. $\square$

The theorem above shows a first step in identifiability which holds up to a linear transformation. This is similar to Hyvärinen et al. (2019), but here we allow for dependencies between components. We can further sharpen the result, in line with Khemakhem et al. (2020) even in this non-independent case as follows:

**Theorem 4b.** *Assume assumptions (i)-(v) hold. Further assume that either assumptions (vi)-(viii) or assumptions (vi')-(viii') hold. Then equation (75) can be reduced to the component level, i.e. for each $i \in [\![1, d]\!]$:*

$$\mathbf{H}_i(f_i(\mathbf{x})) = A_i \mathbf{T}_{\gamma(i)}(z_{\gamma(i)}) + \mathbf{b}_i \tag{83}$$

*where $\gamma$ is a permutation of $[\![1, d]\!]$ such that $\dim(\mathbf{H}_i) = \dim(\mathbf{T}_{\gamma(i)})$ and $A_i$ a square invertible matrix.*

*Proof.* We prove this theorem separately for both sets of assumptions.

**Multi-dimensional sufficient statistics: assumptions (vi)-(viii)**   We suppose that $k_i \geq 2$, $\forall i$. The assumptions of Theorem 4a hold, and so we have

$$\mathbf{H}(\mathbf{f}(\mathbf{h}(\mathbf{z}))) = \mathbf{A}\mathbf{T}(\mathbf{z}) + \mathbf{c} \tag{84}$$

for an invertible $\mathbf{A} \in \mathbb{R}^{k \times k}$. We will index $\mathbf{A}$ by four indices $(i, l, a, b)$, where $1 \le i \le d, 1 \le l \le k_i$ refer to the rows and $1 \le a \le d, 1 \le b \le k_a$ to the columns.

Let $\mathbf{y} = \mathbf{f} \circ \mathbf{h}(\mathbf{z})$. Since both $\mathbf{f}$ and $\mathbf{h}$ are $\mathcal{D}^2$-diffeomorphisms (assumptions (vii), (viii)), we can invert this relation and write $\mathbf{z} = \mathbf{v}(\mathbf{y})$. We introduce the notations $v_i^s(\mathbf{y}) := \frac{\partial v_i}{\partial y_s}(\mathbf{y})$, $v_i^{st}(\mathbf{y}) := \frac{\partial^2 v_i}{\partial y_s \partial y_t}(\mathbf{y})$, $T'_{a,b}(z) = \frac{\mathrm{d} T_{a,b}}{\mathrm{d} z}(z)$, $T''_{a,b}(z) = \frac{\mathrm{d}^2 T_{a,b}}{\mathrm{d} z}(z)$ and $H'_{a,b}(y) = \frac{\mathrm{d} H_{a,b}}{\mathrm{d} y}(y)$. Each line of equation (84) can be written as:

$$H_{i,l}(y_i) = \sum_{a=1}^{d} \sum_{b=1}^{k_i} A_{i,l,a,b} T_{a,b}(v_a(\mathbf{y})) + c_{a,b} \tag{85}$$

for $i \le d$, $l \le k_i$. The first step is to show that $v_i(\mathbf{y})$ is a function of only one $y_{j_i}$, for all $i \le d$. by differentiating (85) with respect to $y_s$, $s \le d$:

$$\delta_{is} H'_{i,l}(y_i) = \sum_{a=1}^{d} \sum_{b=1}^{k_i} A_{i,l,a,b} T'_{a,b}(v_a(\mathbf{y})) v_a^s(\mathbf{y}) \tag{86}$$

and by differentiating (86) with respect to $y_t$, $s < t \le d$:

$$0 = \sum_{a,b} A_{i,l,a,b} \left( T'_{a,b}(v_a(\mathbf{y})) v_a^{s,t}(\mathbf{y}) + T''_{a,b}(v_a(\mathbf{y})) v_a^s(\mathbf{y}) v_a^t(\mathbf{y}) \right) \tag{87}$$

This equation is valid for all pairs $(s,t), t > s$. Define $\mathbf{B}_a(\mathbf{y}) := \left( v_a^{1,2}(\mathbf{y}), \ldots, v_a^{d-1,d}(\mathbf{y}) \right) \in \mathbb{R}^{\frac{d(d-1)}{2}}$, $\mathbf{C}_a(\mathbf{y}) := \left( v_a^1(\mathbf{y}) v_a^2(\mathbf{y}), \ldots, v_a^{d-1}(\mathbf{y}) v_a^d(\mathbf{y}) \right) \in \mathbb{R}^{\frac{d(d-1)}{2}}$, $\mathbf{M}(\mathbf{y}) := (\mathbf{B}_1(\mathbf{y}), \mathbf{C}_1(\mathbf{y}), \ldots, \mathbf{B}_d(\mathbf{y}), \mathbf{C}_d(\mathbf{y}))$, $\mathbf{e}^{(a,b)} := (0, \ldots, 0, T'_{a,b}, T''_{a,b}, 0, \ldots, 0) \in \mathbb{R}^{2d}$, such that the non-zero entries are at indices $(2a, 2a + 1)$ and $\overline{\mathbf{e}}(\mathbf{y}) := (\mathbf{e}^{(1,1)}(y_1), \ldots, \mathbf{e}^{(1,k_1)}(y_1), \ldots, \mathbf{e}^{(d,1)}(y_d), \ldots, \mathbf{e}^{(d,k_d)}(y_d)) \in \mathbb{R}^{2d \times k}$. Then by grouping equation (87) for all valid pairs $(s,t)$ and pairs $(i,l)$ and writing it in matrix form, we get:

$$\mathbf{M}(\mathbf{y}) \overline{\mathbf{e}}(\mathbf{y}) \mathbf{A} = 0 \tag{88}$$

Now by Lemma 14, we know that $\overline{\mathbf{e}}(\mathbf{y})$ has rank $2d$ almost surely on $\mathcal{Z}$. Since $\mathbf{A}$ is invertible, it is full rank, and thus $\mathrm{rank}(\overline{\mathbf{e}}(\mathbf{y})\mathbf{A}) = 2d$ almost surely on $\mathcal{Z}$. It suffices then to multiply by its pseudo-inverse from the right to get

$$\mathbf{M}(\mathbf{y}) = 0 \tag{89}$$

In particular, $C_a(\mathbf{y}) = 0$ for all $1 \le a \le d$. This means that the Jacobian of $\mathbf{v}$ at each $\mathbf{y}$ has at most one non-zero entry in each row. By invertibility and continuity of $J_{\mathbf{v}}$, we deduce that the location of the non-zero entries are fixed and do not change as a function of $\mathbf{y}$. We deduce that there exists a permutation $\sigma$ of $[\![1, d]\!]$ such that each of the $v_i(\mathbf{y}) = v_i(y_{\sigma(i)})$, and the same would apply to $\mathbf{v}^{-1}$. Without any loss of generality, we assume that $\sigma$ is the identity.

Now let $\overline{\mathbf{H}}(\mathbf{z}) = \mathbf{H} \circ \mathbf{v}^{-1}(\mathbf{y}) - \mathbf{c}$. This function is a pointwise function because $\mathbf{H}$ and $\mathbf{v}^{-1}$ are such functions. Plugging this back into equation (84) yields:

$$\overline{\mathbf{H}}(\mathbf{z}) = \mathbf{A} \mathbf{T}(\mathbf{z}) \tag{90}$$

The last equation is valid for every component:

$$\overline{H}_{i,l}(z_i) = \sum_{a,b} A_{i,l,a,b} T_{a,b}(z_a) \tag{91}$$

By differentiating both sides with respect to $z_s$ where $s \ne i$ we get

$$0 = \sum_b A_{i,l,s,b} T'_{s,b}(z_s) \tag{92}$$

By Lemma 10, we get $A_{i,l,s,b} = 0$ for all $1 \le b \le k$. Since (92) is valid for all $l$ and all $s \ne i$, we deduce that the matrix $\mathbf{A}$ has a block diagonal form:

$$\mathbf{A} = \begin{pmatrix} \mathbf{A}_1 & & \\ & \ddots & \\ & & \mathbf{A}_n \end{pmatrix} \tag{93}$$

which achieves the proof.

**One-dimensional sufficient statistics: assumptions (vi')-(viii')**  We now suppose that $k_i = 1$, $\forall i$. The proof of Khemakhem et al. (2020, Theorem 3) can be used here, where we define $\mathbf{v} = (\mathbf{f} \circ \mathbf{h})^{-1}$ and $h_{i,a} = D_{i,a} H_a(y_a) - D_{i,a} c_a$, where $\mathbf{D} = \mathbf{A}^{-1}$. We can then rewrite equation (85) for every component as:

$$T_i(v_i(\mathbf{z})) = \sum_{a=1}^{d} h_{i,a}(z_a) \tag{94}$$

which is the same as equation (45) of Khemakhem et al. (2020). All the assumptions required to prove their theorem are met in our case, and the rest of their proof would simply apply here to prove that $\mathbf{A}$ is a permutation matrix. $\qquad\square$

In practice, it is a natural desire to have the feature extractor reduce the dimension of the data, as it is usually very large. This has been achieved in nonlinear ICA before (Khemakhem et al., 2020; Hyvärinen and Morioka, 2016). It turns out that we can also incorporate dimensionality reduction in IMCA and its estimation by ICE-BeeM, under some assumptions.

**Theorem 6.** *Assume either of the following hold:*

- *Assumptions (i)-(x).*

- *Assumptions (i)- (v), (vi')- (viii'), and (ix)- (x).*

*Then $\mathbf{f}_1$ recovers only the modulated latent components as per Theorem 4b.*

*Proof.* The proof of Theorem 4a in this case is unchanged. Simply, we update the total dimension of matrix $\mathbf{L}$ here to $k = \sum_{i=1}^{n} \dim(\mathbf{T}_i)$. when we evaluate equation (78) on these points $\mathbf{y}_0, \ldots, \mathbf{y}_k$, the constant term $\mathbf{g}_2$ and the non-modulated components cancel out, and we are left with the equation

$$\tilde{\mathbf{L}}^T \mathbf{H}_{1:n}(\mathbf{f}_1(\mathbf{x})) = \mathbf{L}^T \mathbf{T}_{1:n}(\mathbf{z}) + \mathbf{b} \tag{95}$$

We then use similar arguments to the proof of Theorem 4a to conclude that

$$\mathbf{H}_{1:n}(\mathbf{f}(\mathbf{x})) = \mathbf{A}\mathbf{T}_{1:n}(\mathbf{z}) + \mathbf{c} \tag{96}$$

where $\mathbf{A} \in \mathbb{R}^n$ a square invertible matrix. At this point, we can make the same conclusion as Theorem 4a, while reducing the dimension of the latent space.

We now explain how we can extend Theorem 4b to the lower dimensional latent space case. Note that we still assume that $\mathbf{f} = (\mathbf{f}_1, \mathbf{f}_2)$ is a diffeomorphism per assumptions (viii) and (viii'). We can then still define $\mathbf{v} = (\mathbf{f} \circ \mathbf{h})^{-1}$.

We consider now two cases like in the proof of Theorem 4b.

**One-dimensional sufficient statistics**  Let $\mathbf{D} = \mathbf{A}^{-1}$ and $h_{i,a} = D_{i,a} H_a(y_a) - D_{i,a} c_a$. We can still write equation (96) like equation (94) as

$$T_i(v_i(\mathbf{z})) = \sum_{a=1}^{n} h_{i,a}(z_a) \tag{97}$$

for all $i \leq n$. The assumptions required for the proof are still met, despite reducing the dimension from $d$ to $n$. This interesting fact is also used for the proof of Theorem 2b as well, which achieves this part of the proof.

**Multi-dimensional sufficient statistics**  We rewrite equation (96)

$$H_{i,l}(y_i) = \sum_{a=1}^{n} \sum_{b=1}^{k_i} A_{i,l,a,b} T_{a,b}(v_a(\mathbf{y})) + c_{a,b} \tag{98}$$

for all $i \leq n, l \leq k_i$. We proceed similarly to the proof of Theorem 4b, replacing all mentions of $d$ by $n$ and keeping all differentiations to indices $t, s \leq n$, up to equation (89), after which we can conclude that $v_i^s v_i^t = 0$ for all $i \leq n$, and all $s, t \leq n$. This is not enough to conclude that each of the $v_i$ is only function of one $y_{j_i}$.

For that, we go back to equation (98) and differentiate it with respect to $y_s$, $s > n$:

$$0 = \sum_{a=1}^{d} \sum_{b=1}^{k_i} A_{i,l,a,b} T'_{a,b}(v_a(\mathbf{y})) v_a^s(\mathbf{y}) \tag{99}$$

which is valid for all $i \leq n$, $l \leq k_i$. Since $\mathbf{A}$ is invertible, we can conclude that $T'_{a,b}(v_a(\mathbf{y})) v_a^s(\mathbf{y}) = 0$ for all $a \leq n$ and $s > n$. Since we only consider strongly exponential distributions (assumption (iii)), and using proposition 11, we conclude that $T'_{a,b}(v_a(\mathbf{y})) \neq 0$ almost everywhere, and that $v_a^s(\mathbf{y}) = 0$, for all $s > n$. This, in addition to the fact that $v_i^s v_i^t = 0$ for all $i \leq n$, and all $s, t \leq n$ allows us to conclude that the first $n$ components of $\mathbf{v}$ are each only a function of one different $y_j$ because $\mathbf{v}$ is a diffeomorphism and its Jacobian is continuous. Finally, we can use this fact to deduce that $\mathbf{A}$ is a block permutation matrix, which achieves the proof. $\qquad \square$

# E   Independently modulated component analysis

As mentioned in section 3, linear latent variable models (Hyvärinen and Oja, 2000) and more recently nonlinear latent variable models may be identifiable provided some additional auxiliary variables (Khemakhem et al., 2020; Hyvärinen et al., 2019). The purpose of this auxiliary variable serves to introduce additional constraints over the distribution over latent variables, which are typically required to be conditionally independent given the auxiliary variable. This avenue of research has thus formalized the trade-off between expressivity of the mapping between latents to observations (from linear to nonlinear) and distributional assumptions over latent variables (from independent to conditionally independent given auxiliary variables).

We would like to relax the assumption of independence while maintaining identifiability, resulting in the framework of Independently Modulated Component Analysis (IMCA). In this section of the Appendix, we will give a detailed analysis of the IMCA model independently of any estimation method, drawing parallels to the identifiability results of the nonlinear ICA model presented in Khemakhem et al. (2020).

## E.1   Definition of the generative model

Assume we observe a random variable $\mathbf{x} \in \mathbb{R}^d$ as a result of a nonlinear transformation $\mathbf{h}$ of a latent variable $\mathbf{z} \in \mathbb{R}^d$ whose distribution is conditioned on an auxiliary variable $\mathbf{y}$ that is also observed:

$$\begin{aligned} \mathbf{z} &\sim p(\mathbf{z}|\mathbf{y}) \\ \mathbf{x} &= \mathbf{h}(\mathbf{z}) \end{aligned} \tag{100}$$

The main modelisation assumption we make is on the latent variable distribution, given by the following definition, where $\mathbf{u}$ is a dummy variable.

**Definition 5** (Exponentially factorial distributions). *We say that a multivariate exponential family distribution is **exponentially factorial** if its density $p(\mathbf{u})$ has the form*

$$p(\mathbf{y}) = \mu(\mathbf{y}) \prod_i e^{\mathbf{T}_i(y_i)^T \boldsymbol{\lambda}_i - \Gamma(\boldsymbol{\lambda})}$$

We assume that the latent variable in the IMCA model has a conditional exponentially factorial distribution, where the parameters of the exponential family are a function of the auxiliary variable $\mathbf{y}$:

$$p(\mathbf{z}|\mathbf{y}) = \mu(\mathbf{z}) e^{\sum_i \mathbf{T}_i(z_i)^T \boldsymbol{\lambda}_i(\mathbf{y}) - \Gamma(\mathbf{y})} \tag{101}$$

Equations (100) and (101) together define the nonparametric IMCA model with parameters $(\mathbf{h}, \mathbf{T}, \boldsymbol{\lambda}, \mu)$. Most importantly, we allow for an arbitrary base measure $\mu(\mathbf{z})$, *i.e.* the components of the latent variable must no longer be independent, as $\mu$ doesn't necessarily factorize across dimensions. The crucial assumption is that the components of the latent variables are independently modulated given the auxiliary variable $\mathbf{y}$, and that through the term $\exp(\sum_i \mathbf{T}_i(z_i)^T \boldsymbol{\lambda}_i(\mathbf{y}))$.

## E.2 Identifiability

The concept of identifiability is core to this work. As such, it is important to understand the different views one can have of this concept.

According to the conventional definition, a probabilistic model $\mathcal{P} = \{\mathcal{P}_{\boldsymbol{\theta}} : \boldsymbol{\theta} \in \Theta\}$ is identifiable *iif* the mapping $\boldsymbol{\theta} \mapsto \mathcal{P}_{\boldsymbol{\theta}}$ is bijective, *i.e.* $\mathcal{P}_{\boldsymbol{\theta}_1} = \mathcal{P}_{\boldsymbol{\theta}_2} \implies \boldsymbol{\theta}_1 = \boldsymbol{\theta}_2$. However, this definition is very restrictive and impractical.

Often, the identifiability form we can prove for a model is equality of the parameters *up to some indeterminacies*. This can be understood as an equivalence relation between parameters. Identifiability in this context implies that the equivalence class of the ground truth parameter can be uniquely recovered from observations. This is relevant only if the definition of the equivalence class is sufficiently narrow and specific to be able to make meaningful conclusions. One example of such equivalence relations can be found in linear ICA: the mixing matrix is uniquely recovered up to a scaled permutation. The permutation is irrelevant, and the scaling is circumvented by whitening the data. This is a good example of an equivalence class that doesn't restrict the practical utility of the ICA model.

An example of indeterminacy which is relevant to us here can be found in variational inference of latent variable models: two parameters are equivalent if they map to the same *inference* distribution (Khemakhem et al., 2020). This is the definition we will be using in this work. We will say that a generative model is identifiable if we can uniquely recover the latent variables, as given by the following definition.

**Definition 6.** *Consider two different sets of parameters* $(\mathbf{h}, \mathbf{T}, \boldsymbol{\lambda}, \mu)$ *and* $(\tilde{\mathbf{h}}, \tilde{\mathbf{T}}, \tilde{\boldsymbol{\lambda}}, \tilde{\mu})$, *defining two densities* $p$ *and* $p'$. *We say that the IMCA model is strongly identifiable if*

$$p(\mathbf{x}|\mathbf{y}) = \tilde{p}(\mathbf{x}|\mathbf{y}) \implies \forall i, \mathbf{T}_i(z_i) = \mathbf{A}_i \tilde{\mathbf{T}}_{\gamma(i)}(\tilde{z}_{\gamma(i)}) + \mathbf{b}_i \tag{102}$$

*where* $\gamma$ *is a permutation,* $\mathbf{A}_i$ *is an invertible matrix, and* $\mathbf{b}_i$ *a vector,* $\forall i \in [\![1, d]\!]$.
*We say that it is weakly identifiable if*

$$p(\mathbf{x}|\mathbf{y}) = \tilde{p}(\mathbf{x}|\mathbf{y}) \implies \mathbf{T}(\mathbf{z}) = \mathbf{A}\tilde{\mathbf{T}}(\tilde{\mathbf{z}}) + \mathbf{b} \tag{103}$$

*where* $\mathbf{A}$ *is an invertible matrix, and* $\mathbf{b}$ *a vector.*

## E.3 Theoretical analysis

In this section, we develop the theory of IMCA. We will give sufficient conditions that guarantee a strong identifiability of the latent components, and discuss a degenerate case where we only obtain a weaker form of identifiability.

### E.3.1 Definitions

We will first define some sets of distributions which are subsets of the exponential family distribution. We will use $u$ as a dummy variable, and introduce the definitions for the unconditional case. Note that all these definitions apply to the conditional case, when the parameters of the exponential family are a function of an auxiliary variable $\mathbf{y}$. For completeness, we restate here Definition 4.

**Definition 7** (Strongly exponential distributions). *We say that a univariate exponential family distribution with density* $p(u) = \mu(u)e^{\mathbf{T}(u)^T\boldsymbol{\theta} - \Gamma(\boldsymbol{\theta})}$ *is **strongly exponential** if for any subset* $\mathcal{U}$ *of* $\mathbb{R}$ *the following is true:*

$$\left(\exists \boldsymbol{\theta} \in \mathbb{R}^k \,|\, \forall u \in \mathcal{U}, \langle\, \mathbf{T}(u), \boldsymbol{\theta} \,\rangle = const\right) \implies (\Lambda(\mathcal{U}) = 0 \text{ or } \boldsymbol{\theta} = 0) \tag{104}$$

*where* $\Lambda$ *is the Lebesgue measure.*

*We say that that a multivariate distribution is strongly exponential if all its univariate marginals are.*

In other words, the density of a strongly exponential distribution has almost surely the exponential component in its expression and can only be reduced to the base measure on a set of measure zero. This definition is very general, and is satisfied by all the usual exponential family distributions like the Gaussian, Laplace, Pareto, Chi-squared, Gamma, Beta, *etc.* We will only prove identifiability

results for strongly exponential families. The non-strongly exponential case will be explored in future work.

There is a certain class of exponential families for which we can only prove a weak form of identifiability. Loosely speaking, this is because this class doesn't constrain the latent space enough.

**Definition 8** (Quasi-location exponential distributions). *We say that a univariate exponential family distribution with density $p(u) = \mu(u)e^{\mathbf{T}(u)^T\boldsymbol{\theta} - \Gamma(\boldsymbol{\theta})}$ is in the **quasi-location** family if:*

   *(i)* $\dim(\mathbf{T}) = 1$

   *(ii)* $\mathbf{T}$ *is monotonic (either non-decreasing or non-increasing)*

*We say that that a multivariate distribution is quasi-location exponential if all its univariate marginals are.*

As a simple illustration, the Gaussian family with fixed variance is a quasi-location family, but with fixed mean it is not. This is because in the first case, the sufficient statistic is $T(u) = u$ which is a monotonic scalar function, while in the second case it is $T(u) = u^2$, a non-monotonic scalar function.

### E.3.2    Identifiability of the general case

As mentioned in section 3, the IMCA model described by equations (100) and (101) generalizes previous nonlinear ICA models by relaxing the independence assumption required for the latent variables. We propose here to extend the identifiability theory of nonlinear ICA developed in Hyvärinen et al. (2019); Khemakhem et al. (2020) to this new framework.

We start by providing a weaker form of identifiability guarantee that applies to the general case, including quasi-location families.

**Theorem 7.** *Assume the following:*

   *(I)* *The observed data follows the exponential IMCA model of equations (100)-(101).*

  *(II)* *The mixing function* $\mathbf{h} : \mathbb{R}^d \to \mathbb{R}^d$ *is invertible.*

 *(III)* *The conditional latent distribution* $p(\mathbf{z}|\mathbf{y})$ *is strongly exponential (definition 7), and its sufficient statistic is differentiable.*

 *(IV)* *There exist $k + 1$ distinct points $\mathbf{y}^0, \ldots, \mathbf{y}^k$ such that the matrix*

$$\mathbf{L} = (\boldsymbol{\lambda}(\mathbf{y}_1) - \boldsymbol{\lambda}(\mathbf{y}_0), \ldots, \boldsymbol{\lambda}(\mathbf{y}_k) - \boldsymbol{\lambda}(\mathbf{y}_0))$$

   *of size $k \times k$ is invertible, where $k = \sum_{i=1}^d \dim(\mathbf{T}_i)$.*

*Then the IMCA model is weakly identifiable.*

This theorem extends the basic identifiability result of Khemakhem et al. (2020, Theorem 1). It is fundamental as it proves a general identifiability results without the restriction of having independent latent variables. This was previously not considered to be possible and could only be demonstrated in very specific circumstances and under very restrictive additional assumptions (*e.g.*, Monti and Hyvärinen (2018) require *both* non-negativity and orthonormality of a mixing matrix in the *linear* case). In the nonlinear case, to prove Theorem 7, we still require that the latent variables are only dependent through the base measure, while still being independently modulated through the auxiliary variable $\mathbf{y}$. This (and the necessity of having an auxiliary variable) is the price to pay for obtaining identifiability in a nonlinear setting.

### E.3.3    Identifiability of the non quasi-location family

The identifiability result of Theorem 7 is weak because of the presence of the linear transformation $\mathbf{A}$ in equation (103). It turns out that by excluding the quasi-location family (definition 8), we can remove this matrix and achieve a stronger form of identifiability. The main technical result of this paper is the following.

**Theorem 8.** *Assume that the assumptions of Theorem 7 hold. Further assume one of the two following sets of assumptions:*

*(V) The sufficient statistic in (101) is twice differentiable and* $\dim(\mathbf{T}_l) \geq 2, \forall l$.

*(VI) The mixing function* $\mathbf{h}$ *is a* $\mathcal{D}^2$-*diffeomorphism*[6].

*or*

*(V)'* $\dim(\mathbf{T}_l) = 1$ *and* $\mathbf{T}_l$ *is non-monotonic* $\forall l$.

*(VI)'* *The mixing function* $\mathbf{h}$ *is a* $\mathcal{C}^1$-*diffeomorphism*[7].

*Then the IMCA model is strongly identifiable.*

This form of identifiability mirrors the strongest results proven in the nonlinear ICA (Khemakhem et al., 2020, Theorems 2,3), without requiring that the latent components be independent. As far as we know, this is the first proof of the kind for nonlinear representation learning. We further note that this theorem generalizes even existing identifiability theory of the linear case. The mixed case where we have both cases where some sufficient statistics are of dimension greater than 2 and some are univariate and non-monotonic will be studied in future work.

### E.4    Estimation of IMCA by self-supervised learning

A recent development in nonlinear ICA is given by Hyvärinen et al. (2019) where the authors assume they observe data $\mathbf{x} = \mathbf{h}(\mathbf{z})$ following a noiseless conditional nonlinear ICA model $p(\mathbf{z}|\mathbf{y}) = \prod_i p_i(z_i|\mathbf{y})$ For estimation, they rely on a self-supervised binary discrimination task based on randomization to learn the unmixing function. More specifically, from a dataset of observations and auxiliary variables pairs $\mathcal{D} = \{\mathbf{x}^{(i)}, \mathbf{y}^{(i)}\}$, they construct a randomized dataset $\mathcal{D}^* = \{\mathbf{x}^{(i)}, \mathbf{y}^*\}$ where $\mathbf{y}^*$ is randomly drawn from the observed distribution of $\mathbf{y}$. To distinguish between both datasets, a deep logistic regression is used. The last hidden layer of the neural network is a feature extractor whose purpose is to extract the relevant features which will allow to distinguish between the two datasets. Surprisingly, this estimation technique works for IMCA, and is summarized by the following theorem.

**Theorem 9.** *Self-supervised nonlinear ICA estimation algorithms presented in Hyvärinen and Morioka (2016); Hyvärinen et al. (2019) work for the estimation of IMCA.*

### E.5    Proofs

#### E.5.1    Proof of Theorem 7

Consider two different sets of parameters $(\mathbf{h}, \mathbf{T}, \boldsymbol{\lambda}, \mu)$ and $(\tilde{\mathbf{h}}, \tilde{\mathbf{T}}, \tilde{\boldsymbol{\lambda}}, \tilde{\mu})$, defining two conditional latent densities $p(\mathbf{z}|\mathbf{y})$ and $\tilde{p}(\mathbf{z}|\mathbf{y})$. Suppose that the density of the observations arising from these two different models are equal:

$$p(\mathbf{x}|\mathbf{y}) = \tilde{p}(\mathbf{x}|\mathbf{y}) \tag{105}$$

$$\log p(\mathbf{g}(\mathbf{x})|\mathbf{y}) - \log \left| \det \mathbf{J}_{\mathbf{h}}^{-1}(\mathbf{x}) \right| = \log p(\tilde{\mathbf{g}}(\mathbf{x})|\mathbf{y}) - \log \left| \det \mathbf{J}_{\tilde{\mathbf{g}}}(\mathbf{x}) \right| \tag{106}$$

$$\log \mu(\mathbf{g}(\mathbf{x})) + \mathbf{T}(\mathbf{g}(\mathbf{z}))^T \boldsymbol{\lambda}(\mathbf{y}) - \Gamma(\mathbf{y}) - \log |\det \mathbf{J}_{\mathbf{g}}(\mathbf{x})| =$$
$$\log \tilde{\mu}(\tilde{\mathbf{g}}(\mathbf{x})) + \tilde{\mathbf{T}}(\tilde{\mathbf{g}}(\mathbf{z}))^T \tilde{\boldsymbol{\lambda}}(\mathbf{y}) - \tilde{\Gamma}(\mathbf{y}) - \log |\det \mathbf{J}_{\tilde{\mathbf{g}}}(\mathbf{x})| \tag{107}$$

Let $\mathbf{y}_0, \ldots, \mathbf{y}_k$ be the points provided by assumption (IV) of the theorem for $\mathbf{T}$, where $k = \sum_i k_i$, and $k_i = \dim(\mathbf{T}_i)$. We plug each of those $\mathbf{y}_l$ in (107) to obtain $k + 1$ such equations. Then, we subtract the first equation for $\mathbf{y}_0$ from the remaining $k$ equations to get for $l = 1, \ldots, k$:

$$\mathbf{T}(\mathbf{z})^T(\boldsymbol{\lambda}(\mathbf{y}_l) - \boldsymbol{\lambda}(\mathbf{y}_0)) - G(\mathbf{y}_l) = \tilde{\mathbf{T}}(\mathbf{z})^T(\tilde{\boldsymbol{\lambda}}(\mathbf{y}_l) - \tilde{\boldsymbol{\lambda}}(\mathbf{y}_0)) - \tilde{G}(\mathbf{y}_l) \tag{108}$$

where we grouped terms that are only a function of $\mathbf{y}_l$ in $G$ and $\tilde{G}$.

Most importantly, both base measure terms disappear after taking the differences, which is the key enabler of identifiability in the IMCA framework.

The rest of the proof is similar to the proof of Khemakhem et al. (2020, Theorem 1). The only difference is that we don't restrict the sufficient statistics to have equal dimensions, and so we can't use the proof technique from Khemakhem et al. (2020, Theorem 1) without any modification. We present an alternative technique in the proof of Theorem 4, which we refer too for more details. We then conclude that

$$\mathbf{T}(\mathbf{h}^{-1}(\mathbf{x})) = \mathbf{A}\tilde{\mathbf{T}}(\tilde{\mathbf{h}}^{-1}(\mathbf{x})) + \mathbf{b} \tag{109}$$

which implies that the model is weakly identifiable.  □

### E.5.2  Proof of Theorem 8

The conclusion of Theorem 7 is the same as the conclusion of Khemakhem et al. (2020, Theorem 1). Since we make the same assumptions as Khemakhem et al. (2020, Theorems 2,3), the proof to Theorem 8 is similar to the proof of these theorems, which we refer too for more details. The IMCA model is strongly identifiable under the assumptions of Theorem 8.  □

### E.5.3  Proof of Theorem 9

We will first quickly summarize the method proposed in Hyvärinen et al. (2019), and then show how it works for IMCA.

We consider that we observe data $(\mathbf{x}, \mathbf{y})$ that follows the exponential IMCA model of equations (4)-(5). Following Hyvärinen et al. (2019) we start by constructing new data from the observations $\mathbf{x}$ and $\mathbf{y}$ to obtain two datasets

$$\tilde{\mathbf{x}} = (\mathbf{x}, \mathbf{y}) \tag{110}$$
$$\tilde{\mathbf{x}}^* = (\mathbf{x}, \mathbf{y}^*) \tag{111}$$

where $\mathbf{y}^*$ is a random value from the distribution of $\mathbf{y}$ and independent of $\mathbf{x}$. We then proceed by defining a multinomial classification task, where we consider the set of all $\{\tilde{\mathbf{x}}, \tilde{\mathbf{x}}^*\}$ as data points to be classified, and whether they come from the randomized dataset or not as labels. In particular, we train a deep neural network using multinomial logistic regression to perform this classification task. The last hidden layer of the neural network is a feature extractor denoted $\mathbf{s}(\mathbf{x})$. The purpose of the feature extractor is therefore to extract the relevant features which will allow to distinguish between the true dataset $\tilde{\mathbf{x}}$ and the randomized dataset $\tilde{\mathbf{x}}^*$. The final layer of the network is simply linear, and the regression function takes the form

$$r(\mathbf{x}, \mathbf{y}) = \mathbf{s}(\mathbf{x})^T \mathbf{v}(\mathbf{y}) + a(x) + b(u) \tag{112}$$

We state now the main result.

**Theorem 9** (Hyvärinen et al. (2019), adapted). *Assume that the assumptions of Theorem 7, and the assumptions (V)-(VI) of Theorem 8 hold. Further assume that we train a nonlinear logistic regression with universal approximation capability to discriminate between $\tilde{\mathbf{x}}$ in (110) and $\tilde{\mathbf{x}}^*$ in (111) with the regression function in (112), where the feature extractor has dimension $d$.*

*Then in the limit of infinite data, the components $s_i(\mathbf{x})$ of the regression function give the latent components up to pointwise nonlinearities.*

*Proof.* The proof of this theorem is inspired by Hyvärinen et al. (2019). By well known theory, after convergence of logistic regression, the regression function equals the difference of the log-densities of the two classes:

$$\sum_{i=1}^{d} s_i(\mathbf{x}) v_i(\mathbf{y}) + a(x) + b(u) = \log p_{\tilde{\mathbf{x}}}(\mathbf{x}, \mathbf{y}) - \log p_{\tilde{\mathbf{x}}^*}(\mathbf{x}, \mathbf{y}^*)$$

$$= \log p(\mathbf{z}, \mathbf{y}) + \log \left| \det \mathbf{J}_{\mathbf{h}}^{-1}(\mathbf{x}) \right| - \log p(\mathbf{z}) p(\mathbf{y}) - \log \left| \det \mathbf{J}_{\mathbf{h}}^{-1}(\mathbf{x}) \right|$$

$$= \log p(\mathbf{z}|\mathbf{y}) - \log p(\mathbf{z})$$

$$= \log \mu(\mathbf{z}) - \log Z(\mathbf{y}) + \sum_{i=1}^{d} \mathbf{T}_i(z_i)^T \boldsymbol{\lambda}_i(\mathbf{y}) - \log p(\mathbf{z})$$

$$\tag{113}$$

where $\mathbf{J}_{\mathbf{h}}^{-1}(\mathbf{x})$ is the Jacobian matrix of $\mathbf{h}^{-1}$ at point $\mathbf{x}$. Let $\mathbf{y}_0, \ldots, \mathbf{y}_k$ be the point provided by assumption (iv). We plug each of those $\mathbf{y}_k$ in (113) to obtain $k + 1$ such equations. We subtract the first equation for $\mathbf{y}_0$ from the remaining $k$ equations to get for $l = 1, \ldots, k$:

$$\sum_{i=1}^{d} s_i(\mathbf{x})(v_i(\mathbf{y}_l) - v_i(\mathbf{y}_0)) + (\mathbf{b}(\mathbf{y}_l) - \mathbf{b}(\mathbf{y}_0)) - \log \frac{Z(\mathbf{y}_l)}{Z(\mathbf{y}_0)} = \sum_{i=1}^{d} \mathbf{T}_i(z_i)^T (\boldsymbol{\lambda}_i(\mathbf{y}_l) - \boldsymbol{\lambda}_i(\mathbf{y}_0)) \quad (114)$$

Interestingly, the term $\log \mu(\mathbf{z})$ cancels out. The rest of the proof is similar to Theorems 4a and 4b. The only minor difference is that the matrix $\mathbf{A}$ will not be square, but it is still full rank, and can be used to prove that $\mathbf{s} \circ \mathbf{h}$ is a point-wise nonlinearity. $\qquad \square$

## Footnotes

[5] the bandwidths satisfy $a = a_N$ and $b = b_N$, and are positive bandwidth sequences which decay to 0 as $N \to +\infty$, where $N$ is the size of the dataset $\mathcal{D}$.

[6]invertible, all second order cross-derivatives of the function and its inverse exist but aren't necessarily continuous

[7]invertible, all partial derivatives of the function and its inverse exist and are continuous