[Reviews · NeurIPS 2020]

Review 1

Summary and Contributions: This manuscript proposes a family of conditional energy-based models for which identifiability conditions exist. In other words, the latent representations identified by these models are unique up to a linear transformation (in the case of weak identifiability) or a permutation (in the case of strong identifiability). In particular, the theory developed leads to a new approach termed IMCA. Experimental testing validates the theoretical claims and shows promising results regarding the usefulness of the extracted representation for downstream processing.

Strengths: This manuscript managed to both be clear and well written while at the same time taking the reader through heavily theoretical work. The approach taken is, to my mind, well motivated and executed on. The experimental section does a great job at validating the theoretical work.

Weaknesses: This work falls on the periphery of my area of expertise, so I’ll try to offer the perspective of someone who is interested and somewhat familiar with this family of methods, but who does not have intricate knowledge of it. I agree that seeking identifiable representations, as defined in the manuscript, is a worthwhile goal, and a promising way to improve downstream processing. The authors came very close to proving full identifiability for the proposed model family, having achieved weak (linear mapping) and strong (permutation) identifiability conditions. However, I’m not sure I follow why strong identifiability represents a substantial step after weak identifiability, in the sense that I struggle to think of a downstream task that can handle strong representations, but not weak. As I understand it, the idea of identifiability explored here was introduced in Khemakem er at. 2020. However, I found it somewhat confusing. I tend to think of identifiability as a property of the system, not the model used to fit the system. The notion of identifiability discussed here seems more related to the notion of degeneracy, i.e., whether, for a given model, there are multiple sets of parameters that lead to equivalent solutions (equally good fits with respect to the cost function employed). Different fields have different naming conventions so if this notion of identifiability is commonly used in related work, there is obviously no issue here. Nevertheless, it might be something worth sparing a few words on. While I think the experimental section did a great job at empirically validating the theoretical results, I was somewhat disappointed with the attempts to show that the identifiable representations obtained do indeed improve downstream processing. Not to detract from the theoretical work here, but my interpretation is that obtaining identifiable representations is not the end goal in itself. Rather, these representations should lead to measurable improvements downstream. To my mind, the application to semi-supervised learning section comes closest to doing this, but it’s a rather barebones attempt. There is no shortage of methods designed to extract latent representations that capture the statistical dependence between two sets of variables, so it would have been interesting to see at least one to two other methods compared in this section. Also, the appendix corresponding to this section (A.5) was also sparse in details. For example, it wasn’t clear from there how the classification took place. Overall, I feel this is a rather important but underexplored component.

Correctness: As far as I can tell, the work presented is correct.

Clarity: I found the manuscript to be a well written and polished piece of work. I also appreciated the extensive supplementary information section.

Relation to Prior Work: As I mentioned above, this paper does not fall squarely in my area of expertise, so it’s hard for me to fully comment on this. As far as I could determine, the manuscript did a good job at putting the proposed approach in the context of existing work.

Reproducibility: Yes

Additional Feedback: Line 12 - “study show” -> study shows Line 29 - “which fulfill” -> which can fulfill Line 86 - “estimate such model” -> estimate this model Equation (2) - I take it this equation should hold for all x in \mathcal{X} Line 107 - “very little assumptions” -> very few assumptions Unless something has changed recently, references don’t follow the usual NeurIPS format. ------------------------------------------------------------------------------------------------- Final comments ------------------------------------------------------------------------------------------------- I found the authors' response to be clear, to the point, and well reasoned. I am satisfied with the proposed changes, and believe they will improve the manuscript further. Furthermore, I agree with all points raised by the remaining reviewers. I believe the changes proposed by the authors address them satisfactorily.


Review 2

Summary and Contributions: This work proposes the following "conditional energy-based model" that they call an ICE-BeeM. Given vector-valued "feature extractors" f_{theta}(x) and g_{theta}(y) (for instance f,g could be neural networks with parameters specified by theta) and for any fixed y, these define a conditional Gibbs distribution over x's with density proportional to e^{-<f_{theta}(x),g_{theta}(y)>}. As they show, this model can approximate, in an asymptotic sense, any conditional density p(x|y) that is everywhere positive on compact support X,Y. A basic question to ask about any latent variable model is whether the underlying parameters are identifiable, maybe modulo some basic symmetries, from the distribution. The main theoretical result of this work is to prove that under certain conditions on f,g of varying strength, the theta's are identifiable in the sense that f_theta, g_theta are either identifiable up to affine transformation, or possibly even just up to coordinatewise scaling and permutation. These conditions pertain to the positivity or differentiability of the feature extractors, the non-degeneracy of their Jacobians, etc. To show these conditions are not vacuous, they show they are satisfied by a family of feedforward leakyrelu nets whose weight matrices and all square submatrices thereof are full rank. Their final result is to port some of these insights to the study of a *generative* model. They propose the following extension of nonlinear ICA to non-product measures. We observe r.v. x computed by a nonlinear function h(z) of a latent variable z. We don't see z, but observe another auxiliary variable y for which the conditional density p(z|y) is a member of an exponential family where the sufficient statistic is just an entrywise function on z but, importantly, the base measure is non-product. Motivated by the identifiability conditions from earlier, they argue that estimating such a model with an ICE-BeeM will recover the latent variables provided certain non-degeneracy conditions on the nonlinear ICA model are satisfied. Experimental results: 1) Identifiability: they provide evidence for stronger identifiability of learnt representations for common image datasets under ICE-BeeM versus an unconditional EBM baseline 2) Transfer learning: they show that ICE-BeeM has the appealing property that if you pretrain f_{theta} on some classes, fix it, and train g_{theta} on the unseen classes, you get better performance compared to other EBM baselines. 3) Semi-supervised learning: they show that training an ICE-BeeM on some classes and using the learnt extractor f_{theta} to classify the unseen classes outperforms doing the same with an unconditional EBM. 4) Estimating IMCA's: given synthetic data from an IMCA, they show that estimating with an ICE-BeeM outperforms previous nonlinear ICA methods like TCL and iVAE when the base measure of the IMCA is non-product and is competitive even when the base measure is product. UPDATE: Thanks for the thorough response! The relaxed conditions mentioned in the rebuttal/Appendix C.2 do sound more compelling. Overall it seems like this important line of work is still in its early stages, and this submission seems to be a substantial improvement over earlier works. I am upgrading my score to a 7.

Strengths: The conditional EBM model proposed in this work is quite natural and flexible, and the experimental results, both for transfer/semi-supervised learning as well as for nonlinear ICA with non-product latents, are promising. The condition for strong identifiability based on augmenting f with its entrywise square is also a nice idea, and in general it seems that the identifiability conditions this paper proposes are milder and more principled than those put forth in previous papers in this line of work.

Weaknesses: While the identifiability conditions proposed are milder than existing ones for related models, they still feel quite strong. At least, that the weight matrices and their submatrices in the MLP example should be full rank for Theorems 1/2 to apply (and presumably would need to be well-conditioned for identifiability to hold in a robust sense) seems a bit artificial. It would be great if the authors could add some discussion for why, in some cases, some of these conditions might be necessary. A more general complaint is the focus on exact identifiability, which a priori would not rule out a situation where two distributions that are close but not necessarily identical are realized by very different choices of parameters. No doubt this is a more challenging question, but without any mathematical evidence for the robustness of a proposed identifiability condition, it feels like the theory is not really telling us much about reality. Granted, it appears all works in this literature focus on exact identifiability, so this is not a weakness of this paper specifically but perhaps something to think about more for future work.

Correctness: I have verified that the theorems are correct and that the empirical methodology is sound.

Clarity: The paper is clearly written, and all of the results, both experimental and theoretical, are easy to parse.

Relation to Prior Work: The paper does a good job of situating itself in the nonlinear ICA literature. One thing that could further strengthen the discussion is to explain to readers unfamiliar with the specifics of the previous papers how exactly the identifiability results in this paper differ from previous identifiability results for nonlinear deep latent variable models. It appears that one conceptual win of this paper is the fact that the identifiability conditions only depend on properties of the feature extractors as vector-valued functions. Is this really a new thing in this literature? Given the generality of the definition of an ICE-BeeM, it seems somewhat silly to show identifiability in terms of something else like the architecture of the feature extractors. If this is indeed the case, it would be great to spell it out in a bit more detail to emphasize the conceptual clarity that this paper adds to this line of work.

Reproducibility: Yes

Additional Feedback: Regarding the above complaint about robust identifiability, seems plausible that for your Theorem 1, e.g., replacing invertibility with a condition number bound might yield robust identifiability with a bit more work? If so, it would be good to note this somewhere.


Review 3

Summary and Contributions: This paper is to develop the theory in identifiability which appears to be a new and important concept in machine learning. The contribution of this paper is the theorical guarantees for an energy-based model to arrive at an identifiable solution. The paper proposes two models, (1) ICE-BeeM a model that guarantees a unique representation for their identifiability that are unique up to a linear transformation. (2) ICMA a deep latent variable model where the latent variables are non-independent, which is essentially applying the non-linear ICA equations to ICE-BeeM. These models proposed have the capability to be a universal approximator and theorems on identifiability to ensure that the latent features are meaningful, disentangled and interpretable and be a universal approximator.

Strengths: The authors have introduced and motivated the problem very well. They have given clear definitions of weak and strong identifiability and have produced two theorems which links their proposed model back to these two definitions. The authors have also shown that this model could be a universal approximator. Theorem 1 and Theorem 2 have shown that the energy-based approach creates feature extractors, that will guarantee their identifiability by learning a representation that is unique up to a linear transformation.

Weaknesses: I have tried very hard to find some limitations and weaknesses to this paper. But I was unable to find anything worthwhile mentioning here. Any unanswered questions in the main paper for me appears to be answered in the supplementary materials for me.

Correctness: I found the explanation of the mean correlation coefficient (MCC) in appendix A.2 a bit hard to follow. Without any references and a brief search on the internet, I was unable to improve my understanding of this metric used to evaluate the model in the experiments section. After reading this section, I understand how this metric is calculated, but I have no intuition on why this is a good metric. Are the authors able to elaborate on this? It seems reasonable, but non-standard. I am thinking, are there any other better metrics for this? I do agree if there is an invariant of permutations it is a bit difficult to measure how “good” your results are.

Clarity: The paper is generally well written. However, what is not so clear to me is how the authors went from the equations for non-linear ICA in Equation (4),(69)&(70) to its density equation in Equation (5)&(71) and why it may be appropriate to approximate \Gamma(y) with H(f(x))g(x). This substitution seems very arbitrary to me. But I am sure the authors have a very good reason for doing this. Are the authors able to clarify where the density equation comes from and why the H(f(x))g(x) is a reasonable reformulation?

Relation to Prior Work: Yes. This work appears to be an extension of the work done in Hyvärinen and Morioka (2016, 2017) and Khemakhem et al (2020) for identifiability. From my understand, this appears to be a new, but an important concept faced in machine learning to find meaningful and principled disentanglement. This work is different from the previous work by providing the formal definitions of weak and strong identifiability and providing two theorems to support this discussion.

Reproducibility: Yes

Additional Feedback: Overall, I thought this paper is very novel and addresses a very important and new concept in identifiability. It certainly a very important issue in many unsupervised learning problems where the latent factors cannot be identified. This concept of identifiability was entirely unknown to me before reviewing this paper. But after reading this paper, I do see many important applications which will require the theocratical guarantees presented in this paper. I have little negative comments to say about this paper besides the very minor comments I have made on the correction and clarity section. ------------------------------------------- Post-Author Feedback Comments: ------------------------------------------- The authors' feedback is well written and has mentioned that they will revise the paper to address any of the clarity issues I had with the MCC metric and the steps to obtain Equation (5), which was not clear in the original submission. The concerns raised by other reviewers appear to be minor to me and have been answered very well in the authors' feedback. My overall appreciation for this paper still remains the same. Identification is a problem which is faced in many unsupervised learning problems. I believe this research direction in identification is an extremely valuable for the NeurIPS community. After reading the reviewers from other reviewers and the authors' response, I would like to maintain the same rating for this submission.


Review 4

Summary and Contributions: The paper investigates conditions in which identifiable (only one representation possible in the limit of infinite samples) feature extractors can be learned and proposes a framework for developing non-linear identifiable deep models with arbitrary latent dependency structures.

Strengths: Being able to guarantee the uniqueness of the learned latent representation can be very important in some deep learning applications (as the authors explain in the paper). The theoretical work is sound and continues a line of research investigating identifiable models, which also means that it is not entirely novel. However, the submission extends previous related work to non-linear and therefore more expressive models, which is a significant contribution. The empirical results show that their method ICE-BeeM learns identifiable models and the resuls are sufficiently convincing. Interesting transfer learning and semi-supervised learning applications of the proposed approach are shown in the results section. Using two classic benchmark datasets, the authors convincingly show that pre-training with their method helps the models generalize to previously unseen classes.

Weaknesses: Identifiability can be a very interesting property of deep models and may lead to important discoveries in the future, however the current applications of the results presented in this paper are relatively limited.

Correctness: All the claims seem correct and the methodology used in the paper is sound and convincing.

Clarity: The paper is very clearly written.

Relation to Prior Work: The relation to prior work is clearly discussed in the paper.

Reproducibility: Yes

Additional Feedback: Comments after reading author rebuttal: I think the authors have addressed all the reviewers comments adequately and I see no reason to modify my review. My comments regarding applicability of the method have been addressed by the authors. I think it is fair to have a more theoretical submission now followed by other submissions that dig deeper into the applications of the method. However, I do think that the lack of clear and relevant applications in the present paper limits how impactful/relevant this submission is for the community.

[Author Response · NeurIPS 2020]

We thank the reviewers for their valuable time and comments. All suggestions will be incorporated in the next revision. Before addressing each of the reviewers' individual comments as best as we can, we want to stress the fact that we do provide extensive and comprehensive experimental results, designed to validate the theoretical results, rather than achieving SOTA performance. We do not deny the importance of pushing the performance of any system to its limit, but we preferred to focus on: $(i)$ proving that identifiability can be achieved in practice by our model; $(ii)$ provide evidence that identifiability can improve certain downstream tasks like the transfer learning and semi-supervised learning examples; $(iii)$ provide evidence on the flexibility of our model, and its capacity to solve both nonlinear ICA and IMCA problems. We also want to bring attention to the fact that identifiability is important in its own right. Whilst improving performance of downstream tasks is indeed an important application of deep models, it would also be hugely beneficial to be able to effectively employ such models to perform principled statistical inference. This latter goal cannot be achieved unless we first achieve basic theoretical results such as identifiability.

**Reviewer 1:** We thank the reviewer for their time, and for providing a complementary view on the topic. *Definition of identifiability*: As pointed out by the reviewer, we use the definition of identifiability usually found in statistics. Rightly, in the context of probability densities modeled by neural networks, this can be seen as a study of degeneracy of the networks. To avoid confusion, we will add a discussion of how our definition fits within the spectrum of definitions encountered in other fields. *Strong identifiability*: We will clarify that a case where strong identifiability is crucial is causal discovery (Monti et al UAI2019): the linear indeterminacy of weak identifiability will change the causal ordering, making this task impossible. *Semi-supervised learning experiments*: The classification was performed using a logistic regression. We will add this to the manuscript. The reviewer correctly notes that there is a wide range of representation learning algorithms that could have been used in the semi-supervised learning experiments. However, the purpose of this section was to highlight the benefits of identifiability. If we compared the ICE-BeeM approach to such alternative methods we would not know if the performance difference is due to identifiability or something else.

**Reviewer 2:** We thank the reviewer for their time and suggestions. *Robustness of identifiability*: We find this suggestion by the reviewer fascinating. As mentioned by the reviewer, work in the literature focuses on exact identifiability. Questions like asymptotic variance, performance bounds, and robustness are natural and important directions to follow in the future, to strengthen pre-existing and novel identifiability results. We note that almost no such analysis has been done in the nonlinear ICA literature so far (we are only aware of Sasaki et al, UAI2020). This question certainly merits to be treated as a separate project. Nevertheless, we will add such discussion to the manuscript. *Relation to previous work*: The reviewer is correct about how we relate to previous work. Our conditions are purely "functional" — no assumptions on how the feature distribution are made, contrary to previous work. Our results also extend to the overcomplete case, which has never been covered before. We will add more detail on what was done in previous work to better situate our important contributions. *Strength of conditions*: Condition 2 of Theorem 1 can be replaced by differentiability and "full rankness" of the Jacobian of $\mathbf{g}_\theta$ in just a single point, but this requires the conditioning variable to be continuous. Similarly, Condition 1 of Theorem 1 can be relaxed, and requires the "full rankness" of the Jacobian of $\mathbf{f}_\theta$ to hold only in one point, as we mention in Appendix C.2. In fact, This condition can be scrapped altogether if we relax the definition of the equivalence class in Appendix C.1 to have no conditions on the ranks of matrices $\mathbf{A}_1$ and $\mathbf{A}_2$. This however comes at the expense of a relatively weak, and potentially meaningless, equivalence class. In practice, random initialization of floating point parameters, which are then optimized with stochastic updates (SGD), will result in weights that are almost certainly full rank. We can also encourage this behaviour by adding spectral normalization to the network. *Intuition behind the MLP conditions*: The conditions we present for the MLP example are necessary to satisfy the assumptions of Theorems 1 and 2. They are also necessary to ensure that the learnt representations are not degenerate, since we lose information with low rank matrices. A more detailed discussion of these conditions will be added to the manuscript. The goal behind this example was to translate the functional assumptions of Theorems 1 and 2 into architectural assumptions, and bridge the gap between theoretical models and practical implementations. However, as pointed out by the reviewer, this first iteration might seem artificial, and we intend on improving the assumptions in future work.

**Reviewer 3:** We thank the reviewer for their thorough read. *On the intuition behind the MCC*: The MCC metric between two representations A and B computes the maximum linear correlations up to any permutation of components. The permutation invariance is required as (similar to linear ICA) we do not have any guarantees on the order of components. We will add more details and some examples to section A.2 to make it easier to grasp. *On the intuition behind fitting ICE-BeeM to IMCA*: The output nonlinearities $\mathbf{H}_l$ play the role of sufficient statistics to the learnt representation $\mathbf{f}_\theta(\mathbf{x})$. Their counterpart in equation (71) is the vector-valued sufficient statistics $\mathbf{T}_i$. We use this trick to ensure that the dot products in equations (71) and (74) happen in the same space, so that we can make conclusions involving square matrices. We agree that this trick is not well explained in the manuscript, and we will amend that.

**Reviewer 4:** We thank the reviewer for their time and valuable comments. As pointed out above, we decided to dedicate the present manuscript to the theoretical study and a basic empirical validation of the theorems as well as the utility of identifiability; exploring applications of identifiable models in greater depth is an important topic for future work.

[Meta-Review · NeurIPS 2020]

All reviewers find this work interesting and solid. I agree that this paper studies an important theoretical question of idenfiability of energy-based models, and thus may be of interest to the community. The experiments are not that strong as reviewers point out, but that is understandable for a theoretical paper. I recommend acceptance.